# A draft human pangenome reference

Wen-Wei Liao[1,2,3,60], Mobin Asri[4,60], Jana Ebler[5,6,60], Daniel Doerr[5,6], Marina Haukness[4], Glenn Hickey[4], Shuangjia Lu[1,2], Julian K. Lucas[4], Jean Monlong[4], Haley J. Abel[7], Silvia Buonaiuto[8], Xian H. Chang[4], Haoyu Cheng[9,10], Justin Chu[9], Vincenza Colonna[8,11], Jordan M. Eizenga[4], Xiaowen Feng[9,10], Christian Fischer[11], Robert S. Fulton[12,13], Shilpa Garg[14], Cristian Groza[15], Andrea Guarracino[11,16], William T. Harvey[17], Simon Heumos[18,19], Kerstin Howe[20], Miten Jain[21], Tsung-Yu Lu[22], Charles Markello[4], Fergal J. Martin[23], Matthew W. Mitchell[24], Katherine M. Munson[17], Moses Njagi Mwaniki[25], Adam M. Novak[4], Hugh E. Olsen[4], Trevor Pesout[4], David Porubsky[17], Pjotr Prins[11], Jonas A. Sibbesen[26], Jouni Sirén[4], Chad Tomlinson[12], Flavia Villani[11], Mitchell R. Vollger[17,27], Lucinda L. Antonacci-Fulton[12], Gunjan Baid[28], Carl A. Baker[17], Anastasiya Belyaeva[28], Konstantinos Billis[23], Andrew Carroll[28], Pi-Chuan Chang[28], Sarah Cody[12], Daniel E. Cook[28], Robert M. Cook-Deegan[29], Omar E. Cornejo[30], Mark Diekhans[4], Peter Ebert[5,6,31], Susan Fairley[23], Olivier Fedrigo[32], Adam L. Felsenfeld[33], Giulio Formenti[32], Adam Frankish[23], Yan Gao[34], Nanibaa' A. Garrison[35,36,37], Carlos Garcia Giron[23], Richard E. Green[38,39], Leanne Haggerty[23], Kendra Hoekzema[17], Thibaut Hourlier[23], Hanlee P. Ji[40], Eimear E. Kenny[41], Barbara A. Koenig[42], Alexey Kolesnikov[28], Jan O. Korbel[23,43], Jennifer Kordosky[17], Sergey Koren[44], HoJoon Lee[40], Alexandra P. Lewis[17], Hugo Magalhães[5,6], Santiago Marco-Sola[45,46], Pierre Marijon[5,6], Ann McCartney[44], Jennifer McDaniel[47], Jacquelyn Mountcastle[32], Maria Nattestad[28], Sergey Nurk[44], Nathan D. Olson[47], Alice B. Popejoy[48], Daniela Puiu[49], Mikko Rautiainen[44], Allison A. Regier[12], Arang Rhie[44], Samuel Sacco[30], Ashley D. Sanders[50], Valerie A. Schneider[51], Baergen I. Schultz[33], Kishwar Shafin[28], Michael W. Smith[33], Heidi J. Sofia[33], Ahmad N. Abou Tayoun[52,53], Françoise Thibaud-Nissen[51], Francesca Floriana Tricomi[23], Justin Wagner[47], Brian Walenz[44], Jonathan M. D. Wood[20], Aleksey V. Zimin[49,54], Guillaume Bourque[55,56,57], Mark J. P. Chaisson[22], Paul Flicek[23], Adam M. Phillippy[44], Justin M. Zook[47], Evan E. Eichler[17,58], David Haussler[4,58], Ting Wang[12,13], Erich D. Jarvis[32,58,59], Karen H. Miga[4], Erik Garrison[11✉], Tobias Marschall[5,6✉], Ira M. Hall[1,2✉], Heng Li[9,10✉] & Benedict Paten[4✉]

Here the Human Pangenome Reference Consortium presents a first draft of the human pangenome reference. The pangenome contains 47 phased, diploid assemblies from a cohort of genetically diverse individuals[1]. These assemblies cover more than 99% of the expected sequence in each genome and are more than 99% accurate at the structural and base pair levels. Based on alignments of the assemblies, we generate a draft pangenome that captures known variants and haplotypes and reveals new alleles at structurally complex loci. We also add 119 million base pairs of euchromatic polymorphic sequences and 1,115 gene duplications relative to the existing reference GRCh38. Roughly 90 million of the additional base pairs are derived from structural variation. Using our draft pangenome to analyse short-read data reduced small variant discovery errors by 34% and increased the number of structural variants detected per haplotype by 104% compared with GRCh38-based workflows, which enabled the typing of the vast majority of structural variant alleles per sample.

The human reference genome has formed the backbone of human genomics since its initial draft release more than 20 years ago[2]. The primary sequences are a mosaic representation of individual haplotypes containing one representative scaffold sequence for each chromosome. There are 210 Mb of gap or unknown (151 Mb) or computationally simulated sequences (59 Mb) within the current GRCh38 release, constituting 6.7% of the primary chromosome scaffolds. Missing reference sequences create an observational bias, or streetlamp effect, which limits studies to be within the boundaries of the reference. Recently, the Telomere-to-Telomere (T2T) consortium finished the first complete sequence of a haploid human genome, T2T-CHM13, which provides a contiguous representation of each autosome and of chromosome X, with the exception of some ribosomal DNA arrays that remain to be fully resolved[3]. Using T2T-CHM13 directly improves genomic analyses; for example, discovering 3.7 million additional single-nucleotide polymorphisms (SNPs) in regions non-syntenic to GRCh38 and better representing the true copy number variants (CNVs) of samples from the 1000 Genomes Project (1KG) compared with GRCh38 (refs. 1,4).

Although T2T-CHM13 represents a major achievement, no single genome can represent the genetic diversity of our species. Previous studies have identified tens of megabases of sequence contained within structural variants (SVs) that are polymorphic within the population[5]. Owing to the absence of these alternative alleles from the reference genome, more than two-thirds of SVs have been missed in studies that used short-read data and the human reference assembly[6-8], despite individual SVs being more likely to affect gene function than either individual SNPs or short insertions and deletions (indels)[9,10].

To overcome reference bias, a transition to a pangenomic reference has been envisioned[11,12]. Pangenomic methods have rapidly progressed over the past few years[13-15] such that it is now practical to propose that common genomic analyses use a pangenome. Here we sequence and assemble a set of diverse individual genomes and present a draft human pangenome, the first release from the Human Pangenome Reference Consortium (HPRC)[15]. These genomes represent a subset of the planned HPRC panel, which aims to better capture global genomic diversity across the 700 haplotypes of 350 individuals.

## Assembling 47 diverse human genomes

We assembled 47 fully phased diploid assemblies from genomes selected to represent global genetic diversity (Fig. 1a) and for which consent had been given for unrestricted access. All assemblies have been made publicly available, along with all data and analyses. These assemblies include 29 samples with long and linked read sequencing data generated entirely by the HPRC and 18 samples sequenced by other efforts[16-18]. In some cases, we supplemented the 18 additional samples with further sequencing. We selected the 29 HPRC samples from the 1KG lymphoblastoid cell lines, limiting selection to those lines classified as karyotypically normal and with low passage (to avoid artefacts from cell culture). We also ensured that the cell lines were derived from participants for whom whole-genome sequencing (WGS) data were available for both parents (for haplotype phasing). Cell lines meeting these criteria were prioritized by genetic and biogeographic diversity (Methods).

We created a consistent set of deeply sequenced data types for every sample (Supplementary Table 1). The data included Pacific Biosciences (PacBio) high-fidelity (HiFi) and Oxford Nanopore Technologies (ONT) long-read sequencing, Bionano optical maps and high-coverage Hi-C Illumina short-read sequencing for all HPRC samples. We also gathered previously generated high-coverage Illumina sequencing data for both parents of each participant[19]. We generated on average 39.7× HiFi sequence depth of coverage for the 46 HPRC samples (excluding HG002, which had around 130× coverage). This depth of coverage is consistent with the requirements for high-quality, state-of-the-art assemblies[20] and facilitates comprehensive variant discovery irrespective of allele frequency (AF). The N50 value, which represents the shortest read length at which 50% of the total sequenced bases are covered by considering only equal or longer reads, was 19.6 kb on average for the HiFi reads (Supplementary Table 1; excluding HG002 because it was sequenced using a different library preparation protocol).

For the core assembler, we chose Trio-Hifiasm[20] after detailed benchmarking of several alternatives[21]. Trio-Hifiasm uses PacBio HiFi long-read sequences and parental Illumina short-read sequences to produce near fully phased contig assemblies. The complete assembly pipeline (Supplementary Fig. 1 and Methods) included steps to remove adaptor and nonhuman sequence contamination and to ensure a single mitochondrial assembly per maternal assembly.

### Assembly assessment

We first searched for large-scale misassemblies, looking for gene duplication errors, phasing errors and interchromosomal misjoins (Methods). We manually fixed three large duplication errors and one large phasing error, but left smaller errors, which are difficult to definitively distinguish from SVs. We found 217 putative interchromosomal joins. Only

one of these joins (in the paternal assembly of HG02080) was located in a euchromatic, non-acrocentric region and was manually confirmed to be a misassembly. The remaining joins involved the short arms of the acrocentric chromosomes (Fig. 1b and Supplementary Table 2). This may be the result of misalignment, nonallelic gene conversion or other biological mechanisms that maintain large-scale homology between the short arms of the acrocentrics—a phenomenon that we have studied in an associated paper[22].

To evaluate the resulting assemblies after manual correction of errors, we developed an automated assembly quality control pipeline that combined methods to assess the completeness, contiguity, base level quality and phasing accuracy of each assembly (Supplementary Table 3 and Methods). Haploid assemblies containing an X chromosome averaged a total length of 3.04 Gb, 99.3% the length of T2T-CHM13 (3.06 Gb), which also contains an X chromosome. Haploid assemblies containing a Y chromosome averaged a total length of 2.93 Gb, which reflects the size difference between the sex chromosomes (Fig. 1c). The average NG50 value, a widely used measure of contiguity, was comparable with the contig NG50 of GRCh38 (40 Mb compared with 56 Mb, respectively; Fig. 1d). Using short substrings (k-mer values of 31) derived from Illumina data, the Yak k-mer analyser[20] estimated an average quality value (QV) of 53.57 for the assemblies, which corresponded to an average of 1 base error per 227,509 bases (Fig. 1e). To validate these QV estimates, we benchmarked the HG002 and HG005 assembly-based variant calls against the small variants called using Genome in a Bottle (GIAB; v.4.2.1). We estimated QVs of 54 for HG002 and 55 for HG005, which were highly similar to the k-mer QVs estimated using Yak. Consistent with our manual observation that most errors were primarily small indels in low-complexity regions, we found that approximately 32% of the indel errors were in homopolymers longer than 5 bp and an additional 48% were in tandem repeats and low-complexity regions. Moreover, about 42% of the indel errors were genotype errors, mostly heterozygous variants incorrectly called as homozygous variants due to collapsed haplotypes in the two assemblies of an individual (Supplementary Table 4). We next used Yak to analyse the phasing accuracy between the maternal and paternal assemblies using k-mer values derived from Illumina sequencing of the parents. An average haplotype switch error rate of 0.67% was observed and a Hamming error rate of 0.79% (Fig. 1f). We also calculated phase accuracy using Pstools[23,24], which uses Hi-C sequence data of the sample not used to create the assembly. Pstools reported slightly lower switch error rates than Yak but comparable Hamming error rates (Supplementary Fig. 2). Taken together, the above results indicate that the assemblies are highly contiguous and accurate.

### Regional assembly reliability

To determine which portions of the assemblies are reliable, we developed a read-based pipeline, Flagger, that detects different types of misassemblies within a phased diploid assembly (Fig. 1g and Methods). The pipeline works by mapping the HiFi reads to the combined maternal and paternal assembly in a haplotype-aware manner. It then identifies coverage inconsistencies within these read mappings that are likely to be due to assembly errors. This process is similar to likelihood-based approaches, which assess the assembly given the reads[25], but is adapted to work with long reads and diploid assemblies. Using Flagger, we identified only 0.88% (26.4 Mb) of each assembly as unreliable (Fig. 1h and Supplementary Table 5). Using T2T-CHM13, we estimated that 0.09% of reliably assembled blocks were falsely labelled as unreliable (Methods). Compared with the distribution of contig sizes, the unreliable blocks were short (54.6 kb N50 average). We intersected the unreliable blocks in the assemblies from Flagger with different repeat annotations (Fig. 1i and Supplementary Table 6). We estimated that the following percentage of elements were correctly assembled: 95.4% of alpha satellites; 91.5% of human satellites 2 and 3; 97.7% of segmental duplications (SDs); 94.3% of variable number tandem repeats (VNTRs); 94.2% of short tandem repeats (STRs); and 98.8% of all human repeats[26].

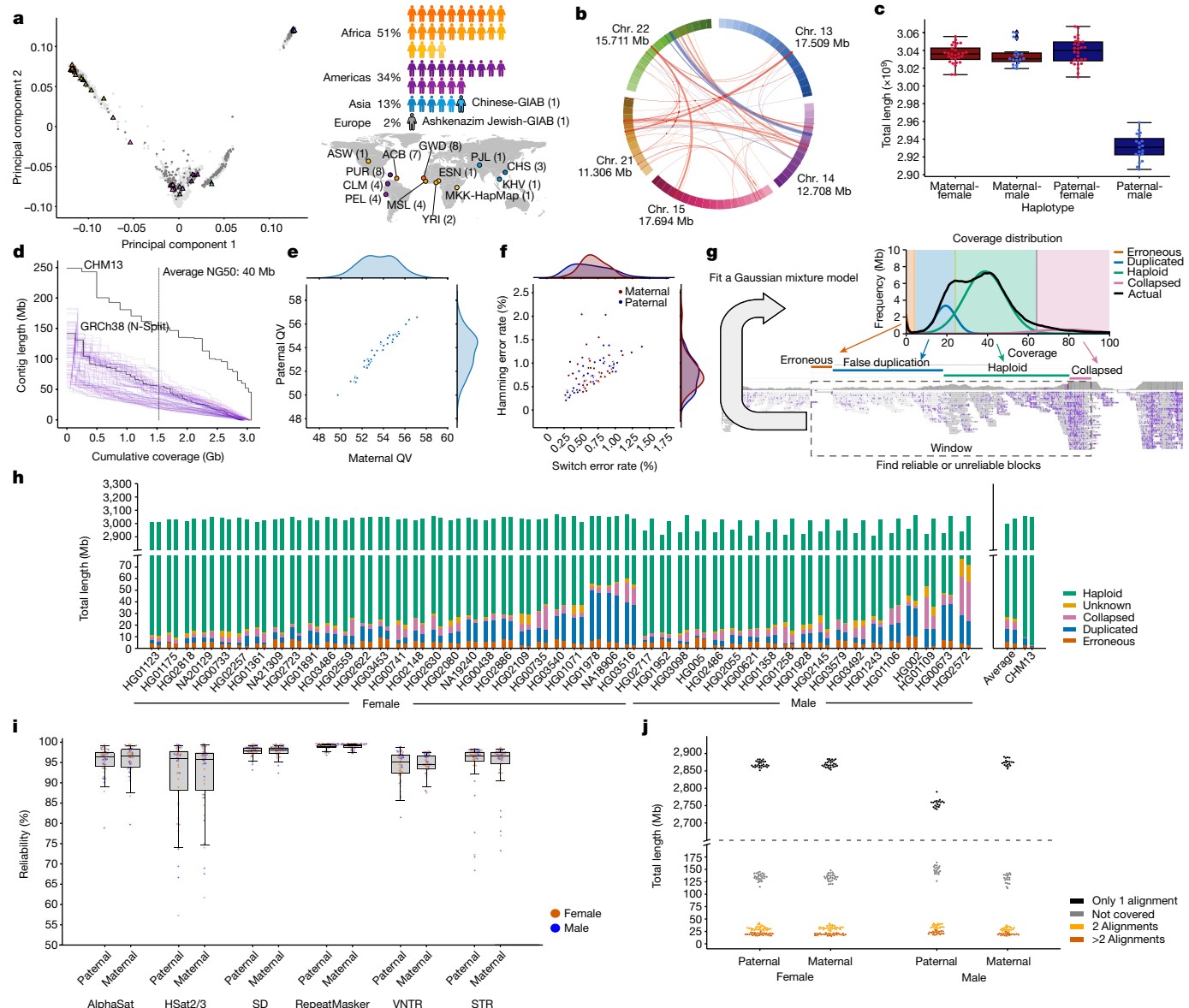

**Fig. 1 | Presenting 47 accurate and near-complete diverse diploid human genome assemblies. a**, Selecting the HPRC samples. Left, the first two principal components of 1KG samples showing HPRC (triangles) samples, excluding HG002, HG005 and NA21309. Right, summary of the HPRC sample subpopulations (three letter abbreviations) on a map of Earth as defined by the 1KG. ACB, African Caribbean in Barbados; ASW, African Ancestry in Southwest US; CHS, Han Chinese South; CLM, Colombian in Medellin, Colombia; ESN, Esan in Nigeria; GWD, Gambian in Western Division; KHV, Kinh in Ho Chi Minh City, Vietnam; MKK, Maasai in Kinyawa, Kenya; MSL, Mende in Sierra Leone; PEL, Peruvian in Lima, Peru; PJL, Punjabi in Lahore, Pakistan; PUR, Puerto Rican in Puerto Rico; YRI, Yoruba in Ibadan, Nigeria. **b**, Interchromosomal joins between acrocentric chromosome short arms. Red, the join is on the same strand; blue, otherwise. **c**, Total assembled sequence per haploid phased assembly. **d**, Assembly contiguity shown as a NGx plot. T2T-CHM13 and GRCh38 contigs are included for comparison. **e**, Assembly QVs showing the base-level accuracy of the maternal and paternal assembly for each sample. **f**, Yak-reported phasing

accuracy showing the switch error percentage versus Hamming error percentage. **g**, Flagger read-based assembly evaluation pipeline. Coverage is calculated across the genome and a mixture model is fit to account for reliably assembled haploid sequence and various classes of unreliably assembled sequence. For each coverage block, a label is assigned according to the most probable mixture component to which it belongs: erroneous, falsely duplicated, (reliable) haploid, collapsed, and unknown. **h**, Reliability of the 47 HPRC assemblies using read mapping. For each sample, the left bar is the paternal and the right bar is the maternal haplotype. Regions flagged as haploid are reliable (green), constituting more than 99% on average of each assembly. The *y* axis is broken to show the dominance of the reliable haploid component and the stratification of the unreliable blocks. **i**, Assembly reliability of six types of repeats. AlphaSat, alpha satellites; HSat2/3, human satellites 2 and 3. **j**, Completeness of the HPRC assemblies relative to T2T-CHM13. The number of reference bases covered by none, by one, by two or by more than two alignments are included.

## Completeness and CNV

To assess the completeness and copy number polymorphism of the assemblies, we aligned them to T2T-CHM13 (Methods). The paternal assemblies of male samples covered about 92.8% of T2T-CHM13

(excluding chromosome X) on average with exactly one alignment. For all other assemblies (excluding chromosome Y), about 94.1% on average was single-copy covered (Fig. 1j and Supplementary Table 7). On average, around 136 Mb (4.4%) of T2T-CHM13 was not covered by any alignment, which indicates that some parts of the genome are either

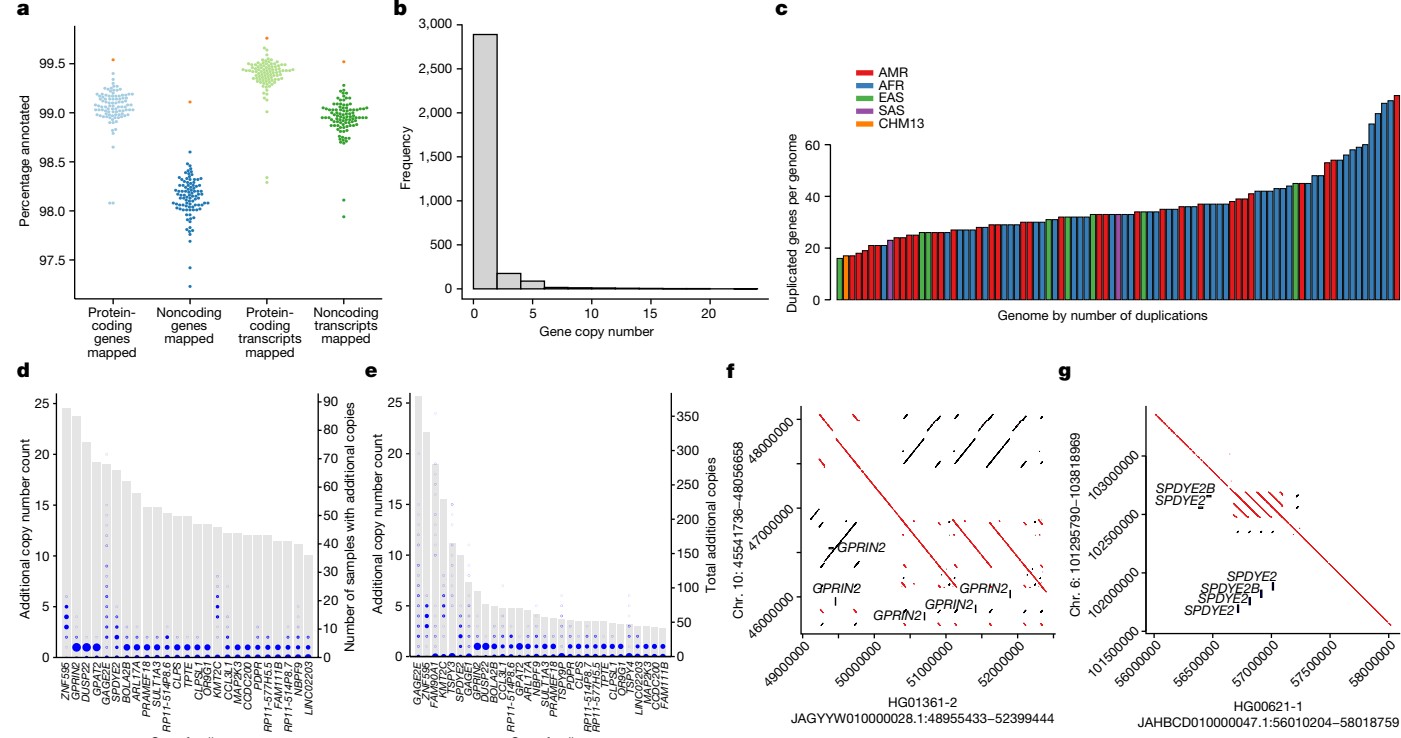

**Fig. 2 | Transcriptome annotation of the assemblies. a**, Ensembl mapping pipeline results. Percentages of protein-coding and noncoding genes and transcripts annotated from the reference set in each of the HPRC assemblies. Orange points represent T2T-CHM13 for comparison. **b**, Frequency of gene copy number. Individual genes may have separate copy number states among genomes, and the frequency reflects 3,210 observed copy number changes among the HPRC genomes. **c**, Number of distinct duplicated genes or gene families per phased assembly relative to the number of duplicated genes annotated in GRCh38 (*n* = 152). The GRCh38 gene duplications reflect families of duplicated genes, whereas the counts in other genomes reflect gene duplication polymorphisms. The assemblies are colour coded according to their population of origin. **d**, The top 25 most commonly CNV genes or gene

families in the HPRC assemblies out of all 1,115 duplicated genes, ordered by the number of samples with additional copies relative to GRCh38. Grey bars, the number of samples with additional copies. Blue circles, the number of additional copies per sample, with the size of the circle proportional to the number of samples. **e**, The top 30 most individually copied CNV genes or gene families in the HPRC assemblies, ordered by total number of additional copies observed. Blue circles, the number of additional copies per sample. Grey bars, the total number of additional copies summed over the samples. **f**, Dotplot illustrating haplotype-resolved *GPRIN2* gains in the HG01361 assembly relative to GRCh38. **g**, Dotplot illustrating *SPDYE2–SPDYE2B* haplotype resolved gains within a tandem duplication cluster of the HG00621 assembly relative to GRCh38.

systematically unassembled or cannot be reliably aligned. About 90% of these regions were centromeric or pericentromeric[27] (Extended Data Fig. 1). Despite the majority of unaligned bases occurring within and around centromeres, on average, 90% of divergent and monomeric alpha satellites, gamma satellites and centromeric transition regions were covered by at least one alignment. Excluding the T2T-CHM13 centromere and satellites[3] and including only the expected sex chromosome for each haploid assembly, on average, around 99.12% of the remaining reference was covered by exactly one alignment (Supplementary Table 7).

The average number of T2T-CHM13 bases with two or with more than two alignments was about 32.4 Mb (around 1.0%) and about 20.0 Mb (around 0.6%), respectively. On average, per haploid assembly, these duplicated regions had about 82.20% and 39.82% overlap with the pericentomeric or centromeric satellites and SDs, respectively, and around 94.62% had overlap with either of them. We characterized the accuracy of regions aligned to SDs in T2T-CHM13 (excluding chromosome Y) using a liftover of the assembly read-depth-based evaluation (Extended Data Fig. 2). On average, we estimated that only 2.5% (4.99 out of 199 Mb) of the SD sequence that could be lifted onto T2T-CHM13 was in error according to the read depth. To identify SDs associated with these errors, we took all 5 kb windows across the unreliable regions and intersected them with the longest and most identical overlapping SD. The median length of SDs overlapping sequences in error was 3.0 times longer (288 kb compared with 96.3 kb) than those in correctly

assembled SDs and 1.8% more identical (98.9 compared with 97.1). This result reinforces earlier findings that the length and identity of SDs play an important part in assembly accuracy[28].

## Annotating 47 diverse genomes

We developed a new Ensembl mapping pipeline to annotate GENCODE[29] genes and transcripts within each new haploid assembly (Methods). A median of 99.07% of protein-coding genes (range = 98.08–99.40%) and 99.42% of protein-coding transcripts (range = 98.29–99.66%) were identified in each of the HPRC assemblies (Fig. 2a and Supplementary Table 8). Similarly, a median of 98.16% of noncoding genes (range = 97.23–98.60%) and 98.96% of noncoding transcripts (range = 97.94–99.28%) were similarly annotated. Running this pipeline on T2T-CHM13 produced similar, slightly higher, results. Intersecting the HPRC annotations with the assembly reliability predictions, a median of 99.53% of gene and 99.79% of transcript annotations occurred wholly within reliable regions, which indicated that most of the annotated transcript haplotypes were structurally correct. To examine transcriptome base accuracy, we looked for nonsense and frameshift mutations in the set of canonical transcripts (one representative transcript per gene; Supplementary Fig. 3, Supplementary Table 8 and Methods). We found a median of 25 nonsense mutations and 72 frameshifts per assembly. A median of 21 (84%) and 58 (80%) of these nonsense mutations and frameshifts per assembly, respectively, were

supported by the independently generated Illumina variant call sets. These numbers were within the range of previously reported numbers of loss-of-function mutations (between 10 and 150 per person, depending on the level of conservation of the mutation)[1,30]. Conservatively, if all the non-confirmed frameshifts and nonsense mutations are assembly errors, this would predict 18 such transcript-altering errors per transcriptome (1 per 1.7 million assembled transcriptome bases).

There were 1,115 protein-coding gene families within the Flagger-predicted reliable regions of the full set of assemblies that had a gain in copy number in at least one genome (Fig. 2b). Each assembly had an average of 36 genes with a gain in copy number relative to GRCh38 within its predicted reliable regions, with a bias towards rare, low-copy CNVs (Fig. 2c). In detail, 71% of CNV genes appeared in a single haplotype. Previous studies using read depth found that rare CNVs generally occur outside regions annotated as being enriched in SDs[31]. The genome assemblies confirmed this observation in sequence-resolved CNVs. When stratifying duplicated genes on the basis of AF into singleton (present in one haplotype), low frequency (<10%) and high frequency, 15% (118 out of 771) of the singleton CNVs mapped to SDs as annotated in GRCh38. Duplicated genes with a higher population frequency had a greater fraction in SDs: 59% (83 out of 140) of low frequency and 81% (44 out of 54) of high frequency. Overall, 58 genes were CNVs in 10% or more of haploid assemblies, and 16 genes were amplified in the majority of individuals relative to GRCh38 (Fig. 2d and Supplementary Table 9). Many of these genes were individually highly copy-number polymorphic and part of complex tandem duplications (Fig. 2e). For example, *GPRIN2* is a copy-number polymorphic[32] based on read depth and has a sequence resolution of one to three additional copies duplicated in tandem in the pangenome (Fig. 2f). *SPDYE2* is similarly resolved as one to four additional copies duplicated in tandem (Fig. 2g). Other CNV genes were not contiguously resolved and reflect limitations of the current assemblies (see the associated article[33]). For example, the defensin gene *DEFB107A* has three to seven additional copies assembled across all samples; however, this gene was assembled into three to seven separate contigs that do not reflect the global organization of this gene.

## Constructing a draft pangenome

We used a sequence graph representation for pangenomes[12,14] in which nodes correspond to segments of DNA. Each node has two possible orientations, forward and reverse, and there are four possible edges between any pair of nodes to reflect all combinations of orientations (bidirected graph). The underlying haplotype sequences can be represented as walks in the graph. The model represents a generalized multiple alignment of the genome assemblies from which we built it, whereby haplotypes are aligned where they co-occur on a given node (Fig. 3a).

The process of generating a combined pangenome representation is an active research area. The problem is nontrivial both because of computational challenges (there are hundreds of billions of bases of sequence to align) and because determining which alignments to include is not always obvious, particularly for recently duplicated and repetitive sequences. We applied three different graph construction methods that have been under active development for this project: Minigraph[34], Minigraph-Cactus (MC)[35] and PanGenome Graph Builder (PGGB)[36] (Extended Data Fig. 3 and Methods). The availability of these three models provided us with multiple views into homology relationships in the pangenome while supporting validation of discovered variation by independent methods. We included GRCh38 and T2T-CHM13 references within the pangenomes, and three samples (HG002, HG005 and NA19240) were held out to permit their use in benchmarking (hence 90 haplotypes total). In brief, Minigraph builds a pangenome by starting from a reference assembly, here GRCh38, and iteratively and progressively adds in additional assemblies, recording only SVs ≥ 50 bases. It admits complex variants, including duplications

and inversions. MC extends the Minigraph pangenome with a base level alignment of the homology relationships between the assemblies using the Cactus genome aligner[37] while retaining the structure of the Minigraph pangenome. PGGB constructs a pangenome from an all-to-all alignment of the assemblies. Although both T2T-CHM13 and GRCh38 are used to partition contigs into chromosomes, the PGGB graph is otherwise reference free (that is, it does not base itself on a chosen reference assembly).

## Measuring pangenome variation

The different algorithmic approaches used to construct a pangenome graph influence graph properties while representing the same underlying sequences. The basic properties of the three graphs produced with the different pangenome methods are shown in Supplementary Table 10. The Minigraph chart, by virtue of being limited to structural variation, is the smallest, with more than two orders of magnitude fewer nodes and edges than the base level graphs. Its length (3.24 Gb), measured as the total bases of all nodes, is similar to the MC graph (3.29 Gb) despite the latter adding many small variants. This difference is due to the MC graph also aligning a significant number of sequences left unaligned by Minigraph. The PGGB graph contains roughly 5 Gb more sequence because it includes highly structurally divergent satellite regions omitted from the other approaches (Methods) and does not implement any trimming or filtering of the input assembly contigs.

To characterize variants in the pangenome graphs, we used graph decomposition to identify 'bubble' subgraphs that correspond to non-overlapping variant sites. We then classified variant sites into small variants (<50 bp) and SVs (≥50 bp) of different types (Methods). We found similar numbers of each variant type in each pangenome, with 22 million small variants in the MC graph (21 million in PGGB) (Fig. 3b), and 67,000 SVs in the MC graph (73,000 in PGGB, 75,000 in Minigraph) (Fig. 3c). We assessed variation in each individual assembly by tracing their paths through the graphs and found similar numbers of small variants and SVs within confident genomic regions defined by Dipcall[38]. Specifically, there were 5.34 million small variants per sample and 16,800 SVs per haplotype on average in the MC graph (5.35 million and 17,400, respectively in PGGB) (Fig. 3e,f). Differences in variant counts among samples from different ancestry groups recapitulated previous observations[1]. There was a total of 90 Mb of non-reference sequence in the SV sites, excluding difficult-to-align centromeric repeats, in the MC graph (55 Mb for PGGB, 86 Mb for Minigraph). Alu, L1 and ERV SVs appeared largely biallelic, whereas VNTRs frequently had three or more distinct alleles per site. The minor AF in the pangenomes of biallelic variants was similar for SNPs and for L1, Alu and VNTR variants, although VNTRs showed a slight shift towards more common alleles (Fig. 3d).

We quantified the amount of euchromatic autosomal non-reference (GRCh38) sequence that each of the 44 diploid genomes incrementally contributes to the pangenome (Fig. 3g and Methods) for both MC and PGGB graphs. We limited the analysis to the euchromatic sequence because we were generally confident in its assembly and alignment, and much of the heterochromatic sequence was omitted from the MC graph (Methods). Overall, the euchromatic autosomal non-reference sequence added up to about 175 Mb in the MC graph (around 190 Mb in PGGB), out of which about 55 Mb (around 105 Mb in PGGB) was observed only on a single haplotype. Our analysis further suggested that about 5 Mb and 70 Mb in the MC graph (around 10 Mb and 60 Mb in PCGB) could be attributed to core (present in ≥95% of all haplotypes) and common genomes (present in ≥5% of all haplotypes), respectively (Supplementary Table 11). We also estimated the growth of the euchromatic autosomal pangenome independent of the order of genomes by sampling 200 permutations (Supplementary Fig. 4) and recording the median pangenome size across all samples in the MC graph. Our results indicated that the second genome added around 23 Mb of euchromatic autosomal sequence to the pangenome, whereas the last genome tended to add only about 0.64 Mb. These numbers are

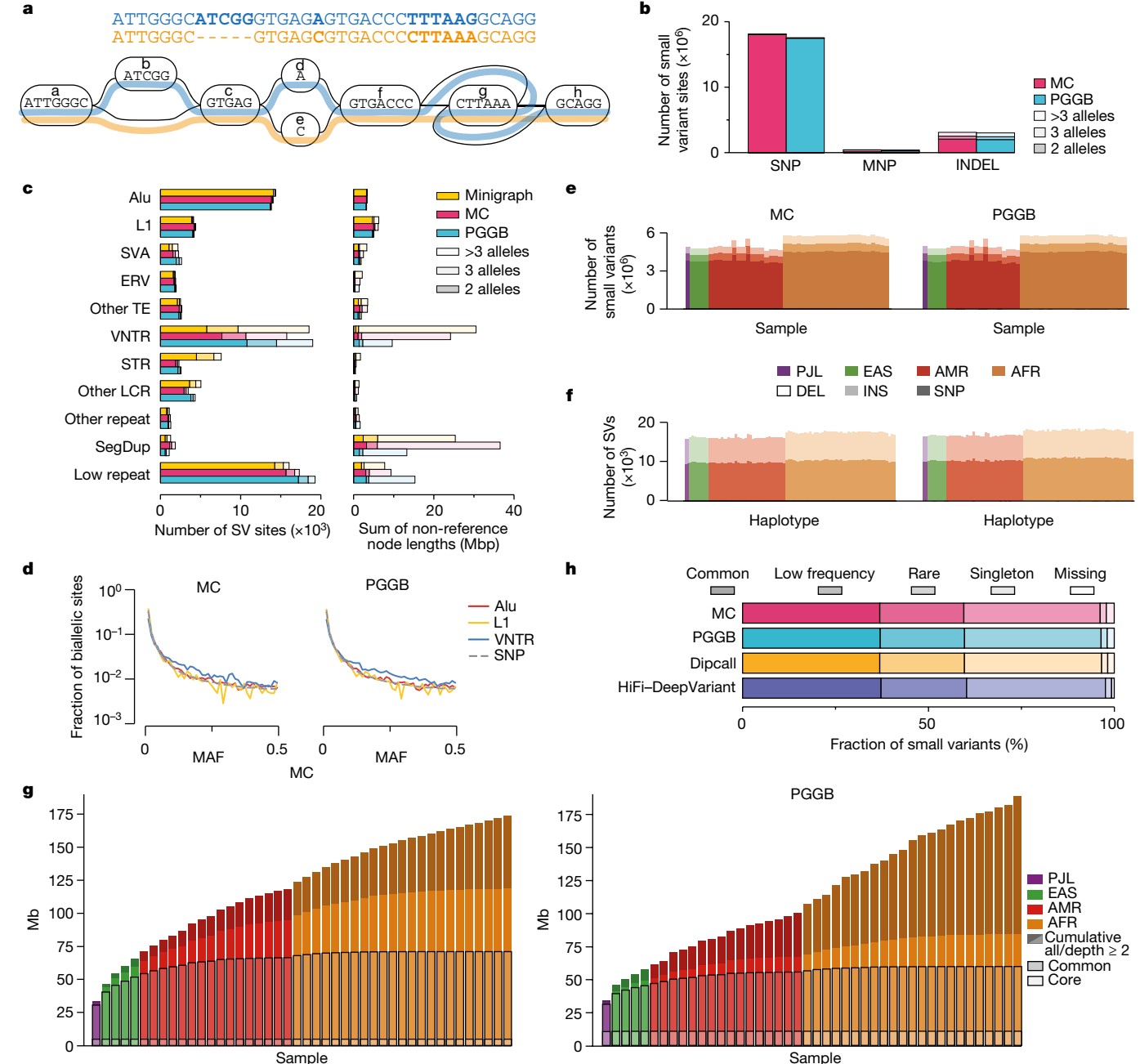

**Fig. 3 | Pangenome graphs represent diverse variation. a**, A pangenome variation graph comprising two elements: a sequence graph, the nodes of which represent oriented DNA strings and bidirected edges represent the connectivity relationships; and embedded haplotype paths (coloured lines) that represent the individual assemblies. **b**, Small variant sites in pangenome graphs stratified by the variant type and by the number of alleles at each site. MNP, multinucleotide polymorphism. **c**, SV sites in the pangenome graphs stratified by repeat class and by the number of alleles at each site. Other TE, a site involving mixed classes of transposable elements (TEs). VNTR, variable-number tandem repeat, a tandem repeat with the unit motif length ≥7 bp. STR, short tandem repeat, a tandem repeat with the unit motif length ≤6 bp. Other LCR, low-complexity regions with mixed VNTR and STR and low-complexity

regions without a clear VNTR or STR pattern. Other repeat, a site involving mixed classes of repeats. SegDup, segmental duplication. Low repeat, a small fraction of the longest allele in a site involving repeats. **d**, Pangenome minor AF (MAF) spectrum for biallelic SNP, VNTR, L1 and Alu variants in the MC and PGGB graphs. **e,f**, Number of autosomal small variants per sample (**e**) and SVs per haplotype (**f**) in the pangenome. Variants were restricted to the Dipcall-confident regions. Samples are organized by 1KG populations. **g**, Pangenome growth curves for MC (left) and PGGB (right). Depth measures how often a segment is contained in any haplotype sequence, whereby core is present in ≥95% of haplotypes, common is ≥5%. **h**, Small variants in the GIAB (v.3.0) 'easy' regions annotated with AFs from gnomAD (v.3.1.2).

conservative owing to additional highly polymorphic sequence residing in assembly gaps. Extrapolating under Heaps' Law[39] (Methods), we anticipate that at least an additional 150 Mb of euchromatic autosomal sequence will be added in the pangenome graph when HPRC produces 700 haplotypes in the future.

We annotated the small variants overlapping the GIAB (v.3.0) 'easy' regions (covering 74.35% of GRCh38) with AFs from gnomAD (v.3.1.2) (Fig. 3h and Supplementary Table 12). In the MC graph, about 60.2% (around 9.7 million variants) had an AF of 1% or greater. About 35.7% were rare, having an AF less than 1% but above zero. About 1.7% were

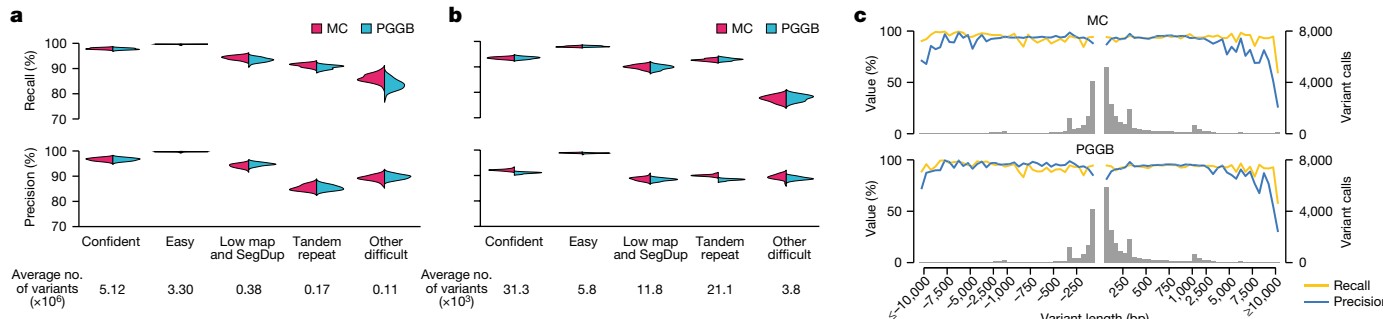

**Fig. 4 | Pangenome graph evaluation. a,b,** Precision and recall of autosomal small variants (**a**) and SVs (**b**) in the pangenomes relative to consensus variant sets. Small variants are compared to HiFi–DeepVariant calls. SVs are compared to the consensus of six reference-based SV callers (Methods). Comparisons are restricted to the Dipcall-confident regions and then stratified by the GIAB (v.3.0) genomic context. **c,** Average SV precision, recall and frequency in the Dipcall-confident regions stratified by length in the MC (top) and PGGB (bottom) graphs relative to consensus SV sets. The histogram bin size is 50 bp for SVs <1 kb and 500 bp for SVs ≥1 kb.

singleton. The remaining 2.4% were missing from gnomAD. Similar results were obtained using the PGGB graph by repeating this exercise with small variant calls detected by pairwise alignment of assemblies to GRCh38 using Dipcall[38] and by calling small variants from the HiFi sequencing data using DeepVariant[40]. Given that 1KG samples are included in gnomAD, these missing variants are expected to be a mixture of false negatives in gnomAD and false positives in the pangenome.

To further explore the quality of variant calls captured by assembly and graph construction, we compared pangenome-decoded variants against GRCh38 to variant sets identified by conventional reference-based genotyping methods (Supplementary Fig. 5 and Methods). These reference-based call sets were generated from the PacBio HiFi reads and haplotype-resolved assemblies using the following different discovery methods: DeepVariant[40], PBSV[41], Sniffles[42] with Iris[43], SVIM[44], SVIM-asm[45], PAV[5] and the Hall-lab pipeline (Methods). For benchmarking small variants, we excluded regions that contained SVs detected or implied by the alignment of the haploid assemblies of that sample to GRCh38, as current benchmarking tools do not account for different representations of small variants inside or near SVs (Methods). Comparing small variants (Fig. 4a) and SVs (Fig. 4b) from the pangenomes to the reference-based sets, we observed a high level of concordance that varied, as expected, by the relative repeat content of the surrounding genome. Overall, variant calling performance was high in both the MC and PGGB graphs. For example, in relatively unique easy genomic regions constituting 75.42% of the autosomal genome, samples showed a mean of 99.64% recall and 99.64% precision for small variants in the MC graph. Meanwhile, in high-confidence regions (around 90% of autosomal genome), samples showed 97.91% recall and 96.66% precision (Fig. 4a). Performance was lower for SVs than for small variants (Fig. 4b), as expected, but was still strong. Variant calling performance was lower in highly repetitive genome regions (3.87% of autosomal genome; Fig. 4a,b), for which more work will be required to achieve high-quality variant maps. These values are likely to be significant underestimates of variant calling quality, considering known errors in the truth set owing to the inherent limitations of reference-based variant callers (see below). Stratifying the insertion and deletion SVs within the pangenome, we observed relatively high levels of agreement with the reference-based methods regardless of length (Fig. 4c).

An independent measure of the quality of the pangenome graphs is the extent to which sample haplotype paths through the graph are well supported by the raw sequencing data. When we calculated the number of supporting reads by aligning them to the MC graph using GraphAligner (Methods), more than 97% of HiFi reads were aligned to the MC graph after filtering (Extended Data Fig. 4, left). We further

calculated the read depth of on-target and off-target edges based on the sample paths in the graph. On average, more than 94% of on-target edges were supported by at least 5 reads, and we observed 2 peaks in the read depth distribution of on-target edges (Extended Data Fig. 4, middle): a minor peak corresponding to the edges in heterozygous regions, and a major peak at twice the minor peak corresponding to the edges in homozygous regions. By contrast, only 7% or fewer off-target edges were supported by at least 5 reads (Extended Data Fig. 4, right). In addition to HiFi reads, we used ONT reads from 29 out of the 44 samples to perform the same analysis. Even though the data were lower in coverage, similar results were obtained (Supplementary Figs. 6 and 7).

These data also show that the pangenome graphs performed better at capturing genome variation than the above benchmarking results imply. For example, a mean of 89.3% of putative false-positive small variant calls were supported by ≥5 HiFi reads, and 75.3% by ≥10 reads (85.9% and 73.8%, respectively, for SVs). This result suggests that most putative errors are in fact real variants that were missed by the reference-based callers used to create the truth set (Supplementary Fig. 8 and Supplementary Table 13).

To assess gene alignments in the pangenome, we used the Comparative Annotation Toolkit (CAT)[46] to liftover GENCODE (v.38) annotations using the MC pangenome alignment onto the individual haplotype assemblies. CAT lifted and annotated a median of 99.1% of 86,757 protein-coding transcripts per assembly (Extended Data Fig. 5, Supplementary Fig. 9, Supplementary Tables 14 and 15 and Methods), making it comparable to the Ensembl-mapping-based pipeline (median of 99.4% per assembly). This result supports the idea that the MC pangenome captures most transcript homologies. When comparing the CAT and Ensembl annotations per assembly, median Jaccard similarities of 0.99 for both genes and transcripts were obtained (Methods). A median of 360 (0.4%) protein-coding transcripts per assembly mapped at different loci between the Ensembl and CAT annotations.

## Pangenomes represent complex loci

We next turned our attention to complex multiallelic SVs, which have historically been difficult to map using reference-based methods. To screen for complex SVs, we identified bubbles >10 kb from Minigraph that exhibited at least five structural alleles among the assembled haplotypes (Methods). We found that 620 out of 76,506 total sites (0.81%) were complex, and 44 of these overlapped with medically relevant protein-coding genes[47] (Supplementary Table 16). Some are well-known complex SV loci, and all are known to be structurally variable based on previous short-read SV mapping studies[10,19,32]. However, whereas previous short-read SV calls at these loci are typically imprecise owing to alignment issues and low-resolution read-depth analysis methods, here we resolved their structure at single-base resolution. We selected

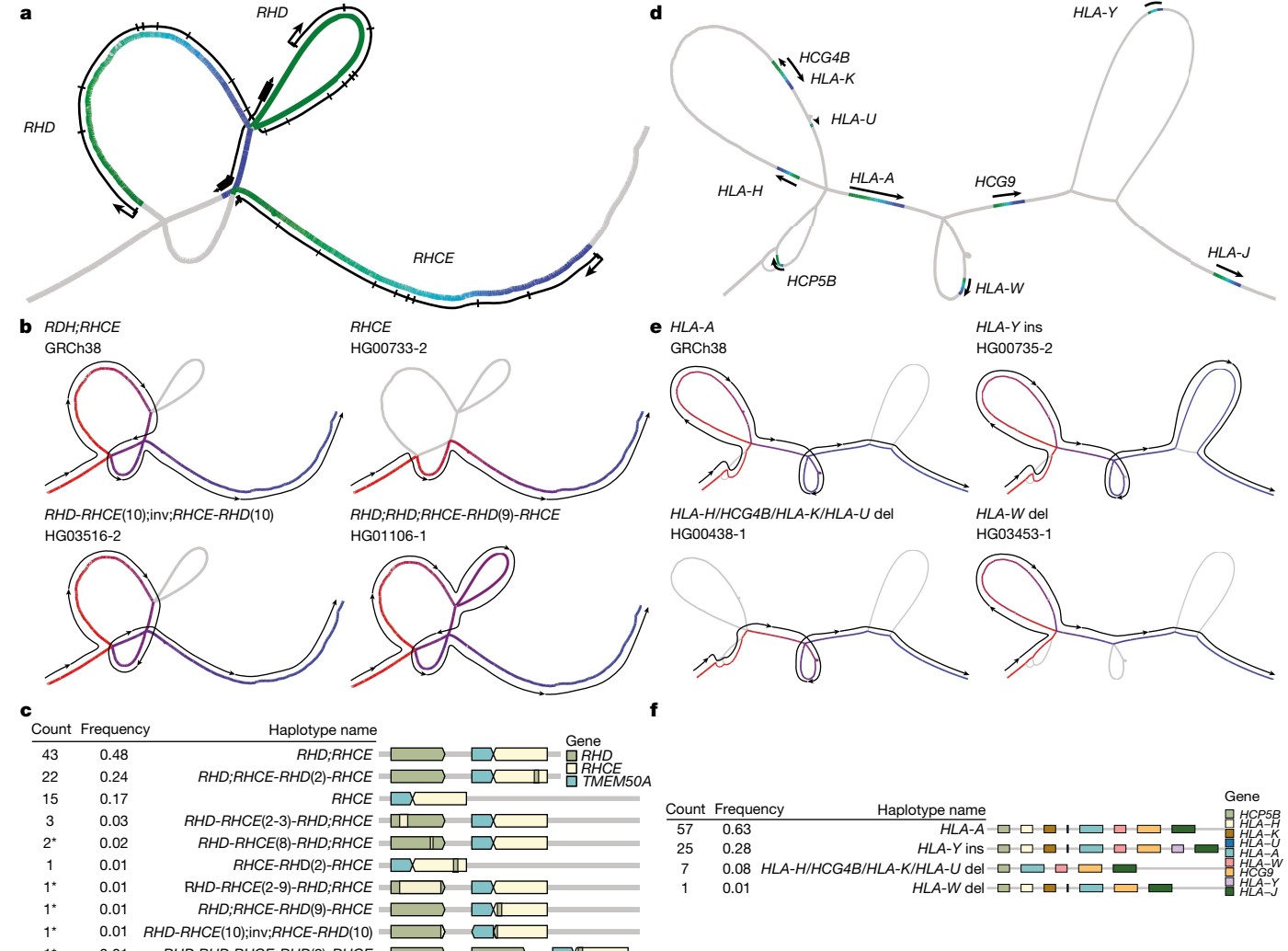

**Fig. 5 | Visualizing complex pangenome loci. a–c**, Structural haplotypes of *RHD* and *RHCE* from the MC graph. Locations of *RHD* and *RHCE* within the graph (**a**). The colour gradient is based on the precise relative position of each gene; green, head of a gene; blue, end of a gene. The lines alongside the graph are based on the approximate position of gene bodies, including exons and transcription start sites. Different structural haplotypes take different paths through the graph (**b**). The colour gradient and lines show the path of each allele; red, start of a path; blue, end of a path. Frequency and linear structural visualization of all structural haplotypes called by the graph among 90 haploid assemblies (**c**). Asterisks indicate newly discovered haplotypes. **d–f**, Structural haplotypes of *HLA-A* from the PGGB graph, visualized using the same conventions as **a–c**. del, deletion; ins, insertion; inv, inversion.

five clinically relevant complex SV loci for detailed structural analysis: *RHD–RHCE*, *HLA-A*, *CYP2D6–CYP2D7*, *C4* and *LPA* (Methods). For each locus and graph, we identified their locations within the graph and then annotated paths within this subgraph with known genes. We traced the individual haplotypes through the subgraph to reveal the structure of each assembly. In *CYP2D6–CYP2D7* (Extended Data Fig. 6), *C4* (Supplementary Fig. 10) and *LPA* (Supplementary Fig. 11), we recapitulated previously described haplotypes. For *CYP2D6–CYP2D7*, our calls matched 96% of haplotypes of 76 assemblies called by Cyrius using Illumina short-read data[48]. Two discrepancies appeared to be caused by errors from Cyrius, and the third was a false duplication in the HG01071-2 pangenome assembly revealed by Flagger. This comparison suggests that the pangenomes faithfully agree with existing knowledge of this complex locus. In *RHD–RHCE* (Fig. 5a–c), in addition to previously described haplotypes, we inferred the presence of five new haplotypes, which included one duplication allele of *RHD* and one inversion allele between *RHD* and *RHCE* that swaps the last exon of both genes. Around *HLA-A* (Fig. 5d–f and Supplementary Fig. 12), two deletion alleles have been previously described—albeit with imprecise breakpoints[10]—but an insertion allele carrying a *HLA-Y* pseudogene was

previously unreported. The long sequence (65 kb) inserted with *HLA-Y* occurred at high frequency (28%) but has little homology to GRCh38.

We also compared the representation of these five loci in the MC and PGGB graphs (Supplementary Fig. 13). Each graph independently recapitulated the same haplotype structures. In general, in the PGGB graph, many SV hotspots, including the centromeres, were transitively collapsed into loops through a subgraph representing a single repeat copy. This feature tends to reduce the size of variants found in repetitive sequences. Assemblies that contained multiple copies of the homologous sequence traversed these nodes a corresponding number of times. By contrast, the MC graph maintained separate copies of these homologous sequences.

## Applications of the pangenome
### Pangenome-based short variant discovery
Our pangenome reference aims to broadly improve downstream analysis workflows by removing mapping biases that are inherent in the use of a single linear reference genome such as GRCh38 or CHM13. As an initial test case, we studied whether mapping against our pangenomes

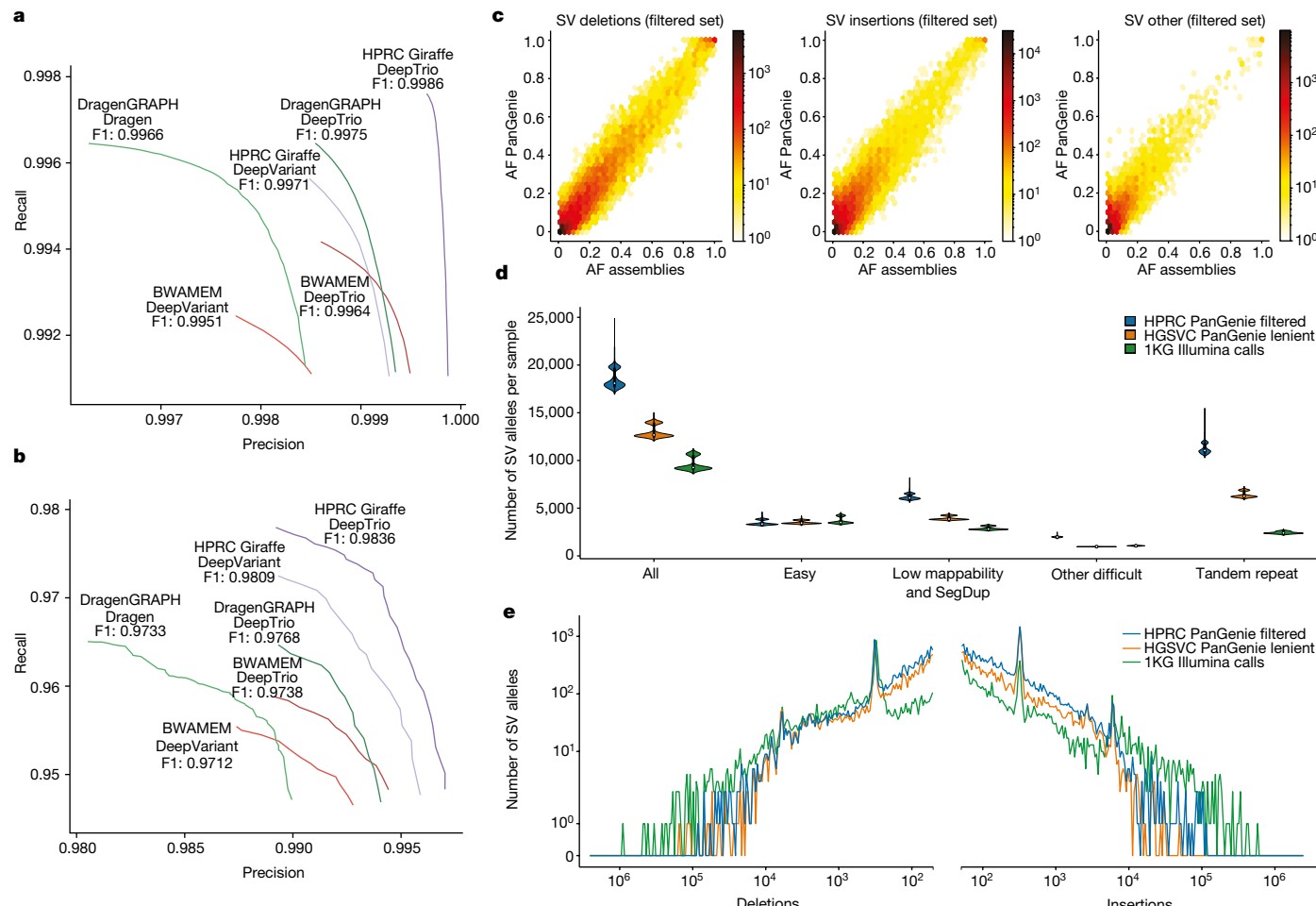

**Fig. 6 | Performance gains for pangenome-aided analysis of short-read WGS data. a,b,** Precision–recall curves showing the performance of different combinations of linear reference and various mappers and variant callers evaluated against the GIAB (v.4.2.1) HG005 benchmark (**a**) and the challenging medically relevant genes (CMRG; v.1.0) benchmark (**b**). Giraffe uses the MC pangenome graph, BWA-MEM uses GRCh38 and Dragen Graph uses GRCh38 with additional alternative haplotype sequences. **c,** Comparison of AFs observed from the PanGenie genotypes for all 2,504 unrelated 1KG samples and the AFs observed across 44 of the HPRC assembly samples in the MC graph. The PanGenie genotypes include all variants contained in the filtered set (28,433 deletions, 84,755 insertions, 32,431 other alleles). **d,** Number of SVs present (genotype 0/1 or 1/1) in each of the 3,202 1KG samples in the filtered HPRC genotypes (PanGenie) after merging similar alleles ($n = 100,442$ SVs), the HGSVC lenient set ($n = 52,659$ SVs) and the 1KG Illumina calls ($n = 172,968$ SVs) in GIAB regions. In the box plots, lower and upper limits represent the first and third quartiles of the data, the white dots represent the median and the black lines mark minima and maxima of the data points. **e,** Length distribution of SV insertions and SV deletions contained in the filtered HPRC genotypes (PanGenie), the HGSVC lenient set and the 1KG Illumina calls. Only variants with a common AF > 5% across the 3,202 samples were considered.

could improve the accuracy of calling small variants from short reads. We used Giraffe[49] to align short reads from the GIAB benchmark samples[18] to the MC pangenome graph. For comparison, we aligned reads to GRCh38 using BWA-MEM[50] and to Dragen Graph[51], which uses GRCh38 augmented with alternative haplotypes at variant sites. We called SNPs and indels with DeepVariant[40] and the Dragen variant caller[51] (Methods). Our pangenomic approach (Giraffe plus DeepVariant) outperformed the other approaches for calling small variants (Fig. 6a), with gains for both SNPs and indels (Supplementary Fig. 14 and Supplementary Table 17). For example, it made 21,700 errors (false positives or false negatives) in the confident regions of the GIAB truth set using 30× reads from HG005. By contrast, 36,144 errors were made when DeepVariant used the reads aligned to GRCh38, and 26,852 errors when using the Dragen pipeline. In challenging medically relevant genes[47], the increase in performance was even larger for both SNPs (F1 score, defined as the harmonic mean of precision and recall, of 0.985 for Giraffe plus DeepVariant compared with <0.976 for the other methods) and indels (F1 score of 0.961 for Giraffe plus DeepVariant compared with <0.958 for the other methods) (Fig. 6b). Many regions benefitted from using

pangenome mapping, but regions with errors in GRCh38 and large L1HS sequences benefitted the most from the pangenomic approach (Supplementary Figs. 15 and 16).

We next benchmarked variant calling using parent–child trios. Using DeepTrio[52] resulted in better performance compared with DeepVariant across all samples of the GIAB (Fig. 6a) and the challenging medically relevant gene benchmarks (Fig. 6b and Supplementary Fig. 14). Moreover, improvements appeared to be additive to those from the pangenome. For example, DeepTrio using Giraffe alignments gave the highest calling accuracy, with the number of errors decreasing from 21,700 (single sample calling) to 10,098 (trio calling) for HG005.

### A pangenome variant resource

To create a community resource to aid the development of methods and the analyses of pangenome-based population genetics, we used Giraffe to align high-coverage short-read data from 3,202 samples of the 1KG[19] to our pangenome graph and DeepVariant to call small variants (Methods). The Mendelian consistency computed across 100 trios from those samples was comparable to the one computed across samples

from the GIAB truth set, which indicated that comparable call set quality was obtained (Supplementary Figs. 17 and 18). The number of small variants called was consistently higher across different ancestries, with on average 64,000 more variants per sample compared with the 1KG catalogue (Extended Data Fig. 7a). Given that our pangenome-based calls showed improved performance in challenging regions (Fig. 6b), this call set across the 1KG cohort now provides the genetics and genomics communities with AF estimates for complex but medically relevant loci. For example, our approach was able to detect the gene conversion event covering the second exon *RHCE*, which was observed in about 25% of assembled haplotypes (Fig. 5c and Extended Data Fig. 8). Moreover, for *KCNE1*, we provide calls and frequencies in a 40 kb region, spanning 3 exons, that could not be previously assessed owing to the presence of a false duplication in GRCh38 (Supplementary Fig. 19; see also an associated article[53] for genome-wide analysis of interlocus gene conversion).

## SV genotyping

The ability to represent polymorphic SVs is a key advantage of a graph-based pangenome reference. To demonstrate the utility of the sequence-resolved SVs inherent to our pangenome, we used PanGenie[54] to genotype the bubbles in the MC graph. We decomposed bubbles into their constituent variant alleles (Supplementary Figs. 20 and 21) and found that 22,133,782 bubbles represented 20,194,117 SNP alleles, 6,848,115 indel alleles and 413,809 SV alleles (Supplementary Fig. 22 and Methods). Of these SV alleles that were non-reference (neither GRCh38 nor T2T-CHM13), 17,720 were observed in biallelic contexts and 396,089 at multiallelic loci with more than 1 non-reference allele, including extreme cases in which all 88 haplotypes showed distinct alleles (Supplementary Fig. 22). To analyse the genotyping performance of PanGenie, we conducted a leave-one-out experiment in which we repeatedly removed one sample from the graph and re-genotyped it using the remaining haplotype paths in the graph and short-read data for the left-out sample (Methods). In line with previous results[5,54], we obtained high genotype concordance across all variant types and genomic contexts (Extended Data Fig. 9). Furthermore, we used PanGenie to genotype HG002 and evaluated genotypes based on SVs at challenging medically relevant loci[47]. This analysis resulted in a precision of 0.74 and an adjusted recall of 0.81 (Methods).

Next we genotyped the 3,202 samples from the 1KG[19] (Methods). We filtered the resulting SV genotypes using a machine-learning approach[5,54] that assessed different statistics, including Mendelian consistency and concordance, to assembly based calls. As a result, we produced a filtered, high-quality subset of SV genotypes containing 28,434 deletion alleles, 84,752 insertion alleles and 26,439 other SV alleles (Supplementary Table 18 and Methods). Many of the alleles not included in the filtered set stemmed from complex, multiallelic loci and were enriched for rare alleles. As independent quality control measures for genotypes in the filtered set, we assessed the Hardy–Weinberg equilibrium values (Supplementary Figs. 23–25) and compared AFs observed across the genotypes of all 2,504 unrelated samples to the respective AFs of the 44 assembly samples (88 haplotypes) contained in the graph. Pearson correlation values of 0.96, 0.93 and 0.90 for the deletion, insertion and other SV alleles, respectively, were observed (Fig. 6c), which indicated the high quality of the genotypes. To quantify our ability to detect additional SVs, we compared our filtered set of genotypes to the HGSVC PanGenie genotypes (v.2.0 'lenient' set)[5] and the Illumina-based 1KG SV call set[19]. We analysed the number of detected SV alleles in each sample (homozygous or heterozygous) and stratified them by genome annotations from GIAB (Fig. 6d and Methods) as well as using our own more detailed annotations (Supplementary Fig. 26). Both of the PanGenie-based call sets detected more SVs (HPRC, 18,483 SVs per sample; HGSVC, 12,997 SVs per sample) than the short-read-based 1KG call set (9,596 SVs per sample), with a particularly substantial advance for deletions <300 bp and insertions (Fig. 6e). The respective average numbers of SVs per haplotype were 12,439 for HPRC,

9,227 for HGSVC and 6,099 for the 1KG calls (Supplementary Fig. 27); that is, a gain of 104.0% for HPRC over 1KG and of 34.8% over HGSVC. This result confirms that short-read-based SV discovery relative to a linear reference genome misses a large proportion of SVs[5,6,8]. As anticipated, the number of SVs per sample within 'easy' genomic regions was consistent across all three call sets, particularly in low-mappability and tandem repeat regions, and the use of our pangenome reference led to substantial gains (Fig. 6d), including for common variants (Fig. 6e and Supplementary Fig. 28). Although the newly identified SVs were harder to genotype because they are primarily located in repetitive regions, genotype concordances were high and close to the ones for known SVs (Supplementary Fig. 29).

## Improved tandem repeat representation

VNTRs are particularly variable among individuals and are challenging to access with short reads. The gains in the number of genotyped SVs in VNTRs (Fig. 6d and Supplementary Fig. 28) prompted us to investigate whether our pangenome reference could also improve read mapping in VNTR regions. We first established orthology mapping between haplotypes in our pangenome reference using danbing-tk[55]. The orthology can be established for 94,452 out of the 98,021 VNTR loci (96.4%) discovered by TRF[56]. When mapping simulated short reads to GRCh38 with BWA-MEM, the rate of unmapped reads was 6.6–8.5 times greater compared with mapping to the MC graph with Giraffe (Extended Data Fig. 10a, Supplementary Fig. 30 and Supplementary Table 19). The true negatives were on average 1.9% higher than the GRCh38 approach, and the true positives were on average 0.087% higher. The graph approach also reduced false negatives by 2.1-fold. Read depth over a locus is correlated with the copy number of a duplication and we evaluated how well length variants in VNTR regions can be estimated using either the MC graph or GRCh38. The graph approach performed better for 80% of the loci (48,085 out of 60,386) and increased the median $r^2$ from 0.58 to 0.70 (Supplementary Fig. 31).

## Improved RNA sequencing mapping

To evaluate the benefit of our pangenome reference on transcriptomics, we simulated RNA sequencing (RNA-seq) reads and mapped them to a pangenome and to a standard reference genome (Methods). The pangenome-based pipeline using vg mpmap[57] achieved significantly lower false mapping rates than a linear reference pipeline using either vg mpmap or STAR[58] (Extended Data Fig. 10b). Compared with the linear reference pipelines, the pangenome pipeline also showed reduced allelic bias and increased mapped coverage on heterozygous variants, which could benefit studies of allele-specific expression (Supplementary Fig. 32). With real sequencing data, mapping rates were more difficult to interpret in the absence of a ground truth (Supplementary Fig. 33). Instead, we focused on the correlation in exon coverage to independent PacBio long-read Iso-Seq data. The analysis showed that the correlation was highest when mapping to a spliced pangenome graph derived from the MC graph (Supplementary Fig. 34). The pangenome pipeline showed a modest increase in correlation over the linear reference pipelines (0.006–0.011). In addition, mapping the simulated reads to the MC graph led to improved gene expression estimates relative to the linear GRCh38, regardless of whether alternative contigs were included in GRCh38 (Supplementary Fig. 35).

## Improved chromatin immunoprecipitation and sequencing analysis

We used the pangenome to re-analyse H3K4me1 and H3K27ac data from chromatin immunoprecipitation and sequencing (ChIP-seq) and assay for transposase-accessible chromatin with high-throughput sequencing (ATAC-seq) of monocyte-derived macrophages from 30 individuals with African ancestry or European ancestry[59]. Overall, we observed a net increase in the number of peak calls, whereby, on average, 2–3% of peaks were found only when using the MC pangenome (Extended Data

Fig. 10c). Moreover, the newly found peaks were replicated in more samples than expected by chance (Supplementary Fig. 36). In addition, we recovered epigenomic features that were specific to SV alleles not present in GRCh38 (termed non-reference). For example, across all H3K4me1 datasets, we assigned 1,326 events to the non-reference SV allele, 1,443 to the reference allele and 2,008 to both alleles within heterozygous SVs (Extended Data Fig. 10d), with some replicated multiple times across samples (Supplementary Fig. 37). Of these, there were 194 SVs with peaks that were observed only in African ancestry genomes, 150 that were observed only in European ancestry genomes and 216 that were observed in both African and European ancestry genomes. As expected, rare alleles were enriched for ancestry-specific events (Supplementary Fig. 37).

## Discussion

We have publicly released 94 de novo haplotype assemblies from a diverse group of 47 individuals. This provides a large set of fully phased human genome assemblies and outperforms earlier efforts on many levels of assembly quality[5,16,60]. For example, compared with Ebert et al.[5], the average median base level accuracy is nearly an order of magnitude higher, the N50 contiguity is nearly double and the structural accuracy is higher[33]. These improvements are the result of recent improvements in de novo assembly driven both by better sequencing technology and coordinated innovations in assembly algorithms[20,21]. To validate assembly structural accuracy, we developed a new pipeline that maps low error, long reads to each diploid assembly to support the predicted haplotypes. This pipeline indicated that more than 99% of each assembly, and greater than 90% of the assembled sequence representing highly repetitive arrays, was structurally correct. Some challenges around loci that harbour copy number polymorphisms and/or inversions remained[33]. Although the focus of this effort was to build a reference resource, highly accurate haplotype-resolved assemblies enabled us to access previously inaccessible regions, highlighting new forms of genetic variation and providing new insights into mutational processes such as interlocus gene conversion[53].

Accompanying these assemblies are 94 sets of Ensembl gene annotations, representing a large collection of de novo assembled human transcriptome annotations. Each transcriptome annotation is highly complete, particularly for protein-coding transcripts. These putative transcriptome annotations enabled us to analyse sequence-resolved CNVs. In detail, we assembled genic CNVs (mostly singletons) for 1,115 different protein-coding genes, confirming earlier mapping-based analyses that predicted that the majority of rare genic CNVs occur outside known SDs[31]. These CNV genes accounted for 0.6–4.4 Mb of additional genic sequences per haplotype compared with GRCh38. These contained genes known to have CNVs associated with human health, including amylase[61] (four to ten copies), β-defensin[62] (three to seven copies, *DEFB107A*) and *NOTCH2NLC–NOTCH2NLB*[63] (one additional copy).

The pangenomes presented here are both a set of individual haploid genome assemblies and an alignment of these assemblies. The combination can be efficiently described as a variation graph[14,64]. A new set of exchange formats for pangenomics, including extensions of the graphical fragment format (GFA) that encode variation graphs, are emerging[34]. An associated article[65] to this work demonstrated that the pangenomes presented here can be losslessly stored using a compressed, binary representation of GFA in just 3–6 GB despite representing more than 282 billion bases of individual sequence, with strongly sublinear scaling as new genomes are added. Creating pangenome graphs is an active research topic, so we developed multiple pipelines, and details of these methods are further explored in companion papers[35,36]. We found concordance between the different construction approaches used here, whereby the MC and PGGB pangenomes contained nearly the same number of small variants and SVs of various types. Furthermore, these encoded pangenome variants showed high levels of agreement with existing linear reference-based methods for variant discovery, particularly within the non-repetitive fraction of the genome. Our study of complex and medically relevant loci showed that the pangenomes faithfully recapitulated existing knowledge and will enable future efforts to study the role of complex variation in human disease. Further work will be required to more comprehensively identify medically relevant complex SVs and to ensure the accuracy of each allele represented in the pangenome.

Where the pangenome graphs differ is principally in how they handle CNV sequences. The PGGB method will frequently merge CNVs, whereas the MC graphs represent CNV copies as independent subgraphs. Both approaches have merits, and which approach to favour will take further experimentation and community input, and may vary by the specific application. The PGGB method retained all centromeric and satellite sequences, whereas the MC graph pruned much of this sequence. This made it practical with current methods to use the MC graphs for read alignment applications. However, pruning these sequences is not a satisfactory solution. Longer term, more work is needed to determine how best to align and represent these large repeat arrays within pangenomes, particularly as T2T assembly becomes commonplace and these arrays therefore completed. Furthermore, although the PGGB graph retained centromeric and satellite sequences, in principle, by enabling analysis of previously inaccessible parts of the pangenome, our initial population-genetic analysis of these regions (Methods) leaves open questions about assembly accuracy and alignment, especially in areas of the genome where mutation rates are thought to be an order of magnitude greater[66]. This suggests that significant care must be taken when studying them, and new methods may need to be developed to fully understand and characterize this component of the human pangenome.

A near-term application of pangenome references will be to improve reference-based sequence mapping workflows. In these workflows, the pangenome can act as a drop-in replacement for existing references, with the read mappings projected from the pangenome space back onto an existing linear reference for downstream processing. This is how the Giraffe–DeepVariant workflow functions: DeepVariant, the variant caller, never needs to consider the complexity of the pangenome, but the workflow benefits from a mapping step that accounts for sequences that are missing from the linear reference. Making the switch to using pangenome mapping is not significantly more computationally expensive[49] and resulted in an average 34% reduction in false-positive and false-negative errors compared with using the standard reference methods (Supplementary Fig. 38). These benefits were also greatest at complex loci[47]. Pangenomes not only improve variant calling but also improve transcript mapping accuracy[57] and detection of ChIP-seq peaks[67].

SVs have been mostly excluded from short-read studies because methods to genotype them using a linear reference have limited accuracy and sensitivity. Previous short-read, linear reference studies have discovered 7,500–9,500 SVs per sample[19,68], whereas long-read sequencing efforts have routinely discovered around 25,000. Ebert et al.[5] showed that using PanGenie, a pangenomic approach, with 32 samples, a subset of these variants could be genotyped in short-read genomes (about 13,000 genotyped on average, ranging from 12,000 to 15,000 per sample). Using the same PanGenie method, the HPRC pangenome increases this to around 18,500 (ranging from 16,900 to 24,900) per sample, enabling the genotyping of the substantial majority of SVs discovered using long-reads per sample. The draft pangenome therefore delivers better SV calling than previous approaches, extracting latent information from short-read samples that are already available. So, in the future, the pangenome will enable the inclusion of tens of thousands of additional SV alleles into genome-wide association studies. Looking beyond short reads, in the future, the combination of the pangenome and low-cost long-read sequencing should prove to be a potent combination for comprehensive SV genotyping.

These new pangenomic workflows could benefit individuals of different ancestries differently. For read mapping and small variant calling, we observed a consistent improvement across individuals (Extended Data Fig. 7). Moreover, the pangenome might improve SV genotyping differently across individuals owing to the stronger divergence of the alleles from the reference. In the 1KG cohort, we observed that the genotyped samples clustered by super-population labels (Extended Data Fig. 11), which would suggest different levels of detection bias that are mitigated with the pangenome. However, we caution that the composition of the samples underlying the pangenome relative to the composition of the set of samples genotyped could potentially influence these results; an analysis with more samples is warranted.

The openly accessible, diverse assemblies and pangenome graphs we present here form a draft of a pangenome reference. There are many remaining challenges to growing and refining this reference. For example, assembly reliability analysis revealed roughly an order of magnitude more erroneously assembled sequences in the HPRC assemblies than in the T2T-CHM13 complete assembly. Similarly, in a companion analysis, Strand-seq data from a subset of assemblies revealed 6–7 Mb of incorrectly oriented sequence per haplotype[33], which indicates that there is room to structurally improve the assemblies. Furthermore, despite being predicted to have less than 1 base error per around 200,000 assembled bases, base level sequencing errors are still an issue. For example, we identified more than a dozen apparent frameshifts and nonsense mutations per genome annotation that are probably the result of sequencing errors. The cohort we present is also relatively small notwithstanding the significant effort to generate the underlying long-read sequencing resource. Our near-term goal is to expand the pangenome to a diverse cohort of 350 individuals (which should capture most common variants), to push towards T2T genomes for this cohort (to properly represent the entire genome in almost all individuals) and to refine the pangenome alignment methods (so that telomere-to-telomere alignment is possible, capturing more complex regions of the genome). This will give us a more comprehensive representation of all types of human variation.

We acknowledge that references generated from the 1KG samples alone are insufficient to capture the extent of sequence diversity in the human population. To ensure that we are able to maximize our surveys of sample diversity while abiding by principles of community engagement and avoiding extractive practices[14,15], we will broaden our efforts to recruit new participants to improve the representation of human genetic diversity. A richer human reference map promises to improve our understanding of genomics and our ability to predict, diagnose and treat disease. A more diverse human reference map should also help ensure that the eventual applications of genomic research and precision medicine are effective for all populations. We recognize that the value of this project will partly be in the future establishment of new standards for how we capture variant diversity, the opportunity to disseminate science into diverse communities and continued efforts to engage with diverse voices in this ambitious goal to build a common global reference resource. The methods we are developing should prove valuable for other species. Indeed, other groups are pioneering such efforts[69,70]. In parallel with our efforts to obtain a more comprehensive collection of diverse and highly accurate human reference genomes, we anticipate further optimization and rapid improvement of the pangenome reference, enabling an increasingly broad set of applications and use cases for both the research and clinical communities.

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

¹Department of Genetics, Yale University School of Medicine, New Haven, CT, USA. ²Center for Genomic Health, Yale University School of Medicine, New Haven, CT, USA. ³Division of Biology and Biomedical Sciences, Washington University School of Medicine, St. Louis, MO, USA. ⁴Genomics Institute, University of California, Santa Cruz, CA, USA. ⁵Institute for Medical Biometry and Bioinformatics, Medical Faculty, Heinrich Heine University, Düsseldorf, Germany. ⁶Center for Digital Medicine, Heinrich Heine University, Düsseldorf, Germany. ⁷Division of Oncology, Department of Internal Medicine, Washington University School of Medicine, St. Louis, MO, USA. ⁸Institute of Genetics and Biophysics, National Research Council, Naples, Italy. ⁹Department of Data Sciences, Dana-Farber Cancer Institute, Boston, MA, USA. ¹⁰Department of Biomedical Informatics, Harvard Medical School, Boston, MA, USA. ¹¹Department of Genetics, Genomics and Informatics, University of Tennessee Health Science Center, Memphis, TN, USA. ¹²McDonnell Genome Institute, Washington University School of Medicine, St. Louis, MO, USA. ¹³Department of Genetics, Washington University School of Medicine, St. Louis, MO, USA. ¹⁴Novo Nordisk Foundation Center for Biosustainability, Technical University of Denmark, Copenhagen, Denmark. ¹⁵Quantitative Life Sciences, McGill University, Montréal, Québec, Canada. ¹⁶Genomics Research Centre, Human Technopole, Milan, Italy. ¹⁷Department of Genome Sciences, University of Washington School of Medicine, Seattle, WA, USA. ¹⁸Quantitative Biology Center (QBiC), University of Tübingen, Tübingen, Germany. ¹⁹Biomedical Data Science, Department of Computer Science, University of Tübingen, Tübingen, Germany. ²⁰Tree of Life, Wellcome Sanger Institute, Hinxton, Cambridge, UK. ²¹Northeastern University, Boston, MA, USA. ²²Department of Quantitative and Computational Biology, University of Southern California, Los Angeles, CA, USA. ²³European Molecular Biology Laboratory, European Bioinformatics Institute, Wellcome Genome Campus, Hinxton, Cambridge, UK. ²⁴Coriell Institute for Medical Research, Camden, NJ, USA. ²⁵Department of Computer Science, University of Pisa, Pisa, Italy. ²⁶Center for Health Data Science, University of Copenhagen, Copenhagen, Denmark. ²⁷Division of Medical Genetics, University of Washington School of Medicine, Seattle, WA, USA. ²⁸Google, Mountain View, CA, USA. ²⁹Barrett and O'Connor Washington Center, Arizona State University, Washington, DC, USA. ³⁰Department of Ecology and Evolutionary Biology, University of California, Santa Cruz, CA, USA. ³¹Core Unit Bioinformatics, Medical Faculty, Heinrich Heine University, Düsseldorf, Germany. ³²Vertebrate Genome Laboratory, The Rockefeller University, New York, NY, USA. ³³National Institutes of Health (NIH)–National Human Genome Research Institute, Bethesda, MD, USA. ³⁴Center for Computational and Genomic Medicine, The Children's Hospital of Philadelphia, Philadelphia, PA, USA. ³⁵Institute for Society and Genetics, College of Letters and Science, University of California, Los Angeles, CA, USA. ³⁶Institute for Precision Health, David Geffen School of Medicine, University of California, Los Angeles, CA, USA. ³⁷Division of General Internal Medicine and Health Services Research, David Geffen School of Medicine, University of California, Los Angeles, CA, USA. ³⁸Department of Biomolecular Engineering, University of California, Santa Cruz, CA, USA. ³⁹Dovetail Genomics, Scotts Valley, CA, USA. ⁴⁰Division of Oncology, Department of Medicine, Stanford University School of Medicine, Stanford, CA, USA. ⁴¹Institute for Genomic Health, Icahn School of Medicine at Mount Sinai, New York, NY, USA. ⁴²Program in Bioethics and Institute for Human Genetics, University of California, San Francisco, CA, USA. ⁴³Genome Biology Unit, European Molecular Biology Laboratory, Heidelberg, Germany. ⁴⁴Genome Informatics Section, Computational and Statistical Genomics Branch, National Human Genome Research Institute, National Institutes of Health, Bethesda, MD, USA. ⁴⁵Computer Sciences Department, Barcelona Supercomputing Center, Barcelona, Spain. ⁴⁶Departament d'Arquitectura de Computadors i Sistemes Operatius, Universitat Autònoma de Barcelona, Barcelona, Spain. ⁴⁷Material Measurement Laboratory, National Institute of Standards and Technology, Gaithersburg, MD, USA. ⁴⁸Department of Public Health Sciences, University of California, Davis, CA, USA. ⁴⁹Department of Biomedical Engineering, Johns Hopkins University, Baltimore, MD, USA. ⁵⁰Berlin Institute for Medical Systems Biology, Max Delbrück Center for Molecular Medicine in the Helmholtz Association, Berlin, Germany. ⁵¹National Center for Biotechnology Information, National Library of Medicine, National Institutes of Health, Bethesda, MD, USA. ⁵²Al Jalila Genomics Center of Excellence, Al Jalila Children's Specialty Hospital, Dubai, UAE. ⁵³Center for Genomic Discovery, Mohammed Bin Rashid University of Medicine and Health Sciences, Dubai, UAE. ⁵⁴Center for Computational Biology, Johns Hopkins University, Baltimore, MD, USA. ⁵⁵Department of Human Genetics, McGill University, Montréal, Québec, Canada. ⁵⁶Canadian Center for Computational Genomics, McGill University, Montréal, Québec, Canada. ⁵⁷Institute for the Advanced Study of Human Biology (WPI-ASHBi), Kyoto University, Kyoto, Japan. ⁵⁸Howard Hughes Medical Institute, Chevy Chase, MD, USA. ⁵⁹Laboratory of Neurogenetics of Language, The Rockefeller University, New York, NY, USA. ⁶⁰These authors contributed equally: Wen-Wei Liao, Mobin Asri, Jana Ebler. ✉e-mail: egarris5@uthsc.edu; tobias.marschall@hhu.de; ira.hall@yale.edu; hli@jimmy.harvard.edu; bpaten@ucsc.edu

## Methods

### Sample selection

We identified parent–child trios from the 1KG in which the child cell line banked within the NHGRI Sample Repository for Human Genetic Research at the Coriell Institute for Medical Research was listed as having zero expansions and two or fewer passages, and rank-ordered representative individuals as follows. Loci with MAFs less than 0.05 were removed. MAFs were measured in the full cohort (that is, 2,504 individuals, 26 subpopulations) regardless of each individual's subpopulation labelling. For each chromosome, principal component analysis (PCA) was performed for dimension reduction. This resulted in a matrix with 2,200 features, which was then centred and scaled using smartPCA normalization. The matrix was further reduced to 100 features through another round of PCA.

We defined the representative individuals of a subpopulation as those who are similar to the other members in the group (which, in this scenario, is the subpopulation they belong to), as well as different from individuals outside the group. Group is defined by previous 1KG population labels (for example, 'Gambian in Western Division'). We did this as follows. For each sample, we first calculated the intragroup distance $d_{intra}$, which is the average of L2-norms between the sample and samples of the same subpopulation. The intergroup distance, $d_{inter}$, was similarly defined as the average of L2-norms between the sample and samples from all other subpopulations. The L2-norms were derived in the feature space of the PCA. The score of this sample was then defined as $10 \times d_{intra} + d_{inter}/(n-1)$, where $n$ is the number of subpopulations. For each subpopulation, if fewer than three trios were available, all were selected. Otherwise, trios were sorted by ranking children with $\max(paternal_{rank}, maternal_{rank})$, where $paternal_{rank}$ and $maternal_{rank}$ are the respective ranks of each parent's score, selecting the three trios with a maximum value. We ranked by parent scores because during the year 1 effort, the child samples did not have sequencing data and therefore had to be represented by the parents.

Ideally, we would have selected the same number of candidates from each subpopulation and have an equal number of candidates from both sexes. To correct for imbalances, we applied the following criteria for each subpopulation's candidate set: (1) when the sex was unbalanced (that is, off by more than one sample), we tried to swap in the next-best candidate of the less represented sex or did nothing if this was not possible; (2) if a subpopulation had fewer individuals than the desired sample selection size (that is, all candidates were selected), their unused slots were distributed to other unsaturated subpopulations. The latter choice is arbitrary but should have little impact on the overall results.

The genetic information used in this study was derived from publicly available cell lines from the NHGRI Sample Repository for Human Genetic Research and the NIGMS Human Genetic Cell Repository at the Coriell Institute for Medical Research. Therefore, this study is exempt from human research approval as the proposed work involved the collection or study of data or specimens that are already publicly available.

### Sequencing

**Cell line expansion and banking.** Lymphoblastoid cell lines (LCLs) used for sequencing from the 1KG collection (Supplementary Table 1) were obtained from the NHGRI Sample Repository for Human Genetic Research at the Coriell Institute for Medical Research. HG002 (GM24385) and HG005 (GM24631) LCLs were obtained from the NIGMS Human Genetic Cell Repository at the Coriell Institute for Medical Research. All expansions for sequencing were derived from the original expansion culture lot to ensure the lowest possible number of passages and to reduce overall culturing time. Cells used for HiFi, Nanopore, Omni-C, Strand-seq, 10x Genomics and Bionano production and for g-banded karyotyping and Illumina Omni2.5 microarray were expanded to a total culture size of $4 \times 10^8$ cells, which resulted in a total of five passages after cell line establishment. Cells were split into production-specific sized

vials as follows: HiFi, $2 \times 10^7$ cells; Nanopore, $5 \times 10^7$ cells; Omni-C, $5 \times 10^6$ cells; Strand-seq, $1 \times 10^7$ cells; 10x Genomics, $4 \times 10^6$ cells; and Bionano, $4 \times 10^6$ cells. Cells for Strand-seq were stored in 65% RPMI-1640, 30% FBS and 5% DMSO and frozen as viable cultures. All other cells were washed in PBS and flash-frozen as dry cell pellets. Cells used for ONT-UL production were separately expanded from the original expansion culture lot to a bank of five vials of $5 \times 10^6$ cells. A single vial was subsequently expanded to a total culture size of $4 \times 10^8$ cells, which resulted in a total of eight passages. Cells were also reserved for g-banded karyotyping and Illumina Omni2.5 microarray.

**Karyotyping and microarray.** G-banded karyotype analysis was performed on $5 \times 10^6$ cells collected at passage five (for HiFi, Nanopore and Omni-C) and passage eight (for ONT-UL). For all cell lines, 20 metaphase cells were counted, and a minimum of 5 metaphase cells were analysed and karyotyped. Chromosome analysis was performed at a resolution of 400 bands or greater. A pass/fail criterion was used before cell lines proceeded to sequencing. Cell lines with normal karyotypes (46,XX or 46,XY) or lines with benign polymorphisms that are frequently seen in apparently healthy individuals were classified as passes. Cell lines were classified as failures if two or more cells harboured the same chromosomal abnormality. DNA used for microarray was isolated from frozen cell pellets ($3 \times 10^6$ to $7 \times 10^6$ cells) using a Maxwell RSC Cultured Cells DNA kit on a Maxwell RSC 48 instrument (Promega). DNA was genotyped at the Children's Hospital of Philadelphia's Center for Applied Genomics using an Infinium Omni2.5-8 v.1.3 BeadChip (Illumina) on an iScan System instrument (Illumina).

**HiFi sequencing.** PacBio HiFi sequencing was distributed between two centres: Washington University in St. Louis and the University of Washington. We describe the protocols used at each centre separately.

**Washington University in St. Louis.** High-molecular-weight DNA was isolated from frozen cell pellets using a Qiagen MagAttract HMW DNA kit and sheared using a Diagenode Megaruptor I to 20 kb mode size. At all steps, DNA quantity was checked on a Qubit Fluorometer I with a dsDNA HS Assay kit (Thermo Fisher), and sizes were examined on a FEMTO Pulse (Agilent Technologies) using a Genomic DNA 165 kb kit. SMRTbell libraries were prepared for sequencing according to the protocol 'Procedure & Checklist—Preparing HiFi SMRTbell Libraries using the SMRTbell Express Template Prep Kit 2.0'. After SMRTbell generation, material was size-selected on a SageELF system (Sage Science) using the '0.75% 1-18 kb' program (target 3,450 bp in well 12), and some combinations of fraction 3 (average size of 15–21 kb), fraction 2 (average size of 16–27 kb) and fraction 1 (average size of 20–31 kb) were selected for sequencing, depending on the empirical size measurements and available mass. The selected library fractions were bound with Sequencing Primer v.2 and Sequel II Polymerase v.2.0 and sequenced on Sequel II instruments (PacBio) on SMRT Cells 8M using Sequencing Plate v.2.0, diffusion loading, 2 h of pre-extension and 30 h of movie times. Samples were sequenced to a minimum HiFi data amount of 108.5 Gbp (35× estimated genome coverage) on four SMRT Cells.

**University of Washington.** High-molecular-weight DNA was isolated from frozen cell pellets using a modified Gentra Puregene method and sheared using gTUBE (Covaris) to 20 kb mode size. At all steps, DNA quantity was checked by fluorometry on a DS-11 FX instrument (DeNovix) with a Qubit dsDNA HS Assay kit (Thermo Fisher), and sizes were examined on a FEMTO Pulse (Agilent Technologies) using a Genomic DNA 165 kb kit. SMRTbell libraries were prepared for sequencing according to the protocol 'Procedure & Checklist—Preparing HiFi SMRTbell Libraries using the SMRTbell Express Template Prep Kit 2.0'. After SMRTbell generation, material was size-selected on a SageELF system (Sage Science) using the '0.75% 1–18 kb' program (target 3,400 bp in well 12), and fraction 2 (average size of 17–20 kb) or fraction 1 (average size of 18–20 kb) was

selected for sequencing, depending on the empirical size measurements and available mass. For some samples, the SageELF program '0.75% agarose, 10 kb–40 kb' (target 10,000 bp in well 10) was used, and fractions 6 and 7 were pooled together for sequencing (average size of 17–21 kb). The selected library fractions were bound with Sequencing Primer v.2 and Sequel II Polymerase v.2.0 and sequenced on Sequel II instruments (PacBio) on SMRT Cells 8M using Sequencing Plate v.2.0, diffusion loading, 3–4 h of pre-extension and 30 h of movie times. Samples were sequenced to a minimum HiFi data amount of 96 Gbp (30× estimated genome coverage) on at least four SMRT Cells.

**Comparisons of HiFi production methods.** Although subtle differences in HiFi data production methods existed between the University of Washington and Washington University in St. Louis, the resulting data were highly similar, with overlapping assembly statistics from most samples. These initial genomes were sequenced at a time when methods were being refined and optimized for HiFi sequencing, as it was a relatively new process. The primary differences in protocols are part of the nucleic acid isolation, fragmentation and size selection, with the downstream sequencing-specific applications being more consistent. Both teams were closely engaged with each other as well as with our company associates, including New England Biolabs (NEB), Qiagen, Diagenode and Sage Science, to provide optimal end products.

**Nanopore ultra-long sequencing protocol.** For the 18 additional samples, we used the nanopore unsheared long-read sequencing protocol[16]. This generated about 60× coverage of unsheared sequencing from 3 PromethION flow cells and a N50 value of around 44 kb. For the 29 newly selected HPRC samples (Results), we used the protocol outlined below.

**DNA extraction.** Around 50 million cells in a pellet were resuspended in 200 µl of PBS, and the resuspended cells were aliquoted (40 µl) into five 1.5 ml DNA Lo-bind Eppendorf tubes. The following procedure for DNA extraction was completed for each of the five aliquots. Each tube contained sufficient DNA for three libraries loaded onto one flow cell. The following reagents were added in sequence to each tube with pipette mixing (10 times up and down) using a P200 wide-bore pipette: 40 µl of proteinase K, 40 µl of buffer CS and 40 µl of CLE3. The samples were then incubated at room temperature (18–25 °C) for 30 min. Next, 40 µl of RNase A was added to each tube with pipette mixing (10 times) with a P200 wide-bore pipette, and samples were incubated at room temperature for 3 min. Two hundred microlitres of BL3 was mixed with 200 µl PBS in a 1.5 ml Eppendorf tube. Four hundred microlitres of this BL3–PBS mixture was then added to each sample and the samples mixed 10 times with a P1000 wide-bore pipette set to 600 µl.

Samples were incubated for 10 min at room temperature and then pipette mixed 5 times, then incubated at room temperature for 10 min and pipette mixed 5 times and then further incubated for 10 min at room temperature. A white precipitate may form after addition of BL3. This is normal. A Nanobind disk was added to the cell lysate, then 600 µl of isopropanol was added. Mixing was performed by inversion of the tube 5 times. Tubes were further mixed on a tube rotator (9 r.p.m. at room temperature for 10 min). The tubes were then placed on a magnetic tube rack, and the Nanobind disk positioned closer to the top of the tube to avoid inadvertent removal of the DNA bound to the Nanobind disk. The supernatant was discarded using a pipette and 700 µl of buffer CW1 was added to each tube. The tube in the magnetic rack was then inverted 4 times for mixing. A second and third wash with 500 µl of buffer CW2 (inversion mix 4 times for each wash) was performed. After the second CW2 wash, liquid was removed from the tube cap and the tubes spun on a mini-centrifuge for 2 s, and replaced on the magnetic rack. Residual liquid was removed from the bottom of the tube, taking care not to remove DNA associated with the Nanobind disk. Elution from the Nanobind disk was accomplished by adding 160 µl Circulomics elution buffer (EB) plus 0.02% Triton X-100 (comprising 316.8 µl EB

and 3.2 µl 2% Triton X-100) and incubating at room temperature for at least 1 h. Tubes were gently tapped halfway through elution. DNA was collected by transferring eluate with a P200 wide-bore pipette to a new 1.5 ml microcentrifuge tube. Some liquid and DNA remained on the Nanobind disk after pipetting. The tube containing the Nanobind disk was spun in a centrifuge at 10,000g for 5 s, and any additional liquid that came off the disk was transferred to the eluate tube. This process was repeated if necessary until all DNA was removed. The samples were pipette mixed 5 times (approximately 10 s to aspirate and 10 s to dispense for each cycle) with a wide-bore P200 pipette to homogenize the sample. Samples were further allowed to rest at room temperature overnight to allow DNA to solubilize (disperse).

### Library preparation

**DNA tagmentation and FRA.** Circulomics EB+ (EB buffer with 0.02% Triton X-100) was prepared, and 140.82 µl EB+ was aliquoted into a 1.5 ml Eppendorf DNA Lo-Bind tube. UHMW DNA (300 µl) from above was aliquoted into the same tube with a wide-bore P200 pipette. The mixture was slowly pipetted up and down 3 times with a wide-bore P200 pipette set to 150 µl. In a separate 1.5 ml Eppendorf DNA Lo-Bind tube, the following reagents were added in sequence: 144 µl of FRA dilution buffer, 9.18 µl of 1 M MgCl$_2$ and 6 µl FRA. The tube was tapped to mix and spun down using a microcentrifuge. The EB–Triton X-100–DNA mixture was added to the FRA dilution buffer–MgCl$_2$–FRA mixture with a wide-bore P200 pipette. This mixture was then pipette mixed 15–20 times with a wide-bore P1000 pipette set to 600 µl. The mixture appeared homogeneous when pipette mixing was finished. The tube was then incubated for 15 min at room temperature. The mixture was then pipette mixed 5 times with a wide-bore P1000 pipette set to 600 µl and incubated at room temperature for an additional 15 min. The mixture was incubated at 30 °C for 1 min, followed by 80 °C for 1 min and then held at 4 °C.

**FRA clean-up.** Clean-up used a Nanobind disk. A 5 mm Nanobind disk was added to the above-described reaction mixture followed by 300 µl of Circulomics buffer NAF10. The tube was gently tapped 10–20 times to mix. The mixture was placed on a platform rocker at 20 r.p.m. for 2 min at room temperature. A DNA 'cloud' was visible on the Nanobind disk. The tube was spun for 1–2 s using a benchtop microcentrifuge and placed on a magnetic rack. The binding solution was removed and discarded. The Nanobind disk was washed by adding 350 µl ONT long fragment buffer (LFB) and gently tapped 5 times to mix. The tube was spun for 1–2 s using a microcentrifuge and placed on a magnetic rack. The ONT LFB was removed and discarded. Care was taken to not pipette DNA attached to the Nanobind disk. This LFB wash was repeated. The tube was then briefly spun (microcentrifuge) to move the Nanobind disk to the bottom of the tube. DNA was eluted from the Nanobind disk by adding 125 µl of ONT EB to the tube. The tube was incubated for 30 min at room temperature then gently tapped 5 times (mixing) and incubated for an additional 30 min at room temperature. Fluid was slowly aspirated 4 times over the Nanobind disk before removing the eluate from the tube. The eluate was transferred to a new 1.5 ml Eppendorf DNA Lo-Bind tube using a wide-bore P200 pipette. The eluate was then pipette mixed 2 times with a wide-bore P200 pipette.

**Adapter attachment and rapid adaptor.** Rapid adaptor (RAP) was added to the DNA preparation. To 120 µl of eluate (from above), 3 µl of ONT RAP was added. The mixture was pipette mixed 8 times with a wide-bore P200 pipette. The mixture was then incubated for 15 min at room temperature and then again pipette mixed 8 times with a wide-bore P200 pipette.

**RAP reaction clean-up with Nanobind.** The final library clean-up step removes unligated adaptor. In brief, 120 µl Circulomics EB was added to 123 µl of the above-described RAP reaction mixture. The mixture

was slowly pipette mixed 3 times with a wide-bore P1000 pipette set to 240 µl. Each aspiration took about 10 s and each dispense took around 10 s. A 5 mm Nanobind disk was added to the reaction mixture followed by 120 µl Circulomics buffer NAF10. Mixing was accomplished by gentle tapping. The tube was incubated for 5 min at room temperature without agitation or rotation. The tube was gently tapped 5 times (each time for 2–3 times) during the 5 min of incubation. The tube was spun for 1–2 s using a microcentrifuge and placed on a magnetic rack. The binding solution was discarded. Next 350 µl of ONT LFB was added to the tube and mixed by gentle tapping 5 times. The tube was then spun for 1–2 s using a microcentrifuge and placed on a magnetic rack. The ONT LFB was removed and discarded. Next the Nanobind disk was washed by adding 350 µl ONT LFB. The tube was gently tapped 5 times to move LFB over the surface of the disk. The tube was then incubated at room temperature for 5 min. The tube was then spun for 1–2 s using a microcentrifuge and placed on a magnetic rack. The ONT LFB was removed and discarded. The tube was briefly spun using a microcentrifuge to move the Nanobind disk to the bottom of the tube. To elute DNA from the Nanobind disk, 126 µl ONT EB was added to the tube. The tube was incubated for 30 min at room temperature, then gently tapped 5–10 times and incubated for an additional 1–2 h at room temperature. The eluate was then transferred to a new 1.5 ml Eppendorf DNA Lo-Bind tube using a wide-bore P200 pipette using the same technique described above for passing the eluate over the Nanobind disk before removing the eluate from the tube. The mixture was then pipette mixed 2–3 times with a wide-bore P200 pipette. The library was stored overnight at 4 °C before sequencing to facilitate maximal dissolution of DNA.

**Flow cell loading and sequencing.** ONT sequencing buffer (SQB) (68 µl) was added to 82 µl of the eluate from above. The mixture was pipette mixed 4 times with a wide-bore P200 pipette set to 150 µl. Each aspiration of 150 µl took 10–20 s, and each dispense of 150 µl took 10–20 s. Samples were then incubated at room temperature for 10 min. Next the samples were pipette mixed 8 times with a wide-bore P200 pipette set to 150 µl. Before loading the library, the flow cell was primed with flush buffer/flush tether mixture per ONT directions. The library was then added to the flow cell. The mixture was viscous but loaded smoothly in about 1 min. Some samples took 2 min maximum to load. The sequencing run had a re-mux time set for every 6 h. Base calling was performed using Guppy (v.4.0.11), with default parameters and the high-accuracy PromethION model (dna_r9.4.1_450bps_hac_prom.cfg).

**Dovetail Omni-C.** We prepared Omni-C libraries from each cell line using a Dovetail Omni-C kit (Dovetail Genomics) with the following modifications as. First, we aliquoted 1 million cells for fixation with formaldehyde and DSG. We digested chromatin with DNAse I until DNA fragments of a desired length were obtained. Per the protocol, we performed end repair on the chromatin, followed by ligation of a biotinylated bridge oligonucleotide, followed by ligation of free chromatin ends. We reversed crosslinks and purified proximity ligated DNA. We converted the DNA into an Illumina sequencing library using a NEB Ultra II library preparation kit (NEB) with a Y-adaptor. We enriched for ligation products using streptavidin bead capture on the final library. Each capture reaction was then split into two replicates before the final PCR enrichment step to preserve complexity. All libraries were uniquely dual indexed and sequenced on an Illumina Novaseq Platform with read lengths of 2 × 150 bp.

## Phased assembly pipeline

We describe the main automated and manual steps taken before, during and after assembly. A combined set of workflow description language (WDL) formatted assembly workflows is available from Dockstore that captures each of the steps for filtering adapter-contained reads and

running Hifiasm (https://dockstore.org/organizations/HumanPangenome/collections/Hifiasm). All assemblies were generated using this workflow, running on AnVIL[71]. Cleaning assemblies and fixing some structural issues were performed through a combination of automated workflows and manual curation as described below. Manual curation was performed using Jupyter notebooks available at GitHub (https://github.com/human-pangenomics/hpp_production_workflows/tree/master/assembly/y1-notebooks).

**Filtering adapter-contained reads and running Hifiasm.** Before producing the assemblies, we detected and removed the reads containing PacBio adapters using a bash script from the HiFiAdapterFilt repository[72] (commit 64d1c7b). This script first creates a database of the PacBio adapter sequences, as illustrated below:

```
>gnl|uv|NGB00972.1:1-45 Pacific Biosciences Blunt Adapter
ATCTCTCTCTTTTCCTCCTCCTCCGTTGTTGTTGTTGAGAGAGAT
>gnl|uv|NGB00973.1:1-35 Pacific Biosciences C2 Primer
AAAAAAAAAAAAAAAAAAAATTAACGGAGGAGGAGGA
```

It then runs blastn with tuned parameters to detect adapter-containing reads as follows:

```
blastn -db ${DATABASE} -query ${HIFI_FASTA} -task blastn -reward
1 -penalty -5 -gapopen 3 -gapextend 3 -dust no -soft_masking true
-evalue 700 -searchsp 1750000000000 -outfmt
```

For 43 samples (out of 47), we removed less than 0.15% of the reads; all of the 29 HPRC-selected samples are among these 43 samples, indicating the low level of adapter contamination in the HiFi data produced by the HPRC. HG005, which is one of the 18 additional samples, had the highest adaptor contamination percentage, at about 1%. (Supplementary Fig. 39)

The removed reads were then aligned to the T2T-CHM13 (v.2.0) reference to ensure that there was no chromosomal or locus-specific bias in the filtering process. Supplementary Fig. 40 shows a snapshot of the IGV browser[73] illustrating the coverage of the adapter-containing reads along the genome. The locations of the reads were almost evenly distributed along the genome and, excluding centromeres, we barely found any region covered with more than two adapter-containing reads, even in HG005, which had the highest contamination percentage.

The trio-binning mode of Hifiasm needs haplotype-specific $k$-mers for trio phasing the assembly graph. To generate these $k$-mers, we used parental Illumina short reads for the 47 HPRC samples, which are publicly available from the 1KG dataset[19]. For each parental short-read sample, we used yak count (v.0.1)[74] to generate the $k$-mer hash tables, running it once for each of the paternal and maternal read sets:

```
yak count -k31 -b37 -o pat.yak paternal.fq.gz
yak count -k31 -b37 -o mat.yak maternal.fq.gz
```

The adapter-filtered HiFi reads along with the parental $k$-mer tables were then given to Trio-Hifiasm (v.0.14) to produce haplotype-resolved assembly graphs. Only the sample HG002 was re-assembled with Trio-Hifiasm (v.0.14.1), which is explained in more detail in the next subsection.

```
hifiasm -o ${SAMPLE_NAME} -t 48 -1 pat.yak -2 mat.yak hifi.fq.gz
```

Hifiasm produces one graph per haplotype in GFA format. Each haplotype-specific GFA file was then converted to FASTA format using Gfatools[75]. The assemblies produced by Trio-Hifiasm (v.0.14) are released under v.2 after performing the three cleaning steps described at the end of this section.

**Manually fixing issues.** We used paftools.js asmgene, from the minimap2 repository (https://github.com/lh3/minimap2/tree/master/misc)[76] to count the number of apparent gene duplications for each of the assemblies produced by Trio-Hifiasm (v.0.14). Asmgene does not distinguish between true duplicates and errors. Looking at its results, we were able to find the duplication trend and detect any outlier. This assessment acted as a proxy for detecting high-level duplication errors. We used the Ensembl v.99 cDNA sequences[77] as the input gene set for running asmgene.

```
# Aligning genes to GRCh38 and each Hifiasm haploid assembly
minimap2 -cx splice:hq hs38.fa cdna.fa > hs38.paf
minimap2 -cx splice:hq ${pat/mat}.fa cdna.fa > ${pat/mat}.paf

# Detecting gene duplications
paftools.js asmgene -a hs38.paf ${pat/mat}.paf
```

Three samples were detected as outliers in terms of the number of gene duplications. To identify the cause of this issue, we aligned back the HiFi reads to those assemblies and checked the depth of coverages and mapping qualities. It showed that the samples HG01358, HG01123 and HG002 contained false duplications in at least one haplotype of length around 55 Mb (in h1tg000058l contig), about 14 Mb (h1tg000013l) and around 70 Mb (h2tg000045l), respectively. In the assembly graphs of HG01358 and HG01123, the duplicated HiFi reads that appeared multiple times were used as anchors to manually determine the exact boundaries of the duplicated regions in the contigs. These two contigs were then manually fixed by breaking the contigs at the duplication start and end points and discarding the duplicated sequence from the assembly. In detail, for HG01123 for h1tg000013l, we discarded the interval [94439457, contig end]. For HG01358 for h1tg000058l, we kept the interval [0, 95732608), renaming the contig to h1tg000058l_1, we discarded the interval [95732608, 150395342) and kept the interval [150395342, contig end], renaming it to h1tg000058l_2. To address the false duplication in HG002 we re-assembled it using a newer version of Trio-Hifiasm (v.0.14.1), which was reported not to have this problem.

We also evaluated the phasing accuracy of the assemblies by using yak trioeval (see below). We detected a single large misjoin in a maternal contig of the HG02080 assembly. It contained an approximately 22-Mb-long paternal block in the middle of the contig, which resulted in two switch errors at the edges of this block. This block was manually discarded from the assembly and the contig was broken into two smaller ones. In detail, in HG02080 for the h2tg000053l contig, we kept the interval [0, 41506503), renaming it to h2tg000053l_1, we discarded the interval [41506503, 63683095) and kept the interval [63683095, contig end], renaming it to h2tg000053l_2.

We finally searched for interchromosomal misjoins using the Minigraph pangenome (see below for construction details). An 'interchromosomal misjoin' was defined by a chimeric Minigraph alignment (see below) consisting of ≥1 Mb subalignments on different chromosomes.

**Cleaning steps.** To clean the raw assemblies, we performed three additional steps: masking the remaining HiFi adapters, dropping the contigs that were contaminated in their entirety and removing any redundant mitochondrial contigs.

In the first cleaning step, the sequence of the PacBio SMRTbell adapter was aligned to each assembly using minimap2 with the parameters -cxsr -f5000 -N2000 --secondary=yes. We extracted only the hits with less than or equal to 2 mismatches and which were longer than 42 nt. In addition, eukaryotic adapters in each assembly were identified by VecScreen[78]. The combined minimap2 and VecScreen adaptor hits (when present) were hard-masked in the assemblies using a WDL of the bedtools maskfasta command (https://dockstore.org/workflows/github.com/human-pangenomics/hpp_production_workflows/MaskAssembly:master?tab=info).

```
bedtools maskfasta \
        -fi ${inputFastaFN} \
        -bed ~{adapterBed} \
        -fo ~{outputFasta}
```

In the second cleaning step, we used VecScreen to detect mitochondrial contigs and contigs consisting of nonhuman sequences from other organisms, such as bacteria, viruses and fungi. These contigs were then dropped from the assemblies using a WDLized version of samtools faidx. It is worth noting that the contigs with nuclear mitochondrial DNAs within them were not dropped.

```
samtools faidx \
        $inputFastaFN \
        `cat contigsToKeep.txt` | gzip \
        > ~{outputFasta}
```

In the last cleaning step, we selected one contig as the best mitochondrial contig per diploid assembly. To do this, selection we aligned the sequence of the mitochondrial DNA (with the RefSeq identifier of NC_012920.1) to each diploid assembly using minimap2 with the parameters -cx asm5 --cs. Then we selected one contig with the highest mapping score and the lowest number of mismatches as the best mitochondrial contig (we selected one randomly if multiple best contigs existed). This contig was then rotated and flipped (if necessary) to match the start and orientation of NC_012920.1.fa and then added to the maternal assembly of the corresponding sample. Only the HG01071 sample did not produce any identifiable mitochondrial contig.

Masked and cleaned mitochondrial assemblies were then accessioned to GenBank, where they underwent another round of adapter masking and removal of contamination, which was mostly Epstein–Barr virus used to generate the LCLs. The final cleaned assemblies in GenBank were downloaded, and the contig identifiers were pre-pended with the sample name and haplotype integer (where 1 = paternal and 2 = maternal). For example, a contig assigned the name JAGYVH010000025 in sample HG02257's maternal assembly was renamed to be HG02257#2#JAGYVH010000025. The renamed assemblies were then released to our Amazon Simple Storage Service (S3) and Google Cloud Platform (GCP) buckets. In the process of downloading from GenBank, three of the assemblies (HG00733 paternal, HG02630 paternal and NA21309 maternal) had their downloads prematurely stopped, which resulted in missing sequences. Notably, NA21309 is missing its mitochondrial contig. Details can be found on the HPRC's year 1 assembly GitHub repository (https://github.com/human-pangenomics/HPP_Year1_Assemblies). The assemblies held at the International Nucleotide Sequence Database Collaboration are not truncated, but the truncated copies were retained in S3 and GCP as they were used in the construction of the pangenomes.

After submission to GenBank, the assemblies were aligned against CHM13 using Winnowmap, and multiple contigs were found to be unmapped. These contigs were subjected to BLAST and found to be almost exclusively Epstein–Barr virus sequence. GenBank confirmed (personal communication) that these unmapped contigs should have been dropped as contamination, but since the genomes were already in active use, they elected not to remove them at this time. A list of the contigs that should have been dropped can be found on the year 1 assembly GitHub repository (https://github.com/human-pangenomics/HPP_Year1_Assemblies/blob/main/genbank_changes/y1_genbank_remaining_potential_contamination.txt).

## Assembly assessment pipeline

Several steps in assembly assessment were managed through a StandardQC workflow written using WDL, run on AnVIL, and available at Dockstore (https://dockstore.org/workflows/github.com/human-pangenomics/hpp_production_workflows/StandardQC). Individual tools within the workflow were run in Docker containers with specific tool versions installed for consistency and reproducibility. Details are available within the Dockstore-deposited workflow. The StandardQC workflow takes short-read data for parental and child samples, the two assembly haplotypes, and it produces an analysis over various quality metrics produced by the tools described below. For each task, the workflow produced a small human-readable summary file, which is also easy to parse for summarizing steps, as well as the full output from the tool for manual inspection. Specific tool invocations can be determined from the deposited workflow and are described in the subsequent sections.

**Measuring interchromosomal joins.** Contigs were aligned to CHM13 (v.2.0) with Minigraph (v.0.18) and processed with the following command line:

```
minigraph -cxasm chm13v2.0.fa contigs.fa | paftools.js misjoin –
```

The 'misjoin' command reports an interchromosomal join if a contig has two ≥1 Mb alignments to two different chromosomes.

**Assembly contiguity assessment.** Assembly contiguity was assessed for each haplotype using QUAST[79]. These statistics included total sequence assembled, total assembled contigs and contig NG50 (assuming a genome size of 3.1 Gb). All reference-based analyses were skipped.
QUAST was invoked with the following command:

```
python /opt/quast/quast-5.0.2/quast-lg.py -t 16 -o <sample>.quast --large --est-ref-size 3100000000 --no-icarus
```

**Assembly QV assessment.** Assembly QV was determined using two separate $k$-mer-based tools. The first is Yak[20]. Yak's QV estimation happens separately on each haplotype. The $k$-mer databases for Yak were generated using the following command:

```
yak count -t16 -b37 -o <sample>.yak <(cat <read_files>) <(cat <read_files>)
```

QV estimation using Yak was generated with the following command:

```
yak qv -t 32 -p -K 3.2g -l 100k <sample>.yak <sample_assembly_haplotype> > <sample>.<haplotype>.yak.qv.txt
```

Assembly QV was also determined using Meryl and Merqury[80]. Meryl generates $k$-mer databases and Merqury determines haplotype QV jointly with both haplotypes.
The $k$-mer databases with Meryl were generated with the following commands. Databases were generated separately for each read file using meryl count and merge with meryl union-sum. Parental-specific $k$-mers (hap-mers) were generated using merylu hapmer.

```
meryl k=21 threads=64 memory=32 count output <sample>.meryl <read_file>
meryl union-sum output <sample>.meryl <sample_read_meryl_files>
bash hampers.sh maternal.meryl paternal.meryl sample.meryl
```

QV estimation using Merqury was generated with the following command:

```
merqury.sh sample.meryl maternal.meryl paternal.meryl <maternal_haplotype> <paternal_haplotype> <sample>.merqury
```

**GIAB-based assembly quality assessment.** As a complementary and stratified assessment of assembly quality, we used the GIAB assembly benchmarking pipeline to compare assembly-based variant calls to GIAB's small variant benchmarks (v.4.2.1) for two GIAB samples assembled in this work: HG002 and HG005. We evaluated the HG002 and HG005 HPRC assemblies aligned to GRCh38. Variants were called from assemblies using Dipcall (v.0.3) (using mimimap2 (v.2.2.4))[38]. We used -z200000,10000 parameter to improve alignment contiguity, as it was previously shown to improve variant recall in regions with dense variation, such as the major histocompatibility complex[81]. Small variant evaluation was performed using hap.py (v.3.15)[82], benchmarking against v.4.2.1 of high-confidence SNP, small indel and homozygous reference calls for the GIAB samples HG002 and HG005. Comparisons were performed with and without restriction to the associated dipcall region file (dip.bed) to assess recall within and outside assembled regions. For better comparisons of complex variants, hap.py was run using vcfeval[83]. Variant calls were stratified using GIAB stratifications (v.3.0)[84], stratifying true-positive, false-positive and false-negative variant calls in challenging and targeted regions of the genome.

**Trio-based assembly phasing assessment.** Assembly phasing was assessed using Yak and described using two statistics: switch error and Hamming error rates. Switch error describes the number of times two adjacent phased variants incorrectly switch between maternal and paternal haplotypes. Hamming error rate relates to the total number of misphased variants per assembled contig. Yak generates phasing statistics separately for each haplotype using parental $k$-mers gathered from Illumina short-read sequencing of the parents.
Yak generates $k$-mer databases for the sample and both parental haplotypes (as described above). We used Yak to generate phasing metrics with the following command:

```
yak trioeval -t 32 paternal.yak maternal.yak <haplotype_assembly> > <sample>.<haplotype>.yak_phasing.txt
```

**Hi-C-based assembly phasing assessment.** An alternative approach for phasing evaluation is to use Hi-C reads that do not require trio information. We computed the switch error rate for local phasing evaluation and the Hamming error rate for global phasing evaluation. We implemented an efficient $k$-mer-based method in pstools (v.0.1)[24] and used maximum Hi-C read support to detect switch errors on heterozygous positions. In this procedure, we first identified heterozygous $k$-mers (hets) from phased assemblies using 31-mers. Then we mapped Hi-C reads to the assemblies using these 31-mers. If there were >5 reads that supported a switch between consecutive hets in assemblies, we considered a haplotype switch. For each het pair, we noted whether Hi-C reads supported or did not support the phase. We considered a switch error when a het site had a phase switched support relative to that of the previous heterozygous site. The switch error rate is the number of local switches divided by the number of heterozygous sites. We performed this operation for the entire contig over all contigs for switch calculations. For the Hamming error calculations, we considered Hamming distance on the entire contig level divided by the number of heterozygous sites. This measure gives a global view of phasing errors and implicitly penalizes any long switches in contigs.

## Assembly read-based evaluation of Flagger

The following describes the generation and cleaning of the HiFi alignments to the HPRC assemblies and running Flagger (v.0.1), a read-based pipeline for evaluating diploid/dual assemblies. All the WDL-based workflows for running these steps are deposited in the Dockstore collection (https://dockstore.org/organizations/HumanPangenome/collections/Flagger-Secphase).

**Preparing the HiFi alignments.** We aligned back the HiFi reads of each sample to its diploid assembly. The alignments were produced with winnowmap (v.2.03) using the following commands:

```
# making the k-mer table with meryl
meryl count k=15 output merylDB asm.fa
meryl print greater-than distinct=0.9998 merylDB > repetitive_k15.txt

# alignment with winnowmap
winnowmap -W repetitive_k15.txt -ax map-pb -Y -L --eqx --cs -I8g
<(cat pat_asm.fa mat_asm.fa) reads.fq.gz | samtools view -hb > read_alignment.bam
```

For all samples, we used the full HiFi read sets mentioned in Supplementary Table 1, except HG002, for which we downsampled the read set to 35×.

To exclude unreliable alignments, we removed all chimeric alignments and alignments shorter than 2 kb or with a gap-compressed mismatch ratios higher than 1%. As the assembly is diploid and the reads aligned to the homozygous regions are expected to have low mapping qualities, we did not filter alignments on the basis of their mapping qualities. In Supplementary Fig. 41, we plot the histograms of mapping qualities and the distributions of alignment identities for one sample, HG00438, as an example. The statistics of three sets of alignments were plotted: the alignments to the diploid assembly and to each haploid assembly (maternal and paternal) separately. It indicates that the reads have higher identities when the diploid assembly is used as reference but about 20% more reads have mapping qualities lower than 10.

Generally, in highly homozygous regions, the aligner may not be able to select the correct haplotype as the primary alignment because of either read errors or misassemblies. To detect these cases, we searched for secondary alignments for which the scores were almost as high as the primary alignment of the same read. For each such read, we made a pseudo-multiple alignment of the read sequence and the assembly blocks captured by all secondary and primary alignments. Using this alignment, we searched for the read bases that were mismatched in at least one alignment but not all alignments. We called such bases single nucleotide markers. For each alignment, we calculated a consistency score by considering only the single nucleotide markers and taking the summation of their base qualities with a negative sign. We then sorted the alignments (regardless of being primary or secondary) based on this score. If the best alignment was a secondary alignment, we assigned the primary tag to this alignment and removed the other alignments. The percentage of the total reads with swapped alignments ranged from 0.03% (HG03453) to 0.44% (HG005) across 47 HPRC samples. This result shows that only a small percentage of the reads needed to be relocalized using this method. This step was performed through the Secphase (v.0.1) workflow, which is available in the Dockstore collection (https://dockstore.org/organizations/HumanPangenome/collections/Flagger-Secphase).

By calling variants, it is possible to detect the regions that either need polishing (that is, are errors) or that have alignments from the wrong haplotype because of mismappings. We used DeepVariant (v.1.3.0) with the parameter --model_type="PACBIO" to call variants on these alignments. The variants were then filtered to include only the biallelic SNPs with a variant frequency higher than 0.3 and genotype quality higher than 10.

```
bcftools view -Ov -f PASS -m2 -M2 -v snps -e 'FORMAT/VAF <
0.3 || FORMAT/GQ < 10' ${OUTPUT_VCF} > ${SNPS_VCF}
```

Having the biallelic SNPs, we found the alignments with alternative alleles and removed them from the bam file. For this step, we implemented and used the program filter_alt_reads, running the following command:

```
filter_alt_reads -i ${INPUT_BAM} -o ${ALT_FILTERED_BAM} -f
${ALT_BAM} -v ${SNPS_VCF}
```

**Running the evaluation pipeline.** To assess the read mappings resulting from our diploid alignment process, we used the following five steps, which are combined into a pipeline that we refer to as Flagger. Flagger essentially fits a mixture model to successive coverage blocks of the read-to-diploid assembly alignment and then classifies each block to a category predicting the accuracy of the assembly at that location.

**Step 1: calculating depth of coverage.** First, after producing and cleaning the HiFi alignments, we calculated the depth of coverage for each assembly base by samtools depth -aa (the -aa option allows outputting the bases with zero coverage):

```
samtools depth -aa -Q 0 read_alignment.bam > read_alignment.depth
```

The output of samtools depth was then converted into a more efficient format with the .cov suffix. This format is implemented specifically for Flagger and is more efficient, as the consecutive bases with the same coverage take only one line. We implemented a program called depth2cov for converting the output of samtools depth to the .cov format.

```
depth2cov -d read_alignment.depth -f asm.fa.fai -o read_alignment.cov
```

**Step 2: fitting the mixture model.** In the second step, the frequencies of coverages were calculated using cov2counts. The output file with the .count suffix is a two-column tab-delimited file: the first column shows coverages and the second column shows the frequencies of those coverages.

```
cov2counts -i read_alignment.cov -o read_alignment.counts
```

The python script fit_gmm.py takes a file .counts suffix, fits a Gaussian mixture model and finds the best parameters through expectation-maximization. This mixture model consists of four main components and each component represents a specific type of region:
(1) Erroneous component, which is modelled by a Poisson distribution. To avoid overfitting, this mode only uses the coverages below 10 so its mean is limited to be between 0 and 10. It represents the regions with very low read support.
(2) (Falsely) duplicated component, which is modelled by a Gaussian distribution, the mean of which is constrained to be half of mean of the haploid component. It should mainly represent the falsely duplicated regions.
(3) Haploid component, which is modelled by a Gaussian distribution. It represents blocks with the coverages that we expect for the blocks of an error-free assembly.
(4) Collapsed component, which is actually a set of components each of which follows a Gaussian distribution, the mean of which is a multiple of the mean of the haploid component. It represents regions that have additional copies present in the underlying genome that have been 'collapsed' into a single copy.

It was noted that the model components may change for different regions owing to regional coverage differences and that the resulting systematic differences affect the accuracy of the partitioning process. To make the coverage thresholds more sensitive to the local patterns, the diploid assembly was split into windows of length (5–10 Mb) and a distinct model was fit for each window. Before fitting, we split the whole-genome coverage file produced in the first step into multiple coverage files for each window. We implemented and ran split_cov_by_window for splitting:

```
split_cov_by_window -c read_alignment.cov -f asm.fa.fai -s 5000000
-p ${OUTPUT_PREFIX}
```

This produced a list of coverage files, each of which ends with
${CONTIG_NAME}_${WINDOW_START}_${WINDOW_END}.cov

We then repeated the above-described steps for each resulting coverage file.

One important observation is that for short contigs, the coverage distribution is generally too noisy to satisfactorily fit the mixture model. To address this issue, we performed the window-specific coverage analysis only for the contigs longer than 5 Mb. For the shorter contigs, we used the results of the whole-genome analysis.

**Step 3: extracting blocks of each component.** Using the fitted model, we assigned each coverage value to one of the four components (erroneous, duplicated, haploid and collapsed). To do so for each coverage value, we picked the component with the highest probability. For example, the coverage value 0 is frequently assigned to the erroneous component. In Supplementary Fig. 42, the coverage intervals are coloured based on their assigned component.

**Step 4: incorporating coverage biases in HSats.** According to an article describing a complete human genome[3], there are some satellite arrays (for example, HSat1, HSat2 and HSat3) for which the HiFi coverage drops or increases systematically owing to biases in sample preparation and sequencing. Such platform-specific biases mislead the pipeline. As a result, the falsely duplicated component may contain a mixture of falsely duplicated and coverage-biased blocks. Similar effects occur for the collapsed component.

To incorporate such coverage biases and to correct the results in the corresponding regions, we first found the regions of each haploid assembly for which a coverage bias is expected. To find such regions, we lifted over the CHM13 HSat1, HSat2 and HSat3 annotation to each assembly by aligning the assembly contigs to the reference T2T-CHM13 (v.1.1) and GRCh38 (chromosome Y) and projecting the HSat coordinates back to the assembly (using python script project_blocks.py). Then we ran fit_gmm.py to fit a mixture model for the blocks assigned to each HSat type and adjusted the parameter --coverage, the starting point of the expectation-maximization process, based on the expected coverage in the corresponding HSat. For HSat1, HSat2 and HSat3 we set --coverage to 0.75, 1.25 and 1.25 times the average sequencing coverage, respectively. Finally, we decomposed each HSat based on the inferred coverage thresholds and replaced the previous assigned component by the new one.

**Step 5: using high-quality alignments to correct spurious flags.** In some cases, the duplicated component was mixed up with the haploid one. It usually happens when the coverage in the haploid component drops systematically or the majority of a long contig is falsely duplicated. To address this issue, we used another indicator of a false duplication, which is the accumulation of alignments with very low mapping quality (MAPQ). We produced another coverage file using only the alignments with MAPQ > 20. Whenever we found a region flagged as duplicated with more than five high-quality alignments, we changed the flag to haploid.

After the correction made in step 5, we merged blocks from each component closer than 1,000, and the overlap of any two components after merging was flagged as 'unknown' to show that this block could not be properly assigned. The BED files produced by Flagger are available in the HPRC S3 bucket (https://s3-us-west-2.amazonaws.com/human-pangenomics/index.html?prefix=submissions/e9ad8022-1b30-11ec-ab04-0a13c5208311--COVERAGE_ANALYSIS_Y1_GENBANK/FLAGGER/APR_08_2022/FINAL_HIFI_BASED/FLAGGER_HIFI_ASM_SIMPLIFIED_BEDS/).

**Assessing T2T-CHM13 using Flagger.** To estimate the false-positive rate of Flagger, we applied it to the T2T-CHM13 (v.1.1) reference. The direct output of Flagger showed that about 12.77 Mb (around 0.41%) of the T2T-CHM13 reference assembly was flagged as potentially unreliable. The HPRC assemblies were almost free of rDNA arrays, but there were modelled sequences for rDNA arrays in the T2T-CHM13 (v.1.1) reference. These arrays were flagged as falsely duplicated in their entirety, which indicated that Flagger with HiFi reads may not be able to correctly evaluate rDNA arrays. Therefore, to make a fair comparison, we excluded rDNA arrays (about 9.92 Mb in total) from the reference evaluation, which decreased the number of unreliable bases to 5.58 Mb (around 0.18%). We additionally identified about 2.76 Mb of a region beside chromosome 1–HSat2 that was mis-flagged as collapsed. This mis-flagging was the impact of the systematic coverage rise on the neighbouring HSat2 that altered the fitted mixture model. By manually fixing this mis-flagging, we had about 2.82 Mb (0.09%) of unreliable blocks in T2T-CHM13 (v.1.1). This number is about 9.3 times lower than the average for the HPRC assemblies. These unreliable blocks are mainly a combination of 'Unknown' blocks, which could not be properly assigned and the regions with HiFi-specific coverage drops. The results of this analysis are available in the HPRC S3 bucket (https://s3-us-west-2.amazonaws.com/human-pangenomics/index.html?prefix=submissions/e9ad8022-1b30-11ec-ab04-0a13c5208311--COVERAGE_ANALYSIS_Y1_GENBANK/FLAGGER/APR_08_2022/FINAL_HIFI_BASED/T2T-CHM13/).

### Repeat masking
Repeat masking on each assembly was iteratively performed using RepeatMasker (v.4.1.2-p1). The first step masked used the default human repeat library, and the second step used a repeat library augmented by CHM13 satellite DNA sequences on the original assemblies after hard masking the initial repeat masked DNA. The augmented repeat library (final_consensi_gap_nohsat_teucer.embl.txt) is available at Zenodo (https://doi.org/10.5281/zenodo.5537107), and a parallelized repeat masking pipeline (RepeatMaskGenome.snakefile) is available at GitHub (https://github.com/chaissonlab/segdupannotation). The union of the two steps generated the complete repeat masking.

### SD annotation
SDs were annotated using sedef[85] after masking repeats in each assembly. Repeats annotated with more than 20 copies corresponded to unannotated mobile elements and were excluded from the analysis. The pipeline for annotating SDs is available at GitHub (https://github.com/ChaissonLab/SegDupAnnotation/releases/tag/vHPRC).

### SD reliability
The reliable and unreliable regions for all haplotype assemblies were aligned to T2T-CHM13 (v.2.0) and then subdivided into 5 kb windows and intersected with the SD annotations for T2T-CHM13. SD annotations were unavailable for chromosome Y on T2T-CHM13 (v.2.0) at the time of analysis. Furthermore, the chromosome Y added to T2T-CHM13 is from the HPRC HG002 sample; therefore chromosome Y was excluded. For each class of unreliable region (unknown, erroneous, duplicated, collapsed and haploid), we calculated the average number of base pairs overlapping SDs across the haplotype assemblies and annotated each 5 kb window with the most representative overlapping SD (the SD with the highest product of identity and length). Then using the most representative SD, we calculated the average length and identity of SDs overlapping each class of unreliable region for all the 94 haplotypes and compared the length of identity of SDs that overlapped the different types of errors in the assembly. The code for this analysis is available on GitHub (https://gist.github.com/mrvollger/3bdd2d34f312932c12917a4379a55973).

## Ensembl mapping pipeline for annotation

To create high-confidence annotations, a new Ensembl annotation pipeline was developed. The pipeline clusters and maps spatially proximal genes in parallel (to help avoid issues with individually mapping near identical paralogues) and attempts to resolve inconsistent mappings by both considering the synteny of the gene neighbourhood in relation to the GRCh38 annotation and the identity and coverage of the underlying mappings.

A reference gene set was created from a subset of the GENCODE (v.38) genes[29], which was mapped to the HPRC assemblies through a two-pass alignment process. This excluded readthrough genes and genes on patches or haplotypes, and only included one copy of the genes on the X/Y PAR region (only one copy, chromosome X, is modelled in the Ensembl representation of the *PAR* genes).

First, to minimize the difficulty of mapping near identical paralogues, a jumping window of 100 kb in length was used to identify clusters of genes to map in parallel (Supplementary Fig. 43). The initial window was positioned at the start of the most 5′ gene for each chromosome in the GRCh38 reference and extended 100 kb from the start of the gene. Any genes fully or partially overlapping the window were then included in the cluster. The next 3′ gene that did not overlap the previous window was then identified, and a new window was created and the process repeated. This resulted in both clustered genes and non-clustered genes (genes were considered not clustered when there was only one gene within the window). The regions to map were then identified on the basis of the start of the most 5′ gene and the end of the most 3′ gene in each cluster (or simply the 5′ and 3′ end of the gene in the case of non-clustered genes).

For each region defined in the previous step, anchor points were then selected to help map the region on the target genome. Two 10 kb anchor points were created 5 kb from the 5′ and 3′ edge of the region, and a central 10 kb anchor was created around the midpoint of the region in the GRCh38 genome. The sequences of these anchors were them mapped against the target genome using minimap2 (ref. 86) with the following command:

```
minimap2 --cs --secondary=yes -N 10 -x map-ont [genome_index]
[anchor_file] > [alignment_file]
```

The resulting hits were examined to determine high-confidence regions in the target genome. High-confidence regions were ones in which all three anchors were on the same top-level sequence, in colinear order, with ≥99% sequence identity and ≥50% hit coverage, and with a similar distance between the anchors when compared with the reference genome. If no suitable candidate region was found with all three anchors, pairs of mapped anchors were then assessed in a similar manner.

The sequence-selected region or regions were then retrieved and aligned against the corresponding GRCh38 region using MAFFT. For each gene, the corresponding exons were retrieved and the coordinates were projected through the alignment of the two regions. Transcripts were then reconstructed from the projected exons. For each transcript, the coverage and identity when aligned to the parent transcript from GRCh38 were calculated.

If the resulting transcript had either a coverage of <98% or an identity of <99%, the parent transcripts were aligned to the target region using minimap2 in splice-aware mode, with the high-quality setting for Iso-Seq/cDNA transcripts enabled. The maximum intron size was set to 100 kb by default. For transcripts with reference introns larger than 100 kb, the maximum intron size was scaled and set as 1.5 times the length of the longest intron (to allow some variability):

```
minimap2 --cs --secondary=no -G [max_intron_size] -ax splice:hq
-u b [expected_target_region] [transcript_sequences] > [sam_file]
```

For each transcript that mapped to the target genome, the quality of the mapping was assessed on the basis of aligning the original reference sequence with the newly identified target sequence. Again, if the coverage or identity of the aligned sequence was <98% or <99%, respectively, the reference transcript sequence was re-aligned to the target region, this time using Exonerate[87]. Exonerate, although slower than minimap2, has the ability to handle very small exons and can incorporate CDS data to preserve the CDS (introducing pseudo-introns as needed). The following command was used:

```
exonerate -options --model cdna2genome --forwardcoordinates
FALSE --softmasktarget TRUE --exhaustive FALSE --score 500
--saturatethreshold 100 --dnawordlen 15 --codonwordlen 15
--dnahspthreshold 60 --bestn 1 --maxintron [max_intron_size]
-coverage_by_aligned 1 --querytype dna --targettype [target_type]
--query [query_file] --target [target_file] --annotation [annotation_file] >
[output_file]
```

When more than one approach was used to model the transcript, the mapping with the highest combined identity and coverage was selected.

For genes not mapped through the initial regional anchors, a second approach was used. The expected location of the gene was located using high-confidence genes mapped during the first phase. High-confidence mappings were those for which there was a single mapped copy of the gene, all the transcripts had mapping scores of 99% coverage and identity on average and the gene also had a similar gene neighbourhood to the neighbourhood in the reference (at least 80% of the of the same genes in common for the 100 closest neighbouring genes in the reference). After this step, the entire genome region underlying the missing gene, including a 5 kb flanking sequence, was mapped against the target genome using minimap2:

```
minimap2 --cs --secondary=yes -x map-ont [genome_index] [gene_
genomic_sequence] > [alignment_file]
```

The resulting hits were then filtered on the basis of overlap with the expected region that the missing gene should lie in. If there was no expected region calculated (cases in which no pair of high-confidence genes could be found to define the 5′ and 3′ boundaries of the expected location of the missing gene, for example, at the edge of a scaffold) or no hit overlapping the expected region was found, the top reported hit was used providing it passed an identity cut-off of 99%. The selected hit or hits were then extended on the basis of how much of the original reference gene they covered to ensure that minor local variants between the reference and target regions did not lead to the target region being truncated. Once extended, the remaining hits were then clustered on the basis of genomic overlap and merged into unique regions. The missing genes were then attempted to be mapped to these regions using an identical process as described above for the initial mappings, involving MAFFT, minimap2 and Exonerate.

To minimize the occurrences of mis-mapped paralogues, each gene was checked for exon overlap in both the target and the reference. If the overlapping genes were not identical at a locus between the reference and the target, then a conflict was identified. For each gene present, filtering was done to reduce or remove the conflict based on a number of factors, including whether the genes were in the expected location, whether the genes were high-confidence mappings, the average percent identity and coverage of the transcript for the genes and the neighbourhood score. When it was not possible to resolve a conflict between two genes, both were kept. This concluded the primary mapping process.

After this process, potential recent duplications were identified. To search for recent duplications, the canonical transcript of each gene (the longest transcript in the case of noncoding genes, or the transcript with the longest translation followed by the longest overall sequence

for protein-coding genes) was selected and aligned across the genome using minimap2 in a splice-aware manner:

```
minimap2 --cs --secondary=no -G [max_intron_size] -ax splice:hq
-u b [genome_index] [input_file] > [sam_file]
```

Mappings that had exon overlap with existing annotations from the primary mapping process on the target genome were removed. For new mappings that did not overlap existing annotations, the quality of the alignment was then assessed by aligning the mapped transcript sequence to the corresponding reference transcript to calculate the coverage and per cent identity of the mapping. Different coverage and per cent identity cut-offs were used for these mappings on the basis of the type of transcript mapped. Protein-coding and small noncoding transcripts used a coverage and identity cut-off of 95%, whereas long noncoding transcripts used a coverage and identity cut-off of 90%. Pseudogene transcripts had a lower coverage cut-off of 80%, but the same identity cut-off of 90% as long noncoding transcripts.

When looking for new paralogues, for cases in which multiple canonical transcripts mapped to a locus, a single representative transcript was selected. This was based on the following hierarchy of gene biotype groups: coding, long noncoding, pseudogene, small noncoding, and miscellaneous or undefined.

If there were multiple transcripts for the highest represented group, the transcript with the longest sequence was selected as the representative.

### Gene annotation quality analysis
**Frameshifts.** For the Ensembl and CAT gene annotation sets, we identified the locations of frameshifting indels by iterating over the coding sequence of each transcript and looking for any gaps in the alignment. If the gap had a length that was not a multiple of 3, and its length was <30 bp long (to remove probable introns from consideration), the gap was determined to be a frameshift and its location saved to a BED file.

**Nonsense mutations.** We also analysed the number of nonsense mutations that would cause early stop codons in both the Ensembl and CAT gene annotation sets. We identified nonsense mutations by iterating through each codon in the coding sequence of the predicted transcripts. If there was an early stop codon before the canonical stop codon at the end of the transcript, we saved the location in a BED file.

**Validation of mutations using Illumina.** For both sets of mutations, we then lifted over the coordinates of the mutations to be on the GRCh38 reference so that we could use existing variant call sets on GRCh38. We used halLiftover to lift over each set of coordinates, using the GRCh38-based HAL file from the MC alignment. Then we used bedtools intersect to intersect with the variant call file for each of the assemblies.

The following sample commands were used:

```
halLiftover GRCh38-f1g-90-mc-aug11.hal <GENOME_NAME>
<MUTATION_BED_FILE> GRCh38 <LIFTED_OVER_BED_FILE>
```

```
bedtools intersect -wo -a <LIFTED_OVER_BED_FILE> -b <SAMPLE_
MERGED_VCF> > <OVERLAP_OUTPUT_TXT_FILE>
```

The VCF files used in this intersection were downloaded from the 1KG (https://urldefense.com/v3/_https://ftp.1000genomes.ebi.ac.uk/vol1/ftp/data_collections/1000G_2504_high_coverage/working/20201028_3202_raw_GT_with_annot/20201028_CCDG_14151_B01_GRM_WGS_2020-08-05_chr$i.recalibrated_variants.vcf.gz_;!!NLFGqXoFfo8MMQ!r6nD4EtteJ7k2BauOrREfgIrlxEI2Upx35sNHiqyI8Did-a6UUUzzxGVQkwYkb-bE_rlHQN2Jw2cBdlw7te_-Q$).

Where $i was replaced with each chromosome number. From there, each chromosome VCF was split so that each sample was in its own

file using bcftools view. The chromosome files for each sample were combined into one VCF using bcftools concat.

### Gene duplication analysis
Duplicated genes were detected as multi-mapped coding sequences using Liftoff[88] supplemented by a complementary approach (gb-map) with multi-mapped gene bodies. The combined set was formed by including all Liftoff gene duplications and duplicated genes detected by gb-map.

**Liftoff.** We ran Liftoff (v.1.6.3) to annotate extra gene copies in each of the assemblies. Liftoff was run with the flag -sc = 0.90 to find additional copies of genes, with an identity threshold of at least 90%. An example command is below:

```
liftoff -p 10 -sc 0.90 -copies -db <GENCODE_V38_DATABASE> -u
<UNMAPPED_FILE> -o <OUTPUT_GFF3> -polish <GENOME_FASTA>
<GRCh38_FASTA>
```

The additional copies of the genes were identified as such in the output gff3 with the field extra_copy_number (equal to anything other than 0). For this analysis, we also only considered genes that were multi-exon, protein-coding genes. The additional gene copies were further filtered to remove any genes outside the 'reliable' haploid regions as determined by the Flagger pipeline.

**gb-map.** The gene-body mapping pipeline identifies duplicated genes by first aligning transcripts of protein-coding and pseudogenes (GENCODE v.38) to each assembly and then multi-mapping the genomic sequences of each corresponding gene. Alignments of at least 90% identity and 90% of the length of the original duplication were considered candidate duplicated genes. Candidates were removed if they overlapped previously mapped transcripts from other genes, low-quality duplications and genes identified through CAT and Liftoff analysis.

**Gene family analysis.** To account for gene duplications in high-identity gene families, gene families were identified on the basis of sequence alignments from gb-map. Genes that mapped reciprocally with 90% identity and 90% length were considered a gene family. A single gene was selected as the representative gene for the family, and any gene duplication in the family was counted towards that gene.

### Pangenome graph construction
**Minigraph.** Minigraph can rapidly perform assembly-to-graph mappings using a generalization of the minimap2 algorithm[34]. New SVs of at least 50 bp detected in the mapping can then be added to the graph. To construct a pangenome graph, one chosen reference assembly, GRCh38 in this case, was used as a starting graph, and the mapping and SV addition steps were repeated for each additional assembly, greedily. This iterative approach is analogous to partial order alignment (POA)[89]. Graphs constructed in this way describe the structural variation within the samples and provide a coordinate system across the reference and all insertions. Minigraph does not produce self-alignments. That is, it will never align a portion of the reference assembly onto another portion of the reference assembly. In this way all reference positions have a unique location within the created pangenome. Minigraph (v.0.14) was used with -xggs options. The input order was GRCh38, CHM13 then the remainder in lexicographic order by sample name.

**MC.** Graphs constructed by Minigraph only contain structural variation (≥50 bp) by default. The aim of the MC pipeline is to refine the output of Minigraph to include smaller variants, down to the SNP level. Doing so allows the graph to comprehensively represent most variation, as well as to embed the input haplotypes within it as paths,

which is important for some applications[49]. To remove noisy alignments from the MC pangenome, long (≥100 kb) non-reference sequences identified as being satellite, unassignable to a reference chromosome or which appear unaligned to the remainder of the assemblies were removed from the graph. This resulted in a pangenome with significantly reduced complexity that nevertheless maintained all sequences of the starting reference assembly and the large majority of those in the additional haplotypes. The MC pipeline is composed of the five steps described below and in more detail in ref. 35. The script and commands to reproduce this process can be found at GitHub (https://github.com/ComparativeGenomicsToolkit/cactus/blob/81903cb82ae80da342515109cdee5a85b2fde625/doc/pangenome.md#hprc-version-10-graphs). A newer, simpler version of the pipeline that no longer requires satellite masking can be found at GitHub (https://github.com/ComparativeGenomicsToolkit/cactus/blob/5fed950471f04e9892bb90531e8f63be911857e1/doc/pangenome.md#hprc-graph).

Paths from the reference, GRCh38, are acyclic in the MC graph. Paths from any other haplotypes can contain cycles (as a result of different query segments mapping to the same target), but they are relatively rare.

(1) Satellite masking: Minigraph is unable to map through highly repetitive sequences such as centromeres and telomeres and, as these regions are also enriched for misassemblies (see the section 'Assembly assessment' of Results), we decided to exclude them from the MC graphs used in this work. dna-brnn is a tool that uses a recurrent neural network to quickly identify alpha satellites as well as human satellites 1 and 2 (ref. 90). We ran it with its default parameters on all input sequences and cut out any identified regions ≥100 kb, except on the reference. The three satellite families that dna-brnn detects account for the majority of satellite sequences, but not all. As such, gaps ≥100 kb in Minigraph mappings were also removed. They were detected by mapping each assembly, after having removed the dna-brnn regions, to the Minigraph (using the procedure described below). Overall, an average of 188.6 Mb of sequence from each (non-reference) assembly was excluded from the graph.

(2) Assembly-to-graph mapping: Minigraph generalizes the fast seeding and chaining algorithms of minimap2, but it does not currently produce exact alignments in cigar strings or otherwise. For this work, an option, --write-mz, was added to report chains of minimizers, which in this case are 15 bp exact matches, and all assemblies were mapped to the Minigraph graph using it. The resulting minimizers were then converted into PAF files with cigars representing exact pairwise alignments between the query contigs and Minigraph node sequences, and all mappings with MAPQ < 5 were excluded, as were overlapping query regions >10 kb.

(3) Chromosome decomposition: the Minigraph graphs do not contain inter-chromosomal rearrangements, but the mappings performed in the previous steps can imply them. That is, a contig can partially map to multiple chromosomes. In most cases, these mappings involve similarity across different acrocentric short arms. To avoid introducing misleading interchromosomal events, and because it is necessary to run the subsequent steps individually on chromosomes owing to memory requirements, the mappings were divided by reference chromosome. This was done by splitting the Minigraph into connected components and using the RN tags to determine their corresponding chromosome names. The PAF mappings were used to determine the coverage of each query contig with each chromosome component. This coverage was used to assign each query contig to a single chromosome by choosing the chromosome with the highest coverage. Contigs with insufficient coverage to any chromosome (<90% for contigs with lengths in [1,10 kb); <80% for [10 kb,100 kb), <75% for [100 kb,1 Mb) and <70% for ≥mb.) were considered ambiguous and not included in the graph. In the GRCh38-based graph, all unplaced and random contigs were grouped together into the same component.

(4) Cactus base alignment: Cactus is a tool that uses a graph-based approach to combine sets of pairwise alignments obtained from lastz into a multiple genome alignment[37]. When aligning different species, it uses a phylogenetic tree to progressively decompose the alignment into a subproblem for each ancestral node in the tree. We adapted it to also accept chromosome-scale sequence-to-Minigraph mappings as produced above, and improved its runtime on alignments of many sequences by replacing its base aligner with abPOA[91]. The core algorithm described in ref. 37 remains unchanged, whereby the pairwise alignments were used to induce a sequence graph, then filtered using the Cactus alignment filtering algorithm, and components of unaligned sequence were then processed by the base alignment and refinement algorithm. The resulting graph was used to infer an ancestral sequence (not explicitly used in this work) and then exported to a hierarchical alignment (HAL) file[92]. We implemented a converter, hal2vg[93] that converts the HAL alignment into a sequence graph in VG format.

(5) Post-processing and whole-genome indexing: the following post-processing steps were performed on each chromosome graph. First, unaligned sequences >10 kb in length, including sequences not aligned to Minigraph, were removed to filter out any under-alignment artefacts that might later be mistaken for insertions. Next, GFAffix[94] was used to normalize the graphs by merging together redundant node prefixes and suffixes. Nodes were flipped as necessary to ensure that reference paths always visit their forward orientations. The chromosomes were combined into a whole-genome graph, indexed and exported to VCF, all using vg. Patched versions of both the GRCh38-based and CHM13-based graphs were created when it was discovered that short contigs split-mapping to distant locations had induced large deletions. The deletions were removed using vg clip -D 10000000 (and the pipeline has since been corrected to no longer produce them). Allele-filtered graphs, used for short-read mapping, were produced (from the patched graphs) by removing all nodes traversed by fewer than 9 haplotype paths (minimum AF = 10%) using vg clip -d 9 -m 10000. The chromosome HAL files were also combined into a whole-genome HAL file using halMergeChroms, and clipped sequences added back (to facilitate running CAT) using halUnclip.

**PGGB.** The PGGB uses a symmetric, all-to-all comparison of genomes to generate and refine a pangenome. We applied it to build a pangenome graph from all genome assemblies and references (both GRCh38 and CHM13). The resulting PGGB graph represents all alignment relationships between input genomes in a single graph. The PGGB graph is a lossless model of the input assemblies that represents all equivalently. This arrangement enables all of our pangenome assemblies to be used as reference systems, a property that we used to explore the scope of pangenome variation in a total way. Owing to ambiguous placement of variation in all-to-all pairwise alignments, many SV hotspots, including the centromeres, are transitively collapsed into loops through a subgraph representing a single repeat copy, a feature that tends to reduce the size of variants found in repetitive sequences. In contrast to MC, PGGB does not filter rapidly evolving satellite sequences or the regions that do not reliably align. This increases its size and complexity relative to the MC graph and adds a significant amount of singleton sequences relative to the Minigraph and MC graphs. However, this property enables annotations and coordinates of all assemblies in the pangenome to be related to the graph structure and utilized in subsequent downstream analyses. We applied the PGGB model to investigate the full pangenome and integrate annotations established de novo on the diverse assemblies into a single model for analyses of pangenome diversity and of complex structurally variable loci (MHC and 8p inversion).

PGGB generates a pangenome graph in three phases. (1) Alignment: in the first phase, the wfmash aligner[95] is used to generate all-vs-all

alignments of input sequences. This method, wfmash, applies the mapping algorithm of MashMap2 to find homologies at a specified length and per cent identity. It then derives base-level alignments using a high-order version of the WFA algorithm (wflign), which first aligns sequences in segments of 256 bp, then patches up the base-level alignment with local application of WFA. wfmash was designed and developed specifically for the problem of building all-to-all alignments for large pangenomes. (2) Graph induction: the input FASTA sequences and PAF-format alignments produced by wfmash are converted to a graph (in GFA format) using seqwish[96]. This losslessly transforms the input alignments and sequences into a graph. (3) Graph normalization: we applied a normalization algorithm—smoothxg[97]—to simplify complex motifs that occur in STRs and other repetitive sequences, as well as to mitigate underalignment. The graph is first sorted using a path-guided stochastic gradient descent method[98] that organizes the graph in one-dimension to optimize path distances and graph distances. This sort provides a way to partition the graph into smaller pieces over which we applied a multiple sequence alignment algorithm (abPOA)[91]. These pieces were laced back into a final graph. We iterated this process twice using different target POA lengths to remove boundary effects caused at the borders of the MSA problems. Finally, we applied GFAffix[94] to remove redundant furcations from the topology of the graph.

To build the HPRC PGGB graph, we used both the CHM13 and GRCh38 references as a target and mapped all contigs against these with wfmash, requiring a full length mapping at 90% total identity, collecting all contigs that mapped to a given chromosome. Contigs that did not map under this arrangement were then partitioned using a split-mapping approach, requiring 90% identity over 50 kb to seed the mappings, and putting the contig into the chromosome bin for which it had the best split mapping. We thus initially partition the data into 25 chromosome sets: one for each autosome, one for each sex chromosome, and finally the mitochondria.

We then applied PGGB (v.0.2.0+531f85f) to each partition to build a chromosome-specific graph. Run in parallel over 6 PowerEdge R6515 AMD EPYC 7402P 24-core nodes with 384 GB of RAM, this process requires 22.49 system days, or around 3.7 days wallclock. To develop a robust process to build the HPRC graph, the PGGB team iterated the build 88 times. The final chromosome graphs were compacted into a single ID space using vg ids -j, then for each reference (GRCh38 and CHM13) a combined VCF file was generated from the graph with vg deconstruct (v.1.36.0/commit 375cad7).

A handful of key parameters defined the shape of the resulting graph. First, in wfmash, we required >100 kb mappings at 98% identity. We mapped each HPRC assembly contig and reference chromosome (both GRCh38 and CHM13) to all the other 89 input haplotypes. To reduce complexity, and false-positive SNPs resulting from misaligned regions, we applied a minimum match length filter (in seqwish) of 311 bp. This meant that the graph that we induced was relatively 'underaligned' locally, and only through normalization in smoothxg did we compress the bubble structures that are produced. For smoothxg, our first iteration attempts to generate 13,033 bp-long POA problems, whereas the second is 13,177 bp. These lengths provided a balanced trade-off between run time and variant detection accuracy.

In addition to a graph (in GFA), PGGB generates visualizations of the graph in one and two dimensions, which show both the topology (two dimensional) and path-to-graph relationship (one dimensional). A code-level description of the build process is provided at GitHub (https://github.com/pangenome/HPRCyear1v2genbank).

### Pangenome graph assessment
**Annotating variant sites in pangenome graphs.** Variant sites in Minigraph and in MC and PGGB graphs were discovered using gfatools bubble (v.0.5)[75] and vg deconstruct[99], respectively. Large (>10 Mb) spurious deletions in MC and PGGB graphs were removed using vcfbub

(v.0.1.0)[100] with options -l 0 -r 10000000. Next, variant sites were classified into small variant (<50 bp) and SV (≥50 bp) sites. The SV sites were then annotated as described in the methods section of article that describes Minigraph[34]. In brief, the longest allele sequence of each SV site was extracted and stored in the FASTA format. The interspersed repeats, low-complexity regions, exact tandem repeats, centromeric satellites and gaps in the longest allele sequences were then identified using RepeatMasker (v.4.1.2-p1) with the NCBI/RMBLAST (v.2.10.0) search engine and Dfam (v.3.3) database, SDUST (v.0.1)[101], ETRF (commit fc059d5)[102], dna-brnn (v.0.1)[90] and seqtk gap (v.1.3)[103], respectively. SDs were identified if the total node length in a site was ≥1,000 bp and ≥20% of bases of these nodes were annotated as SD in the reference or in individual assembly ('SD annotation' subsection). To find hits to the GRCh38 reference genome, minimap2 (v.2.24) with options -cxasm20 -r2k --cs was used to align the longest allele sequences to the reference genome. Based on the identified features, SV sites were classified into various repeat classes using mgutils.js anno (https://github.com/lh3/minigraph/blob/master/misc/mgutils.js) with minor modifications to enable it to work with the files derived from the MC and PGGB graphs.

**Pangenome size and growth.** We use the heaps tool of the odgi pangenome analysis toolkit[98] to estimate how the euchromatic autosomal pangenome grows with each additional genome assembly added. Here we approximated euchromatic regions by non-satellite DNA, which was identified by dna-brnn in the construction of the MC graph (see the 'MC' subsection). Although the MC non-reference haplotypes of the MC graph do not contain satellite DNA, the PGGB graph does. Consequently, we subset the PGGB graph to segments contained in the MC graph. We additionally excluded reference haplotypes (GRCh38 and CHM13) from the analysis. We then sampled permutations of the 88 non-reference (neither GRCh38 nor T2T-CHM13) haplotypes. In each permutation, we calculated the size of the pangenome after adding the first 1, 2, …, N haplotypes in both graphs. This produced a collection of saturation curves from which we derived a median saturation curve onto which we fitted a power law function known as Heaps' Law. The exponent of this function is generally understood to represent the degree of openness—or diversity—of a pangenome[39]. Summing up, we called odgi heaps -i <graph.gfa> -S -n200 to generate pangenome saturation curves for 200 permutations. Next to calculating a non-permuted cumulative base count, we also counted the number of common (≥5% of all non-reference haplotypes) and core (≥95% of all non-reference haplotypes) bases in the pangenome graphs. To this end, we used a tool called panacus[104] and supplied a list of the samples in which they are grouped according to their assigned superpopulation (pangenome-growth -m -t bp <graph.gfa> <sample order>). We repeated the count, this time including only segments of depth ≥2, that is, contained at least twice in any haplotype sequence.

**Decomposing pangenome graphs based on allele traversals.** Pangenome graphs were decomposed topologically into a set of nested subgraphs, termed snarls, that each correspond to one or a collection of genetic variants. These snarls were then converted to VCF format using vg deconstruct[99]. Large (>100 kb) deletions in MC and PGGB graphs were removed using vcfbub (v.0.1.0)[100] with options -r 100000. To ease the comparison of variants with other call sets for each individual, the multi-sample VCF files were converted to per-sample VCF files using bcftools view -a -I -s <sample name>, and the multiallelic sites were split into biallelic records using bcftools norm -m -any. Owing to the limitations of snarl decomposition, snarls may contain multiple variants that cannot be further decomposed into nested snarls using vg deconstruct. If snarls of this kind are compared with truth calls, the evaluation will not be accurate. We solved this problem by comparing reference and alternate allele traversals for each snarl to infer the minimalist representation of variants (Supplementary Fig. 44).

**Annotating small variants in pangenome graphs with AFs from gnomAD.** Variant sites in MC and PGGB graphs were discovered using vg deconstruct[99]. The resulting VCF files were then decomposed on the basis of allele traversals ('Decomposing pangenome graphs based on allele traversals' subsection). The multi-nucleotide polymorphisms and complex indels were further decomposed into SNPs and simple indels using vcfdecompose --break-mnps --break-indels from RTG tools (v.3.12.1)[83], so that they could be annotated with gnomAD later. For comparison, variants called from PacBio HiFi reads using DeepVariant and from haplotype-resolved assemblies using Dipcall were also used. For each discovery method, small variants (<50 bp) were extracted and normalized using bcftools norm -c s -f <reference sequence in FASTA format> -m -any. Next, all per-sample VCF files were combined into one VCF file using bcftools concat -a -D after dropping individual genotype information using bcftools view -G. To annotate small variants with AFs from gnomAD[105], the gnomAD (v.3.1.2) per-chromosome VCF files were downloaded and concatenated into one VCF file using bcftools concat. The VCF file was then compressed into a file in the gnotate format using make-gnotate from slivar (v.0.2.7)[106] with options --field AC:gnomad_ac --field AN:gnomad_an --field AF:gnomad_af --field nhomalt:gnomad_nhomalt. The small variants were annotated with gnomAD using slivar expr --gnotate <gnotate file>.

## Variant benchmarking

**Calling variants from PacBio HiFi reads.** The PacBio HiFi reads were aligned to the GRCh38 human reference genome with no alternatives using Winnowmap2 (v.2.03)[107] with -x map-pb -a -Y -L --eqx --cs. The MD tags required by Sniffles were calculated using samtools calmd. The resulting BAM files were sorted and indexed using SAMtools.

For small variants, the two-pass mode of DeepVariant (v.1.1.0)[107] with WhatsHap (v.1.1)[108] was used to call SNPs and indels from the PacBio HiFi read alignments. The resulting VCF files were used as truth sets for small variant benchmarking.

Three discovery methods were used to call SVs from the PacBio HiFi read alignments. For PBSV (v.2.6.2)[41], SV signatures were identified using pbsv discover with --tandem-repeats <GRCh38 TRF BED file> to improve the calling performance in repetitive regions. SVs were then detected using pbsv call with --ccs --preserve-non-acgt -t DEL,INS,INV,DUP,BND -m 40 from the signatures. For SVIM (v.2.0.0)[44], SVs were called using svim alignment with --read_names --zmws --interspersed_duplications_as_insertions --cluster_max_distance 0.5 --minimum_depth 4 --min_sv_size 40. In contrast to PBSV and Sniffles, SVIM outputs all calls no matter their quality. To determine the threshold used for filtering low-quality calls, a precision–recall curve was generated across various quality scores by comparing with the GIAB (v.0.6) Tier 1 SV benchmark set for HG002 (Supplementary Fig. 45). Consequently, SVIM calls with a quality score lower than ten were excluded. For Sniffles (v.1.0.12b)[42], SVs were discovered with -s 4 -l 40 -n -1 --cluster --ccs_reads. Unlike PBSV and SVIM, Sniffles does not generate consensus sequences of insertions from aggregating multiple supporting reads. Therefore, Iris (v.1.0.4)[43] was used to refine the breakpoints and insertion sequences with --hifi --also_deletions --rerunracon --keep_long_variants. All resulting VCF files were sorted and indexed using BCFtools.

**Calling SVs from haplotype-resolved assemblies.** Three discovery methods were used to call SVs from the haplotype-resolved assemblies generated using Trio-Hifiasm.

For SVIM-asm (v.1.0.2)[45], assemblies were aligned to the GRCh38 human reference genome with no alternatives using minimap2 (v.2.21)[86] with -x asm5 -a --eqx --cs and then sorted and indexed using SAMtools. SVs were called using svim-asm diploid with --query_names --interspersed_duplications_as_insertions --min_sv_size 40. The resulting VCF files were sorted and indexed using BCFtools.

For PAV (v.0.9.1)[5], assemblies were aligned to the GRCh38 human reference genome with no alternatives using minimap2 (v.2.21)[86] with options -x asm20 -m 10000 -z 10000,50 -r 50000 --end-bonus=100 --secondary=no -a --eqx -Y -O 5,56 -E 4,1 -B 5. These alignments are then trimmed to reduce the redundancy of records and to increase the contiguity of alignments. SVs, indels and SNPs were called by using cigar string parsing of the trimmed alignments. Inversion calling in PAV uses a new *k*-mer density assessment to resolve inner and outer breakpoints of flanking repeats, which does not rely on alignment breaks to identify inversion sites. This is designed to overcome limitations in alignment methodologies and to expand inversion calls, which result in duplications and deletions of sequence on the boundaries.

The Hall-lab pipeline is as documented in the WDL workflow (https://github.com/hall-lab/competitive-alignment/blob/master/call_assembly_variants.wdl) (commit 830260a). In brief, the maternal and paternal assemblies were aligned to the GRCh38 human reference genome using minimap2 (v.2.1)[86] with options -ax asm5 -L --cs. Large indels (>50 bp) were detected using the 'call_small_variants' task, based on paftools (v.2.17-r949-dirty). For large SV, breakpoints were mapped based on split alignments of assembly contigs to the reference genome and classified as SVs using a series of custom python scripts in the 'call_sv' task. The breakpoint-mapped SVs were then filtered on the basis of the coverage of the reference genome by the assembly contigs (calculated using bedtools genomecov (v.2.28.0)). For each haplotype assembly, a BED file of 'excluded regions' was defined comprising genomic regions covered by more than one distinct contig or with more than 3× coverage by a single contig. Breakpoint-mapped SVs where either breakpoint or >50% of the outer span intersected an excluded region were filtered.

**Merging SV call sets.** To integrate per-sample VCF files generated by three HiFi-based and three assembly-based SV callers, svtools[109] was used. For each individual, VCF files from the six callers were jointly sorted and then merged using svtools lsort and lmerge, first using a strict criterion (svtools lmerge -f 20), followed by a more lenient second merge (svtools lmerge -f 100 -w carrier_wt). The autosomal SV calls supported by at least two callers were included in the consensus SV call set for comparison.

**Defining confident regions for variant benchmarking.** For SVs, confident regions were generated using Dipcall. Although useful for small variants, current benchmarking tools such as hap.py/vcfeval cannot properly compare different representations of small variants in and around SVs. Therefore, for each sample, the confident regions from Dipcall were further processed as follows:

(1) Exclude any SD, self-chain, tandem repeat longer than 10 kb or satellite DNA if there are any breaks in the Dipcall BED file in the repeat region +15 kb flanking sequence on each side. The rationale is that breaks in the Dipcall BED file are generally caused by missing sequence or errors in the assembly or reference or by large SVs or CNVs for which we do not have tools to benchmark small variants in these regions.

(2) Exclude 15 kb around all breaks in the Dipcall BED file for the same reason as noted above.

(3) Exclude 15 kb around all gaps in GRCh38 because alignments are unreliable.

(4) Exclude variants >49 bp in the Dipcall VCF file and any tandem repeats overlapping these SVs +50 bp on each side.

**Benchmarking variants.** Variant sites in MC and PGGB graphs were discovered using vg deconstruct[99]. Variant sites with alleles larger than 100 kb in MC and PGGB graphs were then removed using vcfbub (v.0.1.0)[100] with options -l 0 -a 100000. The resulting VCF files were further processed using vcfwave from vcflib[110] with option -l 1000. In brief, vcfwave realigned alternate alleles against the reference allele for

each variant site using the bidirectional wavefront alignment (BiWFA) algorithm[111] to decompose complex alleles into primitive ones. The multi-sample VCF files were then converted to per-sample VCF files using bcftools view -a -I -s <sample name> and the multiallelic sites were splitted into biallelic records using bcftools norm -m -any. Next the autosomal small variants (<50 bp) from a given pangenome graph (query set) were compared with the HiFi-DeepVariant call set (truth set) using vcfeval from RTG tools (v.3.12.1)[83] with options -m annotate --all-records --ref-overlap --no-roc. Note that the multi-nucleotide polymorphisms and complex indels were reduced to SNPs and simple indels using vcfdecompose --break-mnps --break-indels from RTG tools (v.3.12.1)[83] The comparison was performed independently for each individual. Recall and precision were calculated within the refined Dipcall confident regions ('Defining confident regions for variant benchmarking' subsection) and then stratified using the GIAB (v.3.0) genomic context. To evaluate the SV (≥50 bp) calling performance, the autosomal SVs from a given pangenome graph (query set) were compared to the consensus SV call set (truth set) for each individual using truvari bench (v.3.2.0)[112] with options --multimatch -r 1000 -C 1000 -O 0.0 -p 0.0 -P 0.3 -s 50 -S 15 --sizemax 100000 --includebed <Dipcall confident regions>. Recall and precision were then stratified using the GIAB (v.3.0) genomic context and by variant length.

### Alignment of long reads to pangenomes

**PacBio HiFi reads.** PacBio HiFi reads from 44 HPRC samples (excluding the held out samples) were aligned to the MC graph using GraphAligner (v.1.0.13)[113] with option -x vg and stored in the GAF format[34]. For each read that aligned to multiple places in the graph, the alignment with the highest score was retained. To remove low-quality alignments, a read with <80% of read length aligned to the graph was discarded. After filtering the read-to-graph alignments, the read depth of each edge was calculated using vg pack (v.1.33.0)[114] with options -Q -1 -D. Note that the resulting GAF files did not contain a mapping quality (encoded as 255 for missing) for each alignment, therefore the option -Q -1 was given to vg pack to ensure that these alignments were used during read-depth calculation. Next, the edges of each sample were classified into either on-target or off-target depending on whether they were on the sample paths (encoded as W-lines in MC GFA files) or not.

**Oxford Nanopore reads.** ONT reads obtained from 29 HPRC samples (samples labelled HPRC in Supplementary Table 1) were aligned against the MC graph. The alignments were produced using GraphAligner (v.1.0.13) with parameter settings -x vg --multimap-score-fraction 1 --multiseed-DP 1. The number of reads in these datasets range between 1 million and 5.4 million and have an average read length of 28.4 kb. On average, 99.68% of the reads received hits from one or more locations in the graph. For each read, we determined its best hit based on alignment score and discarded all its lower-scoring alignments in subsequent analyses. The alignment identities of these best hits peaked above 95%, with an average ratio of alignment-length-to-read-length (ALRL) of 0.880 (s.d. = 0.302) and average MAPQ value of 59.35. The alignment set was further quality-pruned by discarding alignments that either had an ALRL lower than 0.8 or a MAPQ value lower than 50. The surviving alignments had an overall average ALRL of 0.968 (s.d. = 0.047) and effectuate an overall genome coverage between 10.5-fold and 43-fold across the 29 samples (Supplementary Fig. 46).

### Annotating genes within pangenomes

We ran CAT[46] to annotate each of the genomes within a pangenome graph. CAT projects a reference annotation, in this case GENCODE (v.38), to each of the haplotypes using the underlying alignments within the graph. CAT (commit eb2fc87) was run on both of the GRCh38-based and the CHM13-based MC graphs. For each graph, the autosomes were first run all together, and then the sex chromosomes were run on the appropriate haplotypes. The parameters used were default parameters, except as shown below. An example CAT command run is:

```
luigi --module cat RunCat --hal=CHM13-f1g-90-mc-aug11.hal --ref-
genome=GRCh38 --workers=8 --config=cat-hprc.gencode38.auto-
somes.config --work-dir work-hprc-gencode38-chm13 --out-dir out-
hprc-gencode38-chm13 --local-scheduler --assembly-hub --maxCores
8 --binary-mode local > cat.hprc.gencode38.autosomes.chm13.log
```

Comparisons were made between the resulting CAT annotations and those from the Ensembl pipeline by looking at the parent GENCODE identifiers for each gene and transcript in the sets. Numbers of shared and unique identifiers between the sets were tabulated. Because the two annotation sets used slightly different versions of GENCODE (v.38), only those identifiers attempted to be mapped by both pipelines were considered. Additionally, features were considered to be at the same locus if their genomic intervals overlapped.

### Identifying medically relevant sites

SV sites in Minigraph were discovered using gfatools bubble. To obtain the number of observed alleles per site, per-sample alleles were called using minigraph -cxasm --call (Supplementary Table 21). SV sites with alleles larger than 10 kb and at least five observed alleles were selected as complex SV sites. The complex SV sites were further filtered on the basis of whether they overlapped with medically relevant protein-coding genes[47] using bedtools intersect. To understand whether the medically relevant complex SV sites are known in previous studies, the coverage of SVs from the 1KG call sets[10,19] was computed using bedtools coverage -F 0.1 (Supplementary Table 16). All complex SVs were examined using Bandage[115] and visually compared to previous short-read SV call sets[10,19,68,116] using IGV.

### Analysis of five complex loci

**Visualization of graph structures of five loci.** We extracted subgraphs and paths for five loci in the MC and PGGB graphs using gfabase (v.0.6.0)[117] and odgi (v.0.6.2)[98] with the following example commands:

```
gfabase sub GRCh38-f1g-90-mc-aug11.gfab GRCh38.
chr1:25240000-25460000 --range --connected --view --cutpoints
1 --guess-ranges -o RH_locus.walk.gfa
odgi extract -i chr1.pan.smooth.og -o chr1.pan.RH_locus.og -b
chr1.RH_locus.bed -E -P
```

We then visualized the graph structures of the subgraphs using Bandage (v.0.8.1)[115].

**Alignment of genes to graphs.** We aligned Ensembl (release 106)[77] GRCh38 version gene sequences to the MC graph and PGGB graph using GraphAligner (v.1.0.13)[113] with parameter settings -x vg --try-all-seeds --multimap-score-fraction 0.1 to identify the gene positions within the graphs. To show locations of genes on Bandage plots, we applied colour gradients from green to blue to the nodes of each gene. Lines alongside the Bandage plots showing approximate gene positions, exons and transcription start sites based on Ensembl Canonical transcripts were drawn by hand.

**Structural haplotypes identification.** Sequences of each assembly are represented by paths in a GFA file. We identified SVs in each assembly by tracing these paths through different 'big' bubbles (>5,000 bp) in either the MC graph or PGGB graph within those gene regions. We selected the 5,000 bp bubble size based on manual inspection of Bandage plots. An example command to identify big bubbles at a RH locus is as follows:

```
bcftools filter hprc-v1.0-mc-grch38.vcf.gz -r chr1:25240000-
25460000 | grep LV=0 | awk '{OFS="\t"; print $1,$2,$3,$4,$5}' | tr ","
"\t" | awk '{OFS="\t"; for (i=1;i<=NF;i++) {len=length($i); if (len>5000)
{print $1,$2,$3; next}}}'
```

To identify gene conversion events (as gene conversions are not shown as bubbles in the graphs), we identified nodes that were different between a gene and its homologous gene (for example, *RHD* and *RHCE*) based on the GraphAligner alignments described above. We refer to these as paralogous sequence variants. A gene conversion event was detected if a path of a gene goes through more than four paralogous sequence variants of its homologous gene in a row.

**Visualization of linear gene structures.** We counted the number of assemblies for each structural haplotype and computed their frequency. We visualized linear haplotype structures (for example, in Fig. 5c) using gggenes (v.0.4.1)[118] based on the structural haplotypes determined for each assembly from the pangenome graphs. The length of intervals between genes is fixed (except for *TMEM50A* and *RHCE*, because those two genes are immediately next to each other). Lengths of genes are shown as proportional to gene lengths in GRCh38.

## Pangenome point genotyping

**Alignment of reads to the pangenome.** The short reads were first split into chunks to parallelize the read mapping to the 'allele-filtered graph' pangenome, as defined in the 'MC' subsection. This pangenome is included within the dataset accompanying this paper and can be identified as 'clip.d9.m1000.D10M.m1000'. Mapping was performed with Giraffe[49] from vg release v.1.37.0. For trio-based runs, the trio-sample sets of short reads were mapped to the pangenome using Giraffe from vg release v.1.38.0. Note that the core vg algorithms for Giraffe mapping and surjection (conversion from graph space to linear space) are the same in both vg v.1.37.0 and v.1.38.0. The output alignments, surjected to GRCh38 in BAM format as explained below, are available at https://s3-us-west-2.amazonaws.com/human-pangenomics/index.html?prefix=publications/PANGENOME_2022/DeepTrio/samples in the bam directory of each sample's directory, and are organized by aligner.

**Surjection to GRCh38 and indel realignment.** To perform variant calling, GAM alignments were surjected onto the chromosomal paths from GRCh38 (chromosomes 1–22, X and Y) using vg surject and the --prune-low-cplx option to prune short and low-complexity anchors during realignment. The BAM files were sorted and split by chromosome using SAMtools (v.1.3.1)[119]. The reads were realigned, first using bamleftalign from FreeBayes (v.1.2.0)[120], and then with ABRA (v.2.23)[121] on target regions that were identified using RealignerTargetCreator from GATK (v.3.8.1)[122] and expanded by 160 nucleotides with bedtools slop (v.2.21.0)[123].

**Model training.** To perform variant calling with DeepVariant and Deep-Trio, we trained machine-learning models specific to our graph reference and Giraffe alignment pipeline based on our alignments. For all models, chromosome 20 was entirely held out from all input samples to provide a control.

Training was performed on Google's internal cluster, using unreleased Google tensor processing unit (TPU) accelerators, from a cold start (that is, without using a pre-trained model as input). We believe that nothing about the way in which we executed the training is essential to the results obtained. Cold start training is estimated to be feasible outside the Google environment; therefore the claims we present here are falsifiable, but it is not expected to be cost-effective. Researchers looking to independently replicate our training should consider doing warm start training from a base model trained on other data, using commercially available graphics processing unit (GPU) accelerators. An example procedure can be found in the DeepVariant training tutorial at GitHub (https://github.com/google/deepvariant/blob/r1.3/docs/deepvariant-training-case-study.md). We predict that this more accessible method would produce equivalent results.

For both DeepVariant and DeepTrio, the true variant calls being trained against came from the GIAB benchmark (v.4.2.1).

For DeepVariant, we trained on the HG002, HG004, HG005, HG006 and HG007 samples, with HG003 held out. The trained DeepVariant model is available at https://s3-us-west-2.amazonaws.com/human-pangenomics/index.html?prefix=publications/PANGENOME_2022/DeepVariant/models/DEEPVARIANT_MC_Y1.

For DeepTrio, we trained two sets of models: one on HG002, HG003, HG004, HG005, HG006 and HG007, with HG001 held out; and one on HG001, HG005, HG006 and HG007, with the HG002, HG003 and HG004 trio held out. Each DeepTrio model set included parental and child models. The two trained child deeptrio models are available at https://s3-us-west-2.amazonaws.com/human-pangenomics/index.html?prefix=publications/PANGENOME_2022/DeepTrio/models/deeptrio/child and https://s3-us-west-2.amazonaws.com/human-pangenomics/index.html?prefix=publications/PANGENOME_2022/DeepTrio/models/deeptrio-no-HG002-HG003-HG004/child, respectively. The two trained parental DeepTrio models are available at https://s3-us-west-2.amazonaws.com/human-pangenomics/index.html?prefix=publications/PANGENOME_2022/DeepTrio/models/deeptrio/parent and https://s3-us-west-2.amazonaws.com/human-pangenomics/index.html?prefix=publications/PANGENOME_2022/DeepTrio/models/deeptrio-no-HG002-HG003-HG004/parent, respectively.

**Variant calling with DeepVariant.** DeepVariant (v.1.3) was evaluated on HG003, using the model we trained with HG003 held out (see 'Model training'). We used the --keep_legacy_allele_counter_behavior flag (introduced to support this analysis) and a minimum mapping quality of 1 in the make_examples step, before calling the variants with call_variants. Both VCFs and gVCFs were produced. The WDL workflow used for single sample mapping and variant calling was deposited into Zenodo (https://doi.org/10.5281/zenodo.6655968).

**Variant calling on GRCh38 with BWA-MEM and DeepVariant.** Small variants were also called using a more traditional pipeline. We aligned reads with BWA-MEM[50] to GRCh38 with decoys but no ALTs. DeepVariant then called small variants from the aligned reads. The same version and parameters were used for DeepVariant. Only the model was changed to the default DeepVariant model.

**Variant calling with DeepTrio.** Small variants were also called using DeepTrio (v.1.3). For HG001, we used the DeepTrio models we trained with HG001 held out (see 'Model training'). For the HG002, HG003 and HG004 trio and HG005, HG006 and HG007 trio, we used the models trained with the HG002, HG003 and HG004 trio held out; the HG005, HG006 and HG007 trio (except for chromosome 20) was still included in the training set. We used the --keep_legacy_allele_counter_behavior and a minimum mapping quality of 1 in the make_examples step before calling the variants with call_variants. Both VCFs and gVCFs were produced and are available at https://s3-us-west-2.amazonaws.com/human-pangenomics/index.html?prefix=publications/PANGENOME_2022/DeepTrio/samples in the vcf directory of each sample's directory, and are organized by mapping and calling condition. The WDL workflow used for trio-based mapping and variant calling was deposited into Zenodo (https://doi.org/10.5281/zenodo.6655962).

**Variant calling on GRCh38 with BWA-MEM, Dragen Graph and DeepTrio.** For DeepTrio, small variants were also called using a more traditional pipeline and a graph-based implementation of Illumina's Dragen platform (v.3.7.5). The conditions evaluated were each a combination of a mapper and a reference. The Giraffe-HPRC condition used Giraffe (v.1.38.0)[49] to align reads to the HPRC reference. The BWA-MEM condition used BWA-MEM (v.0.7.17-r1188)[50] to align reads to the hs38d1 human reference genome with decoys but no ALTs. The Dragen-DeepTrioCall condition used Illumina's Dragen platform (v.3.7.5)[51] against their default graph, which was constructed using

the same GRCh38 reference with decoys but no ALTs, and population contigs, SNPs and liftover sequences from datasets internal to their platform. DeepTrio then called small variants from the aligned reads. The same version and parameters were used for DeepTrio (v.1.3). Only the default model was used for these conditions. We also applied the native Dragen caller and joint genotyper to the Dragen-Graph-based alignments for comparison purposes, referred to as Dragen-DragenCall and Dragen-DragenJointCall, respectively. Dragen-DragenCall implements a single-sample based method and is what is the default use-case for processing Dragen-Graph-mapped data. Dragen-DragenJointCall uses a pedigree-backed implementation that informs which variants are likely to be de novo and which are erroneous given the genotype information of the parents. To make a fairer comparison with Dragen, we tested these configurations to assess what implementation of Dragen variant calling produced the best results given the available trio data.

**Evaluation using the GIAB benchmark.** The small variant calls were evaluated using HG001–HG007 with the GIAB benchmark (v.4.2.1)[124], on HG002 in challenging medically relevant autosomal genes[47], and on HG002 using a preliminary draft assembly-based benchmark. For the draft assembly-based benchmark, we used Dipcall[38] to align a scaffolded, high-coverage Trio-Hifiasm assembly[21,38] to GRCh38 and call variants, and then we excluded structurally variant regions from the dip.bed file as described above for the benchmarking of small variants from the pangenome graph. The comparison between the call sets and truth set was made with RTG's vcfeval[83] and Illumina's hap.py tool[82] on confident regions of the benchmark. We used high-coverage read sets of the GIAB HG001, HG002 and HG005 trio child samples and evaluated performance within the held-out chromosome 20 for the GIAB (v.4.2.1) truth set, or the entire genome for the reduced truth set of the challenging medically relevant autosomal genes. The evaluation was also stratified using the set of regions provided by the GIAB at GitHub (https://github.com/genome-in-a-bottle/genome-stratifications)[125].

**Variant calls across samples from the 1KG.** We applied our small variant calling pipeline to the high-coverage read sets for the 3,202 samples of the 1KG[19]. The output alignments, in the GAM format, and the VCFs were saved in public buckets at https://console.cloud.google.com/storage/browser/brain-genomics-public/research/cohort/1KGP/vg/graph_to_grch38. We selected 100 trios among those samples to further evaluate the quality of the calls. We tested all variants that have at least one alternative allele in a trio for Mendelian consistency. In addition, for each variant, we only considered trios for which the child's genotype was different from the genotype of at least one of the parents to minimize bias created by systematic calls (for example, all homozygous or all heterozygous). We looked at the fraction of variants–trios that failed Mendelian consistency in the entire genome and in sites that do not overlap simple repeats as defined by the 'simpleRepeat' track downloaded from the UCSC Genome Browser. The results were compared with Mendelian consistency of calls provided by the 1KG that used GATK HaplotypeCaller on the reads aligned to GRCh38. We also repeated this analysis on the two trios of the GIAB (v.4.2.1) benchmark (HG002–HG007) and across the different methods of our evaluation described above (BWA-MEM and DragenGraph mappers; DeepVariant, DeepTrio and Dragen variant callers).

## SV genotyping with PanGenie
**VCF preprocessing.** We used a VCF file created on the basis of snarl traversal of the MC graph as a basis for genotyping. The records contained in this VCF represent bubbles in the underlying pangenome graph and their nested variants, derived from the snarl tree. Each variant was marked according to their level in this tree. Variants annotated by 'LV=0' correspond to the top-level bubbles. We used vcfbub (v.0.1.0)[100] with parameters -l 0 and -r 100000 to filter the VCF. This removed all non-top-level bubbles from the VCF unless they were nested inside

a top-level bubble with a reference length exceeding 100 kb; that is, top-level bubbles longer than that are replaced by their child nodes in the snarl tree. The VCF also contained the haplotypes for all 44 assembly samples, representing paths in the pangenome graph. We additionally removed all records for which more than 20% of all 88 haplotypes carried a missing allele ("."). This resulted in a set of 22,133,782 bubbles. In a next step, we used PanGenie (v.1.0.0)[54] to genotype these bubbles across all 3,202 samples from the 1KG based on high-coverage Illumina reads[19].

**Decomposition of variants.** We genotyped all top-level bubbles across all 1KG samples. Whereas biallelic bubbles can be easily classified representing SNPs, indels or SVs, this becomes more difficult for multiallelic bubbles contained in the VCF. In particular, larger multiallelic bubbles can contain a high number of nested variant alleles overlapping across haplotypes, represented as a single bubble in the graph. This is especially problematic when comparing the genotypes computed for the entire bubble to external call sets, as coordinates of the bubble do not necessarily represent the exact coordinates of individual variant alleles carried by a sample in such a region (Supplementary Fig. 20).

To tackle this problem, we implemented a decomposition approach that aimed to detect all variant alleles nested inside multiallelic top-level records. The idea was to detect variants from the node traversals of the reference and alternative alleles of all top-level bubbles. Given the node traversals of a reference and alternative path through a bubble, our approach was to match each reference node to its leftmost occurrence in the alternative traversal, resulting in an alignment of the node traversals (Supplementary Fig. 21a). Nested alleles could then be determined based on indels and mismatches in this alignment. As the node traversals of the alternative alleles can visit the same node more than once (which is not the case for the reference alleles of the MC graph), this approach is not guaranteed to reconstruct the optimal sequence alignment underlying the nodes in these repeated regions.

As an output, the decomposition process generated two VCF files. The first one is a multiallelic VCF that contains exactly the same variant records as the input VCF, just that annotations for all alternative alleles of a record were added to the identifier tag in the INFO field. For each alternative allele, the identifier tag contains identifiers encoding all nested variants it is composed of, separated by a colon. The second VCF is biallelic and contains a separate record for each nested variant identifier defining reference and alternative allele of the respective variant (Supplementary Fig. 21b). Both VCFs are different representations of the same genomic variation, that is, before and after decomposition. We applied this decomposition method to the MC-based VCF file, used the multiallelic output VCF as input for PanGenie to genotype bubbles, and used the biallelic VCF as well as the identifiers to translate PanGenie's genotypes for bubbles to genotypes for all individual nested variant alleles. All downstream analyses of the genotypes are based on this biallelic representation (that is, after decomposition).

Although the majority of short bubbles (<10 bp) are biallelic, particularly large bubbles (>1,000 bp) tend to be multiallelic. Sometimes each of the 88 non-reference (neither GRCh38 nor T2T-CHM13) haplotypes contained in the graph covered a different path through such a bubble (Supplementary Fig. 22a), leading to a VCF record with 88 alternative alleles listed. We determined the number of variant alleles located inside biallelic and multiallelic bubbles in the pangenome after decomposition. As expected, the majority of SV alleles was located inside of the more complex, multiallelic regions of the pangenome (Supplementary Fig. 22b).

**Genotyping evaluation based on assembly samples.** We conducted a leave-one-out experiment to evaluate PanGenie's genotyping performance for the call set samples. For this purpose, we repeatedly removed one of the panel samples from the MC VCF and genotyped it using only the remaining samples as an input panel for PanGenie. We later used the genotypes of the left-out sample as ground truth for evaluation.

We repeated this experiment for five of the call set samples (HG00438, HG00733, HG02717, NA20129 and HG03453) using 1KG high-coverage Illumina reads[19]. PanGenie is a re-genotyping method. Therefore, like every other re-typer, it can only genotype variants contained in the input panel VCF, that is, it is not able to detect variants unique to the genotyped sample. For this reason, we removed all variant alleles (after decomposition) unique to the left-out sample contained in the truth set for evaluation. To evaluate the genotype performance, we used the weighted genotype concordance[54]. Extended Data Fig. 9 shows the results stratified by different regions. Extended Data Fig. 9a shows concordances in biallelic and multiallelic regions of the MC VCF. The biallelic regions include only bubbles with two branches. The multiallelic regions include all bubbles in which haplotypes cover more than two different paths. Extended Data Fig. 9b shows the same results stratified by genomic regions defined by GIAB that we obtained from the following genotypes: easy (https://ftp-trace.ncbi.nlm.nih.gov/ReferenceSamples/giab/release/genome-stratifications/v3.0/GRCh38/union/GRCh38_notinalldifficultregions.bed.gz); low-mappability (https://ftp-trace.ncbi.nlm.nih.gov/ReferenceSamples/giab/release/genome-stratifications/v3.0/GRCh38/union/GRCh38_alllowmapandsegdupregions.bed.gz); repeats (https://ftp-trace.ncbi.nlm.nih.gov/ReferenceSamples/giab/release/genome-stratifications/v3.0/GRCh38/LowComplexity/GRCh38_AllTandemRepeats_gt100bp_slop5.bed.gz); and other-difficult (https://ftp-trace.ncbi.nlm.nih.gov/ReferenceSamples/giab/release/genome-stratifications/v3.0/GRCh38/OtherDifficult/GRCh38_allOtherDifficultregions.bed.gz).

Here and in the following, we considered results for SNPs, indels (1–49 bp), SV deletions, SV insertions and other SV alleles, defined as follows: SV deletions include all alleles for which length(REF) ≥50 bp and length(ALT) = 1 bp; SV insertions include all alleles for which length(REF) = 1 bp and length(ALT) ≥ 50 bp; all other alleles with a length ≥50 bp are included in 'others'.

Overall, weighted genotype concordances were high for all variant types. In particular, variant alleles in biallelic regions of the graph were easily genotypable. Alleles inside multiallelic bubbles were more difficult to genotype correctly as PanGenie needs to decide between several possible alternative paths, whereas there are only two such paths for biallelic regions (Extended Data Fig. 9a). Furthermore, genotyping accuracy depended on the genomic context (Extended Data Fig. 9b). Regions with low mappability, repetitive regions and other difficult regions were harder to genotype than regions classified as 'easy' by GIAB.

**Creating a high-quality subset.** We generated genotypes for all 3,202 1KG samples with PanGenie and defined a high-quality subset of SV alleles that we could reliably genotype. For this purpose, we applied a machine-learning approach similar to what we have previously presented[5,54]. We defined positive and negative subsets of variants based on the following filters: ac0_fail, a variant allele was genotyped with an AF of 0.0 across all samples; mendel_fail, the mendelian consistency across trios is <80% for a variant allele. Here, we use a strict definition of Mendelian consistency, which excludes all trios with only 0/0, only 0/1 and only 1/1 genotypes; gq_fail, <50 high-quality genotypes were reported for this variant allele; self_fail, genotyping accuracy of a variant allele across the panel samples is <90%; nonref_fail, not a single non-0/0 genotype was genotyped correctly across all panel samples.

The positive set included all variant alleles that passed all five filters. The negative set contained all variant alleles that passed the ac0_fail filter but failed at least three of the other filters. We trained a support vector regression approach based on these two sets that used multiple features, including AFs, Mendelian consistencies or the number of alternative alleles transmitted from parents to children. We applied this method to all remaining variant alleles genotyped with an AF > 0, resulting in a score between −1 (bad) and 1 (good) for each. We finally defined a filtered set of variants that included the positive set and all variant alleles with a score of ≥ −0.5.

We show the number of variant alleles contained in the unfiltered set, the positive set and the filtered set in Supplementary Table 18. As our focus is on SVs and as 65% of all SNPs and indels are already contained in the positive set, we applied our machine-learning approach only to SVs. We found that 50%, 33% and 26% of all deletion, insertion and other alleles, respectively, were contained in the final, filtered set of variants. Note that these numbers take all distinct SV alleles contained in the call sets into account. Especially for insertions and other SVs, many of these alleles are highly similar, with sometimes only a single base pair differing. Therefore, it is probable that many of these actually represent the same events. Our genotyping and filtering approach helped to remove such redundant alleles.

To evaluate the quality of the PanGenie genotypes, we compared the AFs observed for the SV alleles across all 2,504 unrelated 1KG samples to their AFs observed across the 44 assembly samples in the MC call set. Supplementary Figs. 23–25 show the results for SV deletions, insertions and other SV alleles. We observed that the AFs between both sets matched well, resulting in Pearson correlations of 0.93, 0.87 and 0.81 for deletions, insertions and other alleles, respectively, contained in the unfiltered set. For the filtered set, we observed correlations of 0.96, 0.93 and 0.90, respectively. We also analysed the heterozygosity of the PanGenie genotypes across all 2,504 unrelated 1KG samples and observed a relationship close to what is expected by Hardy–Weinberg equilibrium (Supplementary Figs. 23–25, lower panels).

**Number of SVs per sample.** We compared our filtered set of variant alleles to the HGSVC PanGenie genotypes (v.2.0 lenient set)[5] and Illumina-based SV genotypes[19]. A direct comparison of the three call sets is difficult. The HGSVC and HPRC call sets are based on variant calls produced from haplotype-resolved assemblies of 32 and 44 samples, respectively[5]. For each call set, variants were re-genotyped across all 3,202 1KG samples. Note that the call set samples for HPRC and HGSVC are disjoint. As re-genotyping cannot discover new variants, both call sets will miss variants carried by 3,202 samples that were not seen in the assembly samples. By contrast, the 1KG call set contains short-read based variant calls produced for each of the 3,202 1KG samples. Another difference between the HGSVC and HPRC call sets is that in the HGSVC call set, highly similar alleles are merged into a single record to correct for representation differences across different samples or haplotypes. The HPRC call set, however, keeps all these alleles separately even if there is only a single base pair difference between them. To make the call sets better comparable, we merged clusters of highly similar alleles in the HPRC filtered set before comparisons with other call sets. This was done with Truvari (v.3.1.0)[112] using the command: truvari collapse -r 500 -p 0.95 -P 0.95 -s 50 -S 100000.

To be able to properly compare the call sets despite their differences, we counted the number of SV alleles present in each sample (genotype 0/1 or 1/1) in each call set and plotted the corresponding distributions stratified by genome annotations from GIAB (same as above, Fig. 6d). We also generated the same plot including only common SV alleles with an AF > 5% across all 3,202 samples (Supplementary Fig. 28). Both plots showed that both assembly based call sets (HPRC and HGSVC) were able to access more SVs across the genome than the short-read-based 1KG call set, especially deletions <300 bp and insertions (Fig. 6e). This result confirms that SV callers based on short reads alone miss a large proportion of SVs located in regions inaccessible by short-read alignments, which has been previously reposed by several studies[5,8]. In the 'easy' regions, the number of SVs per sample was consistent across all three call sets. For the other regions, however, results indicated that the HPRC-filtered genotypes provided access to more variant alleles than the HGSVC lenient set, especially insertions and variants in regions of low mappability and tandem repeats (Fig. 6d,e).

To evaluate the new SVs in our filtered HPRC call set, we revisited the leave-one-out experiment we had previously performed on the unfiltered set of variants (see above). We restricted the evaluation to the following subset of variants: (1) those that are in our filtered set but not in the 1KG Illumina calls (novel); (2) those in our filtered set and in the 1KG Illumina call set (known); and (3) all variants in our filtered set. To find matches between our set and the Illumina calls, we used a criterion based on reciprocal overlap of at least 50%. Results are shown in Supplementary Fig. 29. We generated two versions of this figure: the first one (top) excludes variants that are unique to the left-out sample and therefore not typable by any re-genotyping method, and the second one includes these variants (bottom). In general, genotype concordances of all lenient variants (brown, dark purple) were slightly higher compared with the concordances we observed for the unfiltered set (Supplementary Fig. 29). Furthermore, concordances of the known variants were highest. This is expected, as these variants tended to be in regions easier to access by short reads. Concordances for novel variants were slightly worse. This was also expected, as these variants tended to be located in more complex genomic regions that are generally harder to access. However, even for these variants, concordances were still high, which indicated that the PanGenie genotypes for these variants are of high quality.

**Evaluation based on medically relevant SVs.** In addition to all 3,202 1KG samples, we genotyped sample HG002 based on Illumina reads from ref. 18. We used the GIAB CMRG benchmark containing medically relevant SVs[47] downloaded from https://ftp-trace.ncbi.nlm.nih.gov/ReferenceSamples/giab/release/AshkenazimTrio/HG002_NA24385_son/CMRG_v1.00/GRCh38/StructuralVariant/ for evaluation. Similar to the 1KG samples, we used the MC-based VCF (see above) containing variant bubbles and haplotypes of 44 assembly samples as an input panel for PanGenie. We extracted all variant alleles with a length ≥50 bp from our genotyped VCF (biallelic version, after decomposition). We converted the ground truth VCF into a biallelic representation using bcftools norm -m -any and kept all alleles with length ≥50 bp. We used Truvari (v.3.1.0)[112] with parameters --multimatch --includebed <medically-relevant-sv-bed> -r 2000 --no-ref a -C 2000 --passonly to compare our genotype predictions to the medically relevant SVs. Results are shown in Supplementary Table 22. As PanGenie is a re-typing method, it can only genotype variants provided in the input and therefore cannot detect novel alleles. As HG002 is not among the panel samples, the input VCF misses variants unique to this HG002 sample. Thus, these unique variants cannot be genotyped by PanGenie and will be counted as false negatives during evaluation. Therefore, we computed an adjusted version of the recall that excluded SV alleles unique to HG002 (that is, alleles not in the graph) from the truth set for evaluation. To identify which SV alleles were unique, we compared each of the 44 panel samples to the ground truth VCF using Truvari to identify the false negatives for each sample. Then we computed the intersection of false-negative calls across all samples. The resulting set then contained all variant alleles unique to the HG002 ground truth set. We found 15 such unique SV alleles among the GIAB CMRG variants. We removed these alleles from the ground truth set and recomputed precision–recall statistics for our genotypes. Adjusted precision–recall values are shown in Supplementary Table 22.

All genotyping results produced by PanGenie are available at Zenodo (https://doi.org/10.5281/zenodo.6797328).

### Read mapping at VNTR regions
**Simulating and mapping VNTR reads.** Raw VNTR coordinates on GRCh38 (chromosomes 1–22 and sex chromosomes only) were generated using TRF (v.4.09)[56] with command trf hg38.fa 2 7 7 80 10 50 500 -f -d. Only repeats with a period size between 6,000 and 10,000 bp, total length >100 bp and not overlapping with centromeric regions were selected, leaving a total number of 98,021

non-overlapping loci. Using the raw VNTR coordinates on GRCh38 as input, VNTR regions across 96 haplotypes (including GRCh38 and CHM13) were annotated using the build module in danbing-tk (v.1.3)[55] (dist_scan=700, dist_merge=1, TRwindow=100000, MBE_th1=0.3, MBE_th2=0.6).

Whole-genome paired-end error-free short reads were simulated at around 30× for each genome, or equivalently about 15× for each haplotype. A read pair was generated for every 20 bp with fragment size of 500 bp and read length of 150 bp. Paired-end read mapping to the MC graph was done using Giraffe (v.1.39.0)[49] using the command vg giraffe -x $pref.xg -g $pref.gg -H $pref.gbwt -m $pref.min -d $pref.dist -p -f <(zcat $h1 $h2) -i -t 16, whereas mapping to GRCh38 was done using BWA-MEM (v.0.7.17-r1188)[50] using the command bwa mem -t 16 -Y -K 100000000 -p $ref <(zcat $h1 $h2). For a fair comparison, GRCh38 plus decoy minus ALT/HLA contigs were used as reference to match the paths included in the MC graph.

**Evaluating read mapping accuracy at VNTR regions.** To evaluate the performance of read mapping using the MC graph plus Giraffe, the VNTR information from danbing-tk were used to annotate each node in the graph by traversing each haplotype path. Every node that covers a VNTR region has a tuple that denotes the intersected interval; any aligned reads overlapping with the interval were considered mapped to the VNTR. Similarly, a read simulated from an interval overlapping with a VNTR was considered derived from the VNTR. To evaluate the performance of GRCh38 plus BWA-MEM, the mapped region by each read was obtained using the bamtobed submodule in BEDTools (v.2.30.0)[123]. The VNTR annotations on GRCh38 were used to determine whether a read was mapped to a VNTR.

For each read, we tracked its source and mapped VNTR and VNTRs, and used this information to compute accuracy. Only VNTRs present in danbing-tk's annotations were tracked; otherwise they were labelled untracked, the same as non-VNTR regions. A true positive denotes mapping from a VNTR to its original VNTR. An exogenous false positive denotes mapping from untracked regions to a VNTR. An endogenous false positive denotes mapping from a VNTR to another VNTR. A true negative denotes mapping from untracked regions to untracked regions. A false negative denotes mapping from a VNTR to untracked regions (Supplementary Fig. 30). Any alignments in the JSON output of Giraffe that did not contain the mapping field were considered unmapped. The two ends of a read pair that did not map to the same chromosome by BWA-MEM were also considered unmapped.

**Estimating VNTR length variants from read depths.** The WGS samples for 35 genomes (HG00438, HG00621, HG00673, HG00733, HG00735, HG01071, HG01106, HG01109, HG01175, HG01243, HG01258, HG01361, HG01891, HG01928, HG01952, HG01978, HG02055, HG02080, HG02145, HG02148, HG02257, HG02572, HG02622, HG02630, HG02717, HG02723, HG02818, HG02886, HG03098, HG03453, HG03486, HG03492, HG03579, NA18906 and NA19240) were mapped to the MC graph using Giraffe as described in the 'Pangenome point genotyping' subsection. Using the VNTR annotations described in the previous section, the number of reads mapped to each VNTR region in the MC graph was calculated as a proxy for VNTR length. VNTRs with invariant length across the 35 genomes were removed from analysis, leaving a total of 60,861 loci. The ground truth for a VNTR in a genome was computed from the number of bases spanned by the VNTR, averaged from the two haplotypes.

As a baseline control, the read depth of each VNTR region for the 35 WGS samples produced by mapping reads to GRCh38 was also computed with mosdepth (v.0.3.1)[126] using the command mosdepth -t 4 -b $VNTR_bed -x -f $hg38 $pref $cram. To be able to compare with the graph-based approach, VNTRs with missing annotation on GRCh38 were further removed, leaving a total of 60,386 VNTRs.

### RNA-seq mapping evaluation

We augmented the allele-filtered graph (see the 'Pangenome point genotyping' subsection) with edges for splice junctions to create a spliced pangenome graph using the rna subcommand in the vg toolkit (v.1.38.0) with a maximum node length set to 32 (vg rna -k 32)[57]. The transcript annotations that were used to define the splice junctions consisted of the CAT transcript annotations on each assembly together with splice junctions from the GENCODE (v.38) annotation[29]. Transcripts from the GENCODE (v.38) annotation were further added as paths to the spliced pangenome graph. For comparison, we also created a spliced reference constructed from the reference sequence, once again using the GENCODE (v.38) transcript annotation. For both graphs, we created the indices needed for mapping using the vg toolkit (v.1.38.0) with default parameters, except when pruning, for which edges on embedded paths were restored (vg prune -r). Furthermore, for the spliced HPRC pangenome graph, it was necessary to use stricter pruning parameters (vg prune -r -k 64 -M 64). For the spliced reference, we also created the index needed by the RNA-seq mapper STAR using default parameters.

We simulated RNA-seq reads with a pipeline that was designed to preserve complex genome variation in the simulated data. The transcript sequences used for the simulation were derived from the GENCODE (v.38) transcript annotations projected onto assembled haplotypes from HG002. Specifically, we used MC to create an alignment between GRCh38 and the two HPRC HG002 assembly haplotypes, which were held out of the main pangenome graph for benchmarking. We then used CAT to lift the transcript annotations over to these haplotypes. We constructed a spliced personal genome graph using the vg rna subcommand, and then we simulated reads using vg sim (commit 2cea1e2) using an Illumina NovaSeq cDNA read set (SRR18109271) to fit model parameters. This essentially amounts to simulating directly from the projected transcript sequences. The transcripts were simulated with uniform expression, split evenly between the two haplotypes, keeping the reads from each haplotype separate. This expression profile is not biologically realistic, but it avoids the difficulty of choosing a particular expression profile as representative for all tissues and stages of development. Moreover, existing estimated profiles would be biased towards the tools that were used to estimate them. We simulated 5,000,000 paired-end 150 bp RNA-seq reads.

Both simulated and real Illumina RNA-seq reads were mapped to the graphs using vg mpmap (commit c0c4816) with default parameters. In addition, the reads were mapped to the spliced reference using STAR (v.2.7.10a) with default parameters[58]. For the real data, we used previously published data NA12878 RNA-seq data (SRR1153470)[127] and the ENCODE project (ENCSR000AED, replicate 1)[128,129].

We used the same approach as previously described[57] to evaluate the alignments. In brief, for the simulated data, the graph alignments were compared with the truth alignments by estimating their overlap on the reference genome paths. The graph alignments were projected to the reference paths using vg surject -S. A uniquely aligned read (one with primary MAPQ ≥ 30) was considered correct if it overlapped 90% of the truth alignment. A multi-mapped read was considered correct if any of the multi-mappings was correct under the same criterion. For the real data, the average read coverage of each exon on the reference path calculated from the projected graph alignments were compared with the corresponding coverages estimated from long-read alignments. For the long-read data, we used PacBio Iso-Seq alignments from the ENCODE project (ENCSR706ANY, all replicates), which come from the same cell line as the Illumina data. The long-read alignments were used to define the exons, and only primary long-read alignments with a mapping quality of at least 30 were used. The alignments for the four Iso-Seq replicates (ENCFF247TLH, ENCFF431IOE, ENCFF520MMC and ENCFF626GWM) were combined and filtered using SAMtools (v.1.15)[130]. BEDTools (v.2.30.0) was used to convert the alignments to exons coordinates[123].

We also used the results of the mapping experiment to quantify allelic bias. We used vg deconstruct to identify sites of variation in the MC graph of haplotypes from HG002, with deletions greater than 10 kb removed to avoid spurious variants. Variants overlapping exons were selected and normalized using BCFtools (v.1.16)[130]. Next we counted the number of mapped reads from each of the two haplotypes that overlapped each heterozygous exonic variant[57]. Specifically, the read count for each allele was calculated as the average count across the two breakpoints of an allele. This was done to treat different variant types and lengths equally. We then tested all variants with a read coverage of at least 20 for allelic bias using a two-sided binomial test at $\alpha = 0.01$. All tests that reject the null hypothesis are false positives, as the reads were simulated without allelic bias. The results were split into different classes of variants and plotted against the number of sites that achieved a coverage of at least 20, which is a rough indicator of mapping sensitivity. Indels larger than 50 bp were excluded.

We also compared gene expression inference using the mapped reads. For the vg mpmap (v.1.43.0) graph mappings, we used RPVG (commit 1d91a9e)[57] in transcript inference mode to quantify expression. Two different transcript annotations were used as input to RPVG. The MC pantranscriptome was created from the CAT transcript annotations on each assembly and the GENCODE (v.38) transcriptome. The mpmap-RPVG pipeline was compared with Salmon (v.1.9.0)[131] and RSEM (v.1.3.3)[132]. These methods were provided the GENCODE (v.38) transcriptome as input, both with and without transcripts on the GRCh38 alternative contigs. Any genes unique to the alternative contigs were filtered. For Salmon, the GRCh38 reference was used as a decoy and duplicate transcripts were kept. Bowtie2 (v.2.4.5)[133] was used as a mapper for RSEM. Gene expression values were calculated by summing the corresponding transcript level values for each gene. GffRead (v.0.12.7) was used to create a table of gene names and transcript identifiers from the transcript annotation[134]. The accuracy was measured in two ways: (1) the Spearman correlation between the simulated and inferred expression values, and (2) the mean absolute relative difference between the simulated and inferred expression values.

The scripts that were used for graph construction, read simulation, mapping and evaluation are available at GitHub (https://github.com/jonassibbesen/hprc-rnaseq-analyses-scripts).

### ChIP-seq analysis

We aligned H3K4me1, H3K27ac and ATAC-seq obtained from monocyte-derived macrophages from 30 individuals[59] using vg map[13] to the GRCh38 reference genome graph and to the HPRC genome graph. Then we called peaks using Graph Peak Caller (v.1.2.3)[135] on both sets of alignments for each of the 30 H3K4me1, H3K27ac and ATAC-seq samples. To identify HPRC-only peaks, we projected HPRC coordinates to the GRCh38 path using Graph Peak Caller and compared intervals using BEDTools[123]. We named HPRC peaks that overlapped GRCh38 peaks as common peaks and those that do not as HPRC-only. We calculated the expected frequency (as inverse cumulative distributions) of common and HPRC-only peaks among the 30 samples by resampling the peaks of each sample from the peaks of all the samples and re-counted the number of overlaps. We repeated this simulation 100 times and plotted the average curves. We determined heterozygous variants in our samples by aligning WGS datasets for each sample to the HPRC graph using vg map and genotyping the variants using vg call -a. We narrowed the list of heterozygous SVs above 50 bp in each sample with the aim of looking for allelic-specific peaks. For each epigenomic sample, we obtained allelic-specific read counts within peaks that lie on the previously identified loci by running vg call -a on the epigenomic HPRC alignments, which outputs the numbers of reads on each path in a bubble (DP and AD fields in the VCF output). We then assigned peaks to the SV or reference allele, or both alleles with a two-tailed binomial test parameterized on the sum of reads on both alleles and $P = 0.05$. Any peak with reads on one allele, but not on the other was assigned to the

allele with the reads. Read counts were proportionally adjusted for the difference in length between the reference and SV alleles.

Processed data, scripts and code for the above steps are available at Zenodo (https://doi.org/10.5281/zenodo.6564396).

### Population genetic analyses

Although the size of the population sample represented in our pangenome is small, it provides access to previously under-ascertained regions of the genome. We sought to understand the potential utility of these regions for future population genetic studies using regional PCA based on variants called compared to the CHM13 and GRCh38 references. For these analyses, we considered both the PGGB (whole pangenome, combined) and MC (reference-based, distinct CHM13 and GRCh38) graphs. For both graph models, the CHM13 VCFs provide access to regions that were not previously observed by studies based on GRCh38, for which short-read-based studies may have difficulty reliably aligning and calling variants. In combination, these two graphs provide cross-validation of implied population genetic patterns in these new regions, which we explore here.

To understand chromosome-specific patterns of variation, we applied PCA to each autosomal chromosome independently to the VCFs from PGGB (PGGB-CHM13, PGGB-GRCh38).

To ensure that observed patterns were not derived from higher rates of assembly error in the repetitive regions of acrocentric p-arms, we used our Flagger-confident region annotations to prune the PGGB graph (using odgi inject to inject the confident regions as subpaths and then odgi prune to remove the full original paths that were including unreliable regions) to only confident regions of assemblies. We then reapplied vg deconstruct to this graph to obtain a new set of SNPs (the code for the PGGB graphs pruning and variant calling on the pruned graphs can be found at the following link: https://github. com/pangenome/HPRCyear1v2genbank/blob/main/workflows/confident_variants.md). Genome-wide, we found that pruning reduced the number of called SNPs by only 1.188% (previous $N$ = 23,272,652, pruned $N$ = 22,996,113). The total reduction in the acrocentrics was higher, with 6.29% fewer SNPs (previous $N$ = 3,735,605, pruned $N$ = 3,676,746), which indicated the difficulty in assembling these regions. We note that the PCA sample distributions remained almost identical (data not shown), which indicated that the patterns observed in the full graph are maintained despite assembly issues. In these filtered PGGB-CHM13 and PGGB-GRCh38 VCFs, we considered all biallelic SNPs relative to the chosen reference, regardless of variant nesting level (data not shown; filtering for only SNPs LV = 0 or LV > 0 produced nearly identical results). A qualitative evaluation suggested no significant differences in PCA patterns across the metacentric chromosomes (Supplementary Fig. 47). However, in the p-arms of the acrocentrics (chromosomes 13, 14, 15, 21 and 22), which are accessible in the PGGB-CHM13 VCF, we observed a reduction in population differentiation and a higher rate of variance explained in the lowest principal component.

To investigate this quantitatively, we measured the number of clusters implied by the PCA for the PGGB-CHM13 VCFs using $k$-means clustering to automatically determine the optimal number of clusters for each PCA (gap_stat clustering in the fviz_nbclust function of the factoextra R package) (analysis code available at: https://github.com/ SilviaBuonaiuto/hprcPopGenAnalysis). Applying this approach to three PCAs per chromosome VCF, we obtained optimal cluster counts for the p-arm, q-arm and entire chromosome. In metacentric chromosomes, we usually observed optimal numbers of clusters approximately corresponding to the number of expected world population groupings in the input genomes (3–5, as in Supplementary Fig. 48). However, in the p-arms of the acrocentrics, we observed much fewer, in general only one cluster, indicative of reduced population differentiation compared with other parts of the acrocentric chromosomes. This pattern was only apparent in the PGGB graph based on CHM13. To evaluate the difference quantitatively, we applied a Wilcoxon rank-sum test to compare the differences between cluster count distributions in metacentric versus acrocentric chromosomes across the entire chromosome, the q-arm and the p-arm. There were non-significant differences between the distributions between acrocentric and metacentric chromosomes at a chromosome scale and in the q-arms, but a significant difference (Wilcoxon $P$ = 0.013) in the case of acrocentric p-arms (Supplementary Fig. 49).

This analysis indicates that significant challenges remain for the use of these new regions in population genetic studies. Patterns observed in PCA projections of the pangenome across all chromosomes suggests a distinct process of variation sharing between populations within the short arms of the acrocentrics. In effect, we observed a more homogenous population in these regions when using the CHM13 assembly as a reference. This reference contains real sequences in these regions, whereas GRCh38 contains gaps, which render analysis impossible. The apparent population homogenization could be driven by error. We mitigated this issue by utilizing only SNPs found in Flagger-confident regions, but this does not guard against potential sources of alignment error that are likely to be amplified by the repetitive sequences in these loci. It is also possible that the chromosome-specific partitioning process applied by both graph models is failing to correctly partition contigs on these short arms. The known homology between the short arms bolsters the possibility of ongoing sequence information exchange between non-homologous chromosomes[3], which would be consistent with the patterns we observed. In summary, this analysis shows that when using CHM13 as a reference, the behaviour of sequences on the short arms of the acrocentrics in the PGGB graph is not similar to that of other sequences in the pangenome.

### Reporting summary

Further information on research design is available in the Nature Portfolio Reporting Summary linked to this article.

## Data availability

Sequencing data, assemblies and pangenomes produced by the HPRC are available at AnVIL (https://anvilproject.org) in the AnVIL_HPRC workspace. Data are also available as part of the AWS Open Data Program (https://registry.opendata.aws) in the human-pangenomics S3 bucket (https://s3-us-west-2.amazonaws.com/human-pangenomics/ index.html). In addition, data have been uploaded to the International Nucleotide Sequence Database Collaboration for long-term storage and availability. Supporting information about the data (including index files with S3 and GCP file locations) can be found in our GitHub repositories (see below). Sequencing data for 29 selected HPRC samples from the 1KG cohort (Results) are uploaded to BioProject PRJNA701308 (https://www.ncbi.nlm.nih.gov/bioproject/PRJNA701308). Sequencing data created by the HPRC for samples in the cohort of 18 additional samples are uploaded to BioProject PRJNA731524 (https://www.ncbi.nlm. nih.gov/bioproject/PRJNA731524). Both sets of assemblies are grouped by an Umbrella BioProject PRJNA730822. Data used in this paper have additional information available at AnVIL and at GitHub (https://github. com/human-pangenomics/HPP_Year1_Data_Freeze_v1.0). Assemblies along with assembly annotations, such as RepeatMasker and Ensembl gene annotations, can be viewed in an assembly hub in the UCSC Genome Browser (http://hprc-browser.ucsc.edu). The data and annotations can also be accessed through the Ensembl Rapid Release Genome Browser (https://rapid.ensembl.org) and a dedicated Ensembl project page for centralized access to HPRC data (https://projects. ensembl.org/hprc), which includes links to download the data locally. File locations for the assemblies (and select annotation files) that are stored in S3 and GCP can also be found at AnVIL or GitHub (https:// github.com/human-pangenomics/HPP_Year1_Assemblies). Pangenomes were uploaded to the European Nucleotide Archive as analysis objects and are organized under Umbrella BioProject PRJNA850430

(https://www.ncbi.nlm.nih.gov/bioproject/PRJNA850430). File locations for pangenomes, indices and variant calls derived from the pangenomes can be found at AnVIL or GitHub (https://github.com/human-pangenomics/hpp_pangenome_resources). Variant calls produced by the analysis performed in this paper are available at Amazon S3 (https://s3-us-west-2.amazonaws.com/human-pangenomics/index.html?prefix=publications/PANGENOME_2022) or where indicated in the relevant sections below. PanGenie genotypes produced for the 1KG samples based on the MC graph are available at Zenodo (https://doi.org/10.5281/zenodo.6797328).

## Code availability

The list of all tools, including versions and/or code commits, used for this study are available in Supplementary Table 20.

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

**Acknowledgements** We would like to acknowledge S. Bidwell and other members of the GenBank staff at the National Center for Biotechnology Information (NCBI; NLM/NIH) for their work to release the assemblies into GenBank. Certain commercial equipment, instruments or materials are identified to adequately specify experimental conditions or reported results. Such identification does not imply recommendation or endorsement by the National Institute of Standards and Technology nor does it imply that the equipment, instruments or materials identified are necessarily the best available for the purpose. Computational infrastructure and support were provided by the Centre for Information and Media Technology at Heinrich Heine University Düsseldorf. This work was funded in part by the National Human Genome Research Institute of the National Institutes of Health under award numbers U41HG010972, 1U01HG010973, U41HG007234, 1R01HG011274, R01HG010485, U24HG010262 U01HG010963, U24HG007497 and R01HG011649. This work was funded in part by the National Institutes of Health under award numbers U01HG010961, OT2OD033761, U24HG011853, R01-HG006677, R35-GM130151, R01HG002385, R01HG010169, U01HG01973, 5U01HG010971, R01GM123489, U24HG009081 and 1ZIAHG200398. This work was funded in part by the Intramural Research Program of the National Human Genome Research Institute, National Institutes of Health.

The work of F.T.-N. and V.A.S. was supported by the National Center for Biotechnology Information of the National Library of Medicine (NLM), National Institutes of Health. This work was funded in part by the USDA National Institute of Food and Agriculture, grant number 2018-67015-28199, and the National Science Foundation (NSF), grant IOS-1744309, and NSF PPoSS award number 2118709 (E.G. and P.P.). This work was funded in part by the Natural Sciences and Engineering Research Council of Canada (NSERC). G.Bourque is supported by a Canada Research Chair Tier 1 award, a FRQ-S, Distinguished Research Scholar award and by the World Premier International Research Center Initiative (WPI), MEXT, Japan. J.S. was supported by the Carlsberg Foundation. This work was funded in part by intramural funding at the National Institute of Standards and Technology. E.E.E., D.H. and E.D.J. are investigators of the Howard Hughes Medical Institute. This work was funded in part by an Oxford Nanopore Research grant (SC20130149) awarded to M.Akeson, University of California Santa Cruz. This work was funded in part by Wellcome Trust award numbers WT104947/Z/14/Z, WT222155/Z/20/Z and WT108749/Z/15/Z. This work was funded in part by a Juan de la Cierva fellowship grant (IJC2020-045916-I) funded by MCIN/AEI/ 10.13039/501100011033 and by the European Union NextGenerationEU/PRTR. This work was funded in part by the Novo Nordisk Foundation (NNF21OC0069089). S.H. acknowledges funding from the Central Innovation Programme (ZIM) for SMEs of the Federal Ministry for Economic Affairs and Energy of Germany. This work was supported by the BMBF-funded de.NBI Cloud within the German Network for Bioinformatics Infrastructure (de.NBI) (031A532B, 031A533A, 031A533B, 031A534A, 031A535A, 031A537A, 031A537B, 031A537C, 031A537D and 031A538A). This work was funded in part by the German Federal Ministry of Education and Research (BMBF) (031L0184A) and the European Commission, Innovative training network (ITN) (956229). W.-W.L. was supported in part by the Government Scholarship to Study Abroad (GSSA) from the Ministry of Education of Taiwan.

**Author contributions** Pangenome empirical analysis and pangenome quality control: W.-W.L., D.D., M.H., G.H., C.M., J.Monlong, H.J.A., J.M.Z., E.E.E., T.M., I.M.H., P.M. and J.W. Paper writing: W.-W.L., M.A., J.E., D.D., M.H., G.H., S.L., J.Monlong, R.S.F., S.G., T.-Y.L., M.W.M., A.M.N., H.E.O., T.P., J.A.S., M.R.V., G.Bourque, K.H.M., E.G., T.M., I.M.H., B.P., R.E.G. and L.H. Paper editing: W.-W.L., D.D., M.H., G.H., X.H.C., H.C., A.G., A.M.N., P.P., A.M.P., E.E.E., E.D.J., K.H.M., E.G., E.E.K., T.M., I.M.H., H.L., B.P., O.E.C., P.E., G.F., A.N.A.T. and A.V.Z. Assembly creation: M.A., J.K.L., H.C., A.M.P., H.L., D.Puiu, A.A.R. and A.V.Z. Assembly quality control and assembly reliability analysis: M.A., J.K.L., H.C., J.C., S.G., K.Howe, T.P., D.Porubsky, C.T., M.R.V., A.M.P., J.M.Z., E.E.E., K.H.M., H.L., R.E.G., S.K., J.McDaniel, S.N., N.D.O., D.Puiu, M.R., A.A.R., A.R., V.A.S., K.S., F.T.-N., J.W., B.W., J.M.D.W. and A.B.P. Pangenome applications (structural variants): J.E., G.H., H.J.A., W.T.H., P.P., E.E.E., T.M., H.P.J. and H.M. Pangenome graph creation: D.D., G.H., A.G., S.H., M.N.M., F.V., E.G., Y.G. and S.M.-S. Data coordination and management: M.H., J.K.L., R.S.F., W.T.H., M.J., C.T., A.M.P., E.D.J., K.H.M., T.W., L.L.A.-F., S.C., M.D., S.F. and R.E.G. Transcriptome and annotation: M.H., J.M.E., F.J.M., M.R.V., M.J.P.C., K.B., M.D., A.F., C.G.G., L.H., T.H. and F.F.T. Pangenome applications (small variants): G.H., J.Monlong, C.M., A.M.N., P.P., J.M.Z., G.Baid, A.B., A.C., P.-C.C., D.E.C., H.P.J., A.K., M.N., K.S. and J.W. Pangenome visualization and complex loci analysis: S.L., J.C., C.F. and A.G. Population genetic analysis: S.B., A.G. and V.C. Sample selection: X.F., K.M.M., A.M.P., E.E.K., E.E.E., K.H.M., S.F. and J.O.K. Sequencing: R.S.F., M.J., M.W.M., K.M.M., H.E.O., A.M.P., E.E.E., E.D.J., K.H.M., C.A.B., O.F., R.E.G., K.Hoekzema, J.O.K., J.K., A.P.L., J.Mountcastle, S.S. and A.D.S. Pangenome applications (ChIP-seq analysis): C.G. and G.Bourque. Principal investigator and laboratory organizer within the HPRC: M.J., M.W.M., M.J.P.C., P.F., E.E.K., A.M.P., E.E.E., D.H., E.D.J., K.H.M., T.W., T.M., B.P., R.E.G. and V.A.S. Pangenome applications (VNTR analysis): T.-Y.L. and M.J.P.C. Pangenome applications (RNA-seq analysis): J.A.S. and J.M.E. Development of algorithms and software: J.S., H.P.J., H.L., G.H., A.M.N., P.P., A.G., S.H., D.D., M.N.M., F.V., E.G., Y.G., S.M.-S., H.L., J.M.E., J.A.S., B.P., T.M. and X.H.C. Ethical, legal and social implications: R.M.C.-D., N.A.G., B.A.K., A.M. and A.B.P. Programme organization: A.L.F., B.I.S., M.W.S. and H.J.S.

**Competing interests** E.E.E. is a scientific advisory board (SAB) member of Variant Bio. P.F is a member of the SABs of Fabric Genomics and Eagle Genomics. E.E.K. is a member of the SAB of Encompass Biosciences, Foresite Labs and Galateo Bio and has received personal fees from Regeneron Pharmaceuticals, 23&Me and Illumina. A.B., A.C., P.-C.C., D.E.C., G.Baid, A.K., M.N. and K.S. are employees of Google and own Alphabet stock as part of the standard compensation package.

**Additional information**
**Correspondence and requests for materials** should be addressed to Erik Garrison, Tobias Marschall, Ira M. Hall, Heng Li or Benedict Paten.

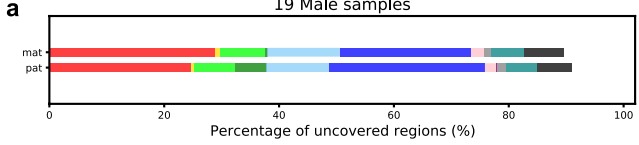

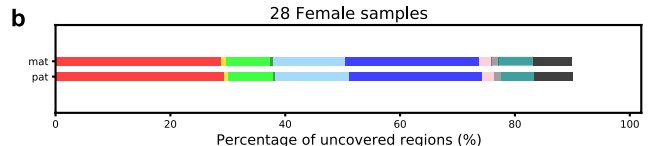

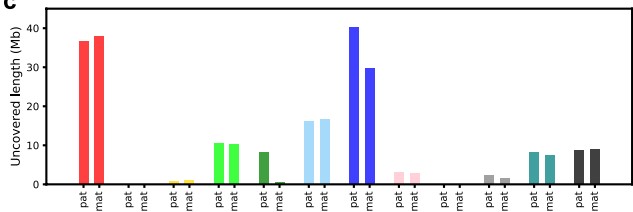

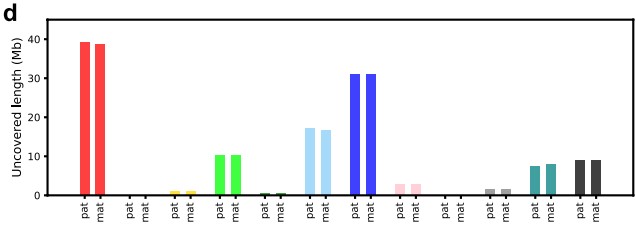

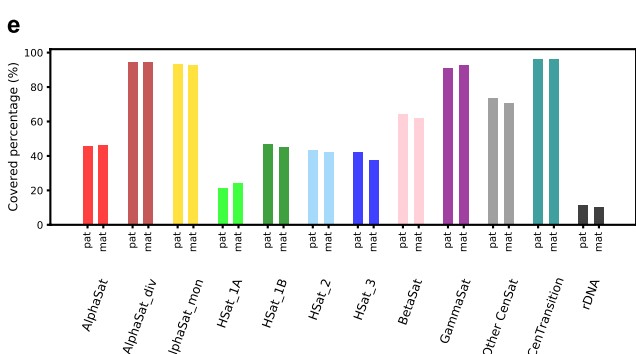

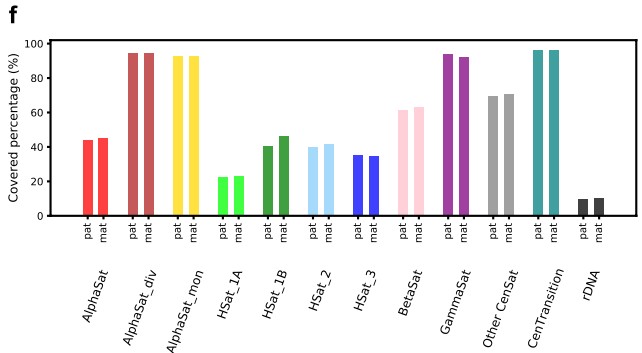

**Extended Data Fig. 1 | Characterizing uncovered reference bases using peri/centromeric annotation and evaluating the completeness of different satellite families.** We characterized the regions not covered by the assembly alignments to the T2T-CHM13 (v.2.0) reference and also investigated the completeness of the peri/centromeric satellites across all HPRC assemblies. We characterized these regions using the peri/centromeric annotation available for the T2T-CHM13 (v.2.0) reference. We made separate bar plots for male and female samples to exclude chromosome X for the paternal assemblies of male samples and exclude chromosome Y for all other assemblies. Panels **a** and **b** indicate that on average ~90% of the uncovered bases are located in peri/centromeric regions with the active/inactive alpha satellites and human satellite 3 comprising ~50% of these bases, mainly due to their highly repetitive composition and also higher frequency compared to other satellites. Other centromeric satellites, centromeric transition regions, and rDNA arrays accounted for another ~40% of the uncovered bases on average. Panels **c** and **d** display the average lengths of uncovered regions located within each satellite family. Panels **e** and **f** show what percentage of each satellite family was covered by at least one assembly alignment. The most complete centromeric regions (~90% coverage) are divergent/monomeric alpha satellites, gamma satellites and centromeric transition regions. The rDNA arrays have been covered by ~8% on average, which made them the least completely assembled repeat arrays.

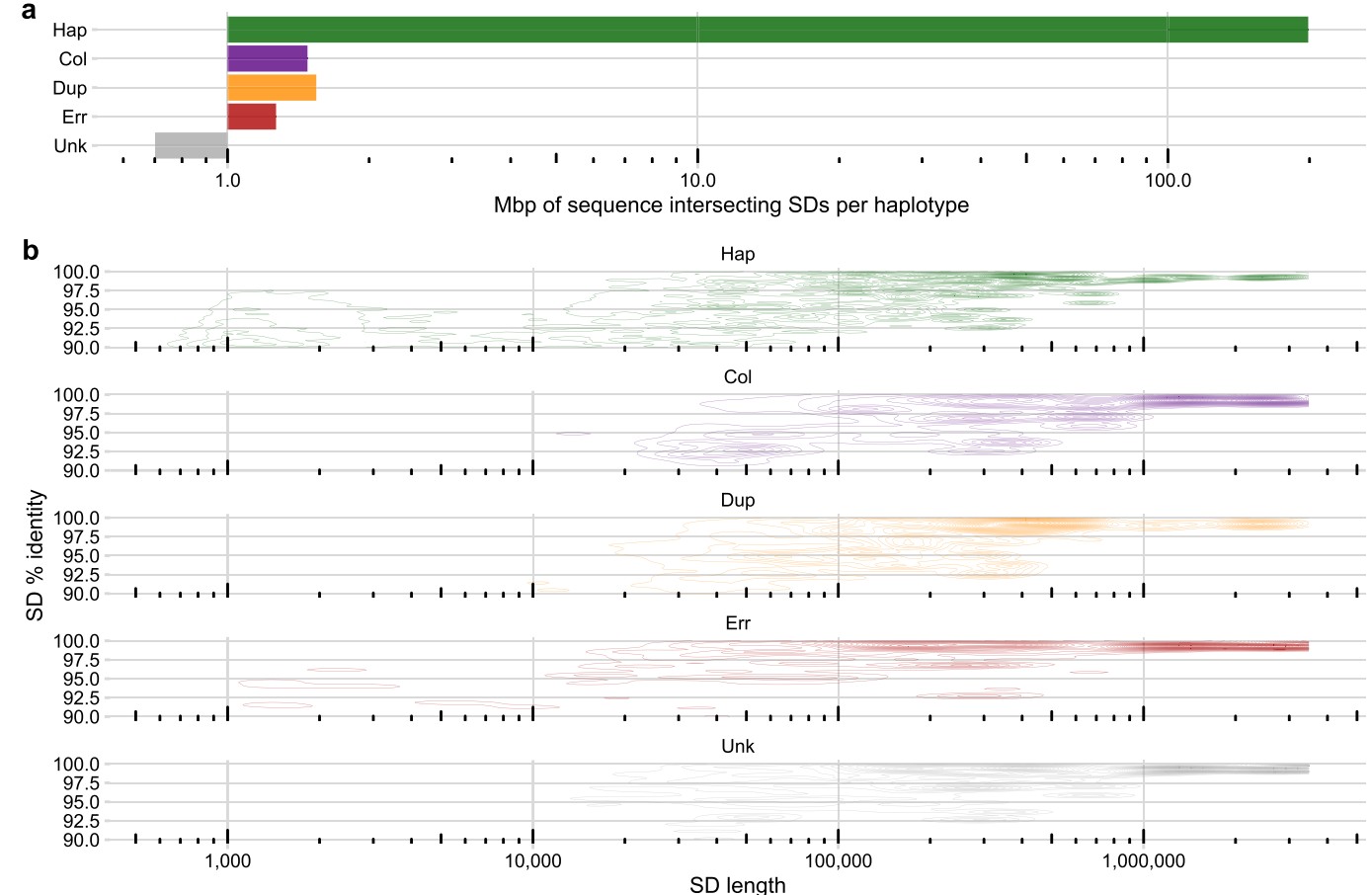

**Extended Data Fig. 2 | Segmental duplication reliability. a**, Average number of Mbp per haplotype of correctly or incorrectly assembled SDs lifted from T2T-CHM13 (v.2.0). **b**, The features of the most identical and longest overlapping SDs for each type of assembly error calculated in 5 kbp windows.

**a**

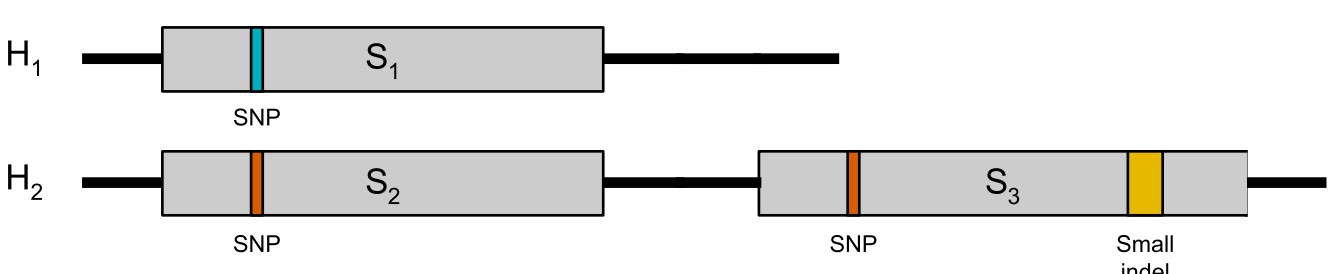

**b**

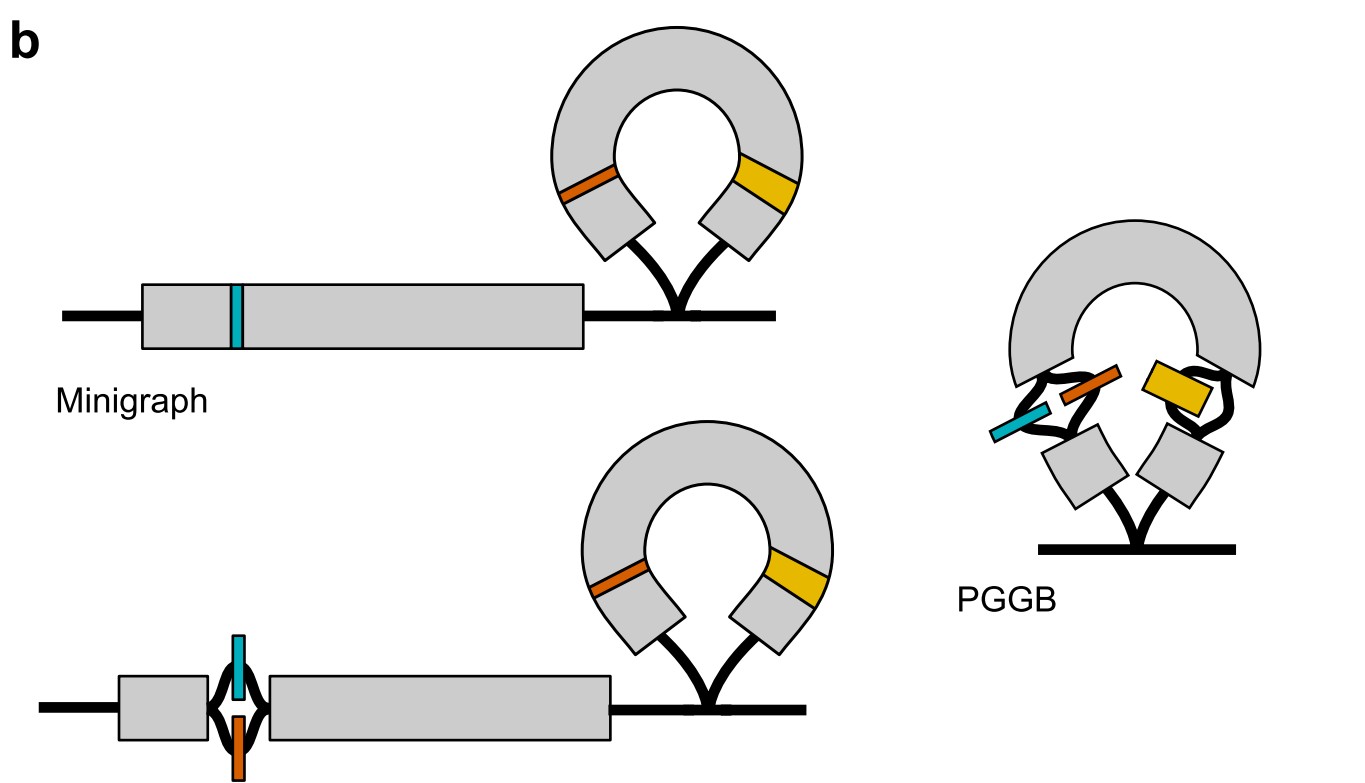

Minigraph

PGGB

MC

**Extended Data Fig. 3 | The differences in pangenome graph construction methods for Minigraph, MC, and PGGB. a**, Two haplotypes ($H_1$ and $H_2$) vary in copy number of a chromosomal segment S. The $S_1$, $S_2$, and $S_3$ segments are highly similar with only a SNP or a small indel. **b**, Pangenome graph structures for Minigraph, MC, and PGGB. Minigraph used $H_1$ as an initial backbone and then augmented with SVs ($\geq 50$ bp) from $H_2$, such that the SNP in $S_2$ is not represented in the pangenome graph. MC added small variants ($<50$ bp) to the pangenome graph constructed by Minigraph. PGGB used a symmetric, all-by-all alignment of haplotypes to build a pangenome graph whose structure is not affected by the order of inputs (unlike Minigraph and MC). The critical difference in graph construction is that, due to ambiguous pairwise relationships of paralogs, PGGB tends to collapse copy-number polymorphic loci like segmental duplications and VNTRs into a single copy through which haplotypes loop, while Minigraph and MC do not.

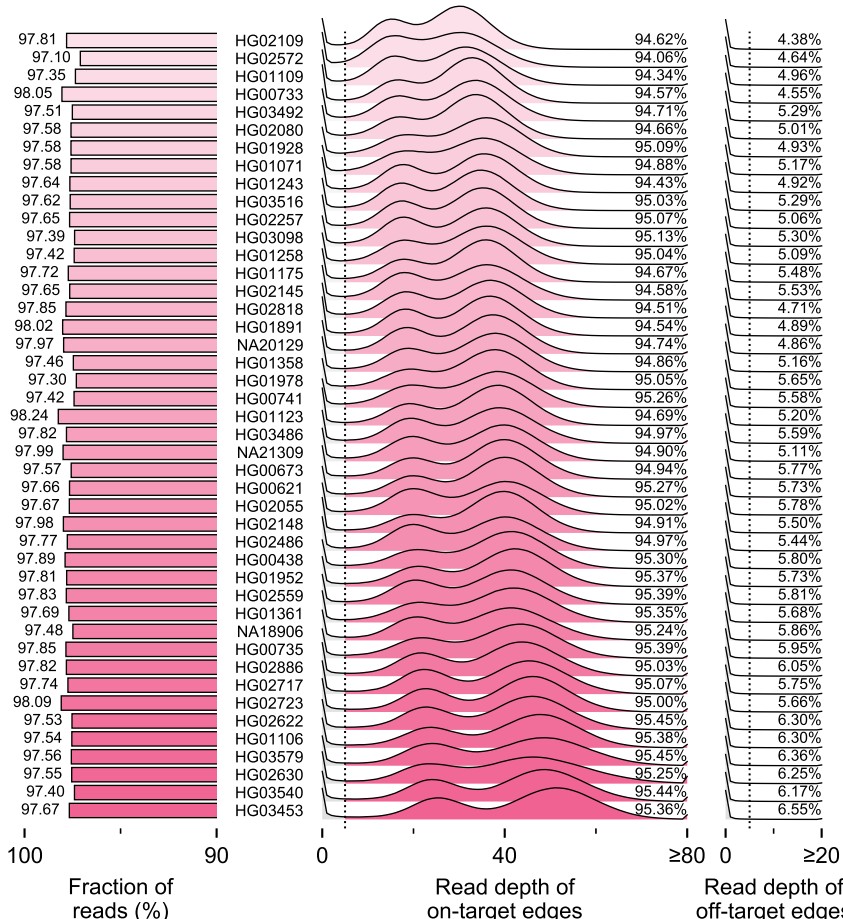

| Fraction of reads (%) | Sample | Read depth of on-target edges | Read depth of off-target edges |
|---|---|---|---|
| 97.81 | HG02109 | 94.62% | 4.38% |
| 97.10 | HG02572 | 94.06% | 4.64% |
| 97.35 | HG01109 | 94.34% | 4.96% |
| 98.05 | HG00733 | 94.57% | 4.55% |
| 97.51 | HG03492 | 94.71% | 5.29% |
| 97.58 | HG02080 | 94.66% | 5.01% |
| 97.58 | HG01928 | 95.09% | 4.93% |
| 97.58 | HG01071 | 94.88% | 5.17% |
| 97.64 | HG01243 | 94.43% | 4.92% |
| 97.62 | HG03516 | 95.03% | 5.29% |
| 97.65 | HG02257 | 95.07% | 5.06% |
| 97.39 | HG03098 | 95.13% | 5.30% |
| 97.42 | HG01258 | 95.04% | 5.09% |
| 97.72 | HG01175 | 94.67% | 5.48% |
| 97.65 | HG02145 | 94.58% | 5.53% |
| 97.85 | HG02818 | 94.51% | 4.71% |
| 98.02 | HG01891 | 94.54% | 4.89% |
| 97.97 | NA20129 | 94.74% | 4.86% |
| 97.46 | HG01358 | 94.86% | 5.16% |
| 97.30 | HG01978 | 95.05% | 5.65% |
| 97.42 | HG00741 | 95.26% | 5.58% |
| 98.24 | HG01123 | 94.69% | 5.20% |
| 97.82 | HG03486 | 94.97% | 5.59% |
| 97.99 | NA21309 | 94.90% | 5.11% |
| 97.57 | HG00673 | 94.94% | 5.77% |
| 97.66 | HG00621 | 95.27% | 5.73% |
| 97.67 | HG02055 | 95.02% | 5.78% |
| 97.98 | HG02148 | 94.91% | 5.50% |
| 97.77 | HG02486 | 94.97% | 5.44% |
| 97.89 | HG00438 | 95.30% | 5.80% |
| 97.81 | HG01952 | 95.37% | 5.73% |
| 97.83 | HG02559 | 95.39% | 5.81% |
| 97.69 | HG01361 | 95.35% | 5.68% |
| 97.48 | NA18906 | 95.24% | 5.86% |
| 97.85 | HG00735 | 95.39% | 5.95% |
| 97.82 | HG02886 | 95.03% | 6.05% |
| 97.74 | HG02717 | 95.07% | 5.75% |
| 98.09 | HG02723 | 95.00% | 5.66% |
| 97.53 | HG02622 | 95.45% | 6.30% |
| 97.54 | HG01106 | 95.38% | 6.30% |
| 97.56 | HG03579 | 95.45% | 6.36% |
| 97.55 | HG02630 | 95.25% | 6.25% |
| 97.40 | HG03540 | 95.44% | 6.17% |
| 97.67 | HG03453 | 95.36% | 6.55% |

100    90    0    40    ≥80    0    ≥20

Fraction of reads (%)        Read depth of on-target edges        Read depth of off-target edges

**Extended Data Fig. 4 | HiFi read depth of on- and off-target edges in the MC graph.** Left: fraction of reads aligned to the pangenome graph after filtering low-quality alignments. Middle: read depth distribution of on-target edges. Right: read depth distribution of off-target edges. Samples are sorted by sequencing coverage (Supplementary Table 1).

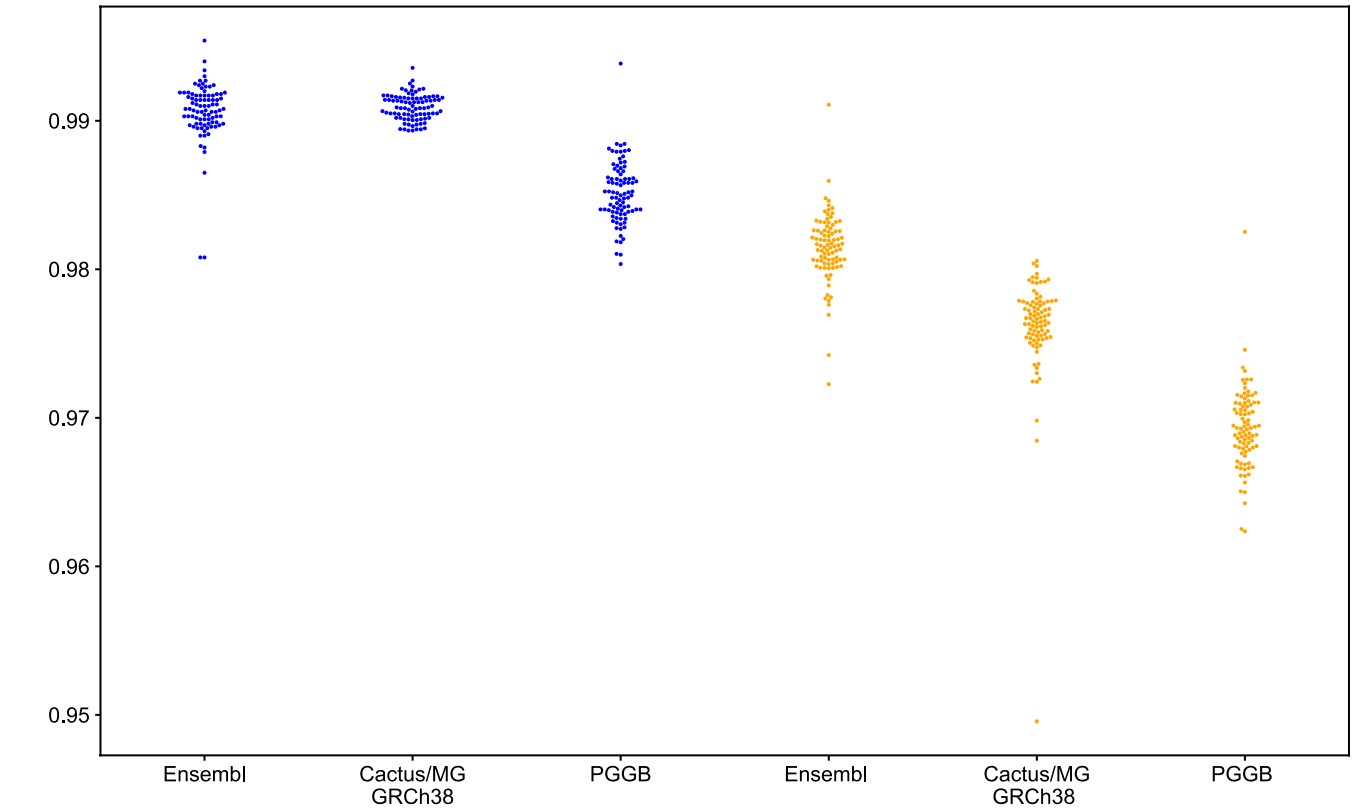

% Protein-coding genes mapped % Noncoding genes mapped

**Extended Data Fig. 5 | Gene mapping in the pangenome graphs.** The first three show the percentage of protein-coding genes from GENCODE (v.38) able to be mapped in the gene annotation sets from Ensembl, CAT run on the MC graph based on GRCh38, and CAT run on the PGGB graph. The second three show the percentage of noncoding genes from GENCODE (v.38) able to be mapped on the same annotation sets.

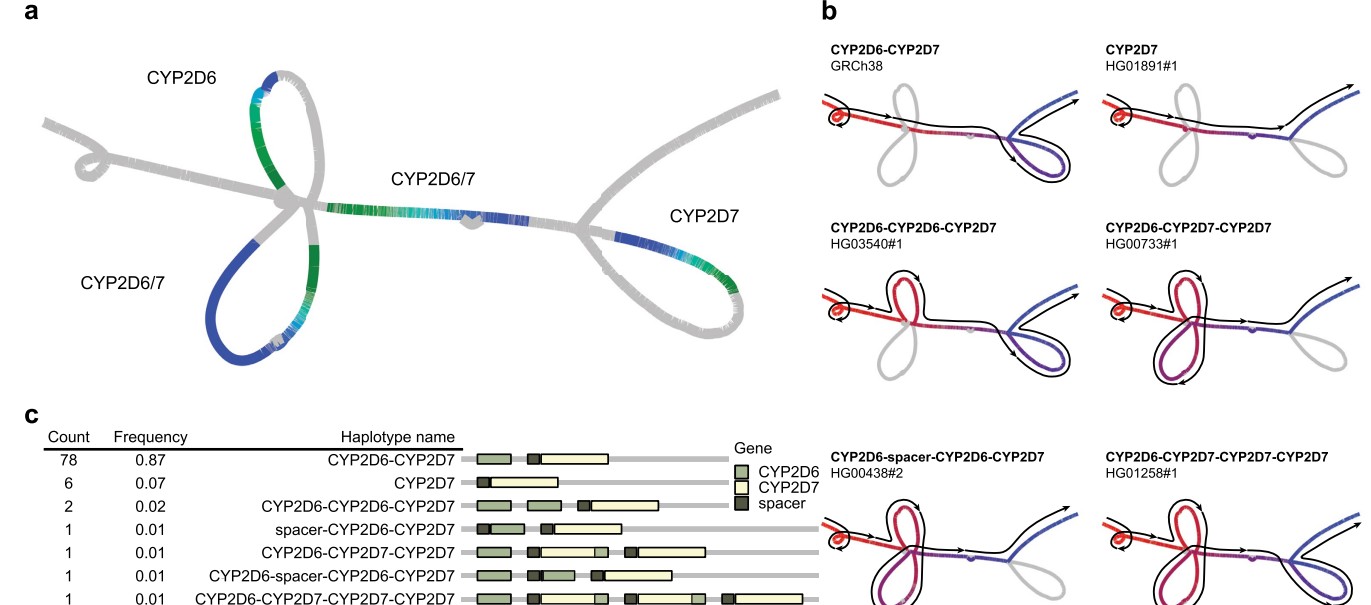

**c**

| Count | Frequency | Haplotype name | | Gene |
|---|---|---|---|---|
| 78 | 0.87 | CYP2D6-CYP2D7 | | CYP2D6 |
| 6 | 0.07 | CYP2D7 | | CYP2D7 |
| 2 | 0.02 | CYP2D6-CYP2D6-CYP2D7 | | spacer |
| 1 | 0.01 | spacer-CYP2D6-CYP2D7 | | |
| 1 | 0.01 | CYP2D6-CYP2D7-CYP2D7 | | |
| 1 | 0.01 | CYP2D6-spacer-CYP2D6-CYP2D7 | | |
| 1 | 0.01 | CYP2D6-CYP2D7-CYP2D7-CYP2D7 | | |

**Extended Data Fig. 6 | Structural haplotypes of *CYP2D6* and *CYP2D7* from the MC graph. a**, Locations of *CYP2D6* and *CYP2D7* within the graph. The colour gradient is based on the precise relative position of each gene; green, head of a gene; blue, end of a gene. **b**, Different structural haplotypes take different paths through the graph. The colour gradient and lines show the path of each allele; red, start of a path; blue, end of a path. **c**, Frequency and linear structural visualization of all structural haplotypes called by the graph among 90 haploid assemblies.

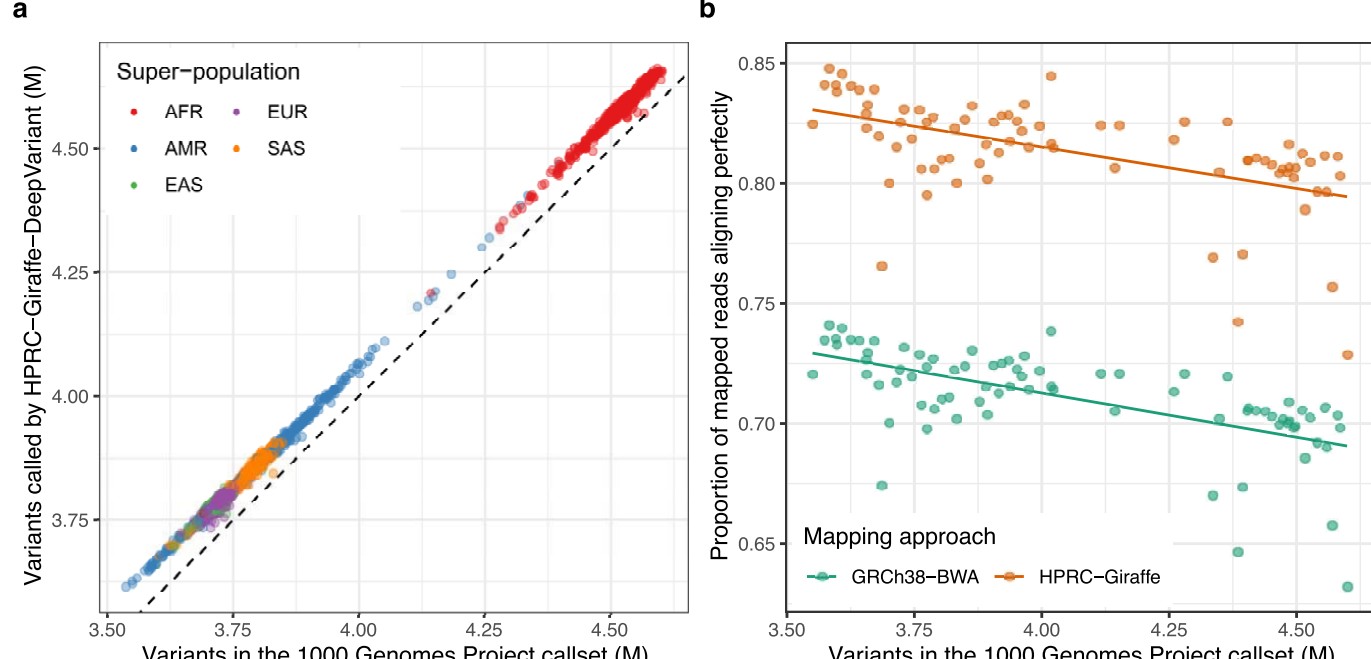

**Extended Data Fig. 7 | Performance comparison of pangenome-based variant calling and read mapping across populations. a**, Number of variants with at least one alternate allele (i.e. excluding homozygous for the reference allele) for each in the 1KG samples. The number of variants in the 1KG callset (x-axis) are compared to the variants found when aligning reads to the HPRC pangenome and calling variants with DeepVariant (y-axis). Points (samples) are coloured by their super-population label from the 1KG. **b**, The proportion of mapped reads that align perfectly (y-axis) is shown for a subset of samples from the 1KG, ordered by the number of variants called (x-axis). Two mapping approaches are compared: mapping short reads to GRCh38 with BWA (green); mapping to the HPRC pangenome with Giraffe (orange). The samples were selected to span the x-axis.

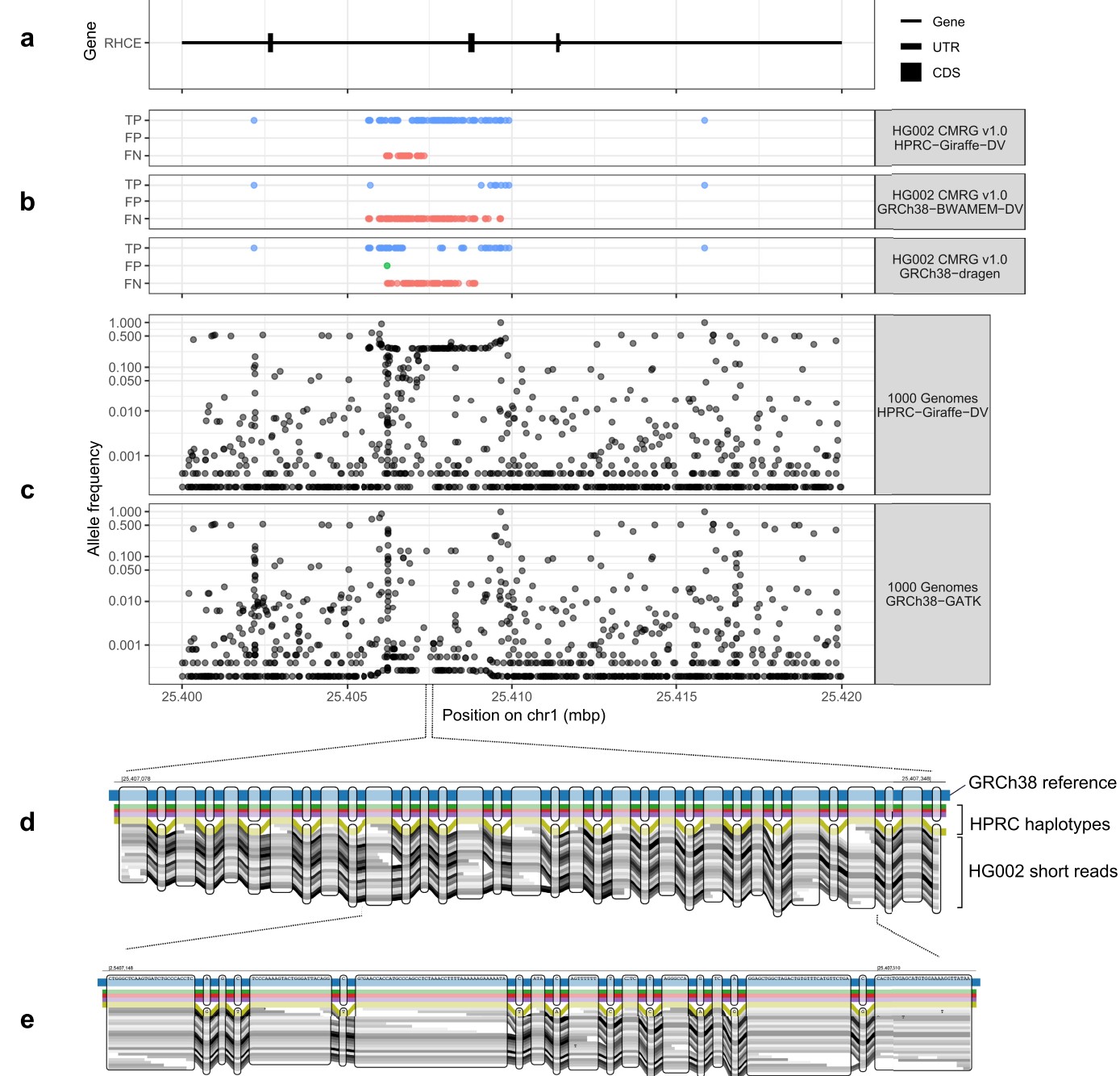

**Extended Data Fig. 8 | Improved genotyping in the challenging medically-relevant gene *RHCE*. a**, Gene annotation of part of the *RHCE* gene. **b**, Genotyping performance in this region for three approaches (horizontal panels). The top panel, using the HPRC pangenome, shows the best performance with most variants being true positives (TP, blue points) based on the CMRG (v.1.0) truth set while more other methods have a higher number of false negatives (FN, red points). **c**, Allele frequency across 2,504 unrelated individuals of the 1KG.

The HPRC-Giraffe-DeepVariant calls show higher frequencies. In particular, the gene-converted alleles, at about 25.406-25.410 Mbp, are observed at ~25% frequency, similar to estimates from the HPRC haplotypes (Fig. 5a–c). **d,e**, A pangenomic view of the gene-converted region showing 1 of 4 haplotypes in the HPRC pangenome supporting the non-reference alleles. The inclusion of this haplotype in the HPRC pangenome enables short sequencing reads, here from HG002, to map along this gene-converted haplotype.

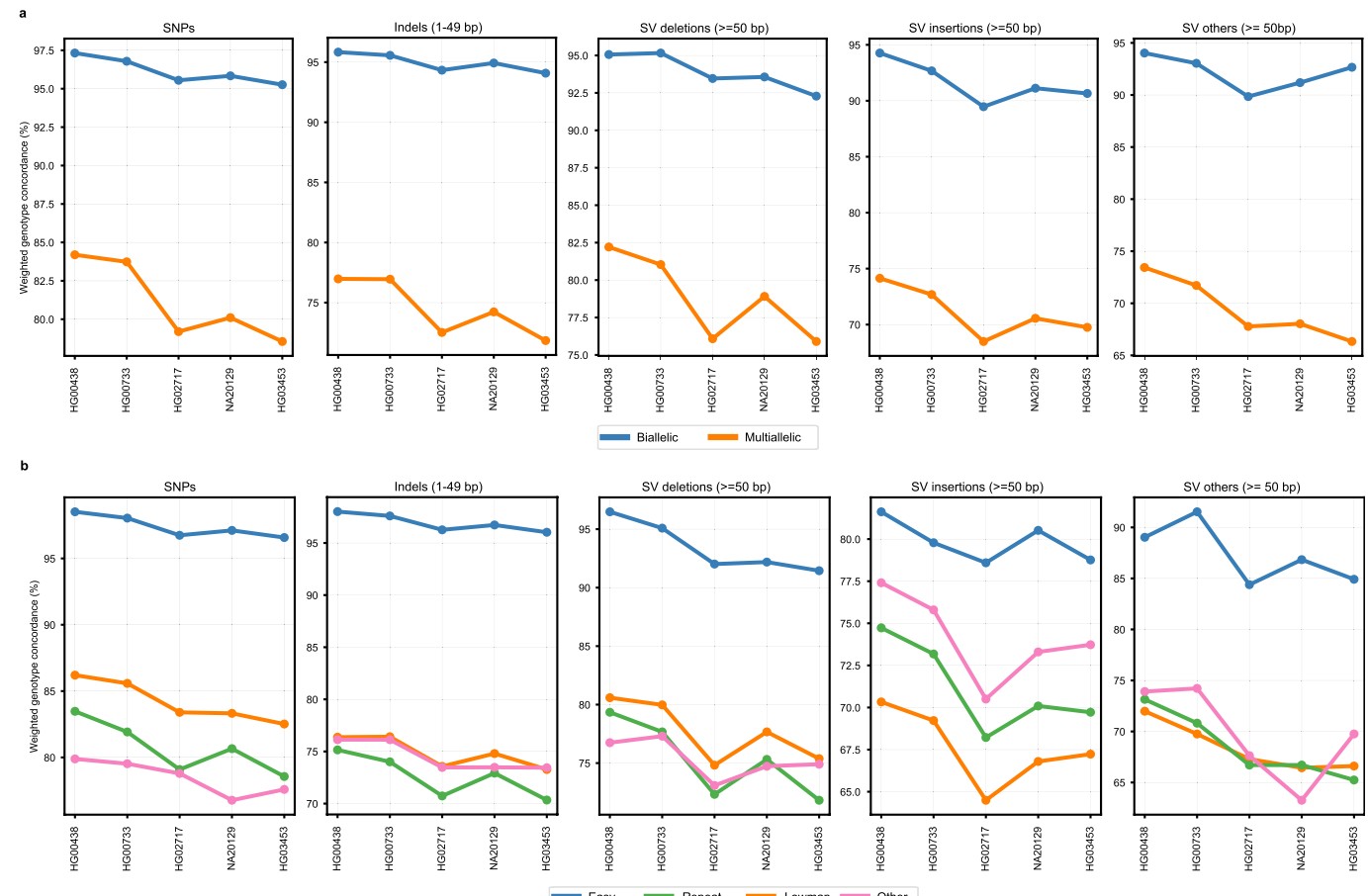

**Extended Data Fig. 9 | Leave-one-out experiment.** A leave-one-out experiment was conducted by repeatedly removing one of the assembly-samples from the panel VCF and genotyping it based on the remaining samples. Plots show the resulting weighted genotype concordances for different variant allele classes. **a**, weighted genotype concordances are stratified by graph complexity: biallelic regions of the MC graph include only bubbles with two branches, and multiallelic regions include all bubbles with > 2 different alternative paths defined by the 88 haplotypes. **b**, results of the same experiment stratified by different genomic regions defined by the GIAB.

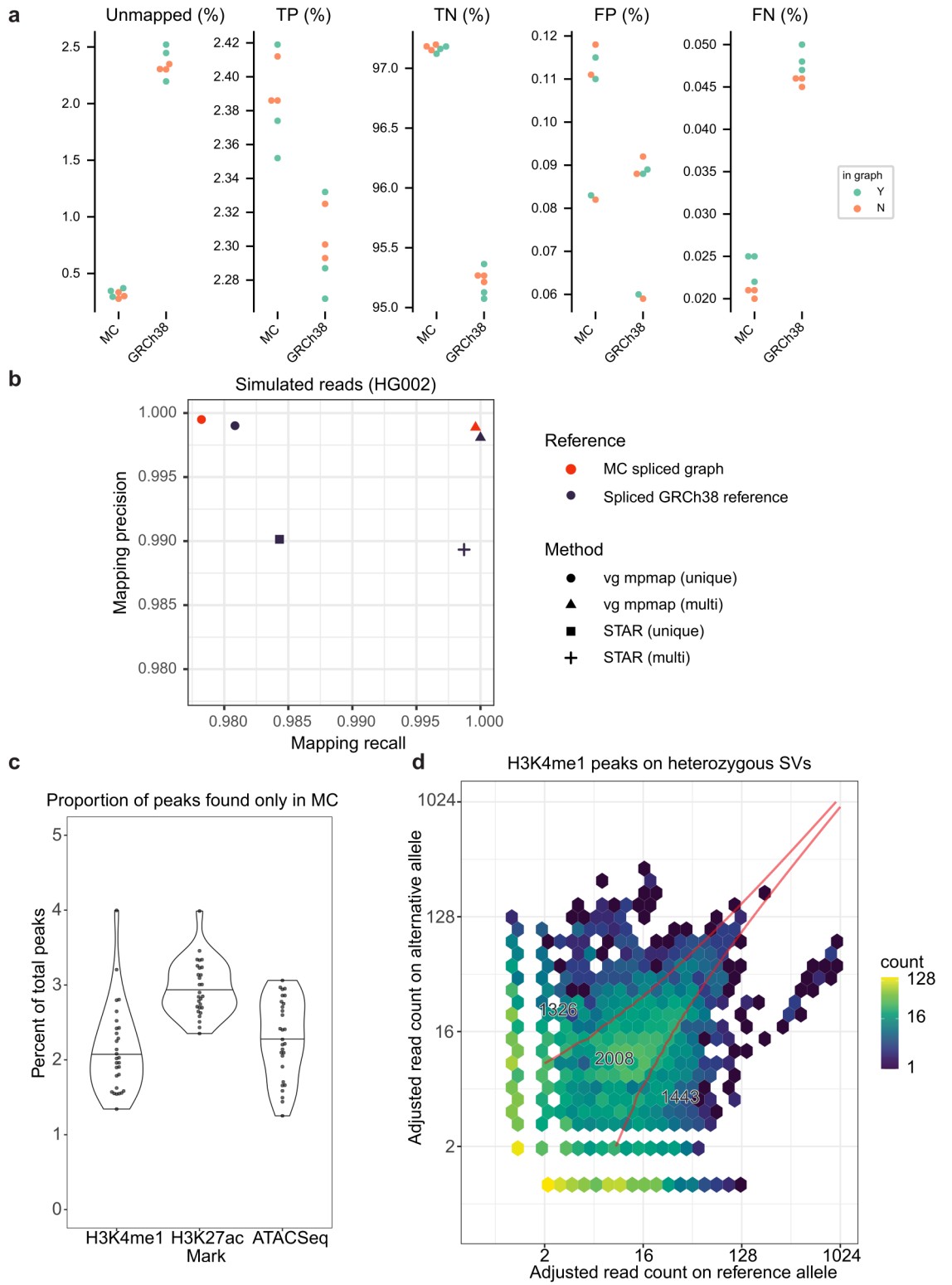

**Extended Data Fig. 10 | Additional applications supported by the pangenome reference. a**, Performance of read alignment in VNTR regions using the MC graph versus GRCh38. All statistics are expressed relative to the total number of reads simulated from each genome. **b**, Performance of RNA-seq read alignment. Mapping rate and false discovery rate are stratified by mapping quality producing the curves shown. The MC graph is compared to a graph derived from the 1KG variant calls and to GRCh38. Each reference is augmented with splice junctions. vg mpmap was used to map to the graphs,

and STAR was used to map to the linear reference. **c**, Proportion of all ChIP-seq peaks that are called only in the MC graph. Each data point represents samples that were assayed for H3K4me1, H3K27ac histone marks or chromatin accessibility using ATAC-seq. **d**, H3K4me1 peaks that overlap an SV for which the sample is heterozygous. The reads within the peak are partitioned between the SV or reference allele. The red boundary represents regions where a binomial test assigns a peak to the SV allele, both alleles, or the reference allele.

Number of SVs per sample

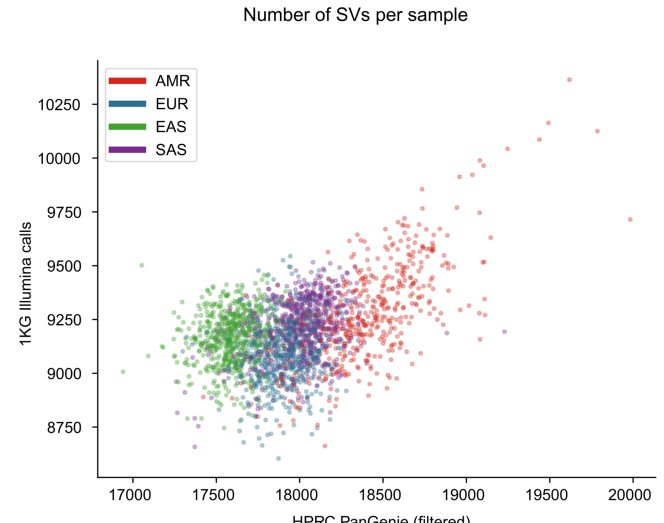

Number of SVs per sample

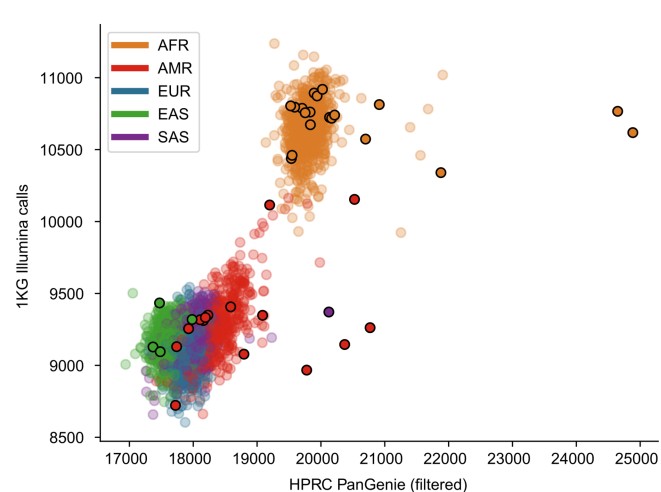

**Extended Data Fig. 11 | Number of SVs per sample in the HPRC PanGenie filtered set as well as the 1KG Illumina calls for all 3,202 1KG samples.** Samples are coloured by superpopulation. The left plot excludes the african superpopulation, while the right plot shows the same results including african samples and including the assembly samples present in the graph (marked by a black circle).

# Reporting Summary

## Statistics

For all statistical analyses, confirm that the following items are present in the figure legend, table legend, main text, or Methods section.

| n/a | Confirmed | |
|-----|-----------|---|
| ☐ | ☒ | The exact sample size (*n*) for each experimental group/condition, given as a discrete number and unit of measurement |
| ☐ | ☒ | A statement on whether measurements were taken from distinct samples or whether the same sample was measured repeatedly |
| ☐ | ☒ | The statistical test(s) used AND whether they are one- or two-sided *Only common tests should be described solely by name; describe more complex techniques in the Methods section.* |
| ☐ | ☒ | A description of all covariates tested |
| ☐ | ☒ | A description of any assumptions or corrections, such as tests of normality and adjustment for multiple comparisons |
| ☐ | ☒ | A full description of the statistical parameters including central tendency (e.g. means) or other basic estimates (e.g. regression coefficient) AND variation (e.g. standard deviation) or associated estimates of uncertainty (e.g. confidence intervals) |
| ☐ | ☒ | For null hypothesis testing, the test statistic (e.g. *F*, *t*, *r*) with confidence intervals, effect sizes, degrees of freedom and *P* value noted *Give P values as exact values whenever suitable.* |
| ☒ | ☐ | For Bayesian analysis, information on the choice of priors and Markov chain Monte Carlo settings |
| ☒ | ☐ | For hierarchical and complex designs, identification of the appropriate level for tests and full reporting of outcomes |
| ☐ | ☒ | Estimates of effect sizes (e.g. Cohen's *d*, Pearson's *r*), indicating how they were calculated |

*Our web collection on statistics for biologists contains articles on many of the points above.*

## Software and code

Policy information about availability of computer code

| Data collection | No software was used |
|---|---|

| Data analysis | abPOA https://github.com/yangao07/abPOA<br>ABRA v2.23 https://github.com/mozack/abra<br>Bandage v0.8.1 https://rrwick.github.io/Bandage/<br>bcftools v1.16 https://samtools.github.io/bcftools/<br>BEDTools v2.28.0, v2.30.0 https://bedtools.readthedocs.io/en/latest/<br>Bowtie2 v2.4.5 https://github.com/BenLangmead/bowtie2<br>BWA-MEMv0.7.17-r1188 https://github.com/lh3/bwa<br>Cactus 6cd9a42 https://github.com/ComparativeGenomicsToolkit/cactus<br>Comparative Annotation Toolkit (CAT) eb2fc87 https://github.com/ComparativeGenomicsToolkit/Comparative-Annotation-Toolkit<br>danbing-tk v1.3 https://github.com/ChaissonLab/danbing-tk<br>DeepVariant v1.3.0 https://github.com/google/deepvariant<br>Dfam v3.3 https://www.dfam.org/home<br>Dipcall v0.3 https://github.com/lh3/dipcall<br>dna-brnn v0.1 https://github.com/lh3/dna-nn<br>Exact Tandem Repeat Finder (ETRF) fc059d5 https://github.com/lh3/etrf<br>Exonerate https://www.ebi.ac.uk/about/vertebrate-genomics/software/exonerate<br>Flagger v0.1 https://github.com/mobinasri/flagger<br>FreeBayes v1.2.0 https://github.com/freebayes/freebayes<br>GATK v3.8.1 https://gatk.broadinstitute.org/hc/en-us |
|---|---|

```
gfabase          v0.6.0        https://github.com/mlin/gfabase
GFAffix          v0.1.3        https://github.com/marschall-lab/GFAffix
GFAtools         v0.5          https://github.com/lh3/gfatools
GffRead          v0.12.7       https://github.com/gpertea/gffread
gggenes          v0.4.1        https://github.com/wilkox/gggenes
Graph Peak Caller    v1.2.3    https://github.com/uio-bmi/graph_peak_caller
GraphAligner     v1.0.13       https://github.com/maickrau/GraphAligner
hal2vg                         https://github.com/ComparativeGenomicsToolkit/hal2vg
Hall-lab pipeline    830260a   https://github.com/hall-lab/competitive-alignment/blob/master/call_assembly_variants.wdl
halLiftover                    https://github.com/ComparativeGenomicsToolkit/hal
hap.py           v3.15         https://github.com/Illumina/hap.py
HiFiAdapterFilt  64d1c7b       https://github.com/sheinasim/HiFiAdapterFilt
hifiasm          v0.14, v0.14.1    https://github.com/chhylp123/hifiasm
Illumina's Dragen platform     v3.7.5     https://support.illumina.com/content/dam/illumina-support/help/
Illumina_DRAGEN_Bio_IT_Platform_v3_7_1000000141465/Content/In/Informatics/DRAGEN/GraphMapper_fDG.htm
Integrative Genome Browser (IGV)          https://software.broadinstitute.org/software/igv/
Iris             v1.0.4        https://github.com/mkirsche/Iris
Liftoff          v1.6.3        https://github.com/agshumate/Liftoff
MAFFT                          https://mafft.cbrc.jp/alignment/software/
Merqury          261b085       https://github.com/marbl/merqury
meryl            6d396a0       https://github.com/marbl/meryl
Minigraph        v0.14, v0.18      https://github.com/lh3/minigraph
minimap2         v2.21, v2.1, v2.2.4    https://github.com/lh3/minimap2
NCBI/RMBLAST     v2.10.0       http://www.repeatmasker.org/rmblast/
odgi pangenome analysis toolkit           https://github.com/pangenome/odgi
panacus          1eeb6d0       https://github.com/marschall-lab/panacus
PanGenie         v1.0.0        https://github.com/eblerjana/pangenie
PanGenome Graph Builder (PGGB)    v0.2.0+531f85f        https://github.com/pangenome/pggb
PAV              v0.9.1        https://github.com/EichlerLab/pav
PBSV             v2.6.2        https://github.com/PacificBiosciences/pbsv
Pstools          v0.1          https://github.com/shilpagarg/pstools
QUAST            v5.0.2        https://quast.sourceforge.net/
RepeatMasker     v4.1.2-p1     https://www.repeatmasker.org/
RPVG             1d91a9e       https://github.com/jonassibbesen/rpvg
RSEM             v1.3.3        https://github.com/deweylab/RSEM
RTG Tools        v3.12.1       https://github.com/RealTimeGenomics/rtg-tools
Salmon           v1.9.0        https://github.com/COMBINE-lab/salmon
Samtools         v1.3.1, v1.15      https://github.com/samtools/samtools
SDUST            v0.1          https://github.com/lh3/sdust
Secphase         v0.1          https://github.com/mobinasri/secphase
sedef                          https://github.com/vpc-ccg/sedef
seqtk            v1.3          https://github.com/lh3/seqtk
seqwish                        https://github.com/ekg/seqwish
slivar           v0.2.7        https://github.com/brentp/slivar
smoothxg                       https://github.com/pangenome/smoothxg
Sniffles         v1.0.12b      https://github.com/fritzsedlazeck/Sniffles
STAR             v2.7.10a      https://github.com/alexdobin/STAR
SVIM             v2.0.0        https://github.com/eldariont/svim
SVIM-asm         v1.0.2        https://github.com/eldariont/svim-asm
svtools                        https://github.com/hall-lab/svtools
Tandem Repeat Finder (TRF)     v4.09      https://tandem.bu.edu/trf/trfdesc.html
truvari          v3.2.0, v3.1.0    https://github.com/ACEnglish/truvari
vcfbub           v0.1.0        https://github.com/pangenome/vcfbub
vcflib                         https://github.com/vcflib/vcflib
vg, vg giraffe   v1.33.0, v1.36.0, v1.37.0, v1.38.0, v1.39.0, v1.43.0, 2cea1e2, c0c4816        https://github.com/vgteam/vg
wfmash                         https://github.com/waveygang/wfmash
WhatsHap         v1.1          https://whatshap.readthedocs.io/en/latest/
winnowmap        v2.03         https://github.com/marbl/Winnowmap
Yak              v0.1          https://github.com/lh3/yak
```

For manuscripts utilizing custom algorithms or software that are central to the research but not yet described in published literature, software must be made available to editors and reviewers. We strongly encourage code deposition in a community repository (e.g. GitHub). See the Nature Portfolio guidelines for submitting code & software for further information.

# Data

Policy information about availability of data

All manuscripts must include a data availability statement. This statement should provide the following information, where applicable:

- Accession codes, unique identifiers, or web links for publicly available datasets
- A description of any restrictions on data availability
- For clinical datasets or third party data, please ensure that the statement adheres to our policy

Sequencing data, assemblies, and pangenomes produced by the HPRC are available in the AnVIL (https://anvilproject.org/) in the AnVIL_HPRC workspace. Data is also available as part of the AWS Open Data Program (https://registry.opendata.aws/) in the human-pangenomics S3 bucket (https://s3-us-west-2.amazonaws.com/human-pangenomics/index.html). In addition, data is uploaded to INSDC for long term storage and availability. Supporting information about the data (including

## Human research participants

Policy information about studies involving human research participants and Sex and Gender in Research.

| | |
|---|---|
| Reporting on sex and gender | This study used cell lines derived from both male (19) and female (28) donors (47 in total). For 31 samples, we performed whole genome sequencing in this study and for all other samples, sequencing data was taken from other studies referenced in this manuscript. Lymphoblastoid cell lines (LCLs) used for sequencing from the 1000 Genomes Project collection were obtained from the NHGRI Sample Repository for Human Genetic Research and HG002 (GM24385) and HG005 (GM24631) LCLs were obtained from the NIGMS Human Genetic Cell Repository at the Coriell Institute for Medical Research. Therefore, this study is exempt from human subjects research since the proposed work involves the collection or study of data or specimens that are already publicly available. |
| Population characteristics | See above |
| Recruitment | See above |
| Ethics oversight | *Identify the organization(s) that approved the study protocol.* |

Note that full information on the approval of the study protocol must also be provided in the manuscript.

# Field-specific reporting

Please select the one below that is the best fit for your research. If you are not sure, read the appropriate sections before making your selection.

☒ Life sciences    ☐ Behavioural & social sciences    ☐ Ecological, evolutionary & environmental sciences

For a reference copy of the document with all sections, see nature.com/documents/nr-reporting-summary-flat.pdf

# Life sciences study design

All studies must disclose on these points even when the disclosure is negative.

| | |
|---|---|
| Sample size | Decided with respect to HPRC's year1-2 funding |
| Data exclusions | Candidates without both parental data, or optimal passage biosamples, were excluded. |
| Replication | Experiments were computational so replication is not applicable. For reproducibility all of the data sets and codes/workflows are publicly available. The corresponding links are mentioned in data availability and code availability statements. |
| Randomization | The samples were not allocated into different experimental groups. |
| Blinding | Blinding is not relevant to this study since the samples in this study were selected solely based on the genetic diversity. Comparison between two or more experimental groups was not the purpose of the study so blinding was not necessary. |

# Behavioural & social sciences study design

All studies must disclose on these points even when the disclosure is negative.

| | |
|---|---|
| Study description | *Briefly describe the study type including whether data are quantitative, qualitative, or mixed-methods (e.g. qualitative cross-sectional, quantitative experimental, mixed-methods case study).* |
| Research sample | *State the research sample (e.g. Harvard university undergraduates, villagers in rural India) and provide relevant demographic information (e.g. age, sex) and indicate whether the sample is representative. Provide a rationale for the study sample chosen. For studies involving existing datasets, please describe the dataset and source.* |
| Sampling strategy | *Describe the sampling procedure (e.g. random, snowball, stratified, convenience). Describe the statistical methods that were used to predetermine sample size OR if no sample-size calculation was performed, describe how sample sizes were chosen and provide a rationale for why these sample sizes are sufficient. For qualitative data, please indicate whether data saturation was considered, and what criteria were used to decide that no further sampling was needed.* |
| Data collection | *Provide details about the data collection procedure, including the instruments or devices used to record the data (e.g. pen and paper, computer, eye tracker, video or audio equipment) whether anyone was present besides the participant(s) and the researcher, and whether the researcher was blind to experimental condition and/or the study hypothesis during data collection.* |
| Timing | *Indicate the start and stop dates of data collection. If there is a gap between collection periods, state the dates for each sample cohort.* |
| Data exclusions | *If no data were excluded from the analyses, state so OR if data were excluded, provide the exact number of exclusions and the rationale behind them, indicating whether exclusion criteria were pre-established.* |
| Non-participation | *State how many participants dropped out/declined participation and the reason(s) given OR provide response rate OR state that no participants dropped out/declined participation.* |
| Randomization | *If participants were not allocated into experimental groups, state so OR describe how participants were allocated to groups, and if allocation was not random, describe how covariates were controlled.* |

# Ecological, evolutionary & environmental sciences study design

All studies must disclose on these points even when the disclosure is negative.

| | |
|---|---|
| Study description | *Briefly describe the study. For quantitative data include treatment factors and interactions, design structure (e.g. factorial, nested, hierarchical), nature and number of experimental units and replicates.* |
| Research sample | *Describe the research sample (e.g. a group of tagged Passer domesticus, all Stenocereus thurberi within Organ Pipe Cactus National Monument), and provide a rationale for the sample choice. When relevant, describe the organism taxa, source, sex, age range and any manipulations. State what population the sample is meant to represent when applicable. For studies involving existing datasets, describe the data and its source.* |
| Sampling strategy | *Note the sampling procedure. Describe the statistical methods that were used to predetermine sample size OR if no sample-size calculation was performed, describe how sample sizes were chosen and provide a rationale for why these sample sizes are sufficient.* |
| Data collection | *Describe the data collection procedure, including who recorded the data and how.* |
| Timing and spatial scale | *Indicate the start and stop dates of data collection, noting the frequency and periodicity of sampling and providing a rationale for these choices. If there is a gap between collection periods, state the dates for each sample cohort. Specify the spatial scale from which the data are taken* |
| Data exclusions | *If no data were excluded from the analyses, state so OR if data were excluded, describe the exclusions and the rationale behind them, indicating whether exclusion criteria were pre-established.* |
| Reproducibility | *Describe the measures taken to verify the reproducibility of experimental findings. For each experiment, note whether any attempts to repeat the experiment failed OR state that all attempts to repeat the experiment were successful.* |
| Randomization | *Describe how samples/organisms/participants were allocated into groups. If allocation was not random, describe how covariates were controlled. If this is not relevant to your study, explain why.* |
| Blinding | *Describe the extent of blinding used during data acquisition and analysis. If blinding was not possible, describe why OR explain why blinding was not relevant to your study.* |

Did the study involve field work?  ☐ Yes  ☐ No

# Field work, collection and transport

| | |
|---|---|
| Field conditions | *Describe the study conditions for field work, providing relevant parameters (e.g. temperature, rainfall).* |
| Location | *State the location of the sampling or experiment, providing relevant parameters (e.g. latitude and longitude, elevation, water depth).* |
| Access & import/export | *Describe the efforts you have made to access habitats and to collect and import/export your samples in a responsible manner and in compliance with local, national and international laws, noting any permits that were obtained (give the name of the issuing authority, the date of issue, and any identifying information).* |
| Disturbance | *Describe any disturbance caused by the study and how it was minimized.* |

# Reporting for specific materials, systems and methods

We require information from authors about some types of materials, experimental systems and methods used in many studies. Here, indicate whether each material, system or method listed is relevant to your study. If you are not sure if a list item applies to your research, read the appropriate section before selecting a response.

## Materials & experimental systems

| n/a | Involved in the study |
|---|---|
| ☒ | ☐ Antibodies |
| ☐ | ☒ Eukaryotic cell lines |
| ☒ | ☐ Palaeontology and archaeology |
| ☒ | ☐ Animals and other organisms |
| ☒ | ☐ Clinical data |
| ☒ | ☐ Dual use research of concern |

## Methods

| n/a | Involved in the study |
|---|---|
| ☐ | ☒ ChIP-seq |
| ☒ | ☐ Flow cytometry |
| ☒ | ☐ MRI-based neuroimaging |

## Antibodies

| | |
|---|---|
| Antibodies used | *Describe all antibodies used in the study; as applicable, provide supplier name, catalog number, clone name, and lot number.* |
| Validation | *Describe the validation of each primary antibody for the species and application, noting any validation statements on the manufacturer's website, relevant citations, antibody profiles in online databases, or data provided in the manuscript.* |

## Eukaryotic cell lines

Policy information about cell lines and Sex and Gender in Research

Cell line source(s)

Lymphoblastoid cell lines used for sequencing from the 1KG collection were obtained from the NHGRI Sample Repository for Human Genetic Research at the Coriell Institute for Medical Research.
HG002 (GM24385) and HG005 (GM24631) lymphoblastoid cell lines were obtained from the NIGMS Human Genetic Cell Repository at the Coriell Institute for Medical Research.

```
Sample Cohort Sex
--------------------------
HG01123 HPRC female
HG01258 HPRC male
HG01358 HPRC male
HG01361 HPRC female
HG01891 HPRC female
HG02257 HPRC female
HG02486 HPRC male
HG02559 HPRC female
HG02572 HPRC male
HG03516 HPRC female
HG00438 HPRC female
HG00621 HPRC male
HG00673 HPRC male
HG00735 HPRC female
HG00741 HPRC female
HG01071 HPRC female
HG01106 HPRC male
HG01175 HPRC female
HG01928 HPRC male
```

HG01952 HPRC male
HG01978 HPRC female
HG02148 HPRC female
HG02622 HPRC female
HG02630 HPRC female
HG02717 HPRC male
HG02886 HPRC female
HG03453 HPRC female
HG03540 HPRC female
HG03579 HPRC male
HG002 HPRC_PLUS male
HG005 HPRC_PLUS male

| | |
|---|---|
| Authentication | This study involved the use of authenticated cell lines that are publicly available from the NHGRI Sample Repository for Human Genetic Research and the NIGMS Human Genetic Cell Repository at the Coriell Institute for Medical Research. Cell line identity was confirmed using a multiplex PCR assay for six autosomal microsatellite markers. |
| Mycoplasma contamination | All cell lines tested negative for mycoplasma contamination. |
| Commonly misidentified lines (See ICLAC register) | No misidentified cell line was used in this study. |

## Palaeontology and Archaeology

| | |
|---|---|
| Specimen provenance | *Provide provenance information for specimens and describe permits that were obtained for the work (including the name of the issuing authority, the date of issue, and any identifying information). Permits should encompass collection and, where applicable, export.* |
| Specimen deposition | *Indicate where the specimens have been deposited to permit free access by other researchers.* |
| Dating methods | *If new dates are provided, describe how they were obtained (e.g. collection, storage, sample pretreatment and measurement), where they were obtained (i.e. lab name), the calibration program and the protocol for quality assurance OR state that no new dates are provided.* |

☐ Tick this box to confirm that the raw and calibrated dates are available in the paper or in Supplementary Information.

| | |
|---|---|
| Ethics oversight | *Identify the organization(s) that approved or provided guidance on the study protocol, OR state that no ethical approval or guidance was required and explain why not.* |

Note that full information on the approval of the study protocol must also be provided in the manuscript.

## Animals and other research organisms

Policy information about studies involving animals; ARRIVE guidelines recommended for reporting animal research, and Sex and Gender in Research

| | |
|---|---|
| Laboratory animals | *For laboratory animals, report species, strain and age OR state that the study did not involve laboratory animals.* |
| Wild animals | *Provide details on animals observed in or captured in the field; report species and age where possible. Describe how animals were caught and transported and what happened to captive animals after the study (if killed, explain why and describe method; if released, say where and when) OR state that the study did not involve wild animals.* |
| Reporting on sex | *Indicate if findings apply to only one sex; describe whether sex was considered in study design, methods used for assigning sex. Provide data disaggregated for sex where this information has been collected in the source data as appropriate; provide overall numbers in this Reporting Summary. Please state if this information has not been collected. Report sex-based analyses where performed, justify reasons for lack of sex-based analysis.* |
| Field-collected samples | *For laboratory work with field-collected samples, describe all relevant parameters such as housing, maintenance, temperature, photoperiod and end-of-experiment protocol OR state that the study did not involve samples collected from the field.* |
| Ethics oversight | *Identify the organization(s) that approved or provided guidance on the study protocol, OR state that no ethical approval or guidance was required and explain why not.* |

Note that full information on the approval of the study protocol must also be provided in the manuscript.

## Clinical data

Policy information about clinical studies
All manuscripts should comply with the ICMJE guidelines for publication of clinical research and a completed CONSORT checklist must be included with all submissions.

| | |
|---|---|
| Clinical trial registration | *Provide the trial registration number from ClinicalTrials.gov or an equivalent agency.* |

| Study protocol | *Note where the full trial protocol can be accessed OR if not available, explain why.* |
| Data collection | *Describe the settings and locales of data collection, noting the time periods of recruitment and data collection.* |
| Outcomes | *Describe how you pre-defined primary and secondary outcome measures and how you assessed these measures.* |

# Dual use research of concern

Policy information about dual use research of concern

## Hazards

Could the accidental, deliberate or reckless misuse of agents or technologies generated in the work, or the application of information presented in the manuscript, pose a threat to:

| No | Yes | |
|---|---|---|
| ☒ | ☐ | Public health |
| ☒ | ☐ | National security |
| ☒ | ☐ | Crops and/or livestock |
| ☒ | ☐ | Ecosystems |
| ☒ | ☐ | Any other significant area |

## Experiments of concern

Does the work involve any of these experiments of concern:

| No | Yes | |
|---|---|---|
| ☒ | ☐ | Demonstrate how to render a vaccine ineffective |
| ☒ | ☐ | Confer resistance to therapeutically useful antibiotics or antiviral agents |
| ☒ | ☐ | Enhance the virulence of a pathogen or render a nonpathogen virulent |
| ☒ | ☐ | Increase transmissibility of a pathogen |
| ☒ | ☐ | Alter the host range of a pathogen |
| ☒ | ☐ | Enable evasion of diagnostic/detection modalities |
| ☒ | ☐ | Enable the weaponization of a biological agent or toxin |
| ☒ | ☐ | Any other potentially harmful combination of experiments and agents |

# ChIP-seq

## Data deposition

☒ Confirm that both raw and final processed data have been deposited in a public database such as GEO.

☒ Confirm that you have deposited or provided access to graph files (e.g. BED files) for the called peaks.

| Data access links | https://doi.org/10.5281/ZENODO.6564396 |
| *May remain private before publication.* | |

| Files in database submission | Rscripts/<br>Rscripts/allele_support.R<br>Rscripts/peak_replication.R<br>peaks/<br>peaks/h3k4me1_ref_peaks/<br>peaks/atac_overlaps/<br>peaks/do.sh<br>peaks/h3k4me1_hprc_peaks/<br>peaks/counts.csv<br>peaks/atac_hprc_peaks/<br>peaks/subtract_peaks.sh<br>peaks/atac_ref_peaks/<br>peaks/h3k27ac_ref_peaks/<br>peaks/h3k27ac_hprc_peaks/<br>peaks/h3k4me1_overlaps/<br>peaks/h3k27ac_overlaps/<br>peaks/h3k27ac_overlaps/EU21_Flu_2-571406_ChIPmentation-<br>H3K27ac_A00266_0263_3_S24_L003_I_peaks.narrowPeak_ref-only.bed<br>peaks/h3k27ac_overlaps/AF34_Flu_2-571349_ChIPmentation-<br>H3K27ac_A00266_0263_3_S17_L003_I_peaks.narrowPeak_pers-only.bed |

```
peaks/h3k27ac_overlaps/AF20_Flu_2-571333_ChIPmentation-H3K27ac_A00266_0263_3_S1_L003_I_peaks.narrowPeak_ref-only.bed
peaks/h3k27ac_overlaps/AF20_Flu_2-571333_ChIPmentation-H3K27ac_A00266_0263_3_S1_L003_I_peaks.narrowPeak_pers-only.bed
peaks/h3k27ac_overlaps/EU05_Flu_2-571412_ChIPmentation-H3K27ac_A00266_0265_2_S6_L002_I_peaks.narrowPeak_pers-only.bed
peaks/h3k27ac_overlaps/EU13_Flu_2-571348_ChIPmentation-H3K27ac_A00266_0263_3_S16_L003_I_peaks.narrowPeak_ref-only.bed
peaks/h3k27ac_overlaps/AF08_Flu_2-571405_ChIPmentation-H3K27ac_A00266_0263_3_S23_L003_I_peaks.narrowPeak_intersect.bed
peaks/h3k27ac_overlaps/EU19_Flu_2-571338_ChIPmentation-H3K27ac_A00266_0263_3_S6_L003_I_peaks.narrowPeak_pers-only.bed
peaks/h3k27ac_overlaps/AF04_Flu_2-642070_ChIPmentation-H3K27ac_A00266_0265_2_S14_L002_I_peaks.narrowPeak_ref-only.bed
peaks/h3k27ac_overlaps/EU27_Flu_2-642077_ChIPmentation-H3K27ac_A00266_0265_2_S21_L002_I_peaks.narrowPeak_intersect.bed
peaks/h3k27ac_overlaps/EU39_Flu_2-571401_ChIPmentation-H3K27ac_A00266_0263_3_S19_L003_I_peaks.narrowPeak_ref-only.bed
peaks/h3k27ac_overlaps/AF30_Flu_2-669476_ChIPmentation-H3K27ac_A00266_0289_1_S12_L001_I_peaks.narrowPeak_intersect.bed
peaks/h3k27ac_overlaps/EU07_Flu_2-642067_ChIPmentation-H3K27ac_A00266_0265_2_S11_L002_I_peaks.narrowPeak_ref-only.bed
peaks/h3k27ac_overlaps/EU15_Flu_2-571342_ChIPmentation-H3K27ac_A00266_0263_3_S10_L003_I_peaks.narrowPeak_intersect.bed
peaks/h3k27ac_overlaps/EU13_Flu_2-571348_ChIPmentation-H3K27ac_A00266_0263_3_S16_L003_I_peaks.narrowPeak_intersect.bed
peaks/h3k27ac_overlaps/AF12_Flu_2-571340_ChIPmentation-H3K27ac_A00266_0263_3_S8_L003_I_peaks.narrowPeak_pers-only.bed
peaks/h3k27ac_overlaps/AF36_Flu_2-571346_ChIPmentation-H3K27ac_A00266_0263_3_S14_L003_I_peaks.narrowPeak_intersect.bed
peaks/h3k27ac_overlaps/AF04_Flu_2-642070_ChIPmentation-H3K27ac_A00266_0265_2_S14_L002_I_peaks.narrowPeak_pers-only.bed
peaks/h3k27ac_overlaps/AF22_Flu_2-669473_ChIPmentation-H3K27ac_A00266_0289_1_S9_L001_I_peaks.narrowPeak_pers-only.bed
peaks/h3k27ac_overlaps/EU47_Flu_2-669480_ChIPmentation-H3K27ac_A00266_0289_1_S16_L001_I_peaks.narrowPeak_pers-only.bed
peaks/h3k27ac_overlaps/AF34_Flu_2-571349_ChIPmentation-H3K27ac_A00266_0263_3_S17_L003_I_peaks.narrowPeak_ref-only.bed
peaks/h3k27ac_overlaps/EU03_Flu_2-571411_ChIPmentation-H3K27ac_A00266_0265_2_S5_L002_I_peaks.narrowPeak_ref-only.bed
peaks/h3k27ac_overlaps/EU21_Flu_2-571406_ChIPmentation-H3K27ac_A00266_0263_3_S24_L003_I_peaks.narrowPeak_intersect.bed
peaks/h3k27ac_overlaps/EU21_Flu_2-571406_ChIPmentation-H3K27ac_A00266_0263_3_S24_L003_I_peaks.narrowPeak_pers-only.bed
peaks/h3k27ac_overlaps/AF22_Flu_2-669473_ChIPmentation-H3K27ac_A00266_0289_1_S9_L001_I_peaks.narrowPeak_intersect.bed
peaks/h3k27ac_overlaps/EU47_Flu_2-669480_ChIPmentation-H3K27ac_A00266_0289_1_S16_L001_I_peaks.narrowPeak_intersect.bed
peaks/h3k27ac_overlaps/AF04_Flu_2-642070_ChIPmentation-H3K27ac_A00266_0265_2_S14_L002_I_peaks.narrowPeak_intersect.bed
peaks/h3k27ac_overlaps/AF36_Flu_2-571346_ChIPmentation-H3K27ac_A00266_0263_3_S14_L003_I_peaks.narrowPeak_pers-only.bed
peaks/h3k27ac_overlaps/AF12_Flu_2-571340_ChIPmentation-H3K27ac_A00266_0263_3_S8_L003_I_peaks.narrowPeak_intersect.bed
peaks/h3k27ac_overlaps/AF18_Flu_2-642068_ChIPmentation-H3K27ac_A00266_0265_2_S12_L002_I_peaks.narrowPeak_ref-only.bed
peaks/h3k27ac_overlaps/EU19_Flu_2-571338_ChIPmentation-H3K27ac_A00266_0263_3_S6_L003_I_peaks.narrowPeak_ref-only.bed
peaks/h3k27ac_overlaps/EU13_Flu_2-571348_ChIPmentation-H3K27ac_A00266_0263_3_S16_L003_I_peaks.narrowPeak_pers-only.bed
peaks/h3k27ac_overlaps/EU15_Flu_2-571342_ChIPmentation-H3K27ac_A00266_0263_3_S10_L003_I_peaks.narrowPeak_pers-only.bed
peaks/h3k27ac_overlaps/AF30_Flu_2-669476_ChIPmentation-H3K27ac_A00266_0289_1_S12_L001_I_peaks.narrowPeak_pers-only.bed
peaks/h3k27ac_overlaps/EU27_Flu_2-642077_ChIPmentation-H3K27ac_A00266_0265_2_S21_L002_I_peaks.narrowPeak_pers-only.bed
peaks/h3k27ac_overlaps/EU19_Flu_2-571338_ChIPmentation-H3K27ac_A00266_0263_3_S6_L003_I_peaks.narrowPeak_intersect.bed
peaks/h3k27ac_overlaps/AF08_Flu_2-571405_ChIPmentation-H3K27ac_A00266_0263_3_S23_L003_I_peaks.narrowPeak_pers-only.bed
peaks/h3k27ac_overlaps/EU33_Flu_2-669466_ChIPmentation-H3K27ac_A00266_0289_1_S2_L001_I_peaks.narrowPeak_ref-only.bed
peaks/h3k27ac_overlaps/AF30_Flu_2-669476_ChIPmentation-H3K27ac_A00266_0289_1_S12_L001_I_peaks.narrowPeak_ref-only.bed
peaks/h3k27ac_overlaps/EU05_Flu_2-571412_ChIPmentation-H3K27ac_A00266_0265_2_S6_L002_I_peaks.narrowPeak_intersect.bed
```

peaks/h3k27ac_overlaps/AF20_Flu_2-571333_ChIPmentation-H3K27ac_A00266_0263_3_S1_L003_I_peaks.narrowPeak_intersect.bed
peaks/h3k27ac_overlaps/AF34_Flu_2-571349_ChIPmentation-H3K27ac_A00266_0263_3_S17_L003_I_peaks.narrowPeak_intersect.bed
peaks/h3k27ac_overlaps/EU39_Flu_2-571401_ChIPmentation-H3K27ac_A00266_0263_3_S19_L003_I_peaks.narrowPeak_intersect.bed
peaks/h3k27ac_overlaps/EU03_Flu_2-571411_ChIPmentation-H3K27ac_A00266_0265_2_S5_L002_I_peaks.narrowPeak_pers-only.bed
peaks/h3k27ac_overlaps/AF24_Flu_2-642069_ChIPmentation-H3K27ac_A00266_0265_2_S13_L002_I_peaks.narrowPeak_intersect.bed
peaks/h3k27ac_overlaps/EU15_Flu_2-571342_ChIPmentation-H3K27ac_A00266_0263_3_S10_L003_I_peaks.narrowPeak_ref-only.bed
peaks/h3k27ac_overlaps/AF12_Flu_2-571340_ChIPmentation-H3K27ac_A00266_0263_3_S8_L003_I_peaks.narrowPeak_ref-only.bed
peaks/h3k27ac_overlaps/AF26_Flu_2-571344_ChIPmentation-H3K27ac_A00266_0263_3_S12_L003_I_peaks.narrowPeak_intersect.bed
peaks/h3k27ac_overlaps/EU33_Flu_2-669466_ChIPmentation-H3K27ac_A00266_0289_1_S2_L001_I_peaks.narrowPeak_pers-only.bed
peaks/h3k27ac_overlaps/EU07_Flu_2-642067_ChIPmentation-H3K27ac_A00266_0265_2_S11_L002_I_peaks.narrowPeak_pers-only.bed
peaks/h3k27ac_overlaps/EU25_Flu_2-571343_ChIPmentation-H3K27ac_A00266_0263_3_S11_L003_I_peaks.narrowPeak_pers-only.bed
peaks/h3k27ac_overlaps/EU41_Flu_2-669478_ChIPmentation-H3K27ac_A00266_0289_1_S14_L001_I_peaks.narrowPeak_intersect.bed
peaks/h3k27ac_overlaps/AF38_Flu_2-669483_ChIPmentation-H3K27ac_A00266_0289_1_S19_L001_I_peaks.narrowPeak_intersect.bed
peaks/h3k27ac_overlaps/AF18_Flu_2-642068_ChIPmentation-H3K27ac_A00266_0265_2_S12_L002_I_peaks.narrowPeak_intersect.bed
peaks/h3k27ac_overlaps/AF08_Flu_2-571405_ChIPmentation-H3K27ac_A00266_0263_3_S23_L003_I_peaks.narrowPeak_ref-only.bed
peaks/h3k27ac_overlaps/EU09_Flu_2-571335_ChIPmentation-H3K27ac_A00266_0263_3_S3_L003_I_peaks.narrowPeak_intersect.bed
peaks/h3k27ac_overlaps/EU29_Flu_2-669467_ChIPmentation-H3K27ac_A00266_0289_1_S3_L001_I_peaks.narrowPeak_ref-only.bed
peaks/h3k27ac_overlaps/AF06_Flu_2-642071_ChIPmentation-H3K27ac_A00266_0265_2_S15_L002_I_peaks.narrowPeak_intersect.bed
peaks/h3k27ac_overlaps/EU29_Flu_2-669467_ChIPmentation-H3K27ac_A00266_0289_1_S3_L001_I_peaks.narrowPeak_intersect.bed
peaks/h3k27ac_overlaps/AF16_Flu_2-571407_ChIPmentation-H3K27ac_A00266_0265_2_S1_L002_I_peaks.narrowPeak_intersect.bed
peaks/h3k27ac_overlaps/EU41_Flu_2-669478_ChIPmentation-H3K27ac_A00266_0289_1_S14_L001_I_peaks.narrowPeak_ref-only.bed
peaks/h3k27ac_overlaps/AF28_Flu_2-571345_ChIPmentation-H3K27ac_A00266_0263_3_S13_L003_I_peaks.narrowPeak_intersect.bed
peaks/h3k27ac_overlaps/AF28_Flu_2-571345_ChIPmentation-H3K27ac_A00266_0263_3_S13_L003_I_peaks.narrowPeak_pers-only.bed
peaks/h3k27ac_overlaps/EU09_Flu_2-571335_ChIPmentation-H3K27ac_A00266_0263_3_S3_L003_I_peaks.narrowPeak_ref-only.bed
peaks/h3k27ac_overlaps/AF16_Flu_2-571407_ChIPmentation-H3K27ac_A00266_0265_2_S1_L002_I_peaks.narrowPeak_pers-only.bed
peaks/h3k27ac_overlaps/EU27_Flu_2-642077_ChIPmentation-H3K27ac_A00266_0265_2_S21_L002_I_peaks.narrowPeak_ref-only.bed
peaks/h3k27ac_overlaps/EU25_Flu_2-571343_ChIPmentation-H3K27ac_A00266_0263_3_S11_L003_I_peaks.narrowPeak_ref-only.bed
peaks/h3k27ac_overlaps/AF22_Flu_2-669473_ChIPmentation-H3K27ac_A00266_0289_1_S9_L001_I_peaks.narrowPeak_ref-only.bed
peaks/h3k27ac_overlaps/AF24_Flu_2-642069_ChIPmentation-H3K27ac_A00266_0265_2_S13_L002_I_peaks.narrowPeak_ref-only.bed
peaks/h3k27ac_overlaps/EU05_Flu_2-571412_ChIPmentation-H3K27ac_A00266_0265_2_S6_L002_I_peaks.narrowPeak_ref-only.bed
peaks/h3k27ac_overlaps/AF36_Flu_2-571346_ChIPmentation-H3K27ac_A00266_0263_3_S14_L003_I_peaks.narrowPeak_ref-only.bed
peaks/h3k27ac_overlaps/EU29_Flu_2-669467_ChIPmentation-H3K27ac_A00266_0289_1_S3_L001_I_peaks.narrowPeak_pers-only.bed
peaks/h3k27ac_overlaps/AF06_Flu_2-642071_ChIPmentation-H3K27ac_A00266_0265_2_S15_L002_I_peaks.narrowPeak_ref-only.bed
peaks/h3k27ac_overlaps/EU09_Flu_2-571335_ChIPmentation-H3K27ac_A00266_0263_3_S3_L003_I_peaks.narrowPeak_pers-only.bed
peaks/h3k27ac_overlaps/AF06_Flu_2-642071_ChIPmentation-H3K27ac_A00266_0265_2_S15_L002_I_peaks.narrowPeak_pers-only.bed
peaks/h3k27ac_overlaps/EU47_Flu_2-669480_ChIPmentation-H3K27ac_A00266_0289_1_S16_L001_I_peaks.narrowPeak_ref-only.bed
peaks/h3k27ac_overlaps/AF38_Flu_2-669483_ChIPmentation-H3K27ac_A00266_0289_1_S19_L001_I_peaks.narrowPeak_ref-only.bed
peaks/h3k27ac_overlaps/AF18_Flu_2-642068_ChIPmentation-H3K27ac_A00266_0265_2_S12_L002_I_peaks.narrowPeak_pers-only.bed

peaks/h3k27ac_overlaps/AF26_Flu_2-571344_ChIPmentation-H3K27ac_A00266_0263_3_S12_L003_I_peaks.narrowPeak_ref-only.bed
peaks/h3k27ac_overlaps/AF38_Flu_2-669483_ChIPmentation-H3K27ac_A00266_0289_1_S19_L001_I_peaks.narrowPeak_pers-only.bed
peaks/h3k27ac_overlaps/EU25_Flu_2-571343_ChIPmentation-H3K27ac_A00266_0263_3_S11_L003_I_peaks.narrowPeak_intersect.bed
peaks/h3k27ac_overlaps/EU41_Flu_2-669478_ChIPmentation-H3K27ac_A00266_0289_1_S14_L001_I_peaks.narrowPeak_pers-only.bed
peaks/h3k27ac_overlaps/EU33_Flu_2-669466_ChIPmentation-H3K27ac_A00266_0289_1_S2_L001_I_peaks.narrowPeak_intersect.bed
peaks/h3k27ac_overlaps/EU07_Flu_2-642067_ChIPmentation-H3K27ac_A00266_0265_2_S11_L002_I_peaks.narrowPeak_intersect.bed
peaks/h3k27ac_overlaps/AF16_Flu_2-571407_ChIPmentation-H3K27ac_A00266_0265_2_S1_L002_I_peaks.narrowPeak_ref-only.bed
peaks/h3k27ac_overlaps/AF28_Flu_2-571345_ChIPmentation-H3K27ac_A00266_0263_3_S13_L003_I_peaks.narrowPeak_ref-only.bed
peaks/h3k27ac_overlaps/AF26_Flu_2-571344_ChIPmentation-H3K27ac_A00266_0263_3_S12_L003_I_peaks.narrowPeak_pers-only.bed
peaks/h3k27ac_overlaps/EU03_Flu_2-571411_ChIPmentation-H3K27ac_A00266_0265_2_S5_L002_I_peaks.narrowPeak_intersect.bed
peaks/h3k27ac_overlaps/AF24_Flu_2-642069_ChIPmentation-H3K27ac_A00266_0265_2_S13_L002_I_peaks.narrowPeak_pers-only.bed
peaks/h3k27ac_overlaps/EU39_Flu_2-571401_ChIPmentation-H3K27ac_A00266_0263_3_S19_L003_I_peaks.narrowPeak_pers-only.bed
peaks/h3k4me1_overlaps/AF38_Flu_2-674837_ChIPmentation-H3K4me1_A00266_0301_2_S14_L002_I_peaks.narrowPeak_ref-only.bed
peaks/h3k4me1_overlaps/AF36_Flu_2-669593_ChIPmentation-H3K4me1_A00266_0289_2_S10_L002_I_peaks.narrowPeak_intersect.bed
peaks/h3k4me1_overlaps/AF24_Flu_2-674804_ChIPmentation-H3K4me1_A00266_0301_1_S5_L001_I_peaks.narrowPeak_intersect.bed
peaks/h3k4me1_overlaps/AF20_Flu_2-669580_ChIPmentation-H3K4me1_A00266_0289_1_S21_L001_I_peaks.narrowPeak_ref-only.bed
peaks/h3k4me1_overlaps/AF18_Flu_2-674803_ChIPmentation-H3K4me1_A00266_0301_1_S4_L001_I_peaks.narrowPeak_ref-only.bed
peaks/h3k4me1_overlaps/AF30_Flu_2-674830_ChIPmentation-H3K4me1_A00266_0301_2_S7_L002_I_peaks.narrowPeak_pers-only.bed
peaks/h3k4me1_overlaps/AF06_Flu_2-674806_ChIPmentation-H3K4me1_A00266_0301_1_S7_L001_I_peaks.narrowPeak_ref-only.bed
peaks/h3k4me1_overlaps/AF16_Flu_2-669654_ChIPmentation-H3K4me1_A00266_0289_2_S21_L002_I_peaks.narrowPeak_pers-only.bed
peaks/h3k4me1_overlaps/EU13_Flu_2-669595_ChIPmentation-H3K4me1A00266_0289_2_S12_L002_I_peaks.narrowPeak_pers-only.bed
peaks/h3k4me1_overlaps/EU29_Flu_2-674821_ChIPmentation-H3K4me1A00266_0301_1_S22_L001_I_peaks.narrowPeak_intersect.bed
peaks/h3k4me1_overlaps/EU27_Flu_2-674815_ChIPmentation-H3K4me1A00266_0301_1_S16_L001_I_peaks.narrowPeak_ref-only.bed
peaks/h3k4me1_overlaps/EU09_Flu_2-669582_ChIPmentation-H3K4me1A00266_0289_1_S23_L001_I_peaks.narrowPeak_intersect.bed
peaks/h3k4me1_overlaps/AF28_Flu_2-669592_ChIPmentation-H3K4me1_A00266_0289_2_S9_L002_I_peaks.narrowPeak_pers-only.bed
peaks/h3k4me1_overlaps/AF12_Flu_2-669587_ChIPmentation-H3K4me1_A00266_0289_2_S4_L002_I_peaks.narrowPeak_intersect.bed
peaks/h3k4me1_overlaps/EU25_Flu_2-669590_ChIPmentation-H3K4me1A00266_0289_2_S7_L002_I_peaks.narrowPeak_pers-only.bed
peaks/h3k4me1_overlaps/AF18_Flu_2-674803_ChIPmentation-H3K4me1_A00266_0301_1_S4_L001_I_peaks.narrowPeak_pers-only.bed
peaks/h3k4me1_overlaps/AF22_Flu_2-674827_ChIPmentation-H3K4me1_A00266_0301_2_S4_L002_I_peaks.narrowPeak_ref-only.bed
peaks/h3k4me1_overlaps/EU13_Flu_2-669595_ChIPmentation-H3K4me1A00266_0289_2_S12_L002_I_peaks.narrowPeak_ref-only.bed
peaks/h3k4me1_overlaps/EU33_Flu_2-674820_ChIPmentation-H3K4me1A00266_0301_1_S21_L001_I_peaks.narrowPeak_pers-only.bed
peaks/h3k4me1_overlaps/AF26_Flu_2-669591_ChIPmentation-H3K4me1_A00266_0289_2_S8_L002_I_peaks.narrowPeak_ref-only.bed
peaks/h3k4me1_overlaps/EU27_Flu_2-674815_ChIPmentation-H3K4me1A00266_0301_1_S16_L001_I_peaks.narrowPeak_intersect.bed
peaks/h3k4me1_overlaps/EU07_Flu_2-674802_ChIPmentation-H3K4me1A00266_0301_1_S3_L001_I_peaks.narrowPeak_pers-only.bed
peaks/h3k4me1_overlaps/AF34_Flu_2-669596_ChIPmentation-H3K4me1_A00266_0289_2_S13_L002_I_peaks.narrowPeak_intersect.bed
peaks/h3k4me1_overlaps/AF04_Flu_2-674805_ChIPmentation-H3K4me1_A00266_0301_1_S6_L001_I_peaks.narrowPeak_ref-only.bed
peaks/h3k4me1_overlaps/AF38_Flu_2-674837_ChIPmentation-H3K4me1_A00266_0301_2_S14_L002_I_peaks.narrowPeak_intersect.bed
peaks/h3k4me1_overlaps/EU05_Flu_2-674801_ChIPmentation-H3K4me1A00266_0301_1_S2_L001_I_peaks.narrowPeak_pers-only.bed

March 2021

```
peaks/h3k4me1_overlaps/EU41_Flu_2-674832_ChIPmentation-
H3K4me1A00266_0301_2_S9_L002_I_peaks.narrowPeak_pers-only.bed
peaks/h3k4me1_overlaps/AF36_Flu_2-669593_ChIPmentation-
H3K4me1_A00266_0289_2_S10_L002_I_peaks.narrowPeak_ref-only.bed
peaks/h3k4me1_overlaps/EU39_Flu_2-669598_ChIPmentation-
H3K4me1A00266_0289_2_S15_L002_I_peaks.narrowPeak_intersect.bed
peaks/h3k4me1_overlaps/EU39_Flu_2-669598_ChIPmentation-
H3K4me1A00266_0289_2_S15_L002_I_peaks.narrowPeak_pers-only.bed
peaks/h3k4me1_overlaps/EU03_Flu_2-674750_ChIPmentation-
H3K4me1A00266_0301_1_S1_L001_I_peaks.narrowPeak_ref-only.bed
peaks/h3k4me1_overlaps/EU29_Flu_2-674821_ChIPmentation-
H3K4me1A00266_0301_1_S22_L001_I_peaks.narrowPeak_ref-only.bed
peaks/h3k4me1_overlaps/EU41_Flu_2-674832_ChIPmentation-
H3K4me1A00266_0301_2_S9_L002_I_peaks.narrowPeak_intersect.bed
peaks/h3k4me1_overlaps/EU05_Flu_2-674801_ChIPmentation-
H3K4me1A00266_0301_1_S2_L001_I_peaks.narrowPeak_intersect.bed
peaks/h3k4me1_overlaps/AF38_Flu_2-674837_ChIPmentation-
H3K4me1_A00266_0301_2_S14_L002_I_peaks.narrowPeak_pers-only.bed
peaks/h3k4me1_overlaps/AF08_Flu_2-669652_ChIPmentation-
H3K4me1_A00266_0289_2_S19_L002_I_peaks.narrowPeak_ref-only.bed
peaks/h3k4me1_overlaps/AF34_Flu_2-669596_ChIPmentation-
H3K4me1_A00266_0289_2_S13_L002_I_peaks.narrowPeak_pers-only.bed
peaks/h3k4me1_overlaps/EU15_Flu_2-669589_ChIPmentation-
H3K4me1A00266_0289_2_S6_L002_I_peaks.narrowPeak_ref-only.bed
peaks/h3k4me1_overlaps/EU27_Flu_2-674815_ChIPmentation-
H3K4me1A00266_0301_1_S16_L001_I_peaks.narrowPeak_pers-only.bed
peaks/h3k4me1_overlaps/EU07_Flu_2-674802_ChIPmentation-
H3K4me1A00266_0301_1_S3_L001_I_peaks.narrowPeak_intersect.bed
peaks/h3k4me1_overlaps/EU47_Flu_2-674834_ChIPmentation-
H3K4me1A00266_0301_2_S11_L002_I_peaks.narrowPeak_ref-only.bed
peaks/h3k4me1_overlaps/AF34_Flu_2-669596_ChIPmentation-
H3K4me1_A00266_0289_2_S13_L002_I_peaks.narrowPeak_ref-only.bed
peaks/h3k4me1_overlaps/EU33_Flu_2-674820_ChIPmentation-
H3K4me1A00266_0301_1_S21_L001_I_peaks.narrowPeak_intersect.bed
peaks/h3k4me1_overlaps/AF18_Flu_2-674803_ChIPmentation-
H3K4me1_A00266_0301_1_S4_L001_I_peaks.narrowPeak_intersect.bed
peaks/h3k4me1_overlaps/EU25_Flu_2-669590_ChIPmentation-
H3K4me1A00266_0289_2_S7_L002_I_peaks.narrowPeak_intersect.bed
peaks/h3k4me1_overlaps/AF12_Flu_2-669587_ChIPmentation-
H3K4me1_A00266_0289_2_S4_L002_I_peaks.narrowPeak_pers-only.bed
peaks/h3k4me1_overlaps/AF28_Flu_2-669592_ChIPmentation-
H3K4me1_A00266_0289_2_S9_L002_I_peaks.narrowPeak_intersect.bed
peaks/h3k4me1_overlaps/EU33_Flu_2-674820_ChIPmentation-
H3K4me1A00266_0301_1_S21_L001_I_peaks.narrowPeak_ref-only.bed
peaks/h3k4me1_overlaps/EU09_Flu_2-669582_ChIPmentation-
H3K4me1A00266_0289_1_S23_L001_I_peaks.narrowPeak_pers-only.bed
peaks/h3k4me1_overlaps/AF16_Flu_2-669654_ChIPmentation-
H3K4me1_A00266_0289_2_S21_L002_I_peaks.narrowPeak_intersect.bed
peaks/h3k4me1_overlaps/EU13_Flu_2-669595_ChIPmentation-
H3K4me1A00266_0289_2_S12_L002_I_peaks.narrowPeak_intersect.bed
peaks/h3k4me1_overlaps/EU29_Flu_2-674821_ChIPmentation-
H3K4me1A00266_0301_1_S22_L001_I_peaks.narrowPeak_pers-only.bed
peaks/h3k4me1_overlaps/EU19_Flu_2-669585_ChIPmentation-
H3K4me1A00266_0289_2_S2_L002_I_peaks.narrowPeak_ref-only.bed
peaks/h3k4me1_overlaps/AF30_Flu_2-674830_ChIPmentation-
H3K4me1_A00266_0301_2_S7_L002_I_peaks.narrowPeak_intersect.bed
peaks/h3k4me1_overlaps/AF24_Flu_2-674804_ChIPmentation-
H3K4me1_A00266_0301_1_S5_L001_I_peaks.narrowPeak_pers-only.bed
peaks/h3k4me1_overlaps/AF36_Flu_2-669593_ChIPmentation-
H3K4me1_A00266_0289_2_S10_L002_I_peaks.narrowPeak_pers-only.bed
peaks/h3k4me1_overlaps/AF08_Flu_2-669652_ChIPmentation-
H3K4me1_A00266_0289_2_S19_L002_I_peaks.narrowPeak_intersect.bed
peaks/h3k4me1_overlaps/AF26_Flu_2-669591_ChIPmentation-
H3K4me1_A00266_0289_2_S8_L002_I_peaks.narrowPeak_pers-only.bed
peaks/h3k4me1_overlaps/AF22_Flu_2-674827_ChIPmentation-
H3K4me1_A00266_0301_2_S4_L002_I_peaks.narrowPeak_pers-only.bed
peaks/h3k4me1_overlaps/EU15_Flu_2-669589_ChIPmentation-
H3K4me1A00266_0289_2_S6_L002_I_peaks.narrowPeak_pers-only.bed
peaks/h3k4me1_overlaps/EU19_Flu_2-669585_ChIPmentation-
H3K4me1A00266_0289_2_S2_L002_I_peaks.narrowPeak_intersect.bed
peaks/h3k4me1_overlaps/EU39_Flu_2-669598_ChIPmentation-
H3K4me1A00266_0289_2_S15_L002_I_peaks.narrowPeak_ref-only.bed
peaks/h3k4me1_overlaps/EU21_Flu_2-669653_ChIPmentation-
H3K4me1A00266_0289_2_S20_L002_I_peaks.narrowPeak_intersect.bed
peaks/h3k4me1_overlaps/AF12_Flu_2-669587_ChIPmentation-
H3K4me1_A00266_0289_2_S4_L002_I_peaks.narrowPeak_ref-only.bed
```

March 2021

peaks/h3k4me1_overlaps/AF06_Flu_2-674806_ChIPmentation-H3K4me1_A00266_0301_1_S7_L001_I_peaks.narrowPeak_intersect.bed
peaks/h3k4me1_overlaps/AF20_Flu_2-669580_ChIPmentation-H3K4me1_A00266_0289_1_S21_L001_I_peaks.narrowPeak_pers-only.bed
peaks/h3k4me1_overlaps/AF04_Flu_2-674805_ChIPmentation-H3K4me1_A00266_0301_1_S6_L001_I_peaks.narrowPeak_pers-only.bed
peaks/h3k4me1_overlaps/EU05_Flu_2-674801_ChIPmentation-H3K4me1A00266_0301_1_S2_L001_I_peaks.narrowPeak_ref-only.bed
peaks/h3k4me1_overlaps/AF16_Flu_2-669654_ChIPmentation-H3K4me1_A00266_0289_2_S21_L002_I_peaks.narrowPeak_ref-only.bed
peaks/h3k4me1_overlaps/EU03_Flu_2-674750_ChIPmentation-H3K4me1A00266_0301_1_S1_L001_I_peaks.narrowPeak_intersect.bed
peaks/h3k4me1_overlaps/EU47_Flu_2-674834_ChIPmentation-H3K4me1A00266_0301_2_S11_L002_I_peaks.narrowPeak_pers-only.bed
peaks/h3k4me1_overlaps/EU41_Flu_2-674832_ChIPmentation-H3K4me1A00266_0301_2_S9_L002_I_peaks.narrowPeak_ref-only.bed
peaks/h3k4me1_overlaps/AF30_Flu_2-674830_ChIPmentation-H3K4me1_A00266_0301_2_S7_L002_I_peaks.narrowPeak_ref-only.bed
peaks/h3k4me1_overlaps/EU47_Flu_2-674834_ChIPmentation-H3K4me1A00266_0301_2_S11_L002_I_peaks.narrowPeak_intersect.bed
peaks/h3k4me1_overlaps/EU25_Flu_2-669590_ChIPmentation-H3K4me1A00266_0289_2_S7_L002_I_peaks.narrowPeak_ref-only.bed
peaks/h3k4me1_overlaps/EU03_Flu_2-674750_ChIPmentation-H3K4me1A00266_0301_1_S1_L001_I_peaks.narrowPeak_pers-only.bed
peaks/h3k4me1_overlaps/AF24_Flu_2-674804_ChIPmentation-H3K4me1_A00266_0301_1_S5_L001_I_peaks.narrowPeak_ref-only.bed
peaks/h3k4me1_overlaps/AF04_Flu_2-674805_ChIPmentation-H3K4me1_A00266_0301_1_S6_L001_I_peaks.narrowPeak_intersect.bed
peaks/h3k4me1_overlaps/AF20_Flu_2-669580_ChIPmentation-H3K4me1_A00266_0289_1_S21_L001_I_peaks.narrowPeak_intersect.bed
peaks/h3k4me1_overlaps/EU07_Flu_2-674802_ChIPmentation-H3K4me1A00266_0301_1_S3_L001_I_peaks.narrowPeak_ref-only.bed
peaks/h3k4me1_overlaps/AF06_Flu_2-674806_ChIPmentation-H3K4me1_A00266_0301_1_S7_L001_I_peaks.narrowPeak_pers-only.bed
peaks/h3k4me1_overlaps/EU21_Flu_2-669653_ChIPmentation-H3K4me1A00266_0289_2_S20_L002_I_peaks.narrowPeak_pers-only.bed
peaks/h3k4me1_overlaps/EU19_Flu_2-669585_ChIPmentation-H3K4me1A00266_0289_2_S2_L002_I_peaks.narrowPeak_pers-only.bed
peaks/h3k4me1_overlaps/EU21_Flu_2-669653_ChIPmentation-H3K4me1A00266_0289_2_S20_L002_I_peaks.narrowPeak_ref-only.bed
peaks/h3k4me1_overlaps/EU15_Flu_2-669589_ChIPmentation-H3K4me1A00266_0289_2_S6_L002_I_peaks.narrowPeak_intersect.bed
peaks/h3k4me1_overlaps/AF28_Flu_2-669592_ChIPmentation-H3K4me1_A00266_0289_2_S9_L002_I_peaks.narrowPeak_ref-only.bed
peaks/h3k4me1_overlaps/AF22_Flu_2-674827_ChIPmentation-H3K4me1_A00266_0301_2_S4_L002_I_peaks.narrowPeak_intersect.bed
peaks/h3k4me1_overlaps/EU09_Flu_2-669582_ChIPmentation-H3K4me1A00266_0289_1_S23_L001_I_peaks.narrowPeak_ref-only.bed
peaks/h3k4me1_overlaps/AF26_Flu_2-669591_ChIPmentation-H3K4me1_A00266_0289_2_S8_L002_I_peaks.narrowPeak_intersect.bed
peaks/h3k4me1_overlaps/AF08_Flu_2-669652_ChIPmentation-H3K4me1_A00266_0289_2_S19_L002_I_peaks.narrowPeak_pers-only.bed
peaks/h3k27ac_hprc_peaks/AF24_Flu_2-642069_ChIPmentation-H3K27ac_A00266_0265_2_S13_L002_I_peaks.narrowPeak
peaks/h3k27ac_hprc_peaks/AF18_Flu_2-642068_ChIPmentation-H3K27ac_A00266_0265_2_S12_L002_I_peaks.narrowPeak
peaks/h3k27ac_hprc_peaks/EU05_Flu_2-571412_ChIPmentation-H3K27ac_A00266_0265_2_S6_L002_I_peaks.narrowPeak
peaks/h3k27ac_hprc_peaks/AF22_Flu_2-669473_ChIPmentation-H3K27ac_A00266_0289_1_S9_L001_I_peaks.narrowPeak
peaks/h3k27ac_hprc_peaks/EU47_Flu_2-669480_ChIPmentation-H3K27ac_A00266_0289_1_S16_L001_I_peaks.narrowPeak
peaks/h3k27ac_hprc_peaks/EU19_Flu_2-571338_ChIPmentation-H3K27ac_A00266_0263_3_S6_L003_I_peaks.narrowPeak
peaks/h3k27ac_hprc_peaks/AF30_Flu_2-669476_ChIPmentation-H3K27ac_A00266_0289_1_S12_L001_I_peaks.narrowPeak
peaks/h3k27ac_hprc_peaks/EU09_Flu_2-571335_ChIPmentation-H3K27ac_A00266_0263_3_S3_L003_I_peaks.narrowPeak
peaks/h3k27ac_hprc_peaks/AF26_Flu_2-571344_ChIPmentation-H3K27ac_A00266_0263_3_S12_L003_I_peaks.narrowPeak
peaks/h3k27ac_hprc_peaks/EU33_Flu_2-669466_ChIPmentation-H3K27ac_A00266_0289_1_S2_L001_I_peaks.narrowPeak
peaks/h3k27ac_hprc_peaks/EU25_Flu_2-571343_ChIPmentation-H3K27ac_A00266_0263_3_S11_L003_I_peaks.narrowPeak
peaks/h3k27ac_hprc_peaks/AF38_Flu_2-669483_ChIPmentation-H3K27ac_A00266_0289_1_S19_L001_I_peaks.narrowPeak
peaks/h3k27ac_hprc_peaks/EU21_Flu_2-571406_ChIPmentation-H3K27ac_A00266_0263_3_S24_L003_I_peaks.narrowPeak
peaks/h3k27ac_hprc_peaks/AF06_Flu_2-642071_ChIPmentation-H3K27ac_A00266_0265_2_S15_L002_I_peaks.narrowPeak
peaks/h3k27ac_hprc_peaks/EU15_Flu_2-571342_ChIPmentation-H3K27ac_A00266_0263_3_S10_L003_I_peaks.narrowPeak
peaks/h3k27ac_hprc_peaks/AF16_Flu_2-571407_ChIPmentation-H3K27ac_A00266_0265_2_S1_L002_I_peaks.narrowPeak
peaks/h3k27ac_hprc_peaks/AF08_Flu_2-571405_ChIPmentation-H3K27ac_A00266_0263_3_S23_L003_I_peaks.narrowPeak
peaks/h3k27ac_hprc_peaks/EU13_Flu_2-571348_ChIPmentation-H3K27ac_A00266_0263_3_S16_L003_I_peaks.narrowPeak
peaks/h3k27ac_hprc_peaks/AF36_Flu_2-571346_ChIPmentation-H3K27ac_A00266_0263_3_S14_L003_I_peaks.narrowPeak
peaks/h3k27ac_hprc_peaks/AF34_Flu_2-571349_ChIPmentation-H3K27ac_A00266_0263_3_S17_L003_I_peaks.narrowPeak
peaks/h3k27ac_hprc_peaks/AF12_Flu_2-571340_ChIPmentation-H3K27ac_A00266_0263_3_S8_L003_I_peaks.narrowPeak
peaks/h3k27ac_hprc_peaks/EU03_Flu_2-571411_ChIPmentation-H3K27ac_A00266_0265_2_S5_L002_I_peaks.narrowPeak
peaks/h3k27ac_hprc_peaks/EU41_Flu_2-669478_ChIPmentation-H3K27ac_A00266_0289_1_S14_L001_I_peaks.narrowPeak
peaks/h3k27ac_hprc_peaks/AF20_Flu_2-571333_ChIPmentation-H3K27ac_A00266_0263_3_S1_L003_I_peaks.narrowPeak

peaks/h3k27ac_hprc_peaks/EU29_Flu_2-669467_ChIPmentation-H3K27ac_A00266_0289_1_S3_L001_I_peaks.narrowPeak
peaks/h3k27ac_hprc_peaks/AF04_Flu_2-642070_ChIPmentation-H3K27ac_A00266_0265_2_S14_L002_I_peaks.narrowPeak
peaks/h3k27ac_hprc_peaks/EU39_Flu_2-571401_ChIPmentation-H3K27ac_A00266_0263_3_S19_L003_I_peaks.narrowPeak
peaks/h3k27ac_hprc_peaks/AF28_Flu_2-571345_ChIPmentation-H3K27ac_A00266_0263_3_S13_L003_I_peaks.narrowPeak
peaks/h3k27ac_hprc_peaks/EU07_Flu_2-642067_ChIPmentation-H3K27ac_A00266_0265_2_S11_L002_I_peaks.narrowPeak
peaks/h3k27ac_hprc_peaks/EU27_Flu_2-642077_ChIPmentation-H3K27ac_A00266_0265_2_S21_L002_I_peaks.narrowPeak
peaks/h3k27ac_ref_peaks/AF24_Flu_2-642069_ChIPmentation-H3K27ac_A00266_0265_2_S13_L002_I_peaks.narrowPeak
peaks/h3k27ac_ref_peaks/AF18_Flu_2-642068_ChIPmentation-H3K27ac_A00266_0265_2_S12_L002_I_peaks.narrowPeak
peaks/h3k27ac_ref_peaks/EU05_Flu_2-571412_ChIPmentation-H3K27ac_A00266_0265_2_S6_L002_I_peaks.narrowPeak
peaks/h3k27ac_ref_peaks/AF22_Flu_2-669473_ChIPmentation-H3K27ac_A00266_0289_1_S9_L001_I_peaks.narrowPeak
peaks/h3k27ac_ref_peaks/EU47_Flu_2-669480_ChIPmentation-H3K27ac_A00266_0289_1_S16_L001_I_peaks.narrowPeak
peaks/h3k27ac_ref_peaks/EU19_Flu_2-571338_ChIPmentation-H3K27ac_A00266_0263_3_S6_L003_I_peaks.narrowPeak
peaks/h3k27ac_ref_peaks/AF30_Flu_2-669476_ChIPmentation-H3K27ac_A00266_0289_1_S12_L001_I_peaks.narrowPeak
peaks/h3k27ac_ref_peaks/EU09_Flu_2-571335_ChIPmentation-H3K27ac_A00266_0263_3_S3_L003_I_peaks.narrowPeak
peaks/h3k27ac_ref_peaks/AF26_Flu_2-571344_ChIPmentation-H3K27ac_A00266_0263_3_S12_L003_I_peaks.narrowPeak
peaks/h3k27ac_ref_peaks/EU33_Flu_2-669466_ChIPmentation-H3K27ac_A00266_0289_1_S2_L001_I_peaks.narrowPeak
peaks/h3k27ac_ref_peaks/EU25_Flu_2-571343_ChIPmentation-H3K27ac_A00266_0263_3_S11_L003_I_peaks.narrowPeak
peaks/h3k27ac_ref_peaks/AF38_Flu_2-669483_ChIPmentation-H3K27ac_A00266_0289_1_S19_L001_I_peaks.narrowPeak
peaks/h3k27ac_ref_peaks/EU21_Flu_2-571406_ChIPmentation-H3K27ac_A00266_0263_3_S24_L003_I_peaks.narrowPeak
peaks/h3k27ac_ref_peaks/AF06_Flu_2-642071_ChIPmentation-H3K27ac_A00266_0265_2_S15_L002_I_peaks.narrowPeak
peaks/h3k27ac_ref_peaks/EU15_Flu_2-571342_ChIPmentation-H3K27ac_A00266_0263_3_S10_L003_I_peaks.narrowPeak
peaks/h3k27ac_ref_peaks/AF16_Flu_2-571407_ChIPmentation-H3K27ac_A00266_0265_2_S1_L002_I_peaks.narrowPeak
peaks/h3k27ac_ref_peaks/AF08_Flu_2-571405_ChIPmentation-H3K27ac_A00266_0263_3_S23_L003_I_peaks.narrowPeak
peaks/h3k27ac_ref_peaks/EU13_Flu_2-571348_ChIPmentation-H3K27ac_A00266_0263_3_S16_L003_I_peaks.narrowPeak
peaks/h3k27ac_ref_peaks/AF36_Flu_2-571346_ChIPmentation-H3K27ac_A00266_0263_3_S14_L003_I_peaks.narrowPeak
peaks/h3k27ac_ref_peaks/AF34_Flu_2-571349_ChIPmentation-H3K27ac_A00266_0263_3_S17_L003_I_peaks.narrowPeak
peaks/h3k27ac_ref_peaks/AF12_Flu_2-571340_ChIPmentation-H3K27ac_A00266_0263_3_S8_L003_I_peaks.narrowPeak
peaks/h3k27ac_ref_peaks/EU03_Flu_2-571411_ChIPmentation-H3K27ac_A00266_0265_2_S5_L002_I_peaks.narrowPeak
peaks/h3k27ac_ref_peaks/EU41_Flu_2-669478_ChIPmentation-H3K27ac_A00266_0289_1_S14_L001_I_peaks.narrowPeak
peaks/h3k27ac_ref_peaks/AF20_Flu_2-571333_ChIPmentation-H3K27ac_A00266_0263_3_S1_L003_I_peaks.narrowPeak
peaks/h3k27ac_ref_peaks/EU29_Flu_2-669467_ChIPmentation-H3K27ac_A00266_0289_1_S3_L001_I_peaks.narrowPeak
peaks/h3k27ac_ref_peaks/AF04_Flu_2-642070_ChIPmentation-H3K27ac_A00266_0265_2_S14_L002_I_peaks.narrowPeak
peaks/h3k27ac_ref_peaks/EU39_Flu_2-571401_ChIPmentation-H3K27ac_A00266_0263_3_S19_L003_I_peaks.narrowPeak
peaks/h3k27ac_ref_peaks/AF28_Flu_2-571345_ChIPmentation-H3K27ac_A00266_0263_3_S13_L003_I_peaks.narrowPeak
peaks/h3k27ac_ref_peaks/EU07_Flu_2-642067_ChIPmentation-H3K27ac_A00266_0265_2_S11_L002_I_peaks.narrowPeak
peaks/h3k27ac_ref_peaks/EU27_Flu_2-642077_ChIPmentation-H3K27ac_A00266_0265_2_S21_L002_I_peaks.narrowPeak
peaks/atac_ref_peaks/EU29_Flu_ATACSeq_1_2-268563_S10_L001_I_peaks.narrowPeak
peaks/atac_ref_peaks/EU09_Flu_ATACSeq_1_2-259471_S11_L001_I_peaks.narrowPeak
peaks/atac_ref_peaks/EU13_Flu_ATACSeq_1_2-259477_S1_L002_I_peaks.narrowPeak
peaks/atac_ref_peaks/EU05_Flu_ATACSeq_1_2-259464_S4_L001_I_peaks.narrowPeak
peaks/atac_ref_peaks/AF10_Flu_ATACSeq_1_2-259491_S15_L002_I_peaks.narrowPeak
peaks/atac_ref_peaks/EU19_Flu_ATACSeq_1_2-259485_S9_L002_I_peaks.narrowPeak
peaks/atac_ref_peaks/AF22_Flu_ATACSeq_1_2-308160_S3_L001_I_peaks.narrowPeak
peaks/atac_ref_peaks/AF14_Flu_ATACSeq_1_2-268556_S3_L001_I_peaks.narrowPeak
peaks/atac_ref_peaks/EU33_Flu_ATACSeq_1_2-308170_S13_L001_I_peaks.narrowPeak
peaks/atac_ref_peaks/EU37_Flu_ATACSeq_1_2-308172_S15_L001_I_peaks.narrowPeak
peaks/atac_ref_peaks/AF18_Flu_ATACSeq_1_2-268567_S14_L001_I_peaks.narrowPeak
peaks/atac_ref_peaks/EU47_Flu_ATACSeq_1_2-308183_S10_L002_I_peaks.narrowPeak
peaks/atac_ref_peaks/EU27_Flu_ATACSeq_1_2-268561_S8_L001_I_peaks.narrowPeak
peaks/atac_ref_peaks/AF26_Flu_ATACSeq_1_2-308164_S7_L001_I_peaks.narrowPeak
peaks/atac_ref_peaks/EU43_Flu_ATACSeq_1_2-308181_S8_L002_I_peaks.narrowPeak
peaks/atac_ref_peaks/EU21_Flu_ATACSeq_1_2-259488_S12_L002_I_peaks.narrowPeak
peaks/atac_ref_peaks/AF34_Flu_ATACSeq_1_2-308174_S17_L001_I_peaks.narrowPeak
peaks/atac_ref_peaks/AF28_Flu_ATACSeq_1_2-308166_S9_L001_I_peaks.narrowPeak
peaks/atac_ref_peaks/EU07_Flu_ATACSeq_1_2-259468_S8_L001_I_peaks.narrowPeak
peaks/atac_ref_peaks/EU41_Flu_ATACSeq_1_2-308179_S6_L002_I_peaks.narrowPeak
peaks/atac_ref_peaks/EU03_Flu_ATACSeq_1_2-259462_S2_L001_I_peaks.narrowPeak
peaks/atac_ref_peaks/EU17_Flu_ATACSeq_1_2-308175_S2_L002_I_peaks.narrowPeak
peaks/atac_ref_peaks/AF16_Flu_ATACSeq_1_2-268565_S12_L001_I_peaks.narrowPeak
peaks/atac_ref_peaks/AF20_Flu_ATACSeq_1_2-308158_S1_L001_I_peaks.narrowPeak
peaks/atac_ref_peaks/EU25_Flu_ATACSeq_1_2-268559_S6_L001_I_peaks.narrowPeak
peaks/atac_ref_peaks/EU15_Flu_ATACSeq_1_2-259481_S5_L002_I_peaks.narrowPeak
peaks/atac_ref_peaks/EU39_Flu_ATACSeq_1_2-308177_S4_L002_I_peaks.narrowPeak
peaks/atac_ref_peaks/AF04_Flu_ATACSeq_1_2-259466_S6_L001_I_peaks.narrowPeak
peaks/atac_ref_peaks/AF08_Flu_ATACSeq_1_2-259483_S7_L002_I_peaks.narrowPeak
peaks/atac_ref_peaks/AF12_Flu_ATACSeq_1_2-268554_S1_L001_I_peaks.narrowPeak
peaks/atac_ref_peaks/AF06_Flu_ATACSeq_1_2-259479_S3_L002_I_peaks.narrowPeak
peaks/atac_ref_peaks/AF38_Flu_ATACSeq_1_2-308187_S14_L002_I_peaks.narrowPeak
peaks/atac_ref_peaks/AF30_Flu_ATACSeq_1_2-308168_S11_L001_I_peaks.narrowPeak
peaks/atac_ref_peaks/AF24_Flu_ATACSeq_1_2-308162_S5_L001_I_peaks.narrowPeak
peaks/atac_hprc_peaks/EU29_Flu_ATACSeq_1_2-268563_S10_L001_I_peaks.narrowPeak
peaks/atac_hprc_peaks/EU09_Flu_ATACSeq_1_2-259471_S11_L001_I_peaks.narrowPeak
peaks/atac_hprc_peaks/EU13_Flu_ATACSeq_1_2-259477_S1_L002_I_peaks.narrowPeak
peaks/atac_hprc_peaks/EU05_Flu_ATACSeq_1_2-259464_S4_L001_I_peaks.narrowPeak
peaks/atac_hprc_peaks/AF10_Flu_ATACSeq_1_2-259491_S15_L002_I_peaks.narrowPeak
peaks/atac_hprc_peaks/EU19_Flu_ATACSeq_1_2-259485_S9_L002_I_peaks.narrowPeak

```
peaks/atac_hprc_peaks/AF22_Flu_ATACSeq_1_2-308160_S3_L001_I_peaks.narrowPeak
peaks/atac_hprc_peaks/AF14_Flu_ATACSeq_1_2-268556_S3_L001_I_peaks.narrowPeak
peaks/atac_hprc_peaks/EU33_Flu_ATACSeq_1_2-308170_S13_L001_I_peaks.narrowPeak
peaks/atac_hprc_peaks/EU37_Flu_ATACSeq_1_2-308172_S15_L001_I_peaks.narrowPeak
peaks/atac_hprc_peaks/AF18_Flu_ATACSeq_1_2-268567_S14_L001_I_peaks.narrowPeak
peaks/atac_hprc_peaks/EU47_Flu_ATACSeq_1_2-308183_S10_L002_I_peaks.narrowPeak
peaks/atac_hprc_peaks/EU27_Flu_ATACSeq_1_2-268561_S8_L001_I_peaks.narrowPeak
peaks/atac_hprc_peaks/AF26_Flu_ATACSeq_1_2-308164_S7_L001_I_peaks.narrowPeak
peaks/atac_hprc_peaks/EU43_Flu_ATACSeq_1_2-308181_S8_L002_I_peaks.narrowPeak
peaks/atac_hprc_peaks/EU21_Flu_ATACSeq_1_2-259488_S12_L002_I_peaks.narrowPeak
peaks/atac_hprc_peaks/AF34_Flu_ATACSeq_1_2-308174_S17_L001_I_peaks.narrowPeak
peaks/atac_hprc_peaks/AF28_Flu_ATACSeq_1_2-308166_S9_L001_I_peaks.narrowPeak
peaks/atac_hprc_peaks/EU07_Flu_ATACSeq_1_2-259468_S8_L001_I_peaks.narrowPeak
peaks/atac_hprc_peaks/EU41_Flu_ATACSeq_1_2-308179_S6_L002_I_peaks.narrowPeak
peaks/atac_hprc_peaks/EU03_Flu_ATACSeq_1_2-259462_S2_L001_I_peaks.narrowPeak
peaks/atac_hprc_peaks/EU17_Flu_ATACSeq_1_2-308175_S2_L002_I_peaks.narrowPeak
peaks/atac_hprc_peaks/AF16_Flu_ATACSeq_1_2-268565_S12_L001_I_peaks.narrowPeak
peaks/atac_hprc_peaks/AF20_Flu_ATACSeq_1_2-308158_S1_L001_I_peaks.narrowPeak
peaks/atac_hprc_peaks/EU25_Flu_ATACSeq_1_2-268559_S6_L001_I_peaks.narrowPeak
peaks/atac_hprc_peaks/EU15_Flu_ATACSeq_1_2-259481_S5_L002_I_peaks.narrowPeak
peaks/atac_hprc_peaks/EU39_Flu_ATACSeq_1_2-308177_S4_L002_I_peaks.narrowPeak
peaks/atac_hprc_peaks/AF04_Flu_ATACSeq_1_2-259466_S6_L001_I_peaks.narrowPeak
peaks/atac_hprc_peaks/AF36_Flu_ATACSeq_1_2-308185_S12_L002_I_peaks.narrowPeak
peaks/atac_hprc_peaks/AF08_Flu_ATACSeq_1_2-259483_S7_L002_I_peaks.narrowPeak
peaks/atac_hprc_peaks/AF12_Flu_ATACSeq_1_2-268554_S1_L001_I_peaks.narrowPeak
peaks/atac_hprc_peaks/AF06_Flu_ATACSeq_1_2-259479_S3_L002_I_peaks.narrowPeak
peaks/atac_hprc_peaks/AF38_Flu_ATACSeq_1_2-308187_S14_L002_I_peaks.narrowPeak
peaks/atac_hprc_peaks/AF30_Flu_ATACSeq_1_2-308168_S11_L001_I_peaks.narrowPeak
peaks/atac_hprc_peaks/AF24_Flu_ATACSeq_1_2-308162_S5_L001_I_peaks.narrowPeak
peaks/h3k4me1_hprc_peaks/AF28_Flu_2-669592_ChIPmentation-H3K4me1_A00266_0289_2_S9_L002_I_peaks.narrowPeak
peaks/h3k4me1_hprc_peaks/AF30_Flu_2-674830_ChIPmentation-H3K4me1_A00266_0301_2_S7_L002_I_peaks.narrowPeak
peaks/h3k4me1_hprc_peaks/EU27_Flu_2-674815_ChIPmentation-
H3K4me1A00266_0301_1_S16_L001_I_peaks.narrowPeak
peaks/h3k4me1_hprc_peaks/EU05_Flu_2-674801_ChIPmentation-H3K4me1A00266_0301_1_S2_L001_I_peaks.narrowPeak
peaks/h3k4me1_hprc_peaks/AF18_Flu_2-674803_ChIPmentation-H3K4me1_A00266_0301_1_S4_L001_I_peaks.narrowPeak
peaks/h3k4me1_hprc_peaks/EU47_Flu_2-674834_ChIPmentation-
H3K4me1A00266_0301_2_S11_L002_I_peaks.narrowPeak
peaks/h3k4me1_hprc_peaks/AF34_Flu_2-669596_ChIPmentation-
H3K4me1_A00266_0289_2_S13_L002_I_peaks.narrowPeak
peaks/h3k4me1_hprc_peaks/EU09_Flu_2-669582_ChIPmentation-
H3K4me1A00266_0289_1_S23_L001_I_peaks.narrowPeak
peaks/h3k4me1_hprc_peaks/AF20_Flu_2-669580_ChIPmentation-
H3K4me1_A00266_0289_1_S21_L001_I_peaks.narrowPeak
peaks/h3k4me1_hprc_peaks/EU39_Flu_2-669598_ChIPmentation-
H3K4me1A00266_0289_2_S15_L002_I_peaks.narrowPeak
peaks/h3k4me1_hprc_peaks/EU03_Flu_2-674750_ChIPmentation-H3K4me1A00266_0301_1_S1_L001_I_peaks.narrowPeak
peaks/h3k4me1_hprc_peaks/EU15_Flu_2-669589_ChIPmentation-H3K4me1A00266_0289_2_S6_L002_I_peaks.narrowPeak
peaks/h3k4me1_hprc_peaks/AF12_Flu_2-669587_ChIPmentation-H3K4me1_A00266_0289_2_S4_L002_I_peaks.narrowPeak
peaks/h3k4me1_hprc_peaks/EU25_Flu_2-669590_ChIPmentation-H3K4me1A00266_0289_2_S7_L002_I_peaks.narrowPeak
peaks/h3k4me1_hprc_peaks/AF24_Flu_2-674804_ChIPmentation-H3K4me1_A00266_0301_1_S5_L001_I_peaks.narrowPeak
peaks/h3k4me1_hprc_peaks/AF36_Flu_2-669593_ChIPmentation-
H3K4me1_A00266_0289_2_S10_L002_I_peaks.narrowPeak
peaks/h3k4me1_hprc_peaks/AF04_Flu_2-674805_ChIPmentation-H3K4me1_A00266_0301_1_S6_L001_I_peaks.narrowPeak
peaks/h3k4me1_hprc_peaks/EU13_Flu_2-669595_ChIPmentation-
H3K4me1A00266_0289_2_S12_L002_I_peaks.narrowPeak
peaks/h3k4me1_hprc_peaks/EU07_Flu_2-674802_ChIPmentation-H3K4me1A00266_0301_1_S3_L001_I_peaks.narrowPeak
peaks/h3k4me1_hprc_peaks/AF38_Flu_2-674837_ChIPmentation-
H3K4me1_A00266_0301_2_S14_L002_I_peaks.narrowPeak
peaks/h3k4me1_hprc_peaks/AF06_Flu_2-674806_ChIPmentation-H3K4me1_A00266_0301_1_S7_L001_I_peaks.narrowPeak
peaks/h3k4me1_hprc_peaks/AF22_Flu_2-674827_ChIPmentation-H3K4me1_A00266_0301_2_S4_L002_I_peaks.narrowPeak
peaks/h3k4me1_hprc_peaks/EU41_Flu_2-674832_ChIPmentation-H3K4me1A00266_0301_2_S9_L002_I_peaks.narrowPeak
peaks/h3k4me1_hprc_peaks/EU29_Flu_2-674821_ChIPmentation-
H3K4me1A00266_0301_1_S22_L001_I_peaks.narrowPeak
peaks/h3k4me1_hprc_peaks/AF26_Flu_2-669591_ChIPmentation-H3K4me1_A00266_0289_2_S8_L002_I_peaks.narrowPeak
peaks/h3k4me1_hprc_peaks/EU19_Flu_2-669585_ChIPmentation-H3K4me1A00266_0289_2_S2_L002_I_peaks.narrowPeak
peaks/h3k4me1_hprc_peaks/AF16_Flu_2-669654_ChIPmentation-
H3K4me1_A00266_0289_2_S21_L002_I_peaks.narrowPeak
peaks/h3k4me1_hprc_peaks/AF08_Flu_2-669652_ChIPmentation-
H3K4me1_A00266_0289_2_S19_L002_I_peaks.narrowPeak
peaks/h3k4me1_hprc_peaks/EU33_Flu_2-674820_ChIPmentation-
H3K4me1A00266_0301_1_S21_L001_I_peaks.narrowPeak
peaks/h3k4me1_hprc_peaks/EU21_Flu_2-669653_ChIPmentation-
H3K4me1A00266_0289_2_S20_L002_I_peaks.narrowPeak
peaks/atac_overlaps/AF10_Flu_ATACSeq_1_2-259491_S15_L002_I_peaks.narrowPeak_intersect.bed
peaks/atac_overlaps/EU17_Flu_ATACSeq_1_2-308175_S2_L002_I_peaks.narrowPeak_ref-only.bed
peaks/atac_overlaps/EU07_Flu_ATACSeq_1_2-259468_S8_L001_I_peaks.narrowPeak_ref-only.bed
```

```
peaks/atac_overlaps/EU47_Flu_ATACSeq_1_2-308183_S10_L002_I_peaks.narrowPeak_pers-only.bed
peaks/atac_overlaps/EU37_Flu_ATACSeq_1_2-308172_S15_L001_I_peaks.narrowPeak_ref-only.bed
peaks/atac_overlaps/EU41_Flu_ATACSeq_1_2-308179_S6_L002_I_peaks.narrowPeak_ref-only.bed
peaks/atac_overlaps/EU47_Flu_ATACSeq_1_2-308183_S10_L002_I_peaks.narrowPeak_ref-only.bed
peaks/atac_overlaps/AF20_Flu_ATACSeq_1_2-308158_S1_L001_I_peaks.narrowPeak_pers-only.bed
peaks/atac_overlaps/AF14_Flu_ATACSeq_1_2-268556_S3_L001_I_peaks.narrowPeak_intersect.bed
peaks/atac_overlaps/EU19_Flu_ATACSeq_1_2-259485_S9_L002_I_peaks.narrowPeak_pers-only.bed
peaks/atac_overlaps/EU15_Flu_ATACSeq_1_2-259481_S5_L002_I_peaks.narrowPeak_ref-only.bed
peaks/atac_overlaps/EU25_Flu_ATACSeq_1_2-268559_S6_L001_I_peaks.narrowPeak_pers-only.bed
peaks/atac_overlaps/EU33_Flu_ATACSeq_1_2-308170_S13_L001_I_peaks.narrowPeak_pers-only.bed
peaks/atac_overlaps/AF04_Flu_ATACSeq_1_2-259466_S6_L001_I_peaks.narrowPeak_intersect.bed
peaks/atac_overlaps/AF22_Flu_ATACSeq_1_2-308160_S3_L001_I_peaks.narrowPeak_pers-only.bed
peaks/atac_overlaps/EU13_Flu_ATACSeq_1_2-259477_S1_L002_I_peaks.narrowPeak_ref-only.bed
peaks/atac_overlaps/AF34_Flu_ATACSeq_1_2-308174_S17_L001_I_peaks.narrowPeak_intersect.bed
peaks/atac_overlaps/EU17_Flu_ATACSeq_1_2-308175_S2_L002_I_peaks.narrowPeak_intersect.bed
peaks/atac_overlaps/EU13_Flu_ATACSeq_1_2-259477_S1_L002_I_peaks.narrowPeak_pers-only.bed
peaks/atac_overlaps/AF38_Flu_ATACSeq_1_2-308187_S14_L002_I_peaks.narrowPeak_intersect.bed
peaks/atac_overlaps/AF22_Flu_ATACSeq_1_2-308160_S3_L001_I_peaks.narrowPeak_ref-only.bed
peaks/atac_overlaps/AF28_Flu_ATACSeq_1_2-308166_S9_L001_I_peaks.narrowPeak_intersect.bed
peaks/atac_overlaps/EU03_Flu_ATACSeq_1_2-259462_S2_L001_I_peaks.narrowPeak_pers-only.bed
peaks/atac_overlaps/EU21_Flu_ATACSeq_1_2-259488_S12_L002_I_peaks.narrowPeak_ref-only.bed
peaks/atac_overlaps/EU03_Flu_ATACSeq_1_2-259462_S2_L001_I_peaks.narrowPeak_intersect.bed
peaks/atac_overlaps/AF28_Flu_ATACSeq_1_2-308166_S9_L001_I_peaks.narrowPeak_pers-only.bed
peaks/atac_overlaps/AF38_Flu_ATACSeq_1_2-308187_S14_L002_I_peaks.narrowPeak_pers-only.bed
peaks/atac_overlaps/EU25_Flu_ATACSeq_1_2-268559_S6_L001_I_peaks.narrowPeak_ref-only.bed
peaks/atac_overlaps/EU05_Flu_ATACSeq_1_2-259464_S4_L001_I_peaks.narrowPeak_ref-only.bed
peaks/atac_overlaps/EU13_Flu_ATACSeq_1_2-259477_S1_L002_I_peaks.narrowPeak_intersect.bed
peaks/atac_overlaps/EU17_Flu_ATACSeq_1_2-308175_S2_L002_I_peaks.narrowPeak_pers-only.bed
peaks/atac_overlaps/AF34_Flu_ATACSeq_1_2-308174_S17_L001_I_peaks.narrowPeak_pers-only.bed
peaks/atac_overlaps/AF26_Flu_ATACSeq_1_2-308164_S7_L001_I_peaks.narrowPeak_ref-only.bed
peaks/atac_overlaps/AF04_Flu_ATACSeq_1_2-259466_S6_L001_I_peaks.narrowPeak_pers-only.bed
peaks/atac_overlaps/AF22_Flu_ATACSeq_1_2-308160_S3_L001_I_peaks.narrowPeak_intersect.bed
peaks/atac_overlaps/AF30_Flu_ATACSeq_1_2-308168_S11_L001_I_peaks.narrowPeak_ref-only.bed
peaks/atac_overlaps/EU33_Flu_ATACSeq_1_2-308170_S13_L001_I_peaks.narrowPeak_intersect.bed
peaks/atac_overlaps/EU09_Flu_ATACSeq_1_2-259471_S11_L001_I_peaks.narrowPeak_ref-only.bed
peaks/atac_overlaps/EU29_Flu_ATACSeq_1_2-268563_S10_L001_I_peaks.narrowPeak_ref-only.bed
peaks/atac_overlaps/AF08_Flu_ATACSeq_1_2-259483_S7_L002_I_peaks.narrowPeak_ref-only.bed
peaks/atac_overlaps/EU25_Flu_ATACSeq_1_2-268559_S6_L001_I_peaks.narrowPeak_intersect.bed
peaks/atac_overlaps/EU19_Flu_ATACSeq_1_2-259485_S9_L002_I_peaks.narrowPeak_intersect.bed
peaks/atac_overlaps/AF14_Flu_ATACSeq_1_2-268556_S3_L001_I_peaks.narrowPeak_pers-only.bed
peaks/atac_overlaps/AF38_Flu_ATACSeq_1_2-308187_S14_L002_I_peaks.narrowPeak_ref-only.bed
peaks/atac_overlaps/AF20_Flu_ATACSeq_1_2-308158_S1_L001_I_peaks.narrowPeak_intersect.bed
peaks/atac_overlaps/EU19_Flu_ATACSeq_1_2-259485_S9_L002_I_peaks.narrowPeak_ref-only.bed
peaks/atac_overlaps/EU47_Flu_ATACSeq_1_2-308183_S10_L002_I_peaks.narrowPeak_intersect.bed
peaks/atac_overlaps/EU33_Flu_ATACSeq_1_2-308170_S13_L001_I_peaks.narrowPeak_ref-only.bed
peaks/atac_overlaps/AF10_Flu_ATACSeq_1_2-259491_S15_L002_I_peaks.narrowPeak_pers-only.bed
peaks/atac_overlaps/EU05_Flu_ATACSeq_1_2-259464_S4_L001_I_peaks.narrowPeak_intersect.bed
peaks/atac_overlaps/EU21_Flu_ATACSeq_1_2-259488_S12_L002_I_peaks.narrowPeak_intersect.bed
peaks/atac_overlaps/AF24_Flu_ATACSeq_1_2-308162_S5_L001_I_peaks.narrowPeak_intersect.bed
peaks/atac_overlaps/AF10_Flu_ATACSeq_1_2-259491_S15_L002_I_peaks.narrowPeak_ref-only.bed
peaks/atac_overlaps/EU41_Flu_ATACSeq_1_2-308179_S6_L002_I_peaks.narrowPeak_pers-only.bed
peaks/atac_overlaps/AF16_Flu_ATACSeq_1_2-268565_S12_L001_I_peaks.narrowPeak_pers-only.bed
peaks/atac_overlaps/EU37_Flu_ATACSeq_1_2-308172_S15_L001_I_peaks.narrowPeak_pers-only.bed
peaks/atac_overlaps/AF30_Flu_ATACSeq_1_2-308168_S11_L001_I_peaks.narrowPeak_intersect.bed
peaks/atac_overlaps/AF06_Flu_ATACSeq_1_2-259479_S3_L002_I_peaks.narrowPeak_intersect.bed
peaks/atac_overlaps/EU43_Flu_ATACSeq_1_2-308181_S8_L002_I_peaks.narrowPeak_ref-only.bed
peaks/atac_overlaps/AF18_Flu_ATACSeq_1_2-268567_S14_L001_I_peaks.narrowPeak_ref-only.bed
peaks/atac_overlaps/AF08_Flu_ATACSeq_1_2-259483_S7_L002_I_peaks.narrowPeak_intersect.bed
peaks/atac_overlaps/EU43_Flu_ATACSeq_1_2-308181_S8_L002_I_peaks.narrowPeak_intersect.bed
peaks/atac_overlaps/AF18_Flu_ATACSeq_1_2-268567_S14_L001_I_peaks.narrowPeak_intersect.bed
peaks/atac_overlaps/EU29_Flu_ATACSeq_1_2-268563_S10_L001_I_peaks.narrowPeak_intersect.bed
peaks/atac_overlaps/EU27_Flu_ATACSeq_1_2-268561_S8_L001_I_peaks.narrowPeak_pers-only.bed
peaks/atac_overlaps/AF26_Flu_ATACSeq_1_2-308164_S7_L001_I_peaks.narrowPeak_pers-only.bed
peaks/atac_overlaps/AF28_Flu_ATACSeq_1_2-308166_S9_L001_I_peaks.narrowPeak_ref-only.bed
peaks/atac_overlaps/AF12_Flu_ATACSeq_1_2-268554_S1_L001_I_peaks.narrowPeak_pers-only.bed
peaks/atac_overlaps/EU15_Flu_ATACSeq_1_2-259481_S5_L002_I_peaks.narrowPeak_pers-only.bed
peaks/atac_overlaps/EU07_Flu_ATACSeq_1_2-259468_S8_L001_I_peaks.narrowPeak_intersect.bed
peaks/atac_overlaps/EU39_Flu_ATACSeq_1_2-308177_S4_L002_I_peaks.narrowPeak_pers-only.bed
peaks/atac_overlaps/EU09_Flu_ATACSeq_1_2-259471_S11_L001_I_peaks.narrowPeak_pers-only.bed
peaks/atac_overlaps/AF34_Flu_ATACSeq_1_2-308174_S17_L001_I_peaks.narrowPeak_ref-only.bed
peaks/atac_overlaps/AF24_Flu_ATACSeq_1_2-308162_S5_L001_I_peaks.narrowPeak_ref-only.bed
peaks/atac_overlaps/EU09_Flu_ATACSeq_1_2-259471_S11_L001_I_peaks.narrowPeak_intersect.bed
peaks/atac_overlaps/EU03_Flu_ATACSeq_1_2-259462_S2_L001_I_peaks.narrowPeak_ref-only.bed
peaks/atac_overlaps/EU39_Flu_ATACSeq_1_2-308177_S4_L002_I_peaks.narrowPeak_intersect.bed
peaks/atac_overlaps/EU07_Flu_ATACSeq_1_2-259468_S8_L001_I_peaks.narrowPeak_pers-only.bed
peaks/atac_overlaps/EU15_Flu_ATACSeq_1_2-259481_S5_L002_I_peaks.narrowPeak_intersect.bed
```

```
peaks/atac_overlaps/AF20_Flu_ATACSeq_1_2-308158_S1_L001_I_peaks.narrowPeak_ref-only.bed
peaks/atac_overlaps/AF12_Flu_ATACSeq_1_2-268554_S1_L001_I_peaks.narrowPeak_intersect.bed
peaks/atac_overlaps/AF14_Flu_ATACSeq_1_2-268556_S3_L001_I_peaks.narrowPeak_ref-only.bed
peaks/atac_overlaps/AF26_Flu_ATACSeq_1_2-308164_S7_L001_I_peaks.narrowPeak_intersect.bed
peaks/atac_overlaps/EU27_Flu_ATACSeq_1_2-268561_S8_L001_I_peaks.narrowPeak_intersect.bed
peaks/atac_overlaps/EU29_Flu_ATACSeq_1_2-268563_S10_L001_I_peaks.narrowPeak_pers-only.bed
peaks/atac_overlaps/AF18_Flu_ATACSeq_1_2-268567_S14_L001_I_peaks.narrowPeak_pers-only.bed
peaks/atac_overlaps/AF16_Flu_ATACSeq_1_2-268565_S12_L001_I_peaks.narrowPeak_ref-only.bed
peaks/atac_overlaps/EU43_Flu_ATACSeq_1_2-308181_S8_L002_I_peaks.narrowPeak_pers-only.bed
peaks/atac_overlaps/AF06_Flu_ATACSeq_1_2-259479_S3_L002_I_peaks.narrowPeak_ref-only.bed
peaks/atac_overlaps/AF08_Flu_ATACSeq_1_2-259483_S7_L002_I_peaks.narrowPeak_pers-only.bed
peaks/atac_overlaps/AF06_Flu_ATACSeq_1_2-259479_S3_L002_I_peaks.narrowPeak_pers-only.bed
peaks/atac_overlaps/AF30_Flu_ATACSeq_1_2-308168_S11_L001_I_peaks.narrowPeak_pers-only.bed
peaks/atac_overlaps/EU37_Flu_ATACSeq_1_2-308172_S15_L001_I_peaks.narrowPeak_intersect.bed
peaks/atac_overlaps/EU27_Flu_ATACSeq_1_2-268561_S8_L001_I_peaks.narrowPeak_ref-only.bed
peaks/atac_overlaps/AF16_Flu_ATACSeq_1_2-268565_S12_L001_I_peaks.narrowPeak_intersect.bed
peaks/atac_overlaps/AF04_Flu_ATACSeq_1_2-259466_S6_L001_I_peaks.narrowPeak_ref-only.bed
peaks/atac_overlaps/EU41_Flu_ATACSeq_1_2-308179_S6_L002_I_peaks.narrowPeak_intersect.bed
peaks/atac_overlaps/AF24_Flu_ATACSeq_1_2-308162_S5_L001_I_peaks.narrowPeak_pers-only.bed
peaks/atac_overlaps/EU21_Flu_ATACSeq_1_2-259488_S12_L002_I_peaks.narrowPeak_pers-only.bed
peaks/atac_overlaps/EU05_Flu_ATACSeq_1_2-259464_S4_L001_I_peaks.narrowPeak_pers-only.bed
peaks/atac_overlaps/EU39_Flu_ATACSeq_1_2-308177_S4_L002_I_peaks.narrowPeak_ref-only.bed
peaks/atac_overlaps/AF12_Flu_ATACSeq_1_2-268554_S1_L001_I_peaks.narrowPeak_ref-only.bed
peaks/h3k4me1_ref_peaks/AF28_Flu_2-669592_ChIPmentation-H3K4me1_A00266_0289_2_S9_L002_I_peaks.narrowPeak
peaks/h3k4me1_ref_peaks/AF30_Flu_2-674830_ChIPmentation-H3K4me1_A00266_0301_2_S7_L002_I_peaks.narrowPeak
peaks/h3k4me1_ref_peaks/EU27_Flu_2-674815_ChIPmentation-H3K4me1A00266_0301_1_S16_L001_I_peaks.narrowPeak
peaks/h3k4me1_ref_peaks/EU05_Flu_2-674801_ChIPmentation-H3K4me1A00266_0301_1_S2_L001_I_peaks.narrowPeak
peaks/h3k4me1_ref_peaks/AF18_Flu_2-674803_ChIPmentation-H3K4me1_A00266_0301_1_S4_L001_I_peaks.narrowPeak
peaks/h3k4me1_ref_peaks/EU47_Flu_2-674834_ChIPmentation-H3K4me1_A00266_0301_2_S11_L002_I_peaks.narrowPeak
peaks/h3k4me1_ref_peaks/AF34_Flu_2-669596_ChIPmentation-H3K4me1_A00266_0289_2_S13_L002_I_peaks.narrowPeak
peaks/h3k4me1_ref_peaks/EU09_Flu_2-669582_ChIPmentation-H3K4me1A00266_0289_1_S23_L001_I_peaks.narrowPeak
peaks/h3k4me1_ref_peaks/AF20_Flu_2-669580_ChIPmentation-H3K4me1_A00266_0289_1_S21_L001_I_peaks.narrowPeak
peaks/h3k4me1_ref_peaks/EU39_Flu_2-669598_ChIPmentation-H3K4me1A00266_0289_2_S15_L002_I_peaks.narrowPeak
peaks/h3k4me1_ref_peaks/EU03_Flu_2-674750_ChIPmentation-H3K4me1A00266_0301_1_S1_L001_I_peaks.narrowPeak
peaks/h3k4me1_ref_peaks/EU15_Flu_2-669589_ChIPmentation-H3K4me1A00266_0289_2_S6_L002_I_peaks.narrowPeak
peaks/h3k4me1_ref_peaks/AF12_Flu_2-669587_ChIPmentation-H3K4me1_A00266_0289_2_S4_L002_I_peaks.narrowPeak
peaks/h3k4me1_ref_peaks/EU25_Flu_2-669590_ChIPmentation-H3K4me1A00266_0289_2_S7_L002_I_peaks.narrowPeak
peaks/h3k4me1_ref_peaks/AF24_Flu_2-674804_ChIPmentation-H3K4me1_A00266_0301_1_S5_L001_I_peaks.narrowPeak
peaks/h3k4me1_ref_peaks/AF36_Flu_2-669593_ChIPmentation-H3K4me1_A00266_0289_2_S10_L002_I_peaks.narrowPeak
peaks/h3k4me1_ref_peaks/AF04_Flu_2-674805_ChIPmentation-H3K4me1_A00266_0301_1_S6_L001_I_peaks.narrowPeak
peaks/h3k4me1_ref_peaks/EU13_Flu_2-669595_ChIPmentation-H3K4me1A00266_0289_2_S12_L002_I_peaks.narrowPeak
peaks/h3k4me1_ref_peaks/EU07_Flu_2-674802_ChIPmentation-H3K4me1_A00266_0301_1_S3_L001_I_peaks.narrowPeak
peaks/h3k4me1_ref_peaks/AF38_Flu_2-674837_ChIPmentation-H3K4me1_A00266_0301_2_S14_L002_I_peaks.narrowPeak
peaks/h3k4me1_ref_peaks/AF06_Flu_2-674806_ChIPmentation-H3K4me1_A00266_0301_1_S7_L001_I_peaks.narrowPeak
peaks/h3k4me1_ref_peaks/AF22_Flu_2-674827_ChIPmentation-H3K4me1_A00266_0301_2_S4_L002_I_peaks.narrowPeak
peaks/h3k4me1_ref_peaks/EU41_Flu_2-674832_ChIPmentation-H3K4me1A00266_0301_2_S9_L002_I_peaks.narrowPeak
peaks/h3k4me1_ref_peaks/EU29_Flu_2-674821_ChIPmentation-H3K4me1A00266_0301_1_S22_L001_I_peaks.narrowPeak
peaks/h3k4me1_ref_peaks/AF26_Flu_2-669591_ChIPmentation-H3K4me1_A00266_0289_2_S8_L002_I_peaks.narrowPeak
peaks/h3k4me1_ref_peaks/EU19_Flu_2-669585_ChIPmentation-H3K4me1A00266_0289_2_S2_L002_I_peaks.narrowPeak
peaks/h3k4me1_ref_peaks/AF16_Flu_2-669654_ChIPmentation-H3K4me1_A00266_0289_2_S21_L002_I_peaks.narrowPeak
peaks/h3k4me1_ref_peaks/AF08_Flu_2-669652_ChIPmentation-H3K4me1_A00266_0289_2_S19_L002_I_peaks.narrowPeak
peaks/h3k4me1_ref_peaks/EU33_Flu_2-674820_ChIPmentation-H3K4me1A00266_0301_1_S21_L001_I_peaks.narrowPeak
peaks/h3k4me1_ref_peaks/EU21_Flu_2-669653_ChIPmentation-H3K4me1A00266_0289_2_S20_L002_I_peaks.narrowPeak
indel_snarls/
indel_snarls/H3K4me1_indel_matched/
indel_snarls/H3K27ac_indel_matched/
indel_snarls/ATACseq_indel_matched/
indel_snarls/ATACseq_indel_matched/0000.vcf.gz
indel_snarls/ATACseq_indel_matched/0001.vcf.gz.tbi
indel_snarls/ATACseq_indel_matched/sites.txt
indel_snarls/ATACseq_indel_matched/0000.vcf.gz.tbi
indel_snarls/ATACseq_indel_matched/0001.vcf.gz
indel_snarls/ATACseq_indel_matched/README.txt
indel_snarls/H3K27ac_indel_matched/0000.vcf.gz
indel_snarls/H3K27ac_indel_matched/0001.vcf.gz.tbi
indel_snarls/H3K27ac_indel_matched/sites.txt
indel_snarls/H3K27ac_indel_matched/0000.vcf.gz.tbi
indel_snarls/H3K27ac_indel_matched/0001.vcf.gz
indel_snarls/H3K27ac_indel_matched/README.txt
indel_snarls/H3K4me1_indel_matched/0000.vcf.gz
indel_snarls/H3K4me1_indel_matched/0001.vcf.gz.tbi
indel_snarls/H3K4me1_indel_matched/sites.txt
indel_snarls/H3K4me1_indel_matched/0000.vcf.gz.tbi
indel_snarls/H3K4me1_indel_matched/0001.vcf.gz
indel_snarls/H3K4me1_indel_matched/README.txt
align.nf
```

call_peaks.nf
figures.R

| Genome browser session (e.g. UCSC) | No longer applicable |

## Methodology

| Replicates | 1 per sample |

| Sequencing depth | Sequencing depth H3K27ac:  Average 257 million reads per sample<br>H3Kme1: Average 288 million reads per sample<br>ATAC-seq; Average of 279 million reads per sample |

| Antibodies | H3K27ac (Diagenode antibody, cat # C15410196) and H3K4me1 (Cell Signaling antibody, cat # CST5326). Sequencing data is produced by another study (https://www.biorxiv.org/content/10.1101/2021.09.29.462206v2) |

| Peak calling parameters | We aligned H3K4me1 and H3K27ac binding sequences obtained from monocyte-derived macrophages from 30 individuals (Groza et al., 2022) using vg map (Garrison et al., 2018) to the hg38 reference genome graph and to the HPRC genome graph. Then, we called peaks using Graph Peak Caller v1.2.3 (Grytten et al., 2019) on both sets of alignments for each of the 30 H3K4me1 and H3K27ac samples. |

| Data quality | 5 million cells per sample, 100 bp paired-end, 150-500bp fragments |

| Software | vg map and Graph Peak Caller v1.2.3 |

# Flow Cytometry

## Plots

Confirm that:

☐ The axis labels state the marker and fluorochrome used (e.g. CD4-FITC).

☐ The axis scales are clearly visible. Include numbers along axes only for bottom left plot of group (a 'group' is an analysis of identical markers).

☐ All plots are contour plots with outliers or pseudocolor plots.

☐ A numerical value for number of cells or percentage (with statistics) is provided.

## Methodology

| Sample preparation | *Describe the sample preparation, detailing the biological source of the cells and any tissue processing steps used.* |

| Instrument | *Identify the instrument used for data collection, specifying make and model number.* |

| Software | *Describe the software used to collect and analyze the flow cytometry data. For custom code that has been deposited into a community repository, provide accession details.* |

| Cell population abundance | *Describe the abundance of the relevant cell populations within post-sort fractions, providing details on the purity of the samples and how it was determined.* |

| Gating strategy | *Describe the gating strategy used for all relevant experiments, specifying the preliminary FSC/SSC gates of the starting cell population, indicating where boundaries between "positive" and "negative" staining cell populations are defined.* |

☐ Tick this box to confirm that a figure exemplifying the gating strategy is provided in the Supplementary Information.

# Magnetic resonance imaging

## Experimental design

| Design type | *Indicate task or resting state; event-related or block design.* |

| Design specifications | *Specify the number of blocks, trials or experimental units per session and/or subject, and specify the length of each trial or block (if trials are blocked) and interval between trials.* |

| Behavioral performance measures | *State number and/or type of variables recorded (e.g. correct button press, response time) and what statistics were used to establish that the subjects were performing the task as expected (e.g. mean, range, and/or standard deviation across subjects).* |

## Acquisition

| | |
|---|---|
| Imaging type(s) | *Specify: functional, structural, diffusion, perfusion.* |
| Field strength | *Specify in Tesla* |
| Sequence & imaging parameters | *Specify the pulse sequence type (gradient echo, spin echo, etc.), imaging type (EPI, spiral, etc.), field of view, matrix size, slice thickness, orientation and TE/TR/flip angle.* |
| Area of acquisition | *State whether a whole brain scan was used OR define the area of acquisition, describing how the region was determined.* |

Diffusion MRI ☐ Used ☐ Not used

## Preprocessing

| | |
|---|---|
| Preprocessing software | *Provide detail on software version and revision number and on specific parameters (model/functions, brain extraction, segmentation, smoothing kernel size, etc.).* |
| Normalization | *If data were normalized/standardized, describe the approach(es): specify linear or non-linear and define image types used for transformation OR indicate that data were not normalized and explain rationale for lack of normalization.* |
| Normalization template | *Describe the template used for normalization/transformation, specifying subject space or group standardized space (e.g. original Talairach, MNI305, ICBM152) OR indicate that the data were not normalized.* |
| Noise and artifact removal | *Describe your procedure(s) for artifact and structured noise removal, specifying motion parameters, tissue signals and physiological signals (heart rate, respiration).* |
| Volume censoring | *Define your software and/or method and criteria for volume censoring, and state the extent of such censoring.* |

## Statistical modeling & inference

| | |
|---|---|
| Model type and settings | *Specify type (mass univariate, multivariate, RSA, predictive, etc.) and describe essential details of the model at the first and second levels (e.g. fixed, random or mixed effects; drift or auto-correlation).* |
| Effect(s) tested | *Define precise effect in terms of the task or stimulus conditions instead of psychological concepts and indicate whether ANOVA or factorial designs were used.* |

Specify type of analysis: ☐ Whole brain ☐ ROI-based ☐ Both

| | |
|---|---|
| Statistic type for inference<br>(See Eklund et al. 2016) | *Specify voxel-wise or cluster-wise and report all relevant parameters for cluster-wise methods.* |
| Correction | *Describe the type of correction and how it is obtained for multiple comparisons (e.g. FWE, FDR, permutation or Monte Carlo).* |

## Models & analysis

| n/a | Involved in the study |
|---|---|
| ☐ | ☐ Functional and/or effective connectivity |
| ☐ | ☐ Graph analysis |
| ☐ | ☐ Multivariate modeling or predictive analysis |

| | |
|---|---|
| Functional and/or effective connectivity | *Report the measures of dependence used and the model details (e.g. Pearson correlation, partial correlation, mutual information).* |
| Graph analysis | *Report the dependent variable and connectivity measure, specifying weighted graph or binarized graph, subject- or group-level, and the global and/or node summaries used (e.g. clustering coefficient, efficiency, etc.).* |
| Multivariate modeling and predictive analysis | *Specify independent variables, features extraction and dimension reduction, model, training and evaluation metrics.* |

