## [Peer Review File · Nature]

Manuscript Title: A Draft Human Pangenome Reference

Reviewer Comments & Author Rebuttals

Reviewer Reports on the Initial Version:

Referees' comments:

Referee #1 (Remarks to the Author):

The manuscript of Liao et al. reports on the assembly, validation and initial analysis of 47 human genome sequence assemblies. The efforts was conducted as part of the human pangenome consortium. The present paper is mainly a technical report on the development and evaluation of method for the construction of pangenome graphs and putting these data structure to good use, a task commonly thought of as the current grand challenge of genome informatics. There is nothing in this manuscript that struck me as a particularly interesting biological finding. What really sets it apart is its methods section, which is both exceptionally detailed and lucid. If all papers had method sections like this, there'd be no reproducibility crisis. The design of the computational experiments to validate pangenomes and their representations in figures is something that will inspire similar analysis in other pan-genome projects (of which there are now many in diverse species). This is a paper that people will read, think about and put to good use in their own research. Another aspect worth mentioning is that the applied software is brand new and under active development. The developers evaluated their respective tools on a common dataset of highest relevance and compared their results in the apparent absence of personal competition between them. This could easily have been otherwise and their decision to join forces will be to the great benefit of the wider pangenome community.

Apart from a question about how chromosome scale assemblies were constructed, I have no technical concerns. The validation approaches build on public data from domains that are ubiquitous in genomics (WGS, RNAseq, ChIPseq) and the rationales for computational experiments are well explained. I do have a few suggestions for improving clarity.

1. At first reading I was a bit disappointed about the lack of interesting biology. Recent papers about crop plant pangenome have illustrated the impact of SVs on agronomic phenotypes and proposed ways how this knowledge can be exploited in crop improvement. On second thought I realized that specific biological findings, which are harder to validate in human beings than in plants because of the need for suitable heterologous systems, might not fit into a technical paper of this breadth, depth and timeliness. The authors operate under the justified assumption that the value of a pangenome infrastructure for biological research is beyond doubt, that said infrastructure (algorithms and data structure) should be as good as possible, and that biological findings will be gleaned from them in future. This implicit argument (which I hope I'm correct to impute to the authors) should be made explicit.

2. One particular thing I wished to have seen was a the impact of pangenome graph on GWAS or other aspect of quantitative genetics. I understand that this entails re-analyzing read data (mapping it to the graph) and phenotypes of real people, something not easily possible because of privacy

rules (I guess). Anyway, I'd appreciate if the authors could give me some insights on one particular matter: GWAS models operate on biallelic variant matrices. How can the pan-genome graph be translated into such matrices and how can graph-derived matrices capture more information than conceptually much simpler methods like kmerGWAS (counting kmers from short reads).

3. I believe that the concluding sentence of the abstract is not the most effective way of getting readers that are disinterested in human genome informatics excited about this manuscript. To understand what reducing errors by 34 % means, the reader needs to know the baseline. The relevance of reducing a, say, 0.5 % error rate by one third, is not self-evident. Another, possibly more relevant, concern which the authors raise themselves, is that the truth sets are not so true after all because they were built from short reads. So the apparent precision (104 %, 34 %) is misleading. Also I misread the second sentence of the abstract in the sense that the 47 assemblies capture 99 % of structural variation (a commonly reported metric in pan-genome papers).

4. One aspect I'd learn more about is how the structural integrity of the haplotype-resolved assemblies was validated at the pseudomolecule level and how much effort was expended in manual curation. How many biologically true inversions and translocations within and between chromosomes were found? Are some of these consequences of tissue culture? Pstools checks haplotype phasing with Hi-C data not used in the assembly process. But how were chromosomal pseudomolecules constructed if no Hi-C data were used?

5. I may have overlooked it, but did not see statements about data and code availability. Obviously, all data (assemblies and underlying raw data and other types of data) need to be publicly released. Also the pangenome graphs should be put up for download.

6. The top most dot in the q arm panel of Supplementary Figure 52 overlaps with the legend (p value).

Referee #2 (Remarks to the Author):

This paper is a feat of science. The team behind this paper have done a tremendous amount of work sequencing, assembling and representing the resulting assemblies as a graph. This is something that the community has talked about for a while, but only now has this been possible thanks to the development of sequencing techniques, algorithmic method development and graph tool building over the last few years.

The results of this paper are first, the pangenome built from the haplotype assemblies generated and second the downstream analysis that show the benefits of moving away from a linear genome reference and using a pangenome approach.

The paper reads quite well, except for two points. First, some sections of the results are about methods used to get results, not the results (see minigraph and pangenome below), merging them with the results part would improve the flow. Second, the choice of MC vs PGGB should be motivated and really shouldn't be an either/or, but rather presented as choosing a representation that fits what your endgoal is. This is addressed in the discussion but would be better to repeat in the results.

I have three major issues that should be easy to fix/address.

1. RNA-seq mapping analysis

This part of the manuscript is a bit thin in my opinion. The role of this part is to show the downstream improvements of using a pangenome graph for the analysis as opposed to a linear reference. The key result is in panel B, Figure 7, where the comparison between STAR and vg mpmmap is used. By using two different programs and two different references it remains unclear whether the improvement is because of the improved reference or the programs. I would suggest running vg mpmmap on the grc38 linear graph with the same splice junctions included as STAR (or at least the same GTF file).

Another simple experiment to run is to generate the truth data from the linear reference and compare the two (as a supp fig) to try to quantify how much of the difference is because of the programs and how much of the difference is due to the reference used.

The use of MAPQ numbers to compare the two is a bit artificial, since STAR has reports only 0,1,255 values (255 being uniquely mapped), using MAPQ as a cutoff makes sense when comparing two outputs from the same program.

For most downstream analysis you would either use all multimapping reads or only uniquely mapping reads.

Finally the authors explicitly mention discarding transcripts from patches or alternative contigs (page 45, ensembl mapping) when mapping genes to the assemblies, this is fine since these are copies of the gene annotations of the reference chromosomes (except maybe for the HLA alt contigs).

But using the patches for analysis has benefits when aligning to the transcriptome and not the genome. This is necessary when quantifying isoforms and can be done using e.g. hisat for

transcriptome alignment followed by rsem. It would be interesting to see the difference between using the transcriptome with patches, e.g. full ensembl GTF, vs transcriptome + pangenome.

2. page 8, 2nd para

"indicating that .. are structurally correct"

this is a bit of a leap in my opinion, the only conclusion is that the regions containing genes are structurally correct, but this analysis does not show that the "vast majority of annotated haplotypes are structurally correct". I would recommend revising this sentence to reflect that for purposes of gene analysis the haplotypes are structurally correct.

3. page 18, figure 5

The bandage layout is convenient when you are working with graphs interactively. For this presentation I would recommend drawing the contigs and the gene locations by hand. It is hard to discern where RHD starts and ends in the graph, also it seems that one of the contigs is traversed backwards and forwards (shown by arrows in panel B).

Once the graph is drawn you can show each of the haplotypes in panel B as a path through the graph. This representation is in line with the language used in the paper and is much easier to follow rather than a color gradient.

Minor concerns

page 2, abstract

- "reduces errors by 34%"

not clear what measurement is used here, sensitivity or specificity?

- boosts the detected structural variants,

the SV's are not boosted, the numbers are (presumably) increased.

page 5, line 6

- Using short substrings (k-mers)

what value of k?

- Yak (Cheng et al. 2021)

Reading this it was unclear that Yak refers to a program

page 6, 3rd para

- The average number of the t2t-chm13 bases with two ...

not clear enough what is being aligned to what and what the numerator is.

Are the sequences in the t2t assembly aligned to the reference?

- (4.99/199Mb,)

skip / , write out of

Figure 2

panel B

this plot is indexed by genes that are duplicated ordered by copy number. The plot does not make good use of the space (large white space) and makes it impossible to see how many of the genes are merely 2,3, copies. I would recommend turning this into a histogram of (x) copy number repeat and (y) number of genes with that copy number

panel C

more or less the same comment, the ordering is fairly arbitrary and it would be

better to turn it into a histogram of duplications (x-axis) colored by ancestry (fill), this way you could glean from the plot whether AFR ancestry have higher duplication calls than others.

page 11, para 1

"generating a combined ... is non-trivial"

There are two ways to read this, it is computationally hard (agreed) and there are trade-offs between representations (also agreed). Perhaps it would be better to rephrase this and emphasize the purpose that the pangenome serves and that different application may benefit from different representations.

page 11, para 2 and 3

"Minigraph" and "pccb"

These paragraphs read more as a methods part than results. The results for these two methods are mentioned on the next page.

page 12, para 3

"dipcall confident regions"

First mention of Dipcall in the paper, the reference and definition of "easy regions" is on the next page

page 13, para 1

"comparative genomics"

odd reference, pdf points to paperpile link

page 13, para 2

"2.4% are missing from gnomAD"

Are they present in the variant calls from 150K sequences of UK Biobank?

page 14, fig 3

panel A, the text should be monospace, the addition of - does not align properly and makes it hard to read and compare to the graph

page 17, para 2

"Figure 5, top"

Which panels

page 21, para 1

1000 Genomes, vs 1KG

There is a naming inconsistency throughout, sometimes the abbreviation is used, sometimes not and in one instance it uses 1000GP (page 23)

page 23, para 4

"is modest but consistent"

modest should be quantified, also make it clear that this derives from supp fig 38

Referee #3 (Remarks to the Author):

This manuscript presents a draft human pan-genome reference derived from diploid assemblies of 47 diverse individuals. It describes multiple applications of and advantages of using a pangenome vs. a linear reference, including improving variant calling particularly for SVs. There are also more modest gains in performance for small variant calling, with most gains in difficult to call regions.

Overall, this study represents a major milestone in moving away from a limited linear reference toward a reference that considers a more complete set of genetic variation. It also produces many valuable community resources (pangenome graphs themselves, improved 1000G callset, and software tools). I have the following suggestions that may help improve the broader utility of this work:

Data availability: there is reference to a data availability section, but I cannot seem to find one. I assume it will be added to the final publication but would be nice to see what those will look like. The HPRC perspective from earlier this year indicates data will be released on S3, google, and other cloud platforms. It will also be helpful to guide users in which of the three pangenomes (minigraph, MC, pggp) might be most useful for different applications.

Applications of pangenomes and showing where they are most useful: My impression has been that some of the main advantages of pangenomes would be: (1) improved structural variation (SV) representation and detection, but also (2) reducing biases in our analyses that stem from mapping to a single (mosaic) linear genome that does not adequately represent diverse human populations. There are many analyses that clearly demonstrate #1, but almost nothing on #2. Although the draft pangenome includes individuals from diverse populations (Fig. 1), and the 2022 perspective mentions "making genotyping accuracy less dependent on ancestry", this is not really highlighted in the current manuscript. For example, is the improvement in variant calling more pronounced for individuals from populations not represented in GRCh38/T2T-CHM13 that are represented in the pangenome?

Also, a large part of the paper focuses on the assemblies (the first two main figures) and it takes a while to get to the actual pangenomes. Even after pangenomes are introduced, a lot of focus (until page ~17/25) is put on small variants, where the gains are real but less exciting than those for the SVs.

Validation of new variants: given that a main selling point of the pangenomes approach is that it enables discovery of novel variants, it would be helpful to have a form of external validation for some of those. For short variants (see Page 16: "suggesting that most putative errors are in fact real variants") -> can some of those be confirmed by e.g. Sanger? For new SVs (e.g. shown in Fig. 6) -> is there a way to confirm some of those? I guess those are difficult to PCR, but perhaps evaluating if there is evidence in existing long read data for some of those samples that the new SVs are real?

The paper nicely shows examples of structurally complex regions (e.g. C4, HLA, LPA) that have already been well characterized. How rare is this type of complexity? Does the pangenome approach enable detection of new regions not known previously to be so complex that are likely to be medically relevant?

Additional points:

The abstract and discussion claim to reduce errors when discovering small variants by 34%, but I am not quite sure where that number comes from. I think it is from Sup. Figure 41 but am not sure how that number was derived.

For VNTRs: I am confused by the notion of TP/TN, etc. which seems to imply the presence or absence of a variant. My impression was that the VNTR itself exists in all genome copies, but then there is variation in copy number. Only upon reading the methods I realized these annotations seem to be based on the read mappings, not discovery of a variant itself. It would be helpful to define in the main text or legend how these “positive” vs. “negative”s are defined.

How does VNTR copy number estimation change as a function of the repeat unit length or total length of the VNTR? Also the performance numbers seem modest compared to other short-read approaches based on coverage (e.g. Mukamel et al Science 2021). Is the poor performance due to lower overall coverage here?

Can VNTR copy number be assessed on the pangenome graphs using something other than coverage? Finally, how was “ground truth” defined for VNTR lengths?

The RNA-seq/ChIP-seq sections are quite limited and have modest results. Related to the comment above on diverse populations, I wonder if this could improve e.g. allele-specific binding or ASE estimates that face a problem with reference bias, especially in diverse genomes.

Minor points:

The asmgene feature of pafTools.js doesn't seem to have usage documentation associated with it. <https://github.com/lh3/minimap2/search?q=asmGene>. How does it distinguish true duplications from errors?

Fig. 3a: would be nicer if the “-”s were monospace so everything is aligned

Fig. 3b: I don't see a bar for minigraph but it is in the legend. I guess it is not included because it focused on larger SVs and not small variants?

Fig. 2c legend: it says duplications are relative to GRCh38 (152) but the y-axis seems to show raw numbers. Also, is it surprising that some African genomes have substantially higher numbers than the other samples?

How are the gene diagrams generated for fig. 5c/f (and similar supplementary figures for other

regions?) Do the segments need to be manually annotated? Can this type of annotation be easily done for newly discovered complex regions? If the manual annotations are taken from elsewhere, it would be helpful to cite where those came from (e.g. it looks like Supp fig. 14 C4 haplotype annotations are based on Sekar et al. 2016).

If there is a concise diagram that could capture it, it could be helpful to somehow conceptually illustrate the differences in how the graphs for the three methods (Minigraph, MC, pggb) are constructed or how they represent certain regions in different ways. This could also help community users decide which graph is most appropriate for their use case.

For Fig. 4 A/B, it would be helpful to label the number of variants falling in each category.

Author Rebuttals to Initial Comments:

Referees' comments:

Referee #1 (Remarks to the Author):

The manuscript of Liao et al. reports on the assembly, validation and initial analysis of 47 human genome sequence assemblies. The efforts was conducted as part of the human pangenome consortium. The present paper is mainly a technical report on the development and evaluation of method for the construction of pangenome graphs and putting these data structure to good use, a task commonly thought of as the current grand challenge of genome informatics. There is nothing in this manuscript that struck me as a particularly interesting biological finding. What really sets it apart is its methods section, which is both exceptionally detailed and lucid. If all papers had method sections like this, there'd be no reproducibility crisis. The design of the computational experiments to validate pangenomes and their representations in figures is something that will inspire similar analysis in other pan-genome projects (of which there are now many in diverse species).

This is a paper that people will read, think about and put to good use in their own research. Another aspect worth mentioning is that the applied software is brand new and under active development. The developers evaluated their respective tools on a common dataset of highest relevance and compared their results in the apparent absence of personal competition between them. This could easily have been otherwise and their decision to join forces will be to the great benefit of the wider pangenome community.

Apart from a question about how chromosome scale assemblies were constructed, I have no technical concerns. The validation approaches build on public data from domains that are ubiquitous in genomics (WGS, RNAseq, ChIPseq) and the rationales for computational experiments are well explained. I do have a few suggestions for improving clarity.

We thank the reviewer for the positive feedback, and have maintained these aspects of the paper.

1. At first reading I was a bit disappointed about the lack of interesting biology. Recent papers about crop plant pangenome have illustrated the impact of SVs on agronomic phenotypes and proposed ways how this knowledge can be exploited in crop improvement. On second thought I realized that specific biological findings, which are harder to validate in human beings than in plants because of the need for suitable heterologous systems, might not fit into a technical paper of this breadth, depth and timeliness. The authors operate under the justified assumption that the value of a pangenome infrastructure for biological research is beyond doubt, that said infrastructure (algorithms and data structure) should be as good as possible, and that biological findings will be gleaned from them in future. This implicit argument (which I hope I'm correct to impute to the authors) should be made explicit.

The reviewer is correct that the focus of this paper is upon building a new reference resource for human genomics rather than exploring the inherent biology. We modified the first paragraph of the discussion to be more explicit about this as follows:

“While the focus of this effort has been to build a reference resource, highly accurate haplotype-resolved assemblies allow us to access previously inaccessible regions highlighting novel forms of genetic variation (as in Figure 5) and providing new insights into mutational processes such as interlocus gene conversion (see Vollger et al, companion).”

We hope that this statement of intent is helpful, and is balanced with the fact that there are some interesting biological results tucked into the paper and within companion manuscripts. Notably, and partly quoting from the paper:

The discovery of several novel haplotypes at complex loci, for example: *“In the RHD/RHCE (Figure 5A-C), in addition to previously described haplotypes, we also inferred the presence of 5 novel haplotypes, which included one duplication allele of the RHD gene, and one inversion allele occurring between the RHD and RHCE gene that results in the swapping of the last exon*

of both genes. Around HLA-A (Figure 5D-F; Supplementary Figure 17), two deletion alleles have been previously described – albeit with imprecise breakpoints (Sudmant et al., 2015) – but an insertion allele carrying an HLA-Y pseudogene is previously unreported. The long sequence (65 kb) inserted with HLA-Y occurs at high frequency (28%) but has little homology to GRCh38.”

The haplotype resolved confirmation of the existence of many rare copy-number variations of genes that lie outside of existing SDs: “There are 1,115 protein-coding gene families within the Flagger predicted reliable regions of the full set of assemblies that have a gain in copy number in at least one genome (Figure 2B). Each assembly has an average of 36 genes with a gain in copy number relative to GRCh38 within its predicted reliable regions, with a bias towards rare, low-copy CNVs (Figure 2C); 71% of CNV genes appear in a single haplotype. Previous studies using read depth found that rare CNVs occur generally outside of regions annotated as being enriched in SDs (Sudmant et al., 2010).”

A significant contribution to understanding how much euchromatic sequence is commonly polymorphic: “Overall, the euchromatic autosomal non-reference sequence adds up to ~175 Mb in MC (and ~190 Mb in PGGB), out of which ~55 Mb (~105 Mb) are observed only on a single haplotype. Our analysis further suggests that ~5 Mb and ~70 Mb (~10 Mb and ~60 Mb) can be attributed to core (present in $\geq 95\%$ of all haplotypes) and common genome (present in $\geq 5\%$ of all haplotypes). ... Extrapolating under Heaps’ Law (Tettelin et al., 2008) (Methods), we expect at least an additional ~150 Mb of euchromatic autosomal sequence in the pangenome graph when HPRC produces 700 haplotypes in future.”

We also provide concrete evidence of strong homology between acrocentric chromosome short arms (Figure 1B). In a companion paper, we investigate further to explore the hypothesis that an unknown mechanism (likely non-homologous recombination) is maintaining homology between these elements.

Finally, with regards to efforts to create pangenomes for other species, we added the following sentence to the discussion with references to the papers mentioned by the reviewer:

“The methods we are developing should prove valuable for other species, and indeed other groups are pioneering such efforts.”

The referenced tomato paper used our vg and vg giraffe, for example.

2. One particular thing I wished to have seen was a the impact of pangenome graph on GWAS or other aspect of quantitative genetics. I understand that this entails re-analyzing read data (mapping it to the graph) and phenotypes of real people, something not easily possible because of privacy rules (I guess). Anyway, I’d appreciate if the authors could give me some insights on one particular matter: GWAS models operate on biallelic variant matrices. How can the pangenome graph be translated into such matrices and how can graph-derived matrices capture more information than conceptually much simpler methods like kmerGWAS (counting kmers from short reads).

The reviewer is absolutely correct that using the pangenome to enable the GWAS of SVs and other variants missed by standard single-reference genome approaches, is an obvious and desirable objective. However, for the reasons the reviewer states, we believe this is well outside of the scope of this current study and worthy of future, standalone efforts. We have now mentioned this in the discussion of the paper as follows:

“The draft pangenome therefore delivers much better SV calling than earlier approaches, extracting latent information from short-read samples that are already available, and so in the future enabling the inclusion of tens of thousands of additional SV alleles into genome-wide association studies (GWAS).”

With respect to the reviewer's question about biallelic variant matrices, the results of the variant calling pipelines presented (both SVs and point variants) are VCF files and can be manipulated as such, as in standard GWAS pipelines. Furthermore, the large majority of the individual variant sites are biallelic (see Figure 3B-C) for both SNPs and SVs. For the remainder, some of which are common (Figures 3E-F), one possible approach is to convert each multi-allelic variant into a set of bi-allelic variants that in combination encode the possible multiple alleles.

3. I believe that the concluding sentence of the abstract is not the most effective way of getting readers that are disinterested in human genome informatics excited about this manuscript. To understand what reducing errors by 34 % means, the reader needs to know the baseline. The relevance of reducing a, say, 0.5 % error rate by one third, is not self-evident. Another, possibly more relevant, concern which the authors raise themselves, is that the truth sets are not so true after all because they were built from short reads. So the apparent precision (104 %, 34 %) is misleading. Also I misread the second sentence of the abstract in the sense that the 47 assemblies capture 99 % of structural variation (a commonly reported metric in pan-genome papers).

We appreciate the reviewers concern. We have modified the sentence to:

“Using our draft pangenome to analyze short-read data reduces combined false positive and false negative errors when discovering small variants by 34% and increases the number of detected structural variants per haplotype by 104% compared to GRCh38-based workflows, allowing the typing of the significant majority of SV alleles present in each genome.”

We hope that these changes make it clearer what the 34% reduction refers to and contextualizes the 104% statistic with regards to SVs. While we appreciate that a 34% reduction of an already small error rate is a small number of errors being removed (order of thousands of variants), we would like to stress that in our view such improvements are important, particularly for rare variant workflows where candidate variant sets are often highly enriched for false positive variants.

With regards to the second sentence, we modified it to be clearer as follows:

*“These assemblies cover more than 99% of the expected sequence **in each genome** and are more than 99% accurate at the structural and base-pair levels.”*

4. One aspect I'd learn more about is how the structural integrity of the haplotype-resolved assemblies was validated at the pseudomolecule level and how much effort was expended in manual curation. How many biologically true inversions and translocations within and between chromosomes were found? Are some of these consequences of tissue culture? Pstools checks haplotype phasing with Hi-C data not used in the assembly process. But how were chromosomal pseudomolecules constructed if no Hi-C data were used?

To confirm inversions, we utilized strand-seq data, mapped to the assemblies. This analysis on a subset of HPRC assemblies is reported in a companion paper <https://doi.org/10.1101/2022.07.06.498874>. This paper reveals that 6-7 Mbp of DNA are incorrectly orientated per haplotype. We have now summarized this finding in the discussion:

*“There are many remaining challenges in growing and refining this reference. For example, assembly reliability analysis revealed roughly an order of magnitude more erroneously assembled sequences in the HPRC assemblies than in the T2T-CHM13 complete assembly. **Similarly, in a companion analysis, Strand-seq data analysis of a subset of assemblies reveals 6-7 Mb of incorrectly oriented sequence per haplotype (see Porubsky et al, companion), indicating that there is room to structurally improve the assemblies.**”*

With respect to tissue culture artifacts, as explained in the text, we attempted to avoid gross chromosomal abnormalities acquired in tissue culture by using minimally passaged lines (1-3 passages) and performing karyotyping and microarray analysis before sequencing. To attempt to actually quantify the effects of tissue culture in the Jarvis et al. companion, which is now published in Nature (<https://www.nature.com/articles/s41586-022-05325-5>), we describe an experiment to sequence the genome of blood from a participant (HG06807) and ask if there were differences with the tissue culture derived genome. While obviously a very limited result, we did not find any change in mosaicism, but three small inversions (1.6-10 kb in size) in one of the haplotypes of the cultured cells. These findings suggest small SV changes in LCL cultures. We pointed readers to this study for cell culture analyses.

With respect to the Hi-C data, we did not use it for phasing the assemblies. Instead we used trio information from the sequencing of the parents of each sample. The algorithmic approach is described in detail in the trio-Hifasm paper (<https://www.ncbi.nlm.nih.gov/pmc/articles/PMC7961889/>). Hifasm Hi-C phasing was not yet available and, furthermore, does not appear to be superior to using paternal and maternal k-mer libraries.

We also did not use Hi-C to create scaffolds (i.e., the pseudomolecules the reviewer refers to), but are rather contig only assemblies. In our bakeoff study

[\(\(https://www.nature.com/articles/s41586-022-05325-5\)\)](https://www.nature.com/articles/s41586-022-05325-5) we show that the scaffolding tools available at the time when we started sequencing (in 2020) were too error prone.

5. I may have overlooked it, but did not see statements about data and code availability. Obviously, all data (assemblies and underlying raw data and other types of data) need to be publicly released. Also the pangenome graphs should be put up for download.

We thank the reviewer for pointing this out and apologize for the unintentional omission. We have now added a “Data Availability” section to the “Methods”. This section details the availability of the sequencing data, assemblies, pangenomes, variant call sets and annotations.

6. The top most dot in the q arm panel of Supplementary Figure 52 overlaps with the legend (p value).

Thanks, this has been fixed.

Referee #2 (Remarks to the Author):

This paper is a feat of science. The team behind this paper have done a tremendous amount of work sequencing, assembling and representing the resulting assemblies as a graph. This is something that the community has talked about for a while, but only now has this been possible thanks to the development of sequencing techniques, algorithmic method development and graph tool building over the last few year.

We thank the reviewer for their positive sentiment; it is collectively appreciated!

The results of this paper are first, the pangenome built from the haplotype assemblies generated and second the downstream analysis that show the benefits of moving away from a linear genome reference and using a pangenome approach.

The paper reads quite well, except for two points. First, some sections of the results are about methods used to get results, not the results (see minigraph and pangenome below), merging them with the results part would improve the flow. Second, the choice of MC vs PGGB should be motivated and really shouldn't be an either or, but rather presented as choosing a representation that fits what your end goal is. This is addressed in the discussion but would be better to repeat in the results.

We have removed duplication in the results section with the methods section by substantially shortening the introductory overview of the different pangenome alignment construction methods. We now spend one or two sentences on each method to give a very high-level overview and have removed the subsection titles that made the section feel like it belonged in the methods.

With regards to the either/or question about which pangenome to prefer, all three (M, MC, and PGGB) methods were actively developed or extended as part of this project and we hope that the results are convincing that there is strong high-level agreement between the graphs within the euchromatic portions of the graph (for example by the analysis of variants and by the analysis of complex loci). Our intention was to show that by completely independent methods we could achieve strong agreement in the shape of the human pangenome alignment. We believe this is an important result. While we appreciate it would be useful to provide guidance on which method to prefer for a particular analysis, with the exception of the mapping-based analyses where we explain that we are limited to using the MC graph, at this point it is unclear if the choice substantially matters.

I have three major issues that should be easy to fix/address.

1. RNA-seq mapping analysis

This part of the manuscript is a bit thin in my opinion. The role of this part is to show the downstream improvements of using a pangenome graph for the analysis as opposed to a linear reference. The key result is in panel B, Figure 7, where the comparison between STAR and vg mpmmap is used. By using two different programs and two different references it remains unclear whether the improvement is because of the improved reference or the programs. I would suggest running vg mpmmap on the grc38 linear graph with the same splice junctions included as STAR (or at least the same GTF file). Another simple experiment to run is to generate the truth data from the linear reference and compare the two (as a supp fig) to try to quantify how much of the difference is because of the programs and how much of the difference is due to the reference used.

The use of MAPQ numbers to compare the two is a bit artificial, since STAR has reports only 0,1,255 values (255 being uniquely mapped), using MAPQ as a cutoff makes sense when comparing two outputs from the same program. For most downstream analysis you would either use all multimapping reads or only uniquely mapping reads.

Finally the authors explicitly mention discarding transcripts from patches or alternative contigs (page 45, ensembl mapping) when mapping genes to the assemblies, this is fine since these are copies of the gene annotations of the reference chromosomes (except maybe for the HLA alt contigs). But using the patches for analysis has benefits when aligning to the transcriptome and not the genome. This is necessary when quantifying isoforms and can be done using e.g. hisat for transcriptome alignment followed by rsem. It would be interesting to see the difference between using the transcriptome with patches, e.g. full ensembl GTF, vs transcriptome + pangenome.

The reviewer makes a good point regarding our comparison with vg mpmmap on the 1000 Genomes Project-derived graph. Comparing instead to vg mpmmap on a linear reference

provides both a more apples-to-apples comparison to STAR and a more meaningful comparison to the HPRC graph in the context of this paper. We have updated the analysis accordingly.

The reviewer is also correct that quantitative MAPQs are less-commonly used in RNA-seq informatics than in DNA-seq informatics. We believe this is in part because many of the most widely-used RNA-seq mapping tools do not adequately estimate mapping quality according to its definition. Nevertheless, we agree that the results will be more accessible to geneticists if we stick to the more familiar distinction between uniquely-mapped and multi-mapped reads. We have replaced the MAPQ-thresholded precision-recall plot with a pair of precision-recall plots stratified only by unique mapping, with no major change in conclusions.

Following the reviewer's suggestion, we also performed an experiment comparing expression inference with a pangenome-based pipeline to linear reference pipelines, both with and without alternative contigs (**Supplementary Figure 43**). In contrast to the reviewer's expectation, we find that the alternative contigs generally lead to reduced accuracy in gene-level expression inference, measured globally across genes. Moreover, the pangenome pipeline's accuracy exceeds that of the linear reference pipelines on the gene-level.

2. page 8, 2nd para

"indicating that .. are structurally correct"

this is a bit of a leap in my opinion, the only conclusion is that the regions containing genes are structurally correct, but this analysis does not show that the "vast majority of annotated haplotypes are structurally correct". I would recommend revising this sentence to reflect that for purposes of gene analysis the haplotypes are structurally correct.

We agree and apologize that this was an overinterpretation. We have changed this sentence to: *"..., indicating that in terms of gene structure the vast majority of the annotated haplotypes are structurally correct."*

3. page 18, figure 5

The bandage layout is convenient when you are working with graphs interactively. For this presentation I would recommend drawing the contigs and the gene locations by hand. It is hard to discern where RHD starts and ends in the graph, also it seems that one of the contigs is traversed backwards and forwards (shown by arrows in panel B). Once the graph is drawn you can show each of the haplotypes in panel B as a path through the graph. This representation is in line with the language used in the paper and is much easier to follow rather than a color gradient.

To address this concern we added lines alongside the Bandage plots in Figure 5 A,B, and E and Supplementary Figures 14, 15, 16, 17, and 18 to show RHD/RHCE and other gene locations and paths more clearly. We hope the inclusion of these lines, in addition to the paths, will allow readers to more clearly discern the features of the genes.

Minor concerns

page 2, abstract

- "reduces errors by 34%"

not clear what measurement is used here, sensitivity or specificity?

To address the mentioned ambiguity this sentence is rephrased to:

"Using our draft pangenome to analyze short-read data reduces combined false-positive and false-negative errors when discovering small variants by 34%...."

-boosts the detected structural variants,

the SV's are not boosted, the numbers are (presumably) increased.

We changed this sentence to: *"... boosts the number of detected structural variants ..."*.

page 5, line 6

- Using short substrings (k-mers)

what value of k?

A k-mer size of 31 was used. We now mention the k-mer size in the text.

- Yak (Cheng et al. 2021)

Reading this it was unclear that Yak refers to a program

Thanks for pointing this out. Yak is a k-mer analyzer. We have rephrased the sentence to make this clearer.

page 6, 3rd para

-The average number of the t2t-chm13 bases with two ...

not clear enough what is being aligned to what and what the numerator is.

Are the sequences in the t2t assembly aligned to the reference?

We aligned our assemblies to the T2T-CHM13 reference. We rephrased the sentence to make this clearer.

- (4.99/199Mb,)

skip / , write out of

Apologies, we replaced "/" by "out of".

Figure 2

panel B

this plot is indexed by genes that are duplicated ordered by copy number. The plot

does not make good use of the space (large white space) and makes it impossible to see how many of the genes are merely 2,3, copies. I would recommend turning this into a histogram of (x) copy number repeat and (y) number of genes with that copy number

As suggested, we have substituted the following histogram with the information suggested. Because any gene may have multiple different copy number states in different individuals, this histogram reflects all observations of copy number combined across assemblies. Note in response to Referee 3, we have applied the Flagger filtering to all gene annotations in addition to duplicated gene annotations, resulting in a change in the distribution of copy number variants towards fewer gene copies.

We have modified the caption for Figure 2B from:

“Assembled gene duplications per genome. The number of genomes containing a duplicated gene for 1529 protein-coding gene duplications indexed by increasing copy number, observed in the predicted reliable regions of the HPRC/HPRC+ genomes.”

To:

“Frequency of gene copy number. Individual genes may have separate copy number states among genomes, and the frequency reflects 3,210 observed copy number changes among the HPRC/HPRC+ genomes.”

panel C

more or less the same comment, the ordering is fairly arbitrary and it would be

better to turn it into a histogram of duplications (x-axis) colored by ancestry (fill), this way you could glean from the plot whether AFR ancestry have higher duplication calls than others.

We feel the pattern of increased number of duplicated genes in genomes with AFR ancestry is largely visible in the existing figure 2C. Because the sample sizes are relatively small, a histogram would require coarse binning of values, hiding some of the information currently shown.

Finally, we have updated the figure legend to clarify that Figure 2D reflects a subset of all duplicated genes.

“The GRCh38 gene duplications reflect families of duplicated genes, while the counts in other genomes reflect gene duplication polymorphisms. The assemblies are color coded according to their population of origin. D) The top 25 most commonly CNV genes or gene-families in the HPRC/HPRC+ assemblies, “

to:

“The GRCh38 gene duplications reflect families of duplicated genes, while the counts in other genomes reflect gene duplication polymorphisms. The assemblies are color coded according to their population of origin. D) The top 25 most commonly CNV genes or gene-families in the HPRC/HPRC+ assemblies out of all 1,115 duplicated genes.”

This conveys the overall fraction of the haplotypes in the pangenome that have at least one duplicated sequence. It also relays which genes are duplicated. It would be difficult to convey the same information in a histogram, and a cutoff is required considering the large number of duplicated genes.

page 11, para 1

"generating a combined ... is non-trivial"

There are two ways to read this, it is computationally hard (agreed) and there are trade-offs between representations (also agreed). Perhaps it would be better to rephrase this and emphasize the purpose that the pangenome serves and that different application may benefit from different representations.

We agree; the mentioned paragraph is rephrased to clarify these points:

“The process of generating a combined pangenome representation is an active research area. The problem is non-trivial both because of the potential computational challenges (there are hundreds of billions of bases of sequence to align) and because determining which alignments to include is not always obvious, particularly for recently duplicated and repetitive sequences. We applied three different graph construction methods that have been under active development for this project: Minigraph (Li et al., 2020), Minigraph-Cactus (MC), and

PanGenome Graph Builder (PGGB) (Methods). The availability of these three models provides us with multiple views into the homology relationships in the pangenome while supporting cross-validation of discovered variation by independent methods."

page 11, para 2 and 3

"Minigraph" and "pggb"

These paragraphs read more as a methods part than results. The results for these two methods are mentioned on the next page.

These two paragraphs have been removed from the results part, with a much briefer overview of the pangenome construction methods now given, and the details that were present have instead been combined with related paragraphs in Methods. (Section "Pangenome Graph Construction")

page 12, para 3

"dipcall confident regions"

First mention of Dipcall in the paper, the reference and definition of "easy regions" is on the next page

We added a reference and changed this sentence to "... confident regions, which were based on alignments of each sample's assembly to the GRCh38 reference using Dipcall (Li et al., 2018): ..."

page 13, para 1

"comparative genomics"

odd reference, pdf points to paperpile link

*We have fixed the reference and now "Tettelin et al., 2008" is used to refer to the article "Tettelin, H., Riley, D., Cattuto, C., & Medini, D. (2008). Comparative genomics: the bacterial pan-genome. *Current Opinion in Microbiology*, 11(5), 472–477."*

page 13, para 2

"2.4% are missing from gnomAD"

Are they present in the variant calls from 150K sequences of UK Biobank?

We unfortunately do not have access to the UK Biobank data currently. We agree that this would be an interesting cross-check and plan to examine the question in more detail in the future.

page 14, fig 3

panel A, the text should be monospace, the addition of - does not align properly and makes it hard to read and compare to the graph

We have updated the figure accordingly.

page 17, para 2
"Figure 5, top"
Which panels

We replaced "Figure 5, top" with "Figure 5A-C" to refer to the panels specifically.

page 21, para 1
1000 Genomes, vs 1KG
There is a naming inconsistency throughout, sometimes the abbreviation is used, sometimes not and in one instance it uses 1000GP (page 23)

We have fixed this and now only use "1KG" to refer to the 1000 Genomes.

page 23, para 4
"is modest but consistent"
modest should be quantified, also make it clear that this derives from supp fig 38

We now clarify that "modest" refers to an increase of 0.006-0.011 across different data sets. We have also altered the presentation to remove the varying MAPQ thresholds. Instead, we use a single threshold of $\text{MAPQ} \geq 30$.

Referee #3 (Remarks to the Author):

This manuscript presents a draft human pan-genome reference derived from diploid assemblies of 47 diverse individuals. It describes multiple applications of and advantages of using a pangenome vs. a linear reference, including improving variant calling particularly for SVs. There are also more modest gains in performance for small variant calling, with most gains in difficult to call regions.

Overall, this study represents a major milestone in moving away from a limited linear reference toward a reference that considers a more complete set of genetic variation. It also produces many valuable community resources (pangenome graphs themselves, improved 1000G callset, and software tools).

We thank the reviewer for their positive words!

I have the following suggestions that may help improve the broader utility of this work:

Data availability: there is reference to a data availability section, but I cannot seem to find one. I assume it will be added to the final publication but would be nice to see what those will look like. The HPRC perspective from earlier this year indicates data will be released on S3, google, and other cloud platforms.

We apologize for the unintentional omission. We have now added a “Data Availability” section to “Methods”. This section details the availability of the sequencing data, assemblies, pangenomes, variant call sets and annotations.

It will also be helpful to guide users in which of the three pangenomes (minigraph, MC, pggb) might be most useful for different applications.

We appreciate the desire for concrete answers here. In the revised discussion we write:

“We demonstrate concordance between these different construction approaches; the MC and PGGB pangenomes contain nearly the same number of small variants and SVs of various types. Further, these encoded pangenome variants show high levels of agreement with existing linear reference-based methods for variant discovery, particularly within the non-repetitive fraction of the genome. Where the pangenome drafts presented differ is principally in how they handle CNV sequences. The PGGB method will frequently merge copies of a CNV, while the MC graphs represent CNV copies as independent subgraphs. Both approaches have merits, and which approach to favor will take further experimentation and community input, and may vary by the specific application. The PGGB method retained all centromeric and satellite sequences, while the MC graph pruned much of this sequence. This made it practical with current methods to use the MC graphs for read alignment applications. However, pruning these sequences is not a satisfactory solution. Longer-term, more work is needed to determine how best to align and represent these large repeat arrays within pangenomes, particularly as T2T assembly becomes commonplace and these arrays are therefore finished.”

We feel this is representative of what we can currently conclude, that is: (i) all three methods are available and applicable to datasets such as that presented, (ii) that they share high-level agreement and that (iii) further work is needed to improve them, particularly with regard to satellite sequences.

Applications of pangenomes and showing where they are most useful: My impression has been that some of the main advantages of pangenomes would be: (1) improved structural variation (SV) representation and detection, but also (2) reducing biases in our analyses that stem from mapping to a single (mosaic) linear genome that does not adequately represent diverse human populations. There are many analyses that clearly demonstrate #1, but almost nothing on #2. Although the draft pangenome includes individuals from diverse populations (Fig. 1), and the 2022 perspective mentions “making genotyping accuracy less dependent on ancestry”, this is not really highlighted in the current manuscript. For example, is the improvement in variant calling more pronounced for individuals from populations not represented in GRCh38/T2T-CHM13 that are represented in the pangenome?

We agree with the reviewer that #2 is very important. We had focused our analyses on #1 because we felt that a full analysis of population specific biases requires pangenomes with more samples in them. To answer the reviewers’ question to the extent possible, we looked at

mappings of short reads to the pangenome relative to GRCh38, as well as comparing variant calls of both small variants and SVs, separately, to the pangenome relative to GRCh38.

The short-read mapping and small variant mapping results show consistent benefits across individuals of different ancestries. For example, for small variants (Supplementary Figure 24: A, pasted below for convenience) we see consistently higher numbers of PASS non-homozygous reference called variants vs. GRCh38, with the increases being constant across ancestry groups. Similarly, for read mapping, we see a consistent improvement of the identities of read alignments relative to the HPRC MC reference vs. GRCh38 (Supplementary Figure 24: B, pasted below). It is worth stressing that this picture may change as we integrate larger numbers of generally rarer variants into the pangenome, and it is also possible that there are specific loci where some samples local ancestries benefit from the move to a pangenome more than others (comparably to the RHCE gene results, shown in Supplementary Figure 26).

Supplementary Figure 24: A) Number of variants with at least one alternate allele (i.e. excluding homozygous for the reference allele) for each in the 1000 Genomes Project samples. The number of variants in the 1000 Genomes Project callset (x-axis) are compared to the variants found when aligning reads to the HPRC pangenome and calling variants with DeepVariant (y-axis). Points (samples) are colored by their “super-population” label from the 1000 Genomes Project. **B)** The proportion mapped reads that align perfectly (y-axis) is shown for a subset of samples from the 1000 Genomes Project, ordered by the number of variant called (x-axis). Two mapping approaches are compared: mapping short reads to GRCh38 with BWA (green); mapping to the HPRC pangenome with vg giraffe (orange). The samples were selected to span the x-axis.

For SVs, where we might expect a larger effect due to stronger divergence from the reference, we compared the number of variants per sample in our SV callsets (i.e. after Pangenie genotyping from short reads) to the respective numbers observed from 1KG short-read callsets.

In **Supplementary Figure 47** (pasted below), we colored the data points according to the superpopulation the sample originates from (AFR, AMR, EAS, EUR and SAS) and observed that they cluster by population, which is indeed consistent with different levels of detection bias per superpopulation. But we caution that the composition of the samples underlying the pangenome relative to the composition of the set of samples genotyped could potentially influence these results and a careful analysis is warranted. We plan to further quantify this once future version of a human pangenome with more samples become available.

Supplementary Figure 47. Number of SVs per sample in the HPRC PanGenie filtered set as well as the 1kGP Illumina calls for all 3,202 1KG samples. Samples are colored by superpopulation. The left plot excludes the african superpopulation, while the right plot shows the same results including african samples and including the assembly samples present in the graph (marked by a black circle).

We have summarised these findings on in the manuscript with the following text:

*“These new pangenomic workflows could potentially benefit individuals of different ancestries differently. For read mapping and small variant calling, we observed a consistent improvement across individuals (Supplementary Figure 24). Still, the pangenome might improve structural variant genotyping differently across individuals due to the stronger divergence of the alleles from the reference. In the 1000 Genomes Project cohort, we observed that the genotyped samples clustered by super-population labels (**Supplementary Figure 47**), which would suggest different levels of detection bias that are mitigated with the pangenome. Still, we caution that the composition of the samples underlying the pangenome relative to the composition of the set of samples genotyped could potentially influence these results and a careful analysis with more samples is warranted.”*

Also, a large part of the paper focuses on the assemblies (the first two main figures) and it takes a while to get to the actual pangenomes. Even after pangenomes are introduced, a lot of focus

(until page ~17/25) is put on small variants, where the gains are real but less exciting than those for the SVs.

We have added several additional analyses involving SVs, as discussed in this response. We believe the assessments of the assemblies, and in particular their structural and base-level correctness, is important and complementary to the later analysis in the paper.

Validation of new variants: given that a main selling point of the pangenomes approach is that it enables discovery of novel variants, it would be helpful to have a form of external validation for some of those. For short variants (see Page 16: “suggesting that most putative errors are in fact real variants”) -> can some of those be confirmed by e.g. Sanger? For new SVs (e.g. shown in Fig. 6) -> is there a way to confirm some of those? I guess those are difficult to PCR, but perhaps evaluating if there is evidence in existing long read data for some of those samples that the new SVs are real?

In the original submitted version of our paper, we had done several quality checks of our filtered set of PanGenie genotypes to ensure that these calls are reliable: (a) we compared the allele frequencies across the PanGenie genotypes for all 2,504 unrelated 1KG samples to the allele frequencies of the 44 HPRC panel haplotypes (Main Figure 6c) and observed high correlation; (b) we additionally plotted the allele frequencies of the computed genotypes vs. the heterozygosity, and observed a relationship close to Hardy-Weinberg Equilibrium (Supplementary Figures 31, 32, and 33). In addition to these experiments, we have now further analyzed the quality of the novel variants (wrt. The 1KG Illumina calls) that are contained in our filtered PanGenie set. For this purpose, we re-visited the leave-one-out experiment that we performed for the first submission of this paper (Supplementary Figure 30, described in detail in the Methods section, “Genotyping Evaluation based on assembly samples” p.66). Briefly, we took the haplotypes of one of the assembly samples out of the MC-VCF, and used the remaining samples to genotype the left-out sample. The predicted genotypes for the left-out sample were then compared to its true genotypes in order to evaluate PanGenie’s genotyping accuracy. While the previous leave-one-out experiment was run on the unfiltered PanGenie genotypes, we have now restricted the evaluation to only the variants contained in the filtered set and among those, separately evaluated performances of novel variants that are not in the 1KG short-read based Illumina calls, and variants that matched between ours and the 1KG callset, respectively. Results are shown in the Figure below (Supplementary Figure 47, pasted below). The top panels excludes variants that are unique to the left-out sample and thus not typable by any re-genotyping method. The bottom panels includes these variants. In general, genotype concordances of all lenient variants (brown, dark purple) are slightly higher compared to the concordances we observed for the unfiltered set (Supplementary Figure 30). Furthermore, concordances of the known variants are highest. This is expected, since these variants tend to be in regions easier to access by short reads. Concordances for novel variants are slightly worse. This is also expected, since these variants tend to be located in more complex genomic regions that are generally harder to access. However, even for these novel variants, concordances are still high, indicating that most of these variants are indeed real.

Supplementary Figure 37. Leave-one-out experiment for novel variants. A leave-one-out experiment was conducted by repeatedly removing one of the assembly-samples from the panel VCF and genotyping it based on the remaining samples. Plots show the resulting weighted genotype concordances for variants in our filtered PanGenie set. The novel variants include only SVs not contained in the 1kGP Illumina set, the known variants include only variants contained in these Illumina calls. Weighted genotype concordances are stratified by graph complexity: biallelic regions of the MC graph include only bubbles with two branches, and multiallelic regions include all bubbles with > 2 different alternative paths defined by the 88 haplotypes. The top panel excludes variants that are unique to the left-out sample and thus not typable by any re-genotyping method. Additionally, we plotted the results including untypables (bottom panel).

The paper nicely shows examples of structurally complex regions (e.g. C4, HLA, LPA) that have already been well characterized. How rare is this type of complexity? Does the pangenome approach enable detection of new regions not known previously to be so complex that are likely to be medically relevant?

To address this comment we have performed a systematic analysis of complex SVs and assessed their overlap with medically relevant genes, and with previously reported SVs from the 1KG project. We have included a new paragraph to describe this analysis, and a new supplementary table 16 to report these loci. We have pasted the new/modified text below for your convenience.

“We next turned our attention to complex multiallelic SVs, which have historically been difficult to map using reference-based methods. To perform an initial screen for complex SVs, we identified bubbles >10 kb size from minigraph that exhibited at least five structural alleles among the assembled haplotypes (Methods). We found that 620 of 76,506 total sites (0.81%) were complex, and that 44 overlap with medically relevant protein coding genes (Wagner et al., 2022) (Supplementary Table 16). Some of these are well known complex SV loci, and all are known to be structurally variable based on prior short-read SV mapping studies (Sudmant et al., 2015; Byrska-Bishop et al., 2022, Handsaker et al, 2015). However, whereas prior short-read SV calls at these loci are typically imprecise due to alignment issues and low resolution read-depth analysis methods, here we have resolved their structure at single base resolution. These results will help enable future efforts to study the role of complex variation in human disease; however, we note that further work will be required to more comprehensively identify medically relevant complex SVs, and to ensure the accuracy of each allele represented in the pangenome.

We then selected five clinically relevant complex SV loci for detailed structural analysis: RHD/RHCE, HLA-A, CYP2D6/CYP2D7, C4, and LPA (Methods). For each locus and graph, we identified its location within the graph and then annotated paths within this subgraph with known genes. We then traced the individual haplotypes through the subgraph to reveal the structure of each assembly. In CYP2D6/7 (Supplementary Figure 14), C4 (Supplementary Figure 15), and LPA (Supplementary Figure 16), we recapitulated previously described haplotypes. For CYP2D6/7, our calls matched 96% of haplotypes of 76 assemblies called by Cyrius using Illumina short-reads data (Chen et al., 2021). Two discrepancies appear to be caused by errors from Cyrius, and the third is a false duplication in the HG01071#2 pangenome assembly revealed by Flagger. This comparison suggests the pangenomes faithfully agree with existing knowledge of this complex loci. In RHD/RHCE (Figure 5A-C), in addition to previously described haplotypes, we also inferred the presence of 5 novel haplotypes, which included one duplication allele of the RHD gene, and one inversion allele occurring between the RHD and RHCE gene that results in the swapping of the last exon of both genes. Around HLA-A (Figure 5D-F; Supplementary Figure 17), two deletion alleles have been previously described – albeit with imprecise breakpoints (Sudmant et al., 2015) – but an insertion allele carrying an HLA-Y pseudogene is previously unreported. The long sequence (65 kb) inserted with HLA-Y occurs at high frequency (28%) but has little homology to GRCh38.

We also compared the representation of these five loci in the MC and PGGB graphs (Supplementary Figure 18). Each graph independently recapitulated the same haplotype structures. In general, in the PGGB graph many SV hotspots, including the centromeres, are transitively collapsed into loops through a subgraph representing a single repeat copy, a feature which tends to reduce the size of variants found in repetitive sequences. Assemblies that contain multiple copies of the homologous sequence traverse these nodes a corresponding number of times. In contrast, MC maintains separate copies of these homologous sequences.”

Additional points:

The abstract and discussion claim to reduce errors when discovering small variants by 34%, but I am not quite sure where that number comes from. I think it is from Sup. Figure 41 but am not sure how that number was derived.

Sup. Figure 41 shows the amount of errors for each approach, relative to the standard method of mapping reads to the reference genome with BWA-MEM. The points corresponding to the HPRC-Giraffe-DeepVariant pipeline are, on average, 34% lower than that GRCh38-BWAMEM-DeepVariant baseline. We have clarified the legend of this figure:

“Mapping to the HPRC pangenome with Giraffe and calling variants with DeepVariant (light blue circle) resulted in a reduction of errors of 34%, on average across samples, compared to mapping reads to the linear reference with BWA-MEM.”

In the main text and in this legend, we have also clarified that errors correspond to false positive and false negative calls.

For VNTRs: I am confused by the notion of TP/TN, etc. which seems to imply the presence or absence of a variant. My impression was that the VNTR itself exists in all genome copies, but then there is variation in copy number. Only upon reading the methods I realized these annotations seem to be based on the read mappings, not discovery of a variant itself. It would be helpful to define in the main text or legend how these “positive” vs. “negative”s are defined.

We have updated the text accordingly to indicate that the performance was measured on VNTR read mapping. Specifically, we updated the following text from:

The graph approach also outperformed the alternative in terms of true positives (TP), true negatives (TN), and false negatives (FN) (Figure 7A): The TN was on average 1.9% higher than the GRCh38 approach, and the TP was on average 0.087% higher. The graph approach also reduced FN by 2.1 fold. The slight increase in FP is possibly due to the boundary annotation of VNTRs on assemblies.

to:

The graph approach also outperformed the alternative in terms of the correctness of VNTR read mapping (Figure 7A): The true negatives were on average 1.9% higher than the GRCh38 approach, and the true positives were on average 0.087% higher. The graph approach also reduced false negatives by 2.1 fold. The slight increase in false positives is possibly due to the boundary annotation of VNTRs on assemblies.

How does VNTR copy number estimation change as a function of the repeat unit length or total length of the VNTR? Also the performance numbers seem modest compared to other short-read approaches based on coverage (e.g. Mukamel et al Science 2021). Is the poor performance due to lower overall coverage here?

The Mukamel et al. study had an emphasis on coding VNTRs and reported the genotyping performance for just 118 loci. In this work, we genotyped VNTRs across the whole genome and measured the performance for 60,861 loci. We provided a simplistic approach for VNTR genotyping as a proof of concept, showing that existing methods operating on hg38 can be improved by considering the read depth in a pangenome graph. We also noted that while methods exist to genotype VNTR length from short reads using sophisticated models (Bakhtiari et al Nat Commun 2021) or intricate bias correction (Mukamel et al Science 2021), it is beyond the scope of this work and deserves dedicated work to explore the utility of the pangenome graph in efficient and accurate VNTR genotyping.

Can VNTR copy number be assessed on the pangenome graphs using something other than coverage? Finally, how was “ground truth” defined for VNTR lengths?

We have added the following description to the methods section:

The ground truth for a VNTR in a genome is defined as the number of bases spanned by the VNTR, averaged from the two haplotypes.

The RNA-seq/ChIP-seq sections are quite limited and have modest results. Related to the comment above on diverse populations, I wonder if this could improve e.g. allele-specific binding or ASE estimates that face a problem with reference bias, especially in diverse genomes.

We agree with the reviewer that allele-specific expression is a natural use case for pangenomes in transcriptomics. We have added an analysis of allelic bias in mapped coverage over heterozygous variants, which shows that the pangenome-based pipeline reduces allelic bias and preserves a greater proportion of mapped coverage (**Supplementary Figure 40**). This is not a full ASE analysis, which would also include further downstream statistical estimation and hypothesis testing. Nevertheless, we believe that it is not a stretch to suppose that it would improve these methods’ statistical validity and power to provide them inputs with higher coverage and reduced bias.

Indeed, in our results, we apply the draft pangenome reference to map epigenomic data to SVs and then we apply the binomial test to identify events that are specific to an SV allele (Fig 7D). At the request of the reviewer, we checked how the draft pangenome reference enables ASE estimates in diverse genomes using the fact that the ChIP-seq came from individuals from African-ancestry (AF) or European-ancestry (EU). To do this, we tabulated how many heterozygous SVs were involved in ASE events observed only in AF genomes, EU genomes or both. We have found 194 SVs that were involved uniquely in African-ancestry genomes, 150 SVs that are uniquely involved in European-ancestry genomes and 216 SVs that were involved in both African and European-ancestry genomes. As expected, rare alleles were enriched for ancestry-specific events. We have amended the text to include this new result.

Supplementary Figure 45: Number of samples in which H3K4me1 peaks were assigned to the SV allele, the reference allele, or both alleles SVs with peaks are stratified into those that are observed only in African-ancestry genomes, only European-ancestry genomes, or both ancestries.

Minor points:

The `asmgene` feature of `paftools.js` doesn't seem to have usage documentation associated with it. <https://github.com/lh3/minimap2/search?q=asmgene>. How does it distinguish true duplications from errors?

As the reviewer correctly points out, `asmgene` does not distinguish between true duplicates and errors. Looking at `asmgene` results we observe the duplication trend and detect any outlier across assemblies. Regarding the documentation we added a citation to `hifiasm` paper where `asmgene` is described.

Fig. 3a: would be nicer if the "-"s were monospace so everything is aligned

Thanks for pointing this out, we have updated the Figure accordingly.

Fig. 3b: I don't see a bar for minigraph but it is in the legend. I guess it is not included because it focused on larger SVs and not small variants?

The reviewer is correct that small variants (<50 bp) are not included in the minigraph, and therefore no bar for minigraph in Figure 3B. To avoid confusion, now Figure 3B and 3C have their own legends.

Fig. 2c legend: it says duplications are relative to GRCh38 (152) but the y-axis seems to show raw numbers. Also, is it surprising that some African genomes have substantially higher numbers than the other samples?

Our pipeline to discover duplicated genes annotates orthologous gene copies to GRCh38, and then identifies additional copies of each gene. Because gene duplications are also enriched in misassemblies, `Flagger` was applied to remove spurious duplications. We investigated our pipeline for calling duplicated genes and found that the original orthology maps included regions annotated by `Flagger` as low quality, and were included in the total duplication count. We apologize for this mistake (and glad you prompted us to investigate!). In this revision we have applied `Flagger` to both the duplication annotations and the original gene counts. This has lowered the total number of duplicated genes from 1,529 to 1,115. The fraction of low-copy (rare) duplicated genes decreased from 80% to 71%, indicating that our results remain in agreement with previous studies (Sudmant 2015). Similarly, the fraction of duplications in segmental duplications remains: low, medium, and high copy-number changes were updated from 13, 40, and 80% were updated to 14, 50, and 81%. The highlighted genes *SPDYE2* and *GPRIN2* were unchanged (Figures 2F and 2G), and the maximum defensin gene copy number changed from 8 to 7. The most commonly duplicated gene families (Figure 2D, 2E) remain largely unchanged; there were some small adjustments to the count and ordering of some genes, but because the majority of the genes found in erroneously duplicated regions were in genes with low copy number changes, this does not affect the high copy-number changes much. Finally, the trend of increasing duplication count in African genomes remains.

How are the gene diagrams generated for fig. 5c/f (and similar supplementary figures for other regions?) Do the segments need to be manually annotated? Can this type of annotation be easily done for newly discovered complex regions? If the manual annotations are taken from elsewhere, it would be helpful to cite where those came from (e.g. it looks like Supp fig. 14 C4 haplotype annotations are based on Sekar et al. 2016).

We first identified structural haplotypes of each assembly by detecting SVs, including CNVs, inversions and gene conversions. We then counted the number of assemblies that had each structural haplotype and computed their frequency. The linear diagrams in Figure 5C/F are generated by the gggenes R package based on the structural haplotypes we identified. We have rephrased the description of this analysis in the Methods (entitled “Analysis of 5 complex loci”) to make this clearer.

The linear haplotype plot of copy number change of each assembly can also be automatically obtained by using odgi (v0.6.2, (Guarracino, Heumos, et al., 2022), https://odgi.readthedocs.io/en/latest/rst/tutorials/injecting_gene_arrows.html), although in this case we did not use this tool.

We manually annotated the ranges of the genomic regions to be shown in the bandage plots, and also the locations of genes and interesting segments, including the HERV-K insertions in C4A/C4B (<https://www.ncbi.nlm.nih.gov/nucore/U07856.1?report=genbank>), the CYP2D7 spacer (PMID: 33462347) and the KIV-2 repeat in LPA (PMID: 3670400). Since only GRCh38 coordinates for the regions and segments are needed as input, this annotation can be easily done for newly discovered complex regions.

If there is a concise diagram that could capture it, it could be helpful to somehow conceptually illustrate the differences in how the graphs for the three methods (Minigraph, MC, pggB) are constructed or how they represent certain regions in different ways. This could also help community users decide which graph is most appropriate for their use case.

Thank you for the suggestion. We have included a new supplementary figure 6 to illustrate the differences in pangenome graph construction between Minigraph, MC, and PGGB. We have pasted the new figure with its caption below for your convenience.

Supplementary Figure 6 | The differences in pangenome graph construction methods for Minigraph, MC, and PGGB. A) Two haplotypes (H_1 and H_2) vary in copy number of a chromosomal segment S . The S_1 , S_2 , and S_3 segments are highly similar with only a SNP or a small indel. **B)** Pangenome graph structures for Minigraph, MC, and PGGB. Minigraph used H_1 as an initial backbone and then merely augmented with SVs (≥ 50 bp) from H_2 , such that the SNP in S_2 is not represented in the pangenome graph. MC added small variants (< 50 bp) to the pangenome graph constructed by Minigraph. PGGB used a symmetric, all-by-all alignment of haplotypes to build a pangenome graph whose structure is not affected by the order of inputs (unlike Minigraph and MC). The critical difference in graph construction is that PGGB collapses highly similar copies of a chromosomal segment, while Minigraph and MC do not.

For Fig. 4 A/B, it would be helpful to label the number of variants falling in each category.

We have updated Figure 4A-B to include the average number of variants for each category.

Reviewer Reports on the First Revision:

Referees' comments:

Referee #1 (Remarks to the Author):

I'm happy with the revision. The authors engaged with the referees' comments and crafted a well-thought-out response letter. The companion papers referred to by the authors clarify some open technical points and dive deeper into biological implications.

Referee #2 (Remarks to the Author):

The authors have addressed all of the concerns I pointed out in my earlier review. In particular I find that figures 2 and 5 are significantly easier to interpret.

I did try to replicate parts of the RNA-seq simulation analysis, having the simulated reads available for download was good. The scripts given on the github site were readable, but not reproducible without significant amount of work, still better than most papers.

Referee #3 (Remarks to the Author):

The authors have done an impressive job addressing the comments raised. Congratulations on this work and I look forward to seeing it published.

While going through the revision, the following typos were noticed. This is probably not an exhaustive list so the text, in particular the supplementary figure legends, should be proofread carefully:

Supplementary Figure 45 legend is missing a period: "the reference allele, or both alleles SVs with peaks are" -> "the reference allele, or both alleles. SVs with peaks are"

Supplementary Figure 24 legend: "The proportion mapped reads" -> "The proportion of mapped reads"

In the discussion: "particularly as T2T assembly becomes" -> "particularly as the T2T assembly becomes"